


# Climate Change in the Baltic Sea Region: A Summary
H. E. Markus Meier[1,2], Madline Kniebusch[1], Christian Dieterich[2,†], Matthias Gröger[1], Eduardo
Zorita[3], Ragnar Elmgren[4], Kai Myrberg[5,6], Markus Ahola[7], Alena Bartosova[2], Erik Bonsdorff[8],
Florian Börgel[1], Rene Capell[2], Ida Carlén[9], Thomas Carlund[10], Jacob Carstensen[11], Ole B.
Christensen[12], Volker Dierschke[13], Claudia Frauen[1,14], Morten Frederiksen[11], Elie Gaget[15,16],
Anders Galatius[11], Jari J. Haapala[17], Antti Halkka[18], Gustaf Hugelius[19], Birgit Hünicke[3], Jaak
Jaagus[20], Mart Jüssi[21], Jukka Käyhkö[22], Nina Kirchner[19], Erik Kjellström[2], Karol Kulinski[23],
Andreas Lehmann[24], Göran Lindström[2], Wilhelm May[25], Paul A. Miller[25,26], Volker Mohrholz[1],
Bärbel Müller-Karulis[27], Diego Pavón-Jordán[28], Markus Quante[28], Marcus Reckermann[29],
Anna Rutgersson[30], Oleg P. Savchuk[27], Martin Stendel[12], Laura Tuomi[17], Markku Viitasalo[5],
Ralf Weisse[3], Wenyan Zhang[3]
[1]Department of Physical Oceanography and Instrumentation, Leibniz Institute for Baltic Sea Research
Warnemünde, Rostock, Germany
[2]Research and Development Department, Swedish Meteorological and Hydrological Institute, Sweden
[3]Institute of Coastal Systems-Analysis and Modeling, Helmholtz-Zentrum Hereon, Geesthacht, Germany
[4]Department of Ecology, Environment and Plant Sciences, Stockholm University, Stockholm, Sweden
[5]Marine Research Centre, Finnish Environment Institute, Finland
[6]Marine Research Institute, University of Klaipeda, Lithuania
[7]Swedish Museum of Natural History, Stockholm, Sweden
[8]Environmental and Marine Biology, Åbo Akademi University, Finland
[9]Coalition Clean Baltic, Uppsala, Sweden
[10]Information and Statistics Department, Swedish Meteorological and Hydrological Institute, Norrköping, Sweden
[11]Department of Bioscience, Aarhus University, Roskilde, Denmark
[12]National Centre for Climate Research, Danish Meteorological Institute, Copenhagen, Denmark
[13]Gavia EcoResearch, Winsen (Luhe), Germany
[14]Deutsches Klimarechenzentrum, Hamburg, Germany
[15]Department of Biology, University of Turku, Turku, Finland
[16]International Institute for Applied Systems Analysis (IIASA), Laxenburg, Austria
[17]Finnish Meteorological Institute, Helsinki, Finland
[18]Department of Biological and Environmental Sciences, University of Helsinki, Helsinki, Finland
[19]Department of Physical Geography, Stockholm University, Stockholm
[20]Department of Geography, Institute of Ecology and Earth Sciences, University of Tartu, Tartu, Estonia
[21]Pro Mare, Estonia
[22]Department of Geography and Geology, University of Turku, Finland
[23]Institute of Oceanology, Polish Academy of Sciences, Gdansk, Poland
[24]GEOMAR Helmholtz Centre for Ocean Research, Kiel, Germany
[25]Centre for Environmental and Climate Science, Lund University, Lund, Sweden
[26]Department of Physical Geography and Ecosystem Science, Lund University, Lund, Sweden
[27]Baltic Sea Centre, Stockholm University, Stockholm, Sweden
[28]Department of Terrestrial Ecology, Norwegian Institute for Nature Research (NINA), P.O. Box 5685 Torgarden,
N-7485 Trondheim, Norway
[29]International Baltic Earth Secretariat, Helmholtz-Zentrum Hereon, Geesthacht, 21502, Germany
[30]Department of Earth Sciences, Uppsala University, Sweden
†Deceased
*Correspondence to*: H. E. Markus Meier (markus.meier@io-warnemuende.de)



**Abstract.** Based on the Baltic Earth Assessment Reports of this thematic issue in Earth System Dynamics and recent peer-reviewed literature, current knowledge about the effects of global warming on past and future changes in climate of the Baltic Sea region is summarized and assessed. The study is an update of the Second Assessment of Climate Change (BACC II) published in 2015 and focusses on the atmosphere, land, cryosphere, ocean, sediments and the terrestrial and marine biosphere. Based on the summaries of the recent knowledge gained in paleo-, historical and future regional climate research, we find that the main conclusions from earlier assessments remain still valid. However, new long-term, homogenous observational records, e.g. for Scandinavian glacier inventories, sea-level driven saltwater inflows, so-called Major Baltic Inflows, and phytoplankton species distribution and new scenario simulations with improved models, e.g. for glaciers, lake ice and marine food web, have become available. In many cases, uncertainties can now be better estimated than before, because more models can be included in the ensembles, especially for the Baltic Sea. With the help of coupled models, feedbacks between several components of the Earth System have been studied and multiple driver studies were performed, e.g. projections of the food web that include fisheries, eutrophication and climate change. New data sets and projections have led to a revised understanding of changes in some variables such as salinity. Furthermore, it has become evident that natural variability, in particular for the ocean on multidecadal time scales, is greater than previously estimated, challenging our ability to detect observed and projected changes in climate. In this context, the first paleoclimate simulations regionalized for the Baltic Sea region are instructive. Hence, estimated uncertainties for the projections of many variables increased. In addition to the well-known influence of the North Atlantic Oscillation, it was found that also other low-frequency modes of internal variability, such as the Atlantic Multidecadal Variability, have profound effects on the climate of the Baltic Sea region. Challenges were also identified, such as the systematic discrepancy between future cloudiness trends in global and regional models and the difficulty of confidently attributing large observed changes in marine ecosystems to climate change. Finally, we compare our results with other coastal sea assessments, such as the North Sea Region Climate Change Assessment (NOSCCA) and find that the effects of climate change on the Baltic Sea differ from those on the North Sea, since Baltic Sea oceanography and ecosystems are very different from other coastal seas such as the North Sea. While the North Sea dynamics is dominated by tides, the Baltic Sea is characterized by brackish water, a perennial vertical stratification in the southern sub-basins and a seasonal sea ice cover in the northern sub-basins.

During the time in which this paper was prepared, shortly before submission, Christian Dieterich passed away (1964-2021). This sad event marked the end of the life of a distinguished oceanographer and climate scientist who made important contributions to the climate modeling of the Baltic Sea, North Sea and North Atlantic regions. This paper is dedicated to him.

# 1 Introduction

## 1.1 Overview

In this study, the results concerning climate change of the various articles of this thematic issue, the so-called Baltic Earth Assessment Reports (BEARs) coordinated by the Baltic Earth program[1] (Meier et al., 2014), and other relevant literature are summarized and assessed. We focus on the knowledge gained during 2013-2020 about past, present and future climate changes in the Baltic Sea region. The methodology of all BEARs follows the earlier assessments of climate change in the Baltic Sea region (BACC Author Team, 2008; BACC II Author Team, 2015). The aim of this review is to inform and update scientists, policymakers and stakeholders about recent research results. The focus is on the atmosphere, hydrosphere, cryosphere, lithosphere and biosphere. In contrast to the earlier assessments, we do not investigate the impact of climate change on human society. We start (Section 1) with a summary of key messages from the earlier assessments of climate change in the Baltic Sea region, a description of the Baltic Sea region and its climate, a comparison of the Baltic Sea with other coastal seas and a summary of current knowledge on global climate change assessed in the latest Intergovernmental Panel on Climate Change (IPCC) reports. In Section 2, the methods for the literature assessment, climate model data and uncertainty

---

[1] https://baltic.earth



estimates are outlined. In Section 3, the results of the assessment for selected variables (Table 1) under past
(paleoclimate), present (historical period with instrumental data) and future (until 2100) climate conditions are
presented, *inter alia* by summarizing the results in various papers of this special issue by Lehmann et al. (2021),
Kuliński et al. (2021), Rutgersson et al. (2021), Weisse et al. (2021), Gröger et al. (2021a), Christensen et al.
(2021), Meier et al. (2021a) and Viitasalo (2021) and by other relevant review studies. In Section 4, the interactions
of climate with other anthropogenic drivers are summarized from Reckermann et al., 2021. As the adjacent North
Sea has different physical characteristics and topographical features but is located in a similar climatic zone as the
Baltic Sea, we compare the results of this assessment with the results of the North Sea Region Climate Change
Assessment (NOSCCA; Quante and Colijn, 2016; Section 5). Knowledge gaps (Section 6), key messages (Section
7) and conclusions (Section 8) finalize the study.
**1.2 The BACC and BEAR projects**
This assessment is an update to the two BACC books, published as comprehensive textbooks in 2008 and 2015
(BACC Author Team, 2008; BACC II Author Team, 2015). The acronym BACC (**B**ALTEX **A**ssessment of
**C**limate **C**hange) refers to the Baltic Earth pre-cursor programme BALTEX (Baltic Sea Experiment; Reckermann
et al., 2011). From the beginning, BALTEX tried to approach three basic questions: 1. What is the evidence for
past and present regional climate change? 2. What are the model projections for future regional climate change?
3. Which impacts can we already observe in terrestrial and marine ecosystems?
First ideas for a comprehensive appraisal of the current knowledge on climate change and its impact on the Baltic
Sea region evolved in 2004 as it became evident that there was a demand for this, in particular by the Baltic Marine
Environment Protection Commission, the Helsinki Commission (HELCOM; BALTEX, 2005). A steering group
of leading experts from the Baltic Sea region was enlisted, which elaborated a grand chapter structure at several
preparatory workshops and meetings and also recruited a group of lead authors. In total, more than 80 scientists
from 12 countries and all relevant scientific disciplines contributed to the first regional climate change assessment
(BACC Author Team, 2008), which underwent a rigorous review process.
In 2011, a second edition of the BACC book was initiated as an update, but also as a complement to the first book,
by including new topics like an overview of changes since the last glaciation, and a new section on regional drivers
and attribution. The Second Assessment of Climate Change for the Baltic Sea Basin (BACC II Author Team,
2015) was published in 2015, used the same procedures and principles, but with a new steering and author group,
and under the auspices of Baltic Earth, the successor of BALTEX. Close collaboration with HELCOM was
envisaged from the very beginning, with HELCOM using material from both BACC assessments for their own
climate change assessment reports (HELCOM, 2007; 2013b).
In 2018, the Baltic Earth Science Steering Group decided to produce a series of new assessment reports, the
BEARs, on the current Baltic Earth Grand Challenges, Earth System models and projections for the Baltic Sea
Region. The BEARs are comprehensive, peer-reviewed review articles in journal format, and the update to BACC
II (this article) is one of the ten envisaged contributions summarizing the current knowledge on regional climate
change and its impacts, knowledge gaps and advice for future work. For further details about our knowledge on
climate change, the reader is referred to the other BEARs. The close collaboration with HELCOM is continued in



the joint HELCOM-Baltic Earth Expert Network of Climate Change (EN CLIME), which was assembled to produce a Baltic Earth – HELCOM Climate Change Fact Sheet for the Baltic Sea region[2].

Hence, this thematic issue comprises nine BEARs and, in addition, this summary of the current knowledge about past, present and future climate changes for the Baltic Sea region ("BACC III"). Below a few key-words characterizing the BEARs' contents are listed:

1. Salinity dynamics of the Baltic Sea (Lehmann et al., 2021): water and energy cycles with focus on Baltic Sea salinity during past climate variability, meteorological patterns at various space and time scales and mesoscale variability in precipitation, variations in river runoff and various types of inflows of saline water, exchange of water masses between various sub-basins and vertical mixing processes. The paper also includes the observed trends of salinity during the last >100 years.

2. Baltic Earth Assessment Report on the biogeochemistry of the Baltic Sea (Kuliński et al., 2021): terrestrial biogeochemical processes and nutrient loads to the Baltic Sea, transformations of C, N, P in the coastal zone, organic matter production and remineralization, oxygen availability, burial and turnover of C, N, P in the sediments, the Baltic Sea $CO_2$ system and seawater acidification, role of specific microorganisms in Baltic Sea biogeochemistry, interactions between biogeochemical processes and chemical contaminants.

3. Natural Hazards and Extreme Events in the Baltic Sea region (Rutgersson et al., 2021): extremes in wind, waves, and sea level, sea-effect snowfall, river floods, hot and cold spells in the atmosphere, marine heat waves, droughts, ice seasons, ice ridging, phytoplankton blooms and some implications of extreme events for society (including forest fires, coastal flooding, offshore wind mills and shipping).

4. Sea Level Dynamics and Coastal Erosion in the Baltic Sea Region (Weisse et al., 2021): sea level dynamics and coastal erosion in past and future climates. The current knowledge about the diverse processes affecting mean and extreme sea level changes is assessed.

5. Coupled regional Earth system modelling in the Baltic Sea Region (Gröger et al., 2021a): status report on coupled regional Earth system modeling with focus on the coupling between atmosphere and ocean, atmosphere and land surface including dynamic vegetation, ocean, sea ice and waves and atmosphere and hydrological components to close the water cycle.

6. Atmospheric regional climate projections for the Baltic Sea Region until 2100 (Christensen et al., 2021): comparison of coupled and uncoupled regional future climate model projections. As the number of atmospheric scenario simulations of the Euro-CORDEX program (Kjellström et al., 2018; Teichmann et al., 2018; Jacob et al., 2018) is large, uncertainties can be better estimated and the effects of mitigation measures can be better addressed compared to earlier assessments.

7. Oceanographic regional climate projections for the Baltic Sea until 2100 (Meier et al., 2021a): new projections with a coupled physical-biogeochemical ocean model of future climate considering global sea level rise, regional climate change and nutrient input scenarios are compared with previous studies and the differences are explained by differing scenario assumptions and experimental setups.

[2] https://helcom.fi/helcom-at-work/groups/state-and-conservation/en-clime/,
https://baltic.earth/projects/en_clime/index.php.en



8.   Climate change and the Baltic Sea ecosystem: direct and indirect effects on species, communities and

ecosystem function (Viitasalo, 2021): impact of past and future climate changes on the marine ecosystem.

9.   Human impacts and their interactions in the Baltic Sea region (Reckermann et al., 2021): interlinkages of

factors controlling environmental changes. Changing climate is only one of the many anthropogenic and

natural impacts that effect the environment. Other investigated factors are coastal processes, hypoxia,

acidification, submarine groundwater discharge, marine ecosystems, non-indigenous species, land use

and land cover (called natural) and agriculture and nutrient loads, aquaculture, fisheries, river regulations

and restorations, offshore wind farms, shipping, chemical contaminants, unexploded and dumped warfare

agents, marine litter and microplastics, tourism, and coastal management (called human-induced).

## 1.3 Summary of BACC I and II key messages

Quotation by the BACC II Author Team (2015):
"The key findings of the BACC I assessment were as follows:
• The Baltic Sea region is warming, and the warming is almost certain to continue throughout the twenty-first
century.
• It is plausible that the warming is at least partly related to anthropogenic factors.
• So far, and as is likely to be the case for the next few decades, the signal is limited to temperature and to directly
related variables, such as ice conditions.
• Changes in the hydrological cycle are expected to become obvious in the coming decades.
• The regional warming is almost certain to have a variety of effects on terrestrial and marine ecosystems—some
will be more predictable (such as the changes in phenology) than others.
The key findings of the BACC II assessment […] are as follows:
1.   The results of the BACC I assessment remain valid.
2.   Significant additional material has been found and assessed. Some previously contested issues have been

resolved (such as trends in sea-surface temperature).

3.   The use of multi-model ensembles seems to be a major improvement; there are first detection studies, but

attribution is still weak.

4.   Regional climate models still suffer from biases related to the heat and water balances. The effect of

changing atmospheric aerosol load to date cannot be described; first efforts at describing the effect of

land-use change have now been done.

5.   Data homogeneity is still a problem and is sometimes not taken seriously enough.
6.   The issue of multiple drivers on ecosystems and socioeconomics is recognized, but more efforts to deal

with them are needed.

7.   In many cases, the relative importance of different drivers of change, not only climate change, needs to

be evaluated (e.g. atmospheric and aquatic pollution and eutrophication, overfishing, and changes in land

cover).

8.   Estimates of future concentrations and deposition of substances such as sulphur and nitrogen oxides,

ammonia/ammonium, ozone, and carbon dioxide depend on future emissions and climate conditions.

Atmospheric warming seems relatively less important than changes in emissions. The specification of



future emissions is plausibly the biggest source of uncertainty when attempting to project future
deposition or ocean acidification.
9. In the narrow coastal zone, the combination of climate change and land uplift acting together creates a
particularly challenging situation for plant and animal communities in terms of adaptation to changing
environmental conditions.
10. Climate change is a compounding factor for major drivers of changes in freshwater biogeochemistry, but
evidence is still often based on small-scale studies in time and space. The effect of climate change cannot
yet be quantified on a basin-wide scale.
11. Climate model scenarios show a tendency towards future reduced salinity, but due to the large bias in the
water balance projections, it is still uncertain whether the Baltic Sea will become less or more saline.
12. Scenario simulations suggest that the Baltic Sea water may become more acidic in the future. Increased
oxygen deficiency, increased temperature, changed salinity, and increased ocean acidification are
expected to affect the marine ecosystem in various ways and may erode the resilience of the ecosystem.
13. When addressing climate change impacts on, for example, forestry, agriculture, urban complexes, and the
marine environment in the Baltic Sea basin, a broad perspective is needed which considers not only
climate change but also other significant factors such as changes in emissions, demographic and economic
changes, and changes in land use.
14. Palaeoecological 'proxy' data indicate that the major change in anthropogenic land cover in the Baltic
Sea catchment area occurred more than two thousand years ago. Climate model studies indicate that past
anthropogenic land-cover change had a significant impact on past climate in the northern hemisphere and
the Baltic Sea region, but there is no evidence that land cover change since AD 1850 was even partly
responsible for driving the recent climate warming."
For comparison, the findings of this assessment study can be found in Section 8.

## 1.4 Baltic Sea region characteristics

### 1.4.1 Climate variability of the Baltic Sea Region

The Baltic Sea region (including the Kattegat) is located between maritime temperate and continental sub-arctic
climate zones, in the latitude–longitude box 54°N–66°N × 9°E–30°E (Fig. 1). The climate of the Baltic Sea region
has a large variability due to the opposing effects of moist and relatively mild marine air flows from the North
Atlantic Ocean and the Eurasian continental climate. The regional weather regimes varies depending on the exact
location of the polar front and the strength of the westerlies, and both seasonal and interannual variations are
considerable. The westerlies are particularly important in winter, when the temperature difference between the
marine and continental air masses is large.
The southern and western parts of the Baltic Sea belong to the Central European mild climate zone in the westerly
circulation. The northern part locates at the polar front and the winter climate is cold and dry due to cold arctic air
outbreaks from the east. In terms of classical meteorology, during winter the polar front fluctuates over the Baltic
Sea region but during summer it is located farther to the north. Depending on the particular year, the central part
of the Baltic Sea can be either on the mild or the cold side of the polar front. The temperature difference between
winter and summer is much larger in the north. During warm summers and cold winters the air pressure field is





smooth and winds are weak, and blocking high pressure situations are common. During such periods, the weather
can be very stable for several weeks.

The climate of the Baltic Sea region is strongly influenced by the large-scale atmospheric variability (e.g.
Andersson, 2002; Tinz, 1996; Meier and Kauker, 2003; Omstedt and Chen, 2001; Zorita and Laine, 2000;
Lehmann et al., 2002). In particular, the North Atlantic Oscillation (NAO), blocking and, on longer time scales,
circulation patterns related to the Atlantic Multidecadal Oscillation (AMO) play important roles for the climate of
the Baltic Sea region. The AMO consists of an unforced component which is the result of atmosphere-ocean
interactions (e.g. Wills et al., 2018) and a forced component such as volcanic eruptions (Mann et al., 2021; Mann
et al., 2020). However, the relative importance of its forced and unforced components is still debated (Mann et al.,

2021).


The NAO is the dominant mode of near-surface pressure variability over the North Atlantic and its influence is
strongest in winter (Hurrell et al., 2003), when it accounts for almost one-third of the sea level pressure (SLP)
variance (e.g. Kauker and Meier, 2003). During the positive (negative) phase of the NAO the Icelandic Low and
Azores High pressure systems are stronger (weaker), leading to a stronger (weaker) than normal westerly flow
(Hurrell, 1995). Positive NAO phases are associated with mild temperatures and increased precipitation and
storminess whereas negative NAO phases are characterized by warm summers, cold winters, and less precipitation
(Hurrell et al., 2003). Increasing winter temperatures in the Baltic Sea have also been linked to an observed shift
in the storm tracks (BACC II Author Team, 2015). There is a large interannual to interdecadal variability in the
NAO, reflecting interactions with and changes in surface properties, including sea surface temperature (SST) and
sea ice cover. This makes it difficult to detect a possible long-term trend in the NAO.

Atmospheric blocking occurs when persistent high pressure systems interrupt the normally westerly flow over the
middle and high latitudes, e.g. the North Atlantic. By redirecting the pathways of midlatitude cyclones, blockings
lead to negative precipitation anomalies in the region of the blocking anticyclone and positive anomalies in the
surrounding areas (Sousa et al., 2017). In this way, blockings can also be associated with extreme events such as
heavy precipitation (Lenggenhager and Martius, 2019) or drought (Schubert et al., 2014).

The AMO describes fluctuations in North Atlantic sea surface temperature (SST) with a period of 50-90 years
(Knight et al., 2006). Thus only a few distinct AMO phases have been observed in the 150-year instrumental
record. However, a recent model study suggests that variations in the AMO may influence atmospheric circulation
that leads to additional precipitation over the Baltic Sea region (Börgel et al., 2018). Further, it was found that the
AMO altered the zonal position of the NAO and affected the regional imprint of the NAO for the Baltic Sea region
(Börgel et al., 2020).
**1.4.2 A unique brackish water basin**
The Baltic Sea is a unique brackish water basin in the World Ocean which has a salinity less than 24.7 g kg$^{-1}$ in all
areas (Leppäranta and Myrberg, 2009; Voipio, 1981; Magaard and Rheinheimer, 1974; Feistel et al., 2008;
Omstedt et al., 2014). The sea is very shallow (with a mean depth of only 54 m), and can be characterized as a
number of sub-basins (Fig. 2). The Baltic Sea has the only connection to the North Sea through the Danish straits



(Fig. 2). The exchange of water between the Baltic Sea and North Sea through the narrow straits is quite limited.
The Baltic Sea has a positive fresh water balance with an average salinity of about 7.4 g kg$^{-1}$ – this being only one-
fifth the salinity of the World Ocean, thus water masses are brackish. The Baltic Sea is located between mild
maritime and continental sub-arctic climate zones and partly ice-covered in every winter. However, it is completely
frozen over only during extremely cold winters. The highly variable coastal geomorphology and the extended
archipelago areas make the Baltic Sea unique (see Section 5).

The World Ocean has only four large brackish water basins (Leppäranta and Myrberg, 2009). These are from the
largest to the smallest the Black Sea (Ivanov and Belokopytov, 2013) located between Europe and Asia Minor, the
Baltic Sea, the Gulf of Ob in the Kara Sea (Volkov et al., 2002) and the Chesapeake Bay (Kjerfve, 1988), on the
east coast of the United States of America. All these sea areas developed into brackish water basins during the
Holocene. During the most recent (Weichselian) glaciation period the Black Sea was a freshwater lake, the Baltic
Sea and the Gulf of Ob were under the Eurasian ice sheet, and the Chesapeake Bay was a river valley (Leppäranta
and Myrberg, 2009). The mean depth of the Black Sea is 1200 m, and due to the strong salinity stratification and
extremely slow deep water renewal the water masses below 200 m are anoxic. The Sea of Azov in the north-
eastern part of the Black Sea is often frozen during the winter. The Gulf of Ob is the long (800 km), narrow estuary
of the River Ob in the Kara Sea in the Russian Arctic, and ice-covered in winter. Finally, Chesapeake Bay is a
small, very shallow basin and a drowned river valley or ria, in the humid subtropical climate zone, with hot
summers and ice formation in river mouths in some winters.

Table 2 gives basic information of the brackish water seas and other basins comparable with the Baltic Sea. Most
similar to the brackish water seas is Hudson Bay (Roff and Legendre, 1986). It is an oceanic, semi-enclosed basin
with a positive fresh water balance, and a salinity of about 30 g kg$^{-1}$. In contrast, small Mediterranean seas with a
negative fresh water balance and salinities above 40 g kg$^{-1}$ are found in the tropical zone; e.g. the Red Sea and
Persian Gulf. The largest lakes are comparable in size to the Baltic Sea, and the Caspian Sea is even larger in
volume.

The Baltic Sea basin is a very old geomorphological depression. Prior to the Weichselian glaciation this basin
contained the Eem Sea, which extended from the North Sea to the Barents Sea, making Fennoscandia an island.
At the end of the Weichselian glaciation, 13,500 years ago, the Baltic Ice Lake was formed by glacier meltwater.
During the Holocene fresh and brackish phases followed dictated by the balance of glacier retreats and
progressions, land uplift and eustatic changes of the global sea level (Tikkanen and Oksanen, 2002). The present
brackish phase commenced 7000 years ago, and since about 2000 years ago the salinity has been close to the
present level. Postglacial land uplift has slowly changed the Baltic Sea landscape, making it possible to observe
how land rises from the sea and how terrestrial life gradually takes over. People living in the region have adapted
to this slow long-term change.

### 324    1.4.3 The Baltic Sea - a specific European sea

The basic features of the European seas reveal key differences, in areal extent, depth profile, salinity level, fresh
water budget, climate, and tidal motions (Table 3). The Baltic Sea and the North Sea are shallow, with a mean
depth of less than 100 m; the Baltic can be described as a "coastal sea", with a mean depth of only 54 m. The Black




Sea and the Mediterranean Sea are much deeper, with mean depths of approximately 1200 m and 1500 m,
respectively, whereas the North-East Atlantic reaches the full oceanic depth of ca. 4 km, fringed by much shallower
continental shelf areas, at about 400 m. These depth differences influence, among other things, the mixing of the
water column, variability in temperature, and distribution of benthic ecosystems (Myrberg et al., 2019).

Among the European Seas, the Baltic Sea physics stands out in terms of its small tidal amplitudes, low salinity,
strong stratification and anoxic conditions. Additionally, frequent and spatially extensive upwelling and regular
seasonal ice cover are typical of the Baltic Sea (Leppäranta and Myrberg, 2009). To summarize:
● The Baltic Sea is permanently stratified due to a large salinity (density) difference between the fresh upper
layer and the more saline bottom layer. This limits ventilation, leading to oxygen deficiency in the bottom
layer. For instance in autumn 2016, some 70 000 km$^2$ of the seabed experienced permanent hypoxia.
Irregular Major Baltic Inflows (MBIs; Matthäus and Franck, 1992; Mohrholz, 2018) are the main
mechanism transporting oxygen-rich waters from the North Sea to Baltic Sea deeps. The associated salt
transport in turn intensifies vertical stratification and eventually enlarges hypoxic area (Conley et al.,
2002).
● In the small, semi-enclosed Baltic Sea, almost any winds are likely to blow parallel to some section of
the coast and thus cause coastal upwelling. At the Swedish south-western coast, upwelling occurs 25-40
% of time (Lehmann et al., 2012). At times, about one third of the entire Baltic Sea may be under the
influence of upwelling.
● Among European seas, ice is a unique feature of the Baltic Sea that strongly limits air-sea interaction and
modifies the Baltic Sea ecosystem in many ways.

The salinity in the Baltic Sea is not only an oceanographic variable as in other more ventilated seas, but also
integrates the complete water and energy cycles, with their specific Baltic Sea features. Baltic Sea salinity, and
especially its low basic value and the large variations, is also an elementary factor controlling the marine
ecosystem. The salinity dynamics is governed by several factors: net precipitation, river runoff, surface outflow of
brackish Baltic Sea water and the compensating deep inflow of higher salinity water from the Kattegat. The latter
is strongly controlled by the prevailing atmospheric forcing conditions. Due to freshwater supply from the Baltic
Sea catchment area and due to the limited water exchange with the World Ocean, surface salinity varies from > 20
g kg$^{-1}$ in Kattegat to < 2 g kg$^{-1}$ in the Bothnian Bay and is close to zero at the mouth of the River Neva, in the
easternmost end of the Gulf of Finland. In the vertical direction, the dynamics of the Baltic Sea is characterized
by a permanent, two-layer system because of a pronounced, perennial vertical gradient in salinity. In summer, a
shallow thermocline is also formed, complicating the vertical structure.
**1.5 Global climate change**
In the following, a brief overview is given of the latest global climate assessments, based on the IPCC Fifth
Assessment Report (AR5; IPCC, 2014b) and results so far available from the current Coupled Model
Intercomparison Project (CMIP) phase 6 (Eyring et al., 2016). The focus is on large-scale changes in climate that
are of particular relevance for the Baltic Sea region (mainly North Atlantic, Arctic). Furthermore, whenever
feasible, changes are described in terms of pattern scaling which relies on the fact that for many quantities the
geographical change patterns are sufficiently consistent across models and scenarios to emerge from the



background noise (IPCC, 2014a). Hence, changes in e.g. local temperatures can be scaled to changes per °C of
global mean temperature change relative to 1981-2005 (Christensen et al., 2019).

Our future climate change assessment relies on the concentration driven scenarios RCP2.6, RCP4.5 and RCP8.5
from the CMIP5 suite (RCP = Representative Concentration Pathway), corresponding to changes in radiative
forcing for the 21st century. Hence, policy targeted goals inspired by the United Nations Framework Convention
on Climate Change (UNFCCC; United Nations Climate Change, 2015) to limit global mean warming below 2.0
or 1.5°C compared to preindustrial level, i.e. prior to the 20th century (the Paris Agreement, PA), are not considered
in many scenario simulations but referred to studies within the Euro-CORDEX framework (Kjellström et al., 2018;
Teichmann et al., 2018; Jacob et al., 2018) and for a broader region. In order to achieve the goal of a significant
reduction of the risks and impacts of climate change, the Paris Agreement commits the participating countries to
aim "to reach global peaking of greenhouse gas emissions as soon as possible" and "to undertake rapid reductions
thereafter in accordance with best available science, so as to achieve a balance between anthropogenic emissions
by sources and removals by sinks of greenhouse gases in the second half of this century". Furthermore, the
countries "should take action to conserve and enhance, as appropriate, sinks and reservoirs of greenhouse gases
[…], including forests".

RCP8.5 is a totally unmitigated scenario and assumes a radiative forcing of +8.3 W m$^{-2}$ in year 2100, as compared
to the preindustrial period. Assumptions for RCP8.5 are described in Riahi et al. (2011). RCP8.5 has been criticized
because it assumes continued use of coal for energy production translating into too high greenhouse gas emissions.
Moderate mitigation actions are reflected by RCP4.5 (Thomson et al., 2011), and RCP2.6 was developed for
effective mitigation scenarios aiming at limiting global mean warming to ~+2°C (van Vuuren et al., 2011). With
respect to global development, RCP2.6 and RCP8.5 might be unrealistic (Hausfather and Peters, 2020). However,
both scenarios can be used as envelopes of plausible pathways of future greenhouse gas emissions.

Confidence levels expressing evidence and agreement are provided following the definitions of the IPCC (see
Method Section 2.3).

**1.5.1 Atmosphere**

**1.5.1.1 Surface air temperature**

For the three considered scenarios, the IPCC AR5 (IPCC, 2014a; 2014b; Collins et al., 2013) reported a likely
increase in global mean air temperature for the period 2081-2100 relative to 1986-2005 in the likely range (5th to
95th percentile of CMIP5 models) between 0.3 to 1.7°C (RCP2.6), 1.1 to 2.6°C (RCP4.5), and 2.6 to 4.8°C
(RCP8.5). The corresponding mean changes are 1.0°C (RCP2.6), 1.8°C (RCP4.5) and 3.7°C (RCP8.5; IPCC,
2014b).

The large-scale geographical patterns of change remain stable among CMIP5 models and are consistent with the
results of the IPCC AR4. The dominant feature is a strong warming of the Arctic north of 67.5 °N that exceeds
global mean warming by a factor 2.2 to 2.4. The Arctic warming is strongest for the winter season, when sea ice
retreat and reduced snow cover provide positive feedbacks (Arctic amplification), and weakest in summer, when
melting sea ice consumes latent heat and the ice free ocean absorbs heat (IPCC, 2014b). Besides these





408 thermodynamic processes, the lateral transport of latent heat into the Arctic increases under global warming.

409 Weakest warming is found over the Southern Ocean and in the North Atlantic south of Greenland with minimum

410 values per degree global warming of about 0.25°C °C$^{-1}$ (Fig. 12.10 in IPCC, 2014b). This is partly due to a deeper

411 ocean mixed layer that promotes vigorous oceanic heat uptake in these regions compared to others. Generally,

412 land masses warm at a rate 1.4 to 1.7 times more than open ocean regions, leading to a pronounced land-sea pattern

413 in the temperature anomaly and indicating to a lower effective heat capacity of continents compared to the ocean.

### 1.5.1.2 Precipitation

415 Projected global precipitation changes scale nearly linear with global mean temperature changes and range from

416 +0.05 mm d$^{-1}$ or ~2% (RCP2.6) to 0.15 mm d$^{-1}$ or ~5% (RCP8.5; IPCC, 2014a). As a result of an accelerated

417 global water cycle, the contrast between dry and wet regions in annual mean precipitation increases. Likewise,

418 there is high confidence that the contrast between wet and dry seasons will become more pronounced (IPCC,

419 2014a). In the mid to high latitudes, yearly mean precipitation generally increases, with the strongest response

420 over the Arctic, exceeding almost everywhere +12% °C$^{-1}$.

422 Precipitation changes vary greatly among models. Under RCP8.5, high latitude land masses will likely get more

423 precipitation, due to higher moisture content of the lower atmosphere and an increased moisture transport from the

424 tropics (IPCC, 2014a). In the northern hemisphere the poleward branch of the Hadley Cell will expand further

425 north, causing a northward expansion of the subtropical dry zone and reducing precipitation in affected regions.

426 Further dynamical changes probably include a poleward shift of mid-latitude storm tracks (Seager et al., 2010;

427 Scheff and Frierson, 2012) which is, however, of low confidence, especially for the North Atlantic region (IPCC,

428 2014a).

### 1.5.2 Cryosphere

430 The IPCC AR5 postulates a reduction of average February Arctic sea ice extent ranging from 8% for RCP2.6 to

431 34% for RCP8.5. For the monthly mean summer minimum in September, reductions range from 43% for RCP2.6

432 to 94% for RCP8.5. These values are given medium confidence, because of biases in the simulation of present day

433 trends and a large spread across models. For September, ice free conditions are reached before 2090 in 90 % of all

434 CMIP5 models.

436 The permafrost area is projected to decrease in a likely range from 24 ± 16% for RCP2.6 to 69 ± 20% for RCP8.5.

438 Arctic autumn and spring snow cover are projected to decrease by 5–10%, under RCP2.6, and 20–35% under

439 RCP8.5 (high confidence). In high mountain areas, projected decreases in mean winter snow depth are in a likely

440 range of 10–40 % for RCP2.6 and 50–90% for RCP8.5. The likely range of projected inland glacier mass

441 reductions (ice sheets excluded) between 2015 and 2100 varies from 18 ± 7% for RCP2.6 to 36 ± 11% for RCP8.5.

442 Regions with mostly smaller glaciers (e.g. Central Europe, Scandinavia) are projected to lose over 80% of their

443 current ice mass by 2100 under RCP8.5 (medium confidence), with many glaciers disappearing regardless of future

444 emissions (very high confidence).



### 1.5.3 Ocean

**1.5.3.1 Sea level**

For 2081-2100, global mean sea level (GMSL) is projected to rise between 0.40 m under RCP2.6 (likely range 0.26-0.55m) and 0.63 m under RCP8.5 (likely range 0.45-0.82 m) relative to 1986-2005 (IPCC, 2014a; their Chapter 13, Table 13.5). In all scenarios, thermal expansion gives the largest contribution to GMSL rise, accounting for about 30 to 55% of the projections. Glaciers are the next largest contributor, accounting for about 15-35%. By 2100, the Greenland Ice Sheet's projected contribution to GMSL rise is 0.07 m (likely range 0.04–0.12 m) under RCP2.6, and 0.15 m (likely range 0.08–0.27 m) under RCP8.5. The Antarctic Ice Sheet is projected to contribute 0.04 m (likely range 0.01–0.11 m) under RCP2.6, and 0.12 m (likely range 0.03–0.28 m) under RCP8.5. Uncertainties concerning the melting of ice sheets are, however, intensively discussed (Bamber et al., 2019).

Based on the same suite of model projections from CMIP5, the IPCC Special Report on the Ocean and Cryosphere in a Changing Climate (IPCC, 2019a) has updated these numbers by including new estimates of the contribution from Antarctica, for which new ice-sheet modelling results were available (Oppenheimer et al., 2019). While the differences in projected changes until 2100 are small for RCP2.6, projected changes for RCP8.5 increased by about 10 cm compared to AR5 (see Section 3.3.5.4).

**1.5.3.2 Water temperature and salinity**

By the end of the century, the projected global ocean warming ranges from about 1°C (RCP2.6) to more than 3°C (RCP8.5) at the surface and from 0.5°C (RCP2.6) to 1.5°C (RCP8.5) at a depth of 1km. The subtropical waters of the Southern Ocean and the North Atlantic are projected to become saltier, whereas almost all other regions become fresher, in particular the northern North Atlantic (IPCC, 2014a). The freshening at high latitudes in the North Atlantic and Arctic basin is consistent with a weaker Atlantic Meridional Overturning Circulation (AMOC), and a decline in the volume of sea ice, as well as with the intensified water cycles (IPCC, 2019a).

By the end of the century, the annual mean stratification of the top 200 m (averaged between 60ºS–60ºN, relative to 1986–2005) is projected to increase in the very likely range of 1–9% for RCP2.6 and 12–30% for RCP8.5 (IPCC, 2019a).

**1.5.3.3 Atlantic Meridional Overturning Circulation**

Based on the CMIP5 models, the AMOC is estimated to be reduced by 11% (1 to 24%) under RCP2.6 and 34% (12 to 54%) under RCP8.5. There is low confidence in the projected evolution of the AMOC beyond the 21st century (IPCC, 2014a).

**1.5.4 Marine biosphere**

By 2081-2100, net primary productivity relative to 2006-2015 will very likely decline by 4–11% for RCP8.5, due to the combined effects of warming, stratification, light, nutrients and predation, with regional variations between low and high latitudes (IPCC, 2019a).



Globally, and relative to 2006-2015, the oxygen content of the ocean by 2081-2100 is very likely to decline by
1.6–2.0% for the RCP2.6 scenario, or by 3.2–3.7% for the RCP8.5 scenario (IPCC, 2019a). While warming is the
primary driver of deoxygenation in the open ocean, eutrophication is projected to increase in estuaries due to
human activities and due to intensified precipitation, which increase riverine nitrogen loads under both RCP2.6
and RCP8.5 scenarios, both by mid-century (2031–2060) and later (2071–2100; Sinha et al., 2017). Moreover,
stronger stratification in estuaries due to warming is expected to increase the risk of hypoxia by reducing vertical
mixing (IPCC, 2019a; Hallett et al., 2018; Warwick et al., 2018; Du et al., 2018).
**1.5.5 Coupled Model Intercomparison Project**
Future climate change assessments like the coming IPCC sixth Assessment Report AR6 (due 2021/2022), will rely
on a new generation of Earth System Models (ESMs), developed during CMIP6, which offers a wider range of
scenarios than during CMIP5. In particular, scenarios aiming to limit global warming to 1.5°C and 2.0°C and
overshoot scenarios including negative emissions in the second part of the century will be available.

A subset of current CMIP6 models have been shown to be more sensitive to greenhouse gases than previous
generations of CMIP models. Thus, the estimated response to an instantaneous doubling of $CO_2$ (equilibrium
climate sensitivity, ECS) is higher in CMIP6 models (1.8 – 5.6°C) than in CMIP5 models (1.5 – 4.5°C) and their
predecessors (Meehl et al., 2020). Indeed, the first transient simulations with the CMIP6 EC-Earth ESM found
stronger warming than with earlier versions, with about half of the increase attributed to differences between
CMIP5 and CMIP6 greenhouse gas forcing (Wyser et al., 2020).

However, it turns out that models with the highest projected warmings fail to capture past warming trends well,
and therefore recent studies argue that those models should not be used for climate assessments and policy
decisions (Forster et al., 2020; Nijsse et al., 2019; Tokarska et al., 2020; Brunner et al., 2020). Furthermore,
systematic errors in many CMIP5 and CMIP6 models prevent the simulation of the observed 1951-2014 summer
warming trend in Western Europe, and that neither higher resolution nor better representation of the sea surface is
likely to improve this (Boé et al., 2020).
**2 Methods**
**2.1 Assessment of literature**
33 variables representing the components of the Earth system (atmosphere, land, terrestrial biosphere, cryosphere,
ocean and sediment, marine biosphere) of the Baltic Sea region were selected (Table 1) and with respect to past,
present and future climate changes corresponding scientific peer-reviewed publications and reports of scientific
institutes since 2013 were assessed by 48 experts (see the section about author contributions). The year 2013 was
chosen as a starting point because earlier material was included in the last assessment by the BACC II Author
Team (2015). Information about climate change available in the BEARs (Section 2.1) was summarized and cross-
references can be found in Table 1.

For the selected 33 variables and even more general, knowledge gaps (Section 6) and key messages (Section 7) as
well as overall conclusions (Section 8) were formulated. Key messages, new compared to the results of the BACC



II Author Team (2015), were marked. The identified changes of the selected variables of the Earth system and
their uncertainties, following the definitions of the IPCC reports as outlined in Section 2.3, are summarized in
Table 10. The attribution of a changing variable to climate change, here the deterministic response to changes in
external anthropogenic forcing such as greenhouse gas and aerosol emissions, is illustrated by Figure 34. This
study does not claim to be complete, neither with regard to the importance of the limited selection of variables for
the Earth system nor with regard to the discussed and assessed publications.

The assessment was done without influence from any political, economic or ideological group or party. The results
of the BEARs including this summary about climate change impacts in the Baltic Sea region were used by the
joint HELCOM-Baltic Earth Expert Network of Climate Change (EN CLIME) for the compilation of a Climate
Change Fact Sheet for the Baltic Sea region.

For further details about the assessment methods, the reader is referred to the BACC Author Team (2008) and the
BACC II Author Team (2015).
**2.2 Proxy data, instrumental measurements and climate model data**
In addition to selected figures that are reproduced from the literature, for the assessment previously published
datasets were analyzed and discussed.
**2.2.1 Past climate**
For the Holocene climate evolution, paleo-pollen data with a decadal resolution, reconstructing seasonal
temperature and precipitation changes compared to preindustrial climate (Mauri et al., 2015), were analyzed (Fig.
3). More accurate tree-ring data, resolving annual summer mean temperatures, are available for the past
millennium (Luterbacher et al., 2016) and have been discussed here (Fig. 4). For further details, the reader is
referred to Section 3.1.2.
**2.2.2 Present climate**
Historical station data of sea level pressure and sea surface temperature (SST) were used to calculate climate
indices such as the NAO (sea level pressure differences, Fig. 5) and the AMO (SST anomalies, Fig. 6), describing
decadal to multidecadal variability of the large-scale atmosphere circulation. Furthermore, selected records of
variables such as air temperature (Fig. 8), river runoff (Fig. 10), land nutrient inputs (Fig. 11, Table 5), glacier
masses (Fig. 12, Table 6), maximum sea ice extent (Fig. 14), ice thickness data (Figs. 15 and 16), length of the ice
season (Fig. 17), sea level (Fig. 24) and gridded data sets of air temperature, e.g. the land-based CRUTEM4 data
(Jones et al., 2012; Fig. 7, Table 5), and of precipitation, e.g. Copernicus data (Fig. 9), were analyzed.

For the Baltic Sea, intensive environmental monitoring started more than 100 years ago. Since 1898 an agreement
between various Baltic Sea countries on simultaneous investigations on a regular basis at a few selected deep
stations was signed and 1902 the International Council of the Exploration of the Sea (ICES) started its work.
Examples from the national monitoring programs for water temperature (Figs. 18, 19, 20) and salinity (Figs. 21
and 22) are shown, illustrating climate variability and climate change of the Baltic Sea.



In addition, some institutes such as the Swedish Meteorological and Hydrological Institute (SMHI) provide
environmental/climate indices, e.g. averaged sea level station data corrected for land uplift (Fig. 23) and hypoxic
and anoxic areas (Fig. 25).

Since 1979 satellite data have become available, complementing traditional Earth observing systems and having
the advantage of spatially high resolution (e.g.Karlsson and Devasthale, 2018).

Atmospheric reanalysis products, i.e. the combination of model data and observations (e.g. NCEP/NCAR, ERA40,
ERA-Interim, ERA5, UERRA), were important for calculating water and energy budgets of the Baltic Sea region
(BACC Author Team, 2008; BACC II Author Team, 2015). More recently, also ocean reanalysis products have
been developed (e.g. Liu et al., 2017; Axell et al., 2019; Liu et al., 2019) and were, for instance, used for the
evaluation of models (e.g. Placke et al., 2018).

Furthermore, various gridded datasets for North Sea SSTs exist and were compared (Fig. 33).

All data sets presented here are publicly online available. For further details on various datasets, the reader is
referred to Rutgersson et al. (2021).
**2.2.3 Future climate**
For the BEARs, regionalizations of Global Climate Models (GCMs) or ESMs from CMIP3 and CMIP5 analyzed
by (IPCC, 2014b) and (IPCC, 2019b) are assessed. The scenario simulations of CMIP5 are driven by greenhouse
gas concentration scenarios, the Representative Concentration Pathways, RCP2.6, 4.5 and 8.5 (see Section 1.5).

Uncoupled atmospheric regional climate simulations for the 21$^{st}$ century from the Euro-CORDEX framework,
calculated with several Regional Climate Models (RCMs) and global ESMs were analyzed by Christensen et al.
(2021) and conclusions are summarized here.

Furthermore, coupled atmosphere – sea ice – ocean simulations for the Baltic Sea and North Sea regions with one,
so-called Regional Climate System Model (RCSM, Dieterich et al., 2013; Bülow et al., 2014; Dieterich et al.,
2019; Wang et al., 2015; Gröger et al., 2015; Gröger et al., 2019; Gröger et al., 2021b) driven by eight ESMs and
three greenhouse gas concentration scenarios, i.e. RCP2.6, 4.5 and 8.5, were compared by (Christensen et al.,
2021). In this study, we present figures of these consistent results from the coupled atmosphere-ice-ocean scenario
simulations, e.g. for air temperature and precipitation (Fig. 26, Tables 7 and 8), and for sea surface temperature
(Fig. 29, Table 9). The state-of-the-art of coupled modeling is discussed by Gröger et al. (2021a). For further
details about the comparison between coupled and uncoupled scenario simulations, the reader is referred to
Christensen et al. (2021).

Novel compared to the assessment by the BACC II Author Team (2015) are high-resolution projections of glacier
masses including Scandinavian glaciers (Hock et al., 2019). In Figure 27 results are reproduced.


Oceanographic regional climate model projections for the Baltic Sea until 2100 driven by the atmospheric surface
fields of the above mentioned RCSM by Dieterich et al. (2019) have been developed and analyzed by Saraiva et
al. (2019a; 2019b) and Meier et al. (2021a; 2021b). In Meier et al. (2021b), global sea level rise was also
considered, a driver of the Baltic Sea climate variability that were previously neglected (cf. Hordoir et al., 2015;
Arneborg, 2016; Meier et al., 2017). Here, we compare the latest scenario simulation results by Saraiva et al.
(2019b) with previous projections by Meier et al. (2011a; 2011c) for, e.g.SST, sea surface and bottom salinities,
sea level (Fig. 30), bottom oxygen concentration (Fig. 31), and Secchi depth (Fig. 32),

For further details about the latest oceanographic regional climate model projections for the Baltic Sea, the reader
is referred to Meier et al. (2021a).
**2.3 Uncertainty estimates**
Uncertainties of future projections were estimated following the IPCC (2014a) guidance note for lead authors of
the Fifth Assessment Report on consistent treatment of uncertainties (Mastrandrea et al., 2010). These uncertainty
estimates are based upon a matrix of consensus and evidence reported in the literature. For the high confidence of
a statement, high levels of both consensus and cases of evidence are required.

In this assessment, we applied a three-level confidence scale measuring low, medium and high confidence of
identified climate changes (as defined in Section 2.1) of the selected 33 Earth system variables according to current
knowledge (Table 10). We assessed the sign of a change but not its magnitude. Only detected or projected changes
undoubtedly attributed to climate change were considered and synthetized in Figure 34. Changes likely not caused
by increasing greenhouse gas concentrations or changing aerosol emissions were not considered. Other external
drivers of climate variability are internal "random" variations of the climate system, land use, eutrophication,
contaminants, litter, river regulations, fishery, aquaculture, underwater noise, traffic, spatial planning, etc. (see
Reckermann et al., 2021).

Key messages of this assessment that are new compared to the previous assessment by the BACC II Author Team
(2015) are specially marked (Section 7).
**3 Current state of knowledge**
**3.1 Past climate change**
**3.1.1 Key messages from previous assessments**
Climate variations may be triggered by changes in drivers external to the climate system or may be due to internal
processes that reflect the non-linear, chaotic interactions between the different components of the climate system.
The analysis of past climate variations is, therefore, useful for two purposes. One is to estimate the reaction of the
climate to changes in the external forcing. The second is to better understand the mechanisms of internal climate
variations. Since future climate change will include a mixture of both types of climate variations, the analysis of
past climate variations is also necessary for better estimations of future climate change.



The past climate of the Baltic Sea region can be reconstructed from paleo-pollen and dendroclimatological records,
with different time resolutions and degrees of accuracy. Paleo-pollen in lake sediments give information about the
dominant plant species of a certain period. Combining the environmental ranges of those species in terms of annual
maximum temperatures, minimum temperatures and total annual precipitation allows an approximate
reconstruction of past climate conditions over the past millennia, with time resolutions of a few decades (e.g. Kühl
et al., 2002). Dendroclimatological data of tree ring widths, wood density and sometimes also carbon and oxygen
isotopic composition in tree-rings can be dated as exactly as at annual scales.

As described by the BACC II Author Team (2015), the climate history of the Baltic Sea region during the
Holocene, i.e. the last 12000 years, involved very large climate changes, much larger than those during the 20th
century. These climate changes were caused by strong changes in external forcing factors, in particular the Earth's
orbit. These changes first brought about a warming that terminated the Last Ice Age about 13000 years BP, then
caused a period of very warm temperatures (~ 3°C above preindustrial levels) centered around 6000 years BP (the
Holocene Thermal Maximum), followed by a slow temperature decline towards preindustrial levels. During this
long period, other shorter-lived climate events, with durations of a few centuries, caused abrupt drops of
temperature. These events, e.g. the Younger Dryas (12000 years BP) or the 8.2K event (8200 years BP) were
possibly related to abrupt changes in the North Atlantic circulation, when sudden melting of portions of the
remnants of the North American ice-sheet disturbed the circulation of the North Atlantic Ocean and disrupted the
poleward heat transport.

In general, annual precipitation is believed to have changed with the slow multicentennial-scale changes in
temperature. Warmer periods, in particular the Mid-Holocene Optimum, tended to be wetter, although the regional
heterogeneity may have been larger than for temperature.

Following the end of the last glaciation, the coastlines of the Baltic Sea underwent changes due to the interplay
between the rising global sea-level and the local rebound of the Earth's crust after the disappearance of the
Fennoscandian ice sheet. The weight of this ice sheet depressed Fennoscandia by about 500 meters, and its slow,
viscous rebound continuous today, with a rate of about 10 mm year$^{-1}$ at the northern Baltic Sea coast. Due to this
interplay, the Baltic Sea experienced periods of open or closed connections to the North Sea that governed the
transport of salinity and heat and the nature of the Baltic Sea ecosystems (Groß et al., 2018).

The climate evolution during more recent historical times – the past 1000 years (Section 3.1.3) - can be
reconstructed with better accuracy and higher time resolution due to better dendrochronological data availability.
These data show the imprint of the Medieval Warm Period (approx. 900-1350 AD), the Little Ice Age
(approx.1550-1850 AD) and the Contemporary Warm Period (1850-present) on the Baltic Sea region. These
periods were likely caused by changes in the external forcing (volcanic eruptions and solar radiation), and during
the Contemporary Warm Period also by the increase in anthropogenic greenhouse gases (Hegerl et al., 2003).

In the Baltic Sea, this succession of warm-cold-warm temperatures was accompanied by changes in the deep water
oxygen content, with low oxygen conditions in warmer periods (next section). The reasons for these oxygen



variations are still not fully understood, but may be relevant for the future, should future warming also cause lower
oxygen concentrations.

### 3.1.2 New paleoclimate reconstructions

Since the publication of the BACC II report (BACC II Author Team, 2015), new reconstructions of the evolution
of the European climate over the Holocene and over the past millennium have become available. Like previous
reconstructions, the new ones are based on paleo-pollen data and now comprise summer and winter temperatures
and summer and winter precipitation. They are available for 1000-year time segments (Mauri et al., 2015). The
reconstructions of the late-spring-summer temperature evolution over the past millennium are based on
dendroclimatological data, as the previous reconstructions, but they are now based on wood density measurements,
which reflect the slow climate variations better than tree-ring width. These reconstructions are available for
Western Europe from 755 AD onwards (Luterbacher et al., 2016). In this study, only the results for a regular
geographical box approximately covering the Baltic Sea region are discussed.

In addition, new regional climate simulations since the publication of the BACC II report better demonstrate the
connections between the Baltic Sea and North Atlantic climates on multidecadal timescales (Börgel et al., 2018;
Börgel et al., 2020; Kniebusch et al., 2019a).

### 3.1.3 Holocene climate evolution

The picture of the Holocene climate evolution from the BACC II report (BACC II Author Team, 2015) is
essentially confirmed, but the regional details are now clearer (Mauri et al., 2015). Between 7000 and 5000 years
BP, the Baltic Sea region (especially the western Baltic Sea) experienced a period with summer temperatures about
2.5-3°C warmer than in the preindustrial reference period (before the 20[th] century). However, according to these
reconstructions, the Eastern Baltic Sea region (Finland and the Baltic Republics) did not experience a Mid-
Holocene Optimum in summer, when temperatures were similar to the preindustrial period. In contrast, winter
temperatures showed a clear Mid-Holocene Optimum over the whole Baltic Sea region, lasting about 8000-4000
BP, with winter temperatures roughly 3°C warmer than during the preindustrial period. In the eastern Baltic Sea,
winter temperatures were even slightly higher, especially between 6000-5000 BP. As a result, annual mean
temperatures during the millennia of the Mid-Holocene optimum, were generally warmer than in the preindustrial
period. This warming was limited to the winter in the eastern Baltic Sea, where the amplitude of the annual
temperature cycle was clearly lower than in the preindustrial period.

The warm temperatures in the Baltic Sea region during the Mid-Holocene-Optimum, are not surprising, and
basically agree with the previous review (BACC II Author Team, 2015). They also agree with evidence from
regions further north, indicating that the Arctic Ocean in summer may have been ice-free during this period
(Jakobsson et al., 2010). These findings do not contradict the anthropogenic effect on climate observed during
recent decades. During the Mid-Holocene Optimum, the orbital configuration of the Earth was different and
favored warmer temperatures at northern high latitudes, especially in summer, as explained later. For the analysis
of climate impacts on ecosystems it is relevant that high latitudes were exposed to warm temperatures and reduced
sea-ice cover just a few millennia ago. However, at that time temperature changed at a much slower pace – around
2-3°C over several millennia – compared to present and projected rates of about 2°C in just a few decades.






For precipitation, the new reconstructions give a regionally more nuanced view of climate evolution during the
Holocene. The BACC II report (BACC II Author Team, 2015) indicated that warmer climates were generally more
humid. The new reconstructions (Mauri et al., 2015) modulate this vision and constrain the wetter conditions to
the eastern Baltic Sea region, both in summer and winter seasons, with a stronger signal in winter. Precipitation
anomalies in the eastern Baltic Sea region were of the order of + 1-2 mm month$^{-1}$ relative to preindustrial climate.
In the western Baltic Sea region, the Mid-Holocene Optimum tended to be slightly drier than the preindustrial
reference period both in summer and winter, with precipitation deficits of the order of 1-2 mm month$^{-1}$.

The main external forcing that drove the millennial climate evolution over the Holocene Period is the changing
orbital configuration of the Earth (the so-called Milanković cycles), as explained in the BACC-II report (BACC II
Author Team, 2015), and especially the variation in the time of the year of the perihelion (when Earth is nearest
to the sun). The perihelion is now at the beginning of January, but ~10000 BP it was in July. This changes the
seasonal distribution of solar insolation and determines the rate of melting of winter snow and its possible survival
into the next winter. The solar insolation at 60°N at the top of the atmosphere during the Holocene, derived from
Laskar et al. (2004) is depicted in Figure 3. The shift of the perihelion from summer to winter diminishes summer
insolation - and in principle summer temperature - and increases winter insolation during the past few millennia.
The long-term evolution of temperatures would, however, not be a linear response to the long-term evolution of
the seasonal insolation. For instance, the presence or absence of ice-sheets may influence the timing of the response
to increasing insolation during the early Holocene, delaying the Holocene temperature maximum with respect to
the annual insolation maximum. In wintertime, the insolation is rather weak, so that its effect may be overwhelmed
by other factors, such as changes in the large-scale atmospheric and oceanic heat transports.

For the last IPCC report, the mid-Holocene climate was simulated with 14 global Earth System models within
CMIP (Schmidt et al., 2011). These models were essentially the same as those used for future climate projections,
although in some cases with a few simplifications required by limitations in computer power and by the long
timescales involved. These models were driven by known external forcings, including the orbital forcing. The
common evaluation of these simulations with reconstructions helps to interpret the reconstructions and sheds light
on model limitations. An important aspect in this comparison was that the spatial resolution of global models was
relatively coarse, about 2 x 2 degrees longitude x latitude, so that smaller details within the Baltic Sea region
cannot be properly represented.

The simulations showed some agreements with the reconstructions, but also clear, not yet resolved disagreements
(Mauri et al., 2014). In summer, all models showed temperatures 2-3°C warmer during the Mid-Holocene
Optimum than in the preindustrial climate (Fig. 3). However, no model showed the gradient with clearer warming
in the western Baltic Sea, seen in the reconstructions. For wintertime, the disagreement was much clearer. Whereas
reconstructions show a clear warming over the whole region, the 14 models displayed widely varying patterns of
temperature change. Only three models agreed with the reconstructions. For precipitation, the models disagreed
with the west (wet) – east (dry) dipole shown by the reconstructions for summer and winter precipitation (Fig. 3).
Not a single simulation showed this pattern of summer precipitation change, and in general the simulated
precipitation deviations were much smaller than in the reconstructions. This disagreement regarding temperature



(especially in winter) and precipitation, known as the mid-Holocene conundrum, is not unique for the Baltic Sea
region, but was also found for the Mediterranean (Mauri et al., 2014; Liu et al., 2014). Errors in the applied external
(orbital) forcing can be ruled out, as this forcing can be accurately calculated during this period. The reasons for
the disagreement are still unknown. They may involve the influence of chaotic internal climate variations (unlikely
over such long time scales), model deficiencies, or reconstruction uncertainties.
**3.1.4 The past millennium**
For shorter periods closer to the present, like the past one or two millennia, the data available for reconstructing
past climate are denser and more accurate. Abundant dendroclimatological information is available, dated to the
exact year, in contrast to the uncertain decadal-scale dating of paleo-pollen data. Recently, temperature
reconstructions for Western Europe, spatially resolved and approximately covering the last 1200 years, have
become available (Luterbacher et al., 2016) and are presented here in some detail for the Baltic Sea region. These
data are based on analysis of wood density in tree rings. Wood density is more sensitive to growing season
temperature than tree-ring width. In addition, tree-ring width variations usually contain too weak multidecadal
scale variations, even when the year-to-year variations in temperature may be well captured. This makes wood
density a better proxy for temperature reconstructions at these latitudes.

Figure 4 shows the reconstructed growing-season temperature (spring-early summer) for the period 755-2000 AD,
averaged over the Baltic Sea region, based on the European reconstructions by Luterbacher et al. (2016). The
reconstructed temperature displays warmer conditions around 950 AD, confirmed also by the previous pollen-
based reconstructions (Mauri et al., 2015), colder conditions between 1200 and 1850 AD, followed by the recent
warming. This temperature evolution confirms that presented by the BACC II Author Team (2015). The relative
temperature difference between the Medieval Warm Period and the Contemporary Warm Period (mid 20th century)
agree within their respective uncertainties. According to these reconstructions, the Little Ice Age was on average
about 0.8 °C colder than the 20th century.

There is no new analysis of the causes of this temperature evolution specific for the Baltic Sea region. For Europe
as a whole, for which the reconstructions display a similar temporal pattern, the main identified forcings were
volcanic activity - more intense during the Little Ice Age and weaker during the Medieval Warm Period - and solar
activity, with roughly the reverse temporal signal (Luterbacher et al., 2016). With industrialization, greenhouse
gases have become dominant.

The CMIP5 project also included simulations of the past millennium with Earth System Models, although with
fewer models than for the mid-Holocene. These simulations have been compared with the temperature
reconstructions for Europe, in general yielding agreement. However, for the Baltic Sea region, the simulated
temperature changes tend to be smaller, especially for the transition between the Medieval Warm Period and the
Little Ice Age, with a modelled temperature difference of only ~0.2°C (compare with Figure 4 by Luterbacher et
al., 2016).

Climate fluctuations are driven not only by the external forcings but also by chaotic internal dynamics of the Earth
system. Regional climate simulations indicate that North Atlantic temperature variability influences Baltic Sea


temperatures (Kniebusch et al., 2019a) and precipitation (Börgel et al., 2018). North Atlantic temperatures tend to
fluctuate internally at multidecadal timescales, the AMO, and influences the atmospheric circulation of the Baltic
Sea region. Further, the interaction between internal modes of climate variability has recently been identified as a
key driver for the state of the Baltic Sea. Internal fluctuations in the North Atlantic are likely to influence the
spatial position of the NAO, affecting the regional importance of this climate mode for the Baltic Sea (Börgel et
al., 2020).

Climate simulations also indicate an impact of internally driven climate variability on the frequency of wind
extremes. In the present climate, the wintertime wind regime in the Baltic Sea is linked to the NAO, but at the
longer time scales of the preindustrial period, variations in wind extremes appear related neither to the mean wind
conditions nor to the external climate forcings (Bierstedt et al., 2015). In the recent centuries, the main driver of
trends of wind extremes over land appears to be land-use changes such as de- and reforestation (Bierstedt et al.,
2015; Gröger et al., 2021a, and references therein).

An important question is how North Atlantic variations can influence the state of the Baltic Sea, especially its
oxygen conditions, since freshwater input and water temperature (less strongly) affect the stratification of the water
column, and therefore the exchange of oxygen between the surface and deeper layers. Temperature also modulates
algal blooms and thus dissolved oxygen, when bacteria use oxygen to decompose dead algae. Analysis of sediment
cores indicated that the mid-Holocene Optimum, the Roman Period (2000 BP), and the Medieval Warm Period
were all periods of oxygen deficiency at the bottom of the Baltic Sea. Low oxygen conditions are also observed
during the Contemporary Warm Period, unique in their extent on a thousand year perspective (Norbäck Ivarsson
et al., 2019, and references therein). Hence, factors other than temperature, like nutrients input into the Baltic Sea,
can also affect oxygen conditions, and thus the reasons for those hypoxic phases during the past millennia are not
yet completely clear (Schimanke et al., 2012). It had been suggested that agricultural nutrient input was large
enough to influence oxygen conditions already during the Medieval Warm Period, perhaps also modulated by
changes in river runoff due to the described climate fluctuations (Zillén and Conley, 2010). However, a detailed
analysis of new sediments records find little evidence of anthropogenic eutrophication before the industrial period
(Norbäck Ivarsson et al., 2019; Ning et al., 2018; van Helmond et al., 2018). In view of the large temperature
increases projected for this region in the next decades, further study of the influence of climate on oxygen
conditions is warranted.

**3.2 Present climate change**

This section assesses our knowledge of Baltic Sea region climate variability during the past ~200 years, based on
instrumental records, model based reconstructions and reanalyses. We focus on changes in means, extremes, trends
and decadal to multidecadal climate variability.

**3.2.1 Atmosphere**

**3.2.1.1 Large-scale atmospheric circulation**

Long-term trends in NAO could not be detected (e.g. Deser et al., 2017; Marshall et al., 2020). For the period
1960-1990, a positive trend in NAO, with more zonal circulation, mild and wet winters and increased storminess
in central and Northern Europe was found (Hurrell et al., 2003; Gillett et al., 2013; Ruosteenoja et al., 2020).



However, after the mid-1990s, there was a tendency towards more negative NAO indices, i.e. a more meridional
circulation and more cold spells in winter (Fig. 5).

There is no consensus on how strongly the interannual NAO variability is forced externally (Stephenson et al.,
2000; Feldstein, 2002; Rennert and Wallace, 2009). Several external forcing mechanisms have been proposed,
most prominently SST (Rodwell et al., 1999; Marshall et al., 2001) and sea ice in the Arctic (Strong and
Magnusdottir, 2011; Peings and Magnusdottir, 2016; Kim et al., 2014; Nakamura et al., 2015). Other authors
(Screen et al., 2013; Sun et al., 2016; Boland et al., 2017) found no dependence on sea-ice extent. Furthermore,
the impact of changes in the Arctic on midlatitude dynamics are still under debate (Dethloff et al., 2006; Francis
and Vavrus, 2012; Barnes, 2013; Cattiaux and Cassou, 2013; Vihma, 2017).

A weakening of the zonal wind, eddy kinetic energy and amplitude of Rossby waves in summer (Coumou et al.,
2015) as well as an increased waviness of the jet stream associated with Arctic warming (Francis and Vavrus,
2015) in winter have been identified, which may be linked to an increase in blocking frequencies. Blackport and
Screen (2020) argued that previously observed correlations between surface temperature gradients and the
amplitude of Rossby waves have broken down in recent years. Therefore, previously observed correlations may
have to be reinterpreted as internal variability. On the other hand, it has been shown that observed trends in
blocking are sensitive to the choice of the blocking index, and that there is a huge natural variability that
complicates the detection of forced trends (Woollings et al., 2018), compromising the robustness of observed
changes in blocking.

With ongoing global warming, the Arctic will warm faster than the rest of the earth. This decrease of the poleward
temperature gradient will tend to weaken the westerlies and increase the likelihood of blockings. On the other
hand, maximum warming (compared to other tropospheric levels) will occur just below the tropical tropopause
due to the enhanced release of latent heat, which tends to increase the poleward gradient, strengthen upper-level
westerlies and affect the vertical stability, thus altering the vertical shear in midlatitudes. It is not clear which of
these two factors will have the largest effect on the jet streams (Stendel et al., 2021).

The atmospheric circulation over Europe naturally varies significantly on decadal time scales (Dong et al., 2017;
Ravestein et al., 2018). Proposed drivers for these circulation changes include polar and tropical amplification,
stratospheric dynamics and the AMOC (Haarsma et al., 2015; Shepherd et al., 2018; Zappa and Shepherd, 2017).
The attribution of drivers is more straightforward for local changes, in particular for the soil-moisture feedback,
for which an enhancement of heat waves due to a lack of soil moisture has been demonstrated (Seneviratne et al.,
2013; Teuling, 2018; Whan et al., 2015). Räisänen (2019) found only a weal effect of circulation changes on the
observed annual mean temperature trends in Finland, but circulation changes have considerably modified the
trends in individual months. In particular, circulation changes explain the lack of observed warming in June, the
very modest warming in October in southern Finland, and about a half of the very large warming in December.

As part of its natural variability, the North Atlantic warmed from the late 1970s to 2014 (Fig. 6). Recently, the
AMO began transitioning to a negative phase again (Frajka-Williams et al., 2017). Paleoclimate reconstructions
and model simulations suggest that the AMO might change its dominant frequency over time (Knudsen et al.,



2011; Wang et al., 2017). The impact of the AMO on climate is, however, independent of its frequency (Börgel et
al., 2018; Börgel et al., 2020). Its influence on regional climate has been analyzed in several studies (Enfield et al.,
2001; Knight et al., 2006; Sutton and Hodson, 2005; Ting et al., 2011; Casanueva et al., 2014; Ruprich-Robert et
al., 2017; Peings and Magnusdottir, 2014), some dealing with the Baltic Sea (Börgel et al., 2018; Börgel et al.,
2020; Kniebusch et al., 2019a). Kniebusch et al. (2019a) suggested that the influence of the AMO on temperature
during 1980-2008 might have been at least as strong as that induced by humans (IPCC, 2014b).
**3.2.1.2 Air temperature**
A significant increase in surface air temperature in the Baltic Sea region during the last century has been shown
previously (e.g. BACC Author Team, 2008; Rutgersson et al., 2014; BACC II Author Team, 2015). The
temperature increase was not monotonous but accompanied by large multidecadal variations that divided the 20th
century into three main phases: (1) warming from the beginning of the century until the 1930s; (2) slight cooling
until 1960s; and (3) a distinct warming during the last decades of the time series that has continued also during
2014-2020 (Figs. 7 and 8 and Table 4).

Linear trends of the annual mean temperature anomalies during 1878−2020 were 0.10 ℃ decade$^{-1}$ north of 60°N
as well as south of 60°N in the Baltic Sea region. This is larger than the global mean temperature trend and slightly
larger compared to the earlier BACC reports. Over the Baltic Sea surface air temperature trends were smaller than
over land. During 1856-2005, surface air temperature over the Baltic Sea increased by 0.06 and 0.08 ℃ decade$^{-1}$
in the central Baltic Sea and in the Bothnian Bay, respectively (Kniebusch et al., 2019a).

There is a large variability in annual and seasonal mean temperatures, in particularly during winter, but the
warming is seen for all seasons (being largest during spring in the northern part of the region).

Both daily minimum and daily maximum temperatures have increased. A decrease in the daily temperature range
(DTR) have been observed in many regions of the world, but there is no clear signal for the entire Baltic Sea region
(see for example, Jaagus et al., 2014, for DTR analysis of the Baltic countries).
These changes have also resulted in seasonality changes: the growing season has lengthened by about 5 days
decade$^{-1}$ in the period 1965–2016 (Cornes et al., 2019). From this follows that the cold season has become shorter.

Extreme air temperatures can be high or low, but extended periods of extreme temperatures (spells or waves) are
often the most influential. Averaged over land areas, warm spell duration increased during recent decades
(Rutgersson et al., 2021). For some regions, the average annual days defined as warm spells increased from 6–8
to 14 during recent decades. Along with more frequent and longer warm spells came decreases in the frequency,
duration and severity of cold spells, based both on observations (Easterling et al., 2016) and model results. The
length of the frost season and the annual number of frost days also decreased (Sillmann et al., 2013).
**3.2.1.3 Solar radiation and cloudiness**
Multidecadal variations of solar radiation at the Earth's surface, called "dimming" and "brightening", have been
observed in Europe and other parts of the world, particularly in the northern hemisphere (Wild et al., 2005; Wild,
2012; Wild et al., 2017).

One of the world's longest time series of global radiation, i.e. incoming solar radiation at the Earth's surface, is
from Stockholm, where measurements started in 1922. Recently, a first attempt to homogenize this time series was
made by Josefsson (2019). No significant trend was found over the whole time series, but there were large
variations over one to three decades. Other long time series of global radiation in Northern Europe are from
Potsdam, Germany, (Wild et al., 2021), and Tõravere, Estonia, (Russak, 2009). All three time series show a
minimum in global radiation around the mid-1980s. Then a clear increase or "brightening" of about 5-8% followed,
until at least 2005. Before the 1980s minimum there was a period of "dimming" at all stations but with differences
in the details. In Potsdam, there was rather stable dimming all the time from the late 1940s. In Tõravere, there was
a maximum dimming around mid-1960s, while Stockholm recorded high values both around 1950 and 1970, with
a minimum dimming in between. The data also suggest an early brightening in Stockholm time series, but this is
still uncertain, especially before 1950.

Current twentieth century reanalyses models provide results for surface solar radiation. However, most of them
fail to capture multidecadal surface radiation variability in central and southern Europe (Wohland et al., 2020).
The model CERA20C, which shows best results for central and southern Europe still gives questionable results
over Scandinavia, showing a weak increase instead of a decrease in surface solar radiation during the presumed
dimming period before 1980.

Satellite data allowing analyses of cloudiness and solar radiation at the Earth's surface are available since the early
1980s. For Europe, important work has been done within the EUMETSAT Satellite Application Facility on
Climate Monitoring (CM SAF). Several satellite data records have been validated and used in climate studies (e.g.
Urraca et al., 2017; Pfeifroth et al., 2018). At the highest latitudes of the Baltic Sea region there are however larger
uncertainties (Riihelä et al., 2015) or often no data at all, due to low standing Sun and slant viewing geometry from
the satellites.

The satellite data only cover the latest brightening period observed at ground-based stations in Europe. While the
geographical patterns of global average cloud conditions agree well among several satellite cloud-data sets, there
are clear differences in the distribution and size of trends (Karlsson and Devasthale, 2018). However, there seems
to be consensus on a decreasing trend in total cloud fraction of about 1-2% per decade over the Baltic Sea region
during 1984-2009.

Recent CM SAF satellite products on solar irradiance at the Earth's surface, the SARAH-2 and CLARA-A2
datasets, both agree well with station data according to Pfeifroth et al. (2018). In many cases this holds both for
climatological averages and for trend detection. The average trend for the period 1983-2015 is about +3 W m$^{-2}$
decade$^{-1}$ both at the stations closest to the Baltic Sea and in the SARAH-2 dataset. The three long-term stations
mentioned above are all used as reference stations for the satellite data validation. For example, the on-going
monitoring at stations spread over all Sweden show an average increase of about 8% (corresponding to +4 W m$^{-2}$
decade$^{-1}$) from 1983 until 2005-2006 (SMHI, 2021). In later years the solar radiation leveled off, but *inter alia* the
extremely sunny 2018 in Northern Europe contributed to keeping the trend increasing over time.

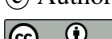

The multidecadal variations in the solar radiation at the Earth's surface were most probably caused by a
combination of changes in cloudiness and in anthropogenic aerosols. Which of the two drivers is the largest
contribution is still an open question, and might differ among regions. Aerosol concentrations over Northern
Europe decreased during the brightening period from the mid-80s onwards (Ruckstuhl et al., 2008; Russak, 2009;
Markowicz and Uscka-Kowalkowska, 2015; Glantz et al., 2019). Russak (2009) considered changes in cloudiness
caused by variations in atmospheric circulation to be the most important factor in Estonia, but aerosol changes also
played a role. In an early study of the modern radiation measurements in Sweden the strong increase in solar
radiation 1983-1997 was also accompanied by a clear decrease in total cloud cover, especially during the half-year
of summer (Persson, 1999). The satellite datasets SARAH-2 and CLARA-A2 where both derived using an aerosol
climatology as input. This underlines the important role of changes in cloudiness for surface solar radiation. Stjern
et al. (2009) also stressed the importance of the contribution of clouds and the atmospheric circulation for dimming
and brightening periods in Northern Europe.
Other studies, e.g. Ruckstuhl et al. (2008) and Wild et al. (2021), concluded that aerosol effects under clear skies
is the main contributor to the multidecadal variations of solar radiation in central Europe. Aerosol-induced
multidecadal variations in surface solar radiation could be expected also over oceans (Wild, 2016), but long-term
measurements are lacking. The interaction between aerosols and clouds, the indirect aerosol effects, needs also be
better understood and quantified.

### 972    3.2.1.4 Precipitation

During the twentieth century in the Baltic Sea region, changes in precipitation were spatially more variable than
for temperature (BACC II Author Team, 2015). Irregularly distributed precipitation measurement stations make it
difficult to determine statistically significant trends and regime shifts. Sweden shows an overall wetting trend since
the 1900s, in particular since the mid 20th century (Chen et al., 2020). In Finland, the overall increase detected for
1961-2010 is neither regionally consistent nor always statistically significant (Aalto et al., 2016). The same holds
for the Baltic States (Jaagus et al., 2018). In the south of the Baltic Sea region, changes were small and not
significant. Nevertheless, precipitation averaged over the Baltic Sea catchment area has increased since 1950 due
to an increase in winter (Fig. 9).
The number of heavy precipitation days is largest in summer. Compared to southern Europe, precipitation extremes
in the Baltic Sea region are not as intense, with daily amounts typically ranging from 8 to 20 mm (Cardell et al.,
2020). Extreme precipitation intensity increased during the period 1960-2018. An index for the maximum annual
five consecutive days of precipitation (Rx5d) shows significant increases of up to 5 mm per decade over the eastern
part of the Baltic Sea catchment (EEA, 2019b). The change is more pronounced in winter than in summer.

### 987    3.2.1.5 Wind

In situ observations allow direct analysis of winds, in particular over sea (e.g. Woodruff et al., 2011). However, in
situ measurements, especially over land, are often locally influenced, and inhomogeneities make the
straightforward use of such data difficult, even for recent decades. Therefore, many studies use reanalyses rather
than direct wind observations. But analysis of storm-track activity for longer periods using reanalysis data suffers
necessarily from uncertainties associated with changing data assimilation and observations before and after the



introduction of satellites, resulting in large variations of storm-track changes across assessments (Wang et al.,
2016; Chang and Yau, 2016).

Owing to inherent inhomogeneities and the large climate variability in the Baltic Sea region, it is unclear whether
there is a general trend in wind speed in the recent climate. Results regarding changes or trends in the wind climate
are strongly dependent on period and region considered (Feser et al., 2015). Due to the strong link to large-scale
atmospheric variability over the North Atlantic, conclusions about changes over the Baltic Sea region are perhaps
best made in a wider spatial context, considering *inter alia* the NAO.

Recent trend estimates for the total number of cyclones over the Northern Hemisphere extratropics during 1979-
2010 revealed a large spread across the reanalysis products, strong seasonal differences, as well as decadal-scale
variability (Tilinina et al., 2013; Wang et al., 2016; Chang et al., 2016; Matthews et al., 2016; Chang et al., 2012).
Common to all reanalysis datasets is a weak upward trend in the number of moderately deep and shallow cyclones
(7 to 11% per decade for both winter and summer), but a decrease in the number of deep cyclones, in particular
for the period 1989-2010. Chang et al. (2016) reported a minor reduction in cyclone activity in the Northern
Hemisphere summer due to a decrease in baroclinic instability as a consequence of Arctic temperatures rising
faster than at low latitudes. Chang et al. (2012) also noticed that state-of-the art models (CMIP5) generally
underestimate this trend. In the Northern Hemisphere winter, recent studies reported a decrease in storm track
activity related to Arctic warming (Ceppi and Hartmann, 2015; Shaw et al., 2016; Wills et al., 2019; Stendel et al.,
2021).


Despite large decadal variations, there is still a positive trend in the number of deep cyclones (< 980 hPa) over the
last six decades, which is consistent with results based on the NCEP reanalysis since 1958 over the northern North
Atlantic Ocean (Lehmann et al., 2011). Using an analogue-based field reconstruction of daily pressure fields over
central to Northern Europe (Schenk and Zorita, 2012), the increase in deep lows over the region might be
unprecedented since 1850 (Schenk, 2015). However, for limited areas the conclusions were rather uncertain.

The effect of differential temperature trends on storm tracks has been recently addressed, both in terms of upper
tropospheric tropical warming (Zappa and Shepherd, 2017) and lower tropospheric Arctic amplification (Wang et
al., 2017), including the direct role of Arctic sea-ice loss (Zappa et al., 2018), and a possible interaction of these
factors (Shaw et al., 2016). The remote and local SST influence has been further examined by Ciasto et al. (2016),
who confirmed the sensitivity of the storm tracks to the SST trends generated by the models and suggested that
the primary greenhouse gas influence on storm track changes was indirect, acting through its influence on SSTs.
The importance of the stratospheric polar vortex for storm track changes has received more attention (Zappa and
Shepherd, 2017). In an aqua planet simulation, Sinclair et al. (2020) found a decrease in the number of extratropical
cyclones and a poleward and downstream displacement due to an increase in diabatic heating.

**3.2.1.6 Air pollution, air quality and atmospheric nutrient deposition**

Air pollution continues to significantly impair the health of the European population, particularly in urban areas.
Brandt et al. (2013) estimated the total number of premature deaths due to air pollution in Europe in the year 2000
to be ~680 000 year$^{-1}$. Although this number was predicted to decrease to approximately 450 000 by 2020, it is



1033 still a matter of grave concern. Particulate matter concentrations were reported to be the primary reason for adverse

1034 health effects. Estimates indicated that PM2.5 concentrations in 2016 were responsible for ~412 000 premature

1035 deaths in Europe, due to long-term exposure (EEA, 2019a).


1037 The state of air pollution is often expressed as air quality, when human health is in focus. The ambient air quality

1038 in the Baltic Sea region is dominated by anthropogenic emissions, and natural emissions play only a minor role.

1039 These emissions show an overall decreasing trend in recent years (EEA, 2019a), as reflected in the ambient

1040 concentrations reported in the EMEP status report (EMEP, 2018). To quantify air quality concentrations of certain

1041 gases and particulate matter are used as measures. The general conclusions in the field of air quality reported by

1042 the BACC II Author Team (2015) still hold today, i.e. that land-based emissions and concentrations of major

1043 constituents continue to decrease due to emission control measures, with the possible exception  of certain

1044 emissions from the shipping sector. Sulphur emissions from shipping have continued to decrease strongly in the

1045 Baltic Sea from 2015, due to much lower limit values for the sulphur content of ship fuel in the emission control

1046 areas. A noticeable decrease in nitrogen emissions due to the newly (2021) implemented nitrogen emission control

1047 area (NECA) is expected in the next decade.

1049 In Europe, the pollutants most harmful to human health are particulate matter (PM), nitrogen dioxide ($NO_2$) and

1050 ground-level ozone ($O_3$). About 14% of the EU-28 urban population was exposed to $O_3$ concentrations above the

1051 EU target value threshold (EEA, 2019a). When compared to other European countries, air pollution was relatively

1052 low in Scandinavia, exception for a few urban traffic hotspots, with annual mean $NO_2$ concentrations elevated to

1053 near the limit value (EEA, 2019b). In this comparison, Northern Germany is located in the lower mid-field, while

1054 northern Poland is among the more polluted countries, especially with PM. Biomonitoring samples analyzed for

1055 toxic metals by Schröder et al. (2016) tended to show lowest concentrations in Northern Europe.

1057 Contributions to air pollution and pollutant deposition in coastal areas by shipping can be substantial. Major

1058 pollutants from shipping are $SO_2$, $NO_X$ and PM (including black carbon). The BACC II assessment estimated

1059 emissions from shipping in the Baltic Sea region. Several studies have recently been published, including Jonson

1060 et al. (2015), Claremar et al. (2017), and Karl et al. (2019b), which use chemistry transport models to predicted

1061 ambient concentrations from known emissions. They show, as expected, that the highest air pollution

1062 concentrations due to ship exhaust are found near major shipping lanes and harbors, but also that considerable

1063 concentrations of $NO_2$ and PM reach populated land areas. This effect is pronounced in the south-western Baltic

1064 Sea area (Quante et al., 2021). Exact numbers from such modelling should still be interpreted with care, as shown

1065 by Karl et al. (2019a), who compared output from three state-of-the-art chemistry transport models for the Baltic

1066 Sea area.

1068 The most important recent change in shipping emissions in the North and Baltic seas are due to the 2015

1069 strengthening of the fuel sulphur content limit for the Sulphur Emission Control Areas (SECAs), by lowering the

1070 maximum allowed sulphur content from 1 to 0.1%. Model calculations indicate large reductions in sulphur

1071 deposition in countries bordering these two sea areas after the implementation of the lowered sulphur limit (Gauss

1072 et al., 2017). Barregard et al. (2019) estimated the contribution of Baltic Sea shipping emissions to PM2.5 before

1073 2014 and after 2016 the new SECA regulation of marine fuel sulphur was implemented. These authors also



estimated human exposure to PM2.5 from shipping and its health effects in the countries around the Baltic Sea. They concluded that PM2.5 emissions from Baltic Sea shipping, and resulting health impacts decreased substantially after the 2015 SECA regulation. Population exposure studies estimating the influence of shipping emissions for selected Baltic Sea harbor cities were published for Rostock, Riga and Gdańsk–Gdynia by Ramacher et al. (2019) and for Gothenburg by Tang et al. (2020). Ramacher et al. (2019) found that shipping emissions strongly influence $NO_2$ exposure in the port areas (50–80 %), while the average influence in home, work and other environments is lower (3–14 %) but still with strong influence close to the ports. It should, however, be noted that reduction of sulphur emissions to the atmosphere by the use of new cleaning techniques (e.g. open loop scrubbers) can increase the risk of acidification and marine pollution (Turner et al., 2017; 2018).

Johansson et al. (2020) published a first comprehensive assessment of emissions from leisure boats in the Baltic Sea. While the modeled $NO_x$ and PM2.5 emissions from leisure boats are clearly lower than those from commercial shipping, these first estimates suggest that carbon monoxide (CO) emissions from leisure boats equal 70 % of the registered shipping emissions and non-methane volatile organic carbon (NMVOC) emissions equal 160 %. It should be noted that most of the leisure boat emissions occur in summer, and often occur near areas for nature conservation and tourism. Most of these emissions can be attributed to Swedish, Finnish and Danish leisure boats, but the leisure boat fleet has the potential for large future increases also in Russia, Estonia, Latvia, Lithuania and Poland.

Air pollution leads to environmental degradation by affecting natural ecosystems and biodiversity. Ground-level ozone ($O_3$) can damage crops, forests and other vegetation, impairing growth and reducing biodiversity. According to a recent study by Proietti et al. (2021), assessed trends of the $O_3$ mean concentration in Northern Europe were not statistically significant for the time period from 2000 to 2014. The annual mean ozone concentration is reported to be slightly below 35 ppb, as compared to 43 to 45 ppb in the Mediterranean Region, for which a significant decreasing trend is found. The exposure index AOT40 (sum of the hourly exceedances above 40 ppb, for daylight hours during the growing season) significantly declined in all European regions except for Northern Europe, for which a positive but not significant trend is seen. On the nation level among the six European countries showing a positive trend were Denmark, Germany, Sweden (Proietti et al., 2021). A clear difference in trends between rural sites and other station typologies is found for Europe for the period 2000 to 2017. I.e. for traffic sites a substantial increase of annual mean $O_3$ concentration was observed, in contrast to rural stations, for which a slight decrease was found (Colette and Rouïl, 2020). Regarding the monitored population exposed to ozone all countries in Europe show a decrease from 2000 to 2014 (NDGT60 > 25 days per year; Fleming et al., 2018).

Harmful exposure and impacts of air pollutants on ecosystems are assessed using the concept of critical loads (CLs; Nilsson and Grennfelt, 1988). The CL is the amount of pollutants that an ecosystem can tolerate without risking unacceptable damage. The most harmful air pollutants in terms of damage to ecosystems in addition to $O_3$ are ammonia ($NH_3$) and nitrogen oxides ($NO_X$). It is estimated that about 62% of the European ecosystem area is still exposed to high levels of $NO_X$, leading to exceedances of CLs for eutrophication in all countries in 2016 (EEA, 2019a). Hotspots of exceedances of CLs for acidification in 2016 were the Netherlands and its borders with Germany and Belgium, southern Germany and also Czechia. However, most of Europe including the Baltic Sea region did not exceed the CLs for acidification (EEA, 2019a).






Since the 1980s, the total nitrogen deposition on the Baltic Sea has decreased substantially, due to an overall
reduction of European emissions, but emission and deposition reductions have stalled since the mid-2000s (Colette
et al., 2015; Gauss et al., 2021). Atmospheric phosphorus deposition remains highly uncertain in amount and trends
(HELCOM, 2015; Kanakidou et al., 2018; Ruoho-Airola et al., 2012).

Air quality and climate interact in several ways. On the one hand, air pollutants can affect climate both directly
and indirectly by changing the radiative balance of the atmosphere. On the other hand, climate change alters
meteorological conditions, which may affect concentrations of air pollutants via several pathways, since air quality
is strongly dependent on weather (Jacob and Winner, 2009). The effects of important meteorological and climate
variables on surface $O_3$ and PM were discussed in a comprehensive review by Doherty et al. (2017). The
connection between high temperatures and increased ground-level ozone concentrations is well established.
Increases in temperature related to climate change (i.e. during heat waves) are expected to lead to higher ozone
concentrations in certain regions with the required precursor concentrations. Other important meteorological
factors influencing air pollution concentrations are a possible change in the number of midlatitude cyclones and
in the number of occurrences and duration of stagnant weather conditions (Jacob and Winner, 2009).
**3.2.2 Land**
**3.2.2.1 River discharge**
The total river discharge to the Baltic Sea is approximately 14,000 m$^3$ s$^{-1}$ (Bergström and Carlsson, 1994). This is
substantially more than the direct net precipitation (precipitation minus evaporation) on the Baltic Sea itself, which
has been estimated at 1,000-2,000 m$^3$ s$^{-1}$ (Meier and Kauker, 2003; Meier and Döscher, 2002; Meier et al., 2019d),
see also the discussion by Leppäranta and Myrberg, (2009). In other words, most of the fresh water entering the
Baltic Sea comes from the terrestrial part of the catchment. Therefore, the fresh water input to the Baltic Sea cannot
be described entirely with only climatic parameters. Non-climatic drivers of runoff include river regulation by
dams and reservoirs, land-use changes in the catchment, and water uptake for irrigation. Although dams are known
to have altered the seasonality of discharge (e.g. McClelland et al., 2004; Adam et al., 2007; Adam and
Lettenmaier, 2008), they do not seem to be responsible for annual discharge changes. In the long-term, net
precipitation over the catchment area and river runoff are strongly correlated (Meier and Kauker, 2003).

For the period 1850-2008, the total river discharge from the Baltic Sea catchment area, reconstructed from
observations (Bergström and Carlsson, 1994; Cyberski et al., 2000; Hansson et al., 2011; Mikulski, 1986) and
hydrological model results (Graham, 1999), showed no statistically significant trend but a pronounced
multidecadal variability, with a period of about 30 years (Meier et al., 2019d). Furthermore, summed river flow
observations in the period 1900-2018 (Lindström, 2019) and a historical reconstruction of the annual river
discharge for the past 500 years showed no statistically significant trend either (Hansson et al., 2011). However,
river runoff from northern Sweden, a part of the catchment area of the Bothnian Bay, significantly increased since
the 1980s compared to 1911-2018 (Lindström, 2019).

There are indeed substantial regional and decadal variations in the river flow. Stahl et al. (2010) studied near-
natural rivers of Europe over the period 1942-2004 and found a clear overall pattern of positive trends in annual





streamflow in the northern areas. (Kniebusch et al., 2019b) also identified a statistically significant positive trend
in the river discharge to the Bothnian Bay for 1921-2004. In Estonian rivers, regime shifts in annual specific runoff
corresponded to the alternation of wet and dry periods (Jaagus et al., 2017). A dry period started in 1963/1964,
followed by a wet period from 1978, with the latest dry period commencing at the beginning of the 21st century.

For the period 1920-2005, positive trends in stream flow at stations of a pan-Nordic dataset dominate annual mean,
winter and spring figures whereas summer trends are statistically not significant (Wilson et al., 2010). A clear
signal of earlier snow-melt floods and a tendency towards more severe summer droughts in southern and eastern
Norway were found.

The observed temperature increases have affected stream flow in the northern Baltic Sea region for 1920-2002 in
a manner corresponding well to the projected consequences of a continued rise in global temperature (Hisdal et
al., 2010). Regarding precipitation, however, the regional impacts of both the observed and projected changes on
stream flow are still unclear.

In the northern Baltic Sea region, all the way south to the Gulf of Finland, runoff is strongly linked to the climate
indices air temperature, wind and rotational circulation components. In the southern region, runoff is associated
more with the strength and torque of the cyclonic or anticyclonic pressure systems (Hansson et al., 2011).

In the Baltic states (Lithuania, Latvia and Estonia), changes in streamflow over the 20th century showed a
redistribution of runoff over the year, with a significant increase in winter and a tendency for decreasing spring
floods (Reihan et al., 2007; Sarauskiene et al., 2015; Jaagus et al., 2017). A similar winter trend was found also
for the reconstructed river discharge to the entire Baltic Sea since the 1970s (Meier and Kauker, 2003).

For the period 1911-2010, a trend of observed annual maximum daily flows in Sweden could not be detected
(Arheimer and Lindström, 2015). However, in particular the annual minimum daily flows in northern Sweden
considerably increased in the period 1911-2018 (Lindström, 2019). Analyzing a pan-European database, Blöschl
et al. (2017) showed that river floods over the past five decades occurred earlier in spring due to (1) an earlier
spring snow melt in northeastern Europe, (2) delayed winter storms associated with polar warming around the
North Sea, and (3) earlier soil moisture maxima in Western Europe.

**3.2.2.2 Land nutrient inputs**

The Baltic Sea catchment area of 1.7 million km$^2$, which is more than four times larger than the sea surface area
(cf. Fig. 1), is populated by over 84 million inhabitants. Stretching between 49° - 69°N and 10° - 38°E, the
catchment exhibits significant gradients in both natural (precipitation, river discharge, temperature, etc.) and
anthropogenic (population density and occupation, agricultural and industrial development, etc.) environmental
factors. These factors change both in time (phenological changes, long-term trends and lags due to land cover
processes) and space (north-south gradients in climate and land use, east-west gradients in socio-economic features
and climate) thus determining heterogeneity and variation of land nutrient inputs that drive long-term
eutrophication of the Baltic Sea (Savchuk, 2018, and references therein; Kuliński et al., 2021).



Estimates of nutrient inputs had been attempted since the 1980s (e.g. Larsson et al., 1985; Stålnacke et al., 1999) and are now being compiled within a permanent process of the HELCOM Pollution Load Compilation (PLC, e.g. HELCOM, 2019). However, these data officially reported to HELCOM by the participating riparian states have been and still are suffering from gaps and inconsistencies. Therefore, the "best available estimates" have been reconstructed in attempts to both fill in such gaps and correct possible sources of inconsistencies (Savchuk et al., 2012; the present study based on Svendsen and Gustafsson, 2020). For long-term studies of the Baltic Sea ecosystem, a historical reconstruction of nutrient inputs since 1850 is available (Gustafsson et al., 2012).

According to HELCOM (2018a; Savchuk, 2018) and updated estimates (HELCOM, 2018c), substantial reductions of land nutrient inputs, comprising riverine inputs and direct point sources at the coast, have been achieved since the 1980s (Fig. 11, Table 5). Since there are no statistically significant trends in annual river discharge (Section 3.2.2.1), these reductions are attributed to socio-economic development, including expansion of the wastewater treatment and reduction of atmospheric nitrogen deposition (Gauss et al., 2021) over the entire Baltic Sea drainage basin, and not to climate related effects (HELCOM, 2018a; Svendsen and Gustafsson, 2020). As an example, the coastal point sources of TN and TP decreased three- and ten-fold, respectively, comparing to the 1990s (Savchuk et al., 2012) and today contribute to the Baltic Sea less nutrients than they did in 1900 (Savchuk et al., 2008; Kuliński et al., 2021).

Agriculture is the main source of anthropogenic diffuse nutrient inputs, which comprise 47% of the riverine nitrogen and 36% of the riverine phosphorus inputs (HELCOM, 2018c). In turn, mineral fertilizer dominates the anthropogenic nutrient inputs to the Baltic Sea catchment, in particular in its intensely farmed southern part (Hong et al., 2017). However, during 2000-2010 only about 17% of the net anthropogenic nitrogen and only about 4.7% of the net anthropogenic phosphorus input to the catchment are currently exported with rivers to the sea (Hong et al., 2017). While denitrification might have removed part of the nitrogen applied in agriculture, the remaining phosphorus has accumulated in the drainage basin. A global budget estimated that agriculture has increased the soil storage of phosphorus in the drainage basin by 50 Mt during 1900-2010 (Bouwman et al., 2013) and a regional approach calculated an increase by 40 Mt during 1900-2013 (McCrackin et al., 2018). However, McCrackin et al. (2018) estimated that about 60% of these phosphorus inputs were retained in a stable pool and did not contribute noticeably to the riverine export. About 40% accumulated in a mobile pool with a residence time of 27 years. McCrackin et al. (2018) suggested that leakage from this mobile legacy pool is, though slowly declining, the dominant source of present riverine phosphorus inputs.

### 3.2.3 Terrestrial biosphere

*Previous assessments*

The comprehensive review of climate-related changes in terrestrial ecosystems in the first BACC report (BACC Author Team, 2008), Smith et al. (2008) concluded that climate change during the preceding 30-50 years had already caused measurable changes in terrestrial ecosystems in the Baltic Sea region, e.g. an advancement of spring phenological phases in some plants, upslope displacement of the alpine tree-line and increased land-surface greenness in response to improved growth conditions and a richer $CO_2$ supply. But as nearly all ecosystems in the region were managed to some extent, the climate impacts might be alleviated or intensified by human interventions, e.g. by choosing favorable tree species in forestry. The observed trends were expected to continue





for at least several decades, assuming that continued future increases in the atmospheric $CO_2$ concentration will
cause continued warming.

In the second BACC report (BACC II Author Team, 2015), climate effects on terrestrial ecosystems were less in
focus, with a section related to forests and natural vegetation in the chapter on environmental impacts on coastal
ecosystems, birds and forests (Niemelä et al., 2015) and as part of the chapter on socioeconomic impacts on forestry
and agriculture (Krug et al., 2015). On the other hand, the second BACC report also considered anthropogenic
land-cover changes as a driver of regional climate change (Gaillard et al., 2015). Niemelä et al. (2015) concluded
that the observed positive effects of climate change on forest growth would continue, in particular for boreal forest
stands that benefitted more than temperate forest stands. The species composition of natural vegetation in the
Baltic Sea region was expected to undergo changes, with a predominantly northward shift of the hemiboreal and
temperate mixed forests. Terrestrial carbon storage was likely to increase in the region, but land-use change could
play an important modifying role, affecting this storage both positively and negatively. Krug et al. (2015)
concluded that there were regional differences in how the vulnerability and productivity of forestry systems were
affected by climate change, with the southern and eastern parts of the Baltic Sea region likely to experience reduced
production and the northern and western parts increased production. Gaillard et al. (2015) found no indication that
deforestation in the Baltic Sea region since 1850 could have been a major cause of the observed climate warming.

Acknowledging the importance of the land component in the climate system, the IPCC recently published its
special report entitled 'Climate Change and Land' on climate change, desertification land degradation, sustainable
land management, food security, and greenhouse gas fluxes in terrestrial ecosystems (IPCC, 2019b). This is
because land, including its water bodies, provides the basis for human livelihoods and well-being through primary
productivity, the supply of food, freshwater, and multiple other ecosystem services.
*Biophysical and biogeochemical interactions*
The land surface and its terrestrial ecosystems in the Baltic Sea region interact with the atmosphere and are, thus,
coupled to the local and regional climate. These interactions determine the exchanges of heat, water and
momentum between the land surface and the atmosphere via biophysical processes, the exchange of greenhouse
gases, e.g. $CO_2$, $CH_4$ and $N_2O$, and emissions of black carbon, aerosol precursors, e.g. biogenic volatile
compounds, or organic carbon aerosols via biogeochemical processes, altering the atmospheric composition.
The nature of the biophysical and biogeochemical interactions between the land surface and the atmosphere and
their effects on climate are studied by comparing the effects of forests with that of open land, e.g. grassland,
pastures or cropland (e.g. Bonan, 2016). Details on the nature of these feedbacks are given in Gröger et al. (2021b).
On average, across the globe, forests absorb atmospheric $CO_2$ and, thus, reduce net radiation and have a cooling
effect on climate. In equilibrium, forest ecosystems, are expected to be carbon neutral, with carbon loss through
phenological turnover, mortality and decomposition over large areas on average matching plant productivity. $CO_2$
fertilization is considered a strong driver for the terrestrial carbon sink, but demographic recovery following past
land use, e.g. afforestation and replanting of harvested forest stands, likely provides an equally important
explanation for net carbon uptake by forests in industrialized regions of North America, Europe and Asia (Pugh et


al., 2019). Deforestation, on the other hand, can lead to a release of carbon and to an increase in net radiation and,
thus, has a warming effect on climate (Friedlingstein et al., 2020, and references therein). In contrast to the
biophysical effects described above and some of the other biogeochemical interactions, the biogeochemical
interactions associated with the carbon cycle have global impacts and operate at very long time scales.
*Anthropogenic land-use and land cover changes*
Using remote sensing data, Jin et al. (2019) investigated recent trends in springtime plant phenology in the Baltic
Sea region and the sensitivities of phenological trends to temperature and precipitation, in spring, winter and
summer. Considering the entire region and combining all vegetation types, the authors found an advancement of
the growing season by 0.30 day year$^{-1}$ over the period 2000-2016. The advancement was particularly strong for
evergreen needle-leaved forests (0.47 day year$^{-1}$) and weaker for cropland and grassland (0.14 day year$^{-1}$).
Evergreen needle-leaved forests, together with deciduous broadleaf forests, dominate the northern part of the Baltic
Sea region, while the southern part is mainly grassland and cropland. Jin et al. (2019) found that the most important
driver of the advancement of the growing season is spring mean temperature, with an advancement rate of 2.47
day (°C)$^{-1}$ of spring warming, considering the entire area and all vegetation types. Spring drying could further
increase the advancement rate by 0.18 day cm$^{-1}$. While the sensitivity of the start of the growing season to climate
conditions in spring is comparable for the entire Baltic Sea region, the sensitivity to climate conditions in the
summer and winter seasons differs between the northern and southern parts of the region. In both seasons an
increase in the mean temperature was found to advance the growing season in the southern part of the Baltic Sea
region but to delay the start of the growing season in the northern part, in contrast to the spring warming. These
sensitivities were markedly stronger for summer than for winter. These trends in plant phenology in spring result
in changes of the land cover early in the year, thus affecting climate through biophysical and biogeochemical
interactions with the atmosphere.
Observations reveal a local impact of changes in forest cover on near-surface temperatures, due to biophysical
effects that depend on the geographical latitude, roughly separating the boreal regions and the temperate zone of
the Northern Hemisphere. When investigating the effects of small-scale clearings at sites in the Americas and Asia,
Zhang et al. (2014a) found on both continents that annual mean temperatures cooled over open land north of about
35°N and warmed south of this latitude. Changes in forest cover have, however, different effects on daily minimum
and daily maximum temperatures. Zhang et al. (2014a) found that the warming effect over open land south of
35°N was related to an increase in daily maximum temperatures, with little change in daily minimum temperatures,
while the cooling effect to the north was due to a decrease in daily minimum temperatures. Lee et al. (2011) showed
consistent results for North America, where the cooling effect of 0.85±0.44°C over non-forested areas north of
45°N was due to a decrease in daily minimum temperatures, associated with the reduced roughness length. At
night, open land cools more than forests, regardless of geographical latitude. This is confirmed by Alkama and
Cescatti (2016), who analyzed the impacts of recent losses in forest cover on near-surface and land-surface
temperatures in the boreal zone. For both, the authors found cooling trends in daily minimum and warming trends
in daily maximum temperatures in response to deforestation and opposite tendencies after afforestation. These
effects were somewhat stronger for land-surface temperatures than air temperatures.



Regional Climate Models (RCMs) have been used to investigate the biophysical effects of changes in forest cover
on climate in Europe. Strandberg and Kjellström (2019), for instance, used simulations with the Rossby Centre
Atmosphere (RCA) RCM to assess the climate effect of maximal afforestation or deforestation in Europe, focusing
on seasonal mean temperatures and precipitation, as well as daily temperature extremes. Maximum afforestation
and deforestation were inferred from a simulation with the LPJ-GUESS dynamical vegetation model, providing a
map for potential natural forest cover for Europe in equilibrium with present-day climate (Gröger et al., 2021a).
To effect maximum afforestation, present-day land cover classes, which represent considerable agricultural
activity in Europe (particularly in western, central and southern Europe), were replaced by the potential natural
forest cover. In the case of deforestation, on the other hand, the potential natural forest cover was converted to
grassland in the model.
The simulations indicated that afforestation in Europe generally increased evapotranspiration, which, in turn, led
to colder near-surface temperatures. In western, central and southern Europe, the cooling in winter due to
afforestation was between 0.5 and 2.5°C. The cooling effect was somewhat stronger in summer, exceeding 2.5°C
in large parts of western and southeastern Europe. Deforestation had the opposite effect, warmer near-surface
temperatures due to decreased evapotranspiration, typically in the range between 0.5 and 2°C in western and
central Europe and reaching up to 3°C in southeastern Europe. In regions with low evapotranspiration, however,
changes in the surface albedo were relatively more important for temperatures. During summer, warming by
deforestation affected the entire Baltic Sea region (in the range between 0.5 and 1.5°C), while the cooling
associated with afforestation only affected its southern part. Over parts of Scandinavia, afforestation actually
resulted in a slight warming of about 0.5°C. In winter, the cooling effect of afforestation was only evident over the
southern part of the Baltic Sea region, while deforestation had no effect on winter temperatures.
Strandberg and Kjellström (2019) found relatively strong biophysical effects of afforestation or deforestation in
Europe on daily maximum temperatures in summer. Deforestation markedly increased daily maximum
temperatures over the entire Baltic Sea region (typically between 2 and 6°C), while afforestation lowered daily
maximum temperatures in the southern part of the region by about 2 to 6°C and slightly increased over parts of
Scandinavia. In contrast to its cooling effect on mean winter temperatures, afforestation lead to a warming of the
daily minimum temperatures in the southern part of the Baltic Sea region in the range between 2 and 6°C.
In a similar study with the Regional Model (REMO) RCM, Gálos et al. (2013) investigated the biophysical effects
of afforestation in Europe on climate, and also compared these effects to the climatic changes expected from future
global warming. Potential afforestation was implemented by specifying deciduous forest cover at all vegetated
areas that were not covered by forests at the end of the 20th century, mainly in western, Central and Eastern Europe.
Given the strong historical deforestation in Central Europe, there is potential for rather extensive afforestation in
the southern part of the Baltic Sea region, but lesser potential in the northern part, i.e. Scandinavia and Finland.
The results indicated a cooling effect of the re-established forests in boreal summer exceeding 0.3°C, mainly
related to increased evapotranspiration from the trees, in combination with intensified fluxes of latent heat due to
stronger vertical mixing (see above). The stronger latent heat fluxes also enhanced precipitation by more than 10%
in some regions. These effects of potential afforestation counteracted the projected future changes in climate, i.e.
somewhat reduced the magnitude of the pronounced future warming and markedly reduced the future drying in
the boreal summer. In some cases, such as in northern Germany, the enhanced precipitation was found to


completely offset the drying effect of future warming. More recent results (Meier et al., 2021c) have used rain-
gauge data to estimate precipitation changes induced by land cover change. Meier et al. (2021c) created a statistical
model to show that reforestation of agricultural land can increase precipitation locally, especially in winter, and
were able to separate the effects on both local and downwind precipitation regionally and seasonally. They also
found that climate change induced summer precipitation reductions could be offset by reforestation, with a
particularly strong effect in southwest Europe, though their analyses also indicate small precipitation increases in
the Baltic Sea region, relative to a baseline scenario with no land cover change, consistent with the results of Gálos
et al. (2013).
In a more regionalized study, Gao et al. (2014) applied the REMO RCM to investigate the biophysical effects of
peatland forestation in Finland before (1920) and after drainage (2000s). In Finland, as in other northern European
countries, vast areas of naturally tree-less or sparsely tree-covered peatland were drained for timber production in
the second half of the 20th century. The total peatland area of Finland was estimated to be 9.7 million ha in the
1950s, but at the beginning of the 21th century the area of peatland drained for forestry was estimated to 5.5 million
ha. The authors found that the peatland forestation caused warming in spring, i.e. during the snow-melt season,
and slight cooling in the growing season (May through October). The spring warming was mainly caused by
decreased surface albedo and the cooling in the growing season by increased evapotranspiration.

### 3.2.4 Cryosphere

### 3.2.4.1 Snow

In the Baltic Sea region, snow cover is an important feature that greatly affects cold season weather conditions. It
is characterized by very high interannual and spatial variability. Snow cover is a sensitive indicator of climate
change, and its variations are closely related to air temperature in many regions. General climate warming is
expected to reduce snow cover. Several thaw periods now interrupt snow cover, making it less stable. Total winter
snowfall in Northern Europe is projected to decrease, but is still expected to increase in mid-winter in the very
coldest regions (Räisänen, 2016).
Previous climate change assessments demonstrated a number of snow-cover trends in recent decades in the Baltic
Sea region (BACC Author Team, 2008; BACC II Author Team, 2015). A decrease in snow cover was observed
in the south, while an increase in snow storage and duration of snow cover was detected in the north-east, and in
the Scandinavian mountains. The spring snow melt has become earlier in most of the region. As a result, the spring
maximum river discharge has become smaller and earlier, in many regions shifting from April to March.
Recent investigations confirm these results. Snow cover in the Northern Hemisphere has decreased since mid-20[th]
century (IPCC, 2014a), as also shown by satellite measurements (Estilow et al., 2015). The largest decline in the
extent of snow cover has been observed in March-April and also in summer. Using the satellite-based NOAA-
CDR data for the period 1970–2019, it was shown that the annual snow cover fraction has reduced over most areas
of the Northern Hemisphere by up to 2% decade$^{-1}$ (Zhu et al., 2021). Thereby, the annual snow cover area has
reduced by $2 \times 10^5$ km$^2$ decade$^{-1}$.



In 1980–2008, snow-cover duration in Northern Europe decreased by about 3-7 days per decade, and the trend was significant at many stations (Peng et al., 2013). Most of the reduction happened in spring, with the end-date of snow cover five days earlier per decade, on average (Peng et al., 2013). Snow-cover variability over Europe is closely related to temperature fluctuations, which, in turn, are determined by large-scale atmospheric circulation during the cold season (Ye and Lau, 2017). A recent study of European snow-depth data in 1951–2017 demonstrated an accelerated decrease after the 1980s (Fontrodona Bach et al., 2018), with an average decline, excluding the coldest climates, of 12.2% per decade for mean snow depth and 11.4% per decade for its maximum. A decreasing trend in snow density was detected in the eastern Baltic Sea region in 1966–2008 (Zhong et al., 2014).

In Poland, rather large changes in snow cover parameters were found for 1952–2013 (Szwed et al., 2017). The duration of snow cover decreased in almost the whole country but this change is mostly not statistically significant. The total reduction in snow cover duration was 1–3 weeks over the 62 years, but the mean and maximum snow depths did not change. The start date of snow cover has not changed, but the end date moved slightly earlier (Szwed et al., 2017). A recent study found a statistically significant decreasing trends in snow cover duration as well as of in snow depth, based on 40 Polish stations in 1967–2020 (Tomczyk et al., 2021). The trend values for the number of days with snow cover were from -3.5 to -4.9 days per decade.

The snow-cover regime at 57 stations in the eastern Baltic Sea region (Lithuania, Latvia and Estonia) in 1961–2015 was analyzed by Rimkus et al. (2018). The mean decrease in snow-cover duration was 3.3 days per decade, and was statistically significant at 35% of the measuring sites, mostly in the southern part of the region. There were no trends in maximum snow depth. An earlier study for Lithuania found similar results (Rimkus et al., 2014).

A detailed study of snow cover data at 22 stations in Estonia during the period 1950/51–2015/16 revealed remarkable decreasing trends (Viru and Jaagus, 2020). Snow-cover duration decreased significantly at 16 stations, and the mean decrease was 4 days per decade. Start dates for permanent snow cover had a non-significant tendency to occur later. Permanent snow cover had a statistically significant trend to end earlier at almost all stations. There were no overall trends in maximum snow depth in Estonia in 1951–2016 (Viru and Jaagus, 2020).

Significant decreases in snow depth parameters were found in Finland in recent decades (Aalto et al., 2016; Luomaranta et al., 2019). Regional differences were substantial. In 1961–2014, the largest decrease in snow depth occurred in the southern, western and central parts of Finland in late winter and early spring. In northern Finland, a decrease in snow depth was most evident in spring, with no change in the winter months, even though the amount of solid precipitation was found to increase in December–February (Luomaranta et al., 2019). Winter mean snow depth (Jylhä et al., 2014) as well as the annual maximum snow depth (Lehtonen, 2015) has decreased significantly at many stations in southern Finland. At the same time, the annual maximum snow depth has not changed in Finnish Lapland (Lépy and Pasanen, 2017; Merkouriadi et al., 2017).

For the century 1909–2008, a general decrease was detected in many snow-cover parameters at three stations in different parts of Finland (Irannezhad et al., 2016). A sharp decline in annual peak snow-water equivalent was detected since 1959. The period of permanent snow cover shortened by 21–32 days per century.






A decline in snow cover parameters was found also in Norway for 1961–2010 (Dyrrdal et al., 2012; Rizzi et al.,
2017).

**3.2.4.2 Glaciers**
A recent basic inventory of Scandinavian glaciers is available through the Randolph Glacier Inventory (RGI
Consortium, 2017; Pfeffer et al., 2014), a collection of digital outlines of the world's glaciers prepared to meet the
needs of the Fifth IPCC Assessment Report (IPCC, 2014b; Vaughan et al., 2014). It has since been updated in
support of the IPCC Special Report on the Ocean and Cryosphere in a Changing Climate (IPCC, 2019a; Hock et
al., 2019). Of the 3417 Scandinavian glaciers reported in the Randolph Glacier Inventory (v6.0), 365, with a
combined area of c. 360 km$^2$, lie within the Baltic Sea Drainage Basin as defined by (Vogt et al., 2007; Vogt et
al., 2008; all in the Scandinavian mountains). The combined glacier volume estimate (following Farinotti et al.,
2019) for the Scandinavian glaciers reported in Hock et al. (Hock et al., 2019) is $0.7 \pm 0.2$ mm global sea level
equivalent or $254 \pm 72$ Gt. The rate of mass loss for all Scandinavian glaciers (not only in the Baltic Sea Drainage
Basin) for the period 2006-2015 is $2 \pm 1$ Gt year$^{-1}$ corresponding to a negligible $0.01 \pm 0.00$ mm year$^{-1}$ of global
sea level rise equivalent, yet with potential importance for local streamflow (Hock et al., 2019; Zemp et al., 2019).
With a 90-100% likelihood, atmospheric warming is the primary driver of glacier mass loss (Hock et al., 2019).

Most of the 365 glaciers in the Baltic Sea Drainage Basin are in Sweden (c. 72% by number, c. 75% by area), with
the remaining ones in Norway. Both Sweden and Norway do nationally coordinated glacier monitoring, with the
most recent results summarized in the Global Glacier Change Bulletin (GGCB) No. 3 (World Glacier Monitoring
Service, 2020; Zemp et al., 2020) as, among others, glacier mass balance changes. A glacier mass balance year
usually covers the period from September 1$^{st}$ to August 31$^{st}$ in the subsequent year. The GGCB includes four
Swedish glaciers, two of which (Rabots glaciär and Storglaciären in the Kebnekaise Massif, which has the world's
longest continuous mass balance record, starting in 1945/46) are so-called reference glaciers, meaning that their
dynamics are not dominated by non-climatically driven dynamics such as calving or surging, and that more than
30 years of ongoing measurements are available (Fig. 12). In Table 6, mass balances of the Swedish GCCB glaciers
following World Glacier Monitoring Service (2020) are summarized. None of the Norwegian GGCB glaciers are
in the Baltic Sea drainage basin.

These with time increasingly negative mass balances coincide with globally increasing air temperatures, with the
latest six years, 2015–2020, the warmest since instrumental recording began (World Meteorological Organization,
2020). Regional and local deviations in mass loss from that expected from long-term global warming is, however,
expected. The slightly positive mass balances for Mårmaglaciär, Storglaciären and Riuojekna in 2016/2017 is the
result of a cold summer, explained by the glacier mass balance years starting on September 1, and that the summer
months (June, July, August) included in the 2016/2017 balance are therefore June, July, and August 2017 – a
period during which average air temperature at Tarfala Research Station was measured to 5.8°C, compared to
6.4°C and 7.4°C in the previous and subsequent mass balance periods (Swedish Infrastructure for Ecosystem
Science, 2020; World Glacier Monitoring Service, 2020).





The ice summit of Kebnekaise Sydtopp (South Peak) lost its status as Sweden's highest in September 2019 and 2020 when due to melting of its ice-covered summit its elevation dropped below that of the rocky, non-ice-covered Kebnekaise Nordtopp (North Peak; Stockholm University - Department of Physical Geography, 2017, 2019, 2020).

**3.2.4.3 Permafrost**

The drainage basin of the Bothnian Bay is characterized by low mean annual air temperatures and includes boreal forest and mountain ecosystems as well as large peatland areas. In this region, permafrost — ground frozen for at least two consecutive years — exists in high alpine environments and in peatlands. While almost all Baltic Sea Drainage Basin permafrost is found in the Bothnian Bay catchment, some isolated occurrences of permafrost are found in the upper reaches of Alpine rivers further south (mainly Umeälven headwaters). Permafrost is a thermal state of ground, rock, soil or sediment, and occurs in regions with low mean annual air temperatures (MAAT). Temperature is the strongest control on permafrost, but thin winter snow depth also favors permafrost aggradation and stability. In alpine permafrost, insolation is an important factor. In the upper Torne River catchment, incoming shortwave summer radiation causes a difference in the altitude of alpine permafrost (permafrost p>0.8) from 850 m a.s.l. on shaded slopes to 1100 m a.s.l. on south-facing slopes (Ridefelt et al., 2008). In lowland areas, permafrost is more likely to form and persist in peatlands, where the low thermal conductivity of peat insulates the ground from warm summer air (Seppälä, 1986). Mire complexes with palsas, elevated peat mounds with an ice-rich permafrost core, are the most common form of lowland permafrost in the Baltic Sea drainage basin (Luoto et al., 2004). Palsa mires in the Baltic Sea basin are predominantly found in regions with MAAT < -3°C and low mean annual precipitation (often <450 mm), based on the 1961-1990 climate period (Fronzek et al., 2006).

Earlier maps of northern hemisphere permafrost extent showed relatively extensive Fennoscandian permafrost, especially in Alpine regions (Brown et al., 1997). Recent advances in permafrost modeling reveal a more nuanced picture, where permafrost is spatially patchy, persisting at high elevations and in lowland regions with low precipitation and large expanses of peat plateaus (Obu et al., 2019; Gisnås et al., 2017). Consistent with observation of permafrost warming and thawing (Biskaborn et al., 2019), models project substantial permafrost losses in recent decades. High resolution modelling (1 km pixels) driven by remotely sensed land surface temperature showed that c. 6,200 km$^2$ of permafrost in 1997, was reduced to 4,800 km$^2$ in 2018 (Obu et al., 2020). Most of this loss is modeled alpine permafrost (here defined as >700 m a.s.l) decreasing from 4,700 to 3,700 km$^2$, while lowland permafrost has decreased from 1,500 km$^2$ to 1,100 km$^2$ (Fig. 13).

**3.2.4.4 Sea ice**

*Introduction*

Sea ice is an essential indicator of climate change and variability in the Baltic Sea region. Not only does existence of sea ice indicate the general severity of the winter due to its close correlation with winter air temperature, but in addition parameters such as annual maximum sea-ice extent of the Baltic Sea (MIB), duration of ice season and maximum thickness of level ice have been monitored regularly in the Baltic Sea since the late 19[th] century.

The BACC II Author Team (2015) concluded that all sea-ice observations demonstrated large inter-annual variations, but with a long-term, statistically significant trend to milder ice conditions that is projected to continue



(Fig. 14). In this section, we review recent ice-climate research in the Baltic Sea and provide updated figures on
sea ice trends and projections that largely confirm previous conclusions.

An important indicator of advancing climate change in the Baltic Sea region is that two of the latest five ice winters
have been extremely mild (2015-2020). In winter 2015, the Bothnian Bay was never fully covered by ice (with a
MIB of 51 000 km$^2$), the first such extreme winter observed with certainty. The winter of 2020 was even milder,
with a MIB of only 37 000 km$^2$, the lowest value in a time series that began in 1720.

*Sea-ice conditions in the Baltic Sea*
On recent average, the northern sea areas of the Baltic Sea are ice-covered every year, from December to May.
During the mildest winters, only the Bothnian Bay (in a few years even only partially) and coastal zones of other
basins are ice-covered. In the past, the entire Baltic Sea was ice-covered only in the most severe winters, e.g. 1940,
1942 and 1947 (Vihma and Haapala, 2009).

In the fast ice regions near the coast, sea ice grows by thermodynamic processes only. Maximum sea ice thickness
in the fast ice regions typically amounts to 40 – 70 cm in the Bothnian Bay and 10 – 40 cm in the Bothnian Sea,
Gulf of Finland and Gulf of Riga. Ice thickness is regularly monitored at tens of fast ice sites.

Observations of sea-ice thickness in drift ice are much more limited. A recent study combined all airborne
electromagnetic ice thickness measurement in the Bothnian Bay and derived the first estimate of basin-scale ice
thickness distribution in the Baltic Sea (Fig. 15; Ronkainen et al., 2018). An important finding of that study was
that mean ice thickness in drift ice regions is greater than the thickness of fast ice, and also greater than the ice
thickness indicated on the ice charts. As expected, the data showed large inter-annual variability, but temporal and
spatial coverage was not sufficient for conclusion on changes in drift ice thickness.

Individual ice ridges caused by compression and shearing of ice drift can be 30 meters thick. The largest gradients
in ice motion are found in the coastal zone (Leppäranta et al., 2012), where mean ice thickness over several km$^2$
can be 1-3 meters (Ronkainen et al., 2018).

In some circumstances, sea ice can accumulate towards the coast and cause spectacular on-shore ridges or ride-up
of several hundred meters from shore to land, causing damage to build structures (Leppäranta, 2013). Such events
have been observed in exposed coastal regions where the stability of fast ice can be overcome by combinations of
storms, currents and water level (Leppäranta, 2013). In the Bothnian Bay, such events can occur regardless of the
severity of ice seasons. In the southern Baltic Sea, on-shore ice has been common during severe winters. During
the last ten years, such events were observed in 2010, 2011, 2012 and 2019 (Girjatowicz and Łabuz, 2020).

*Observed changes*
Long-term changes of the MIB (Seinä and Palosuo, 1996; Niskanen et al., 2009) are shown in Figure 14. The trend
of the MIB during the last 100 years (1921-2020) is -6,400 km$^2$ per decade. This is almost twice the trend reported
by the BACC II Author Team (2015), based on the period 1910-2011. Since 1987 no severe ice winters and since
2012 only average, mild or extremely mild winters have been observed. The latter sea-ice conditions explain the
accelerated trend after 2011.

The recent 30-year period (1991-2020) is definitely the mildest since 1720 (Uotila et al., 2015). The probability
distribution of the MIB has shifted towards zero, with severe winters very rare (Fig. 14). The 30-year mean MIB
is now 139 $10^3$ km$^2$ and the winter 2021 with a MIB of 127 000 km$^2$ on 15 February 2021 (Jouni Vainio, FMI,
personal communication) was close to this mean. During the second mildest 30-year period (1909-1938), the
average MIB was 184 $10^3$ km$^2$ and during the last 100 years it was 182 $10^3$ km$^2$. The shape of the MIB probability
distribution has changed, also indicating a change in sea-ice extremes. According to the ice season classification
(Seinä and Palosuo, 1996), the recent 30-year period includes only one severe or extremely severe ice winter and
13 mild or extremely mild ice winters.

Present ice conditions differ from the past to the extent that Rjazin and Pärn (2020) even suggested defining this
change as a regime shift. They analyzed changes in sea ice extent and air temperature in the Baltic Sea in 1982 –
2016, using a method of splitting the time series in two and concluded that a regime shift towards milder ice
conditions occurred in 2006-2007.

Other studies complement these and BACC II conclusions. Kiani et al. (2018) examined the influence of
atmospheric changes on ice roads between Oulu and Hailuoto. They used air temperature data to calculate freezing
and thawing degree days and found that freezing degree days decreased and thawing degree days increased
significantly during 1974 – 2009. As a consequence, the ice road season started later and ended earlier.

Merkouriadi and Leppäranta (2014) analyzed ice thickness and freezing and breakup dates collected at Tvärminne
Zoological Station, at the entrance to the Gulf of Finland. They found a decrease of almost 30 days in the ice-
covered period and a reduction of 8 cm in maximum annual ice thickness in the last 40 years. Laakso et al. (2018)
used observations from the Utö Atmospheric and Marine Research station during 1914-2016 and concluded that
the length of the ice season has decreased from 10-70 days before 1988 to 0-35 days after 1988 in the northern
Baltic Sea proper. Figures 16 and 17 show level-ice thickness at Kemi and Loviisa and the length of the ice season
at Kemi, Loviisa and Utö. All graphs show statistically significant decreasing trends, except level-ice thickness at
station Kemi, which was probably be influenced by snow cover or changes in measurement location.
**3.2.4.5 Lake ice**
The recent change in ice phenology is probably the single most important climatically induced alteration in lake
environments within the Baltic Sea catchment. New literature demonstrates almost unanimously significant
changes towards earlier ice break-up, later freeze-up, and shorter duration of ice cover across the Baltic Sea
catchment, apart from the coldest climate regime in Lapland. The available centennial data indicate that the ice-
cover duration has decreased by several days per century, whereas the intensified warming in recent decades has
produced a similar change per decade (Efremova et al., 2013; Filazzola et al., 2020; Kļaviņš et al., 2016; Knoll et
al., 2019; Korhonen, 2019; Lopez et al., 2019; Nõges and Nõges, 2014; O'Reilly et al., 2015; Ptak et al., 2020;
Sharma et al., 2016; Sharma et al., 2020; Wrzesiński et al., 2015). Some lakes have, however, responded only
weakly to the warming trend, such a lake Peipsi in Estonia, probably due to increasing snowfall. In individual



years, a positive wintertime NAO seems to be an important factor causing a short ice-cover duration. Among the
main properties that affect the ice cover of individual lakes are size, depth, and shoreline complexity.

### 3.2.5 Ocean and marine sediments

#### 3.2.5.1 Water temperature

The main driver of annual mean water temperature variations and long-term changes is air temperature (Meier et
al., 2019d; 2019c; Kniebusch et al., 2019a). Baltic Sea water temperature has risen fastest at the sea surface (Meier
et al., 2021a). With time the heat spreads downward through different processes, such as lateral inflows, vertical
down-welling and diffusion, and eventually the whole water column warms up, with smallest trends in the cold
intermediate layer between the thermo- and halocline (Meier et al., 2021a).

Since the 1980s, marginal seas around the globe have warmed faster than the global ocean (Belkin, 2009), and the
Baltic Sea has warmed the most (Belkin, 2009). Climate change and decadal variability led to an annual mean,
area averaged increase in Baltic Sea SST of +0.59°C decade$^{-1}$ for 1990-2018 (Siegel and Gerth, 2019) and of
+0.5°C decade$^{-1}$ for 1982-2013 (Stramska and Białogrodzka, 2015). Both figures were derived from satellite data.
In accordance with earlier investigations (BACC Author Team, 2008; BACC II Author Team, 2015), SST
variability in winter can be linked to the NAO (Stramska and Białogrodzka, 2015). However, the spatial maps of
SST trends by Stramska and Białogrodzka (2015) differ from those by Lehmann et al. (2011), perhaps because of
the differing horizontal resolution of the satellite data products. Linear trends for the Baltic Sea during 1982-2012
of 0.41°C decade$^{-1}$ are slightly larger than 0.37°C decade$^{-1}$ for the North Sea (Høyer and Karagali, 2016).

Using monitoring data, Liblik and Lips (2019) found that the upper layer has warmed by 0.3–0.6°C decade$^{-1}$ and
the sub-halocline deep layer by 0.4–0.6°C decade$^{-1}$ in most of the Baltic Sea during 1982-2016. The total warming
in the whole Baltic Sea was 1.07°C over 35 years, approximately twice that of the upper 100 m in the Atlantic
Ocean.

During 1856–2005, the reconstructed Baltic Sea average, annual mean SST increased by 0.03 and 0.06°C decade$^{-1}$
$^{1}$ in the northeastern and southwestern areas, respectively (Kniebusch et al., 2019a). The largest SST increase
trends were found in the summer season in the northern Baltic Sea (Bothnian Bay). Bottom water temperature
trends were smaller than SST trends, with the largest increase in the Bornholm Basin. Independent monitoring
data support the results of the long-term reconstruction (Figs. 18 and 19), see also Meier et al. (2019c; 2019d). The
largest SST warming occurred in summer (May to September), while trends in winter were smaller (Kniebusch et
al., 2019a; Liblik and Lips, 2019).

During the more recent period of 1978–2007, the annual mean SST trend was tenfold higher, with a mean of 0.4°C
decade$^{-1}$ (Kniebusch et al., 2019a). Trends increased more in the northeastern areas than in the southwestern, and
exceeded the contemporary trends in air temperature. See also MARNET station data at Darss Sill and Arkona
Deep in the southwestern Baltic Sea (Fig. 20).

The seasonal ice cover clearly plays an important role in the Baltic Sea by decoupling the ocean and the atmosphere
in winter and spring. Hence, the large trends in air temperature in winter were not reflected by the SST trends


because the air temperature was still below the freezing point. During the melting period, the ice-albedo feedback
led to larger trends in SST than during the ice-covered period, because of a prolonged warming period of sea water.

It has been suggested that the accelerated warming in 1982-2006 might partly be explained by a dominance of the
positive phase of the AMO (Kniebusch et al., 2019a). Historical eutrophication re-distributed the heat in the ocean
by warming the surface layer more than the underlying layers, in particular during spring and summer, because
the increased water turbidity caused an enhanced absorption of sunlight at the sea surface. However, modeling
studies suggest that the historical eutrophication had no impact on SST trends (Löptien and Meier, 2011).

The summer of 2018 was the warmest on instrumental record in Europe, and also the warmest summer in the past
30 years in the southern half of the Baltic Sea (Naumann et al., 2019), with surface-water temperatures 4-5°C
above the 1990-2018 long-term mean. This heat wave was also observed in the bottom temperatures (Humborg et
al., 2019). However, systematic studies on changes in heat waves are not available.
**3.2.5.2 Salinity and saltwater inflows**
During the last decade, many new insights have been gained about the salt balance of the Baltic Sea and the
dynamics of inflow and mixing processes. The Major Baltic Inflow (MBI) in December 2014, in particular,
triggered new investigations and is by far the most intensively observed and modeled inflow event. Pathways and
timing of the inflowing water were tracked by observations (Mohrholz et al., 2015), and numerical modelling
(Gräwe et al., 2015) could reproduce the salt mass and volume of the inflow, calculated from observations. The
inflow was found to be barotropic (pressure) controlled in the Danish straits but dominated by baroclinic (density
stratification) processes on the further pathway into the Baltic proper. At the Bornholm Gat and the Słupsk Furrow
the water exchange showed the clear two-layer flow pattern of an estuarine circulation. The inflow-related studies
were underpinned by theoretical work based on the famous Knudsen Relation (Knudsen, 1900) for estuarine
exchange flow, and its extension to total exchange flow by Burchard et al. (2018). The contribution of the inflowing
saline water to the spatial distribution of salt and the total salt budget depends essentially on mixing with ambient
brackish waters. In the course of the inflow path, the character of the mixing process between the deep salty layer
and the brackish water above changes with increasing depth and decreasing current velocities. In the entrance area
the mixing is dominated by entrainment of brackish water into the eastward spreading saline bottom water, due to
turbulence generated by shear instability. The sills between the consecutive Baltic basins are particular mixing
hotspots (Neumann et al., 2017). In the deeper basins of the Baltic proper, boundary mixing driven by the
interaction of currents and internal waves with the topography, and mixing processes at sill overflows contribute
to the upward salinity flux (Reissmann et al., 2009). Mixing in the eastern Gotland Basin was investigated during
the Baltic Sea Tracer Experiment (BATRE). Using an inert tracer gas, the basin-scale vertical diffusivities were
estimated to $10^{-5}$ m$^2$ s$^{-1}$, whereas the diffusivities in the basins interior were one order of magnitude lower
(Holtermann et al., 2012). This finding holds also for the inflow of saline water in course of an MBI (Holtermann
et al., 2017). The interior mixing is often controlled by double diffusive convection that leads to a typical stair-
case-like vertical stratification structure (Umlauf et al., 2018). The crucial role of boundary mixing at the basin
rim was confirmed by Holtermann and Umlauf (2012), and Lappe and Umlauf (2016). Both studies identified
near-boundary turbulence as the key processes for basin-scale mixing. Main energy sources for boundary mixing
are basin-scale topographic waves, deep rim currents, and near-inertial waves.




The temporal statistics of barotropic saline inflows was reviewed by Mohrholz (2018). In contrast to earlier
investigations he found no long-term trend in inflow frequency, but a pronounced multidecadal variability of 25
to 30 years. Lehmann and Post (2015) and Lehmann et al. (2017) who studied the frequency and intensity of large
volume changes in the Baltic Sea due to inflows, likewise could not find a long-term trend. The distinction between
MBIs and smaller inflows is artificial and does not correspond to the frequency distribution of the inflows, which
shows an exponential decrease in frequency with increasing inflow intensity (Mohrholz, 2018). The classical MBIs
are only responsible for about 20% of the total salt input, while the rest is accounted for by medium and small
inflows with much less pronounced interannual variability.

Paleoclimate simulations covering nearly the recent millennium (Schimanke and Meier, 2016) have provided new
insights into the long-term behavior of the mean salinity of the Baltic Sea. In accordance with previous historical
reconstruction studies (Schimanke and Meier, 2016; Meier and Kauker, 2003), Schimanke and Meier (2016)
identified river discharge, net precipitation and zonal winds as main drivers of the decadal variability in Baltic Sea
salinity. However, their relative contributions are not constant. Extreme periods with strong salinity decrease for
about 10 years occurred once per century. Thus, the long stagnation period from 1976 to 1992 was obviously a
rare but natural event, although its extreme duration might be caused by anthropogenic effects. The Baltic Sea
salinity also has a natural centennial variability. Based on the same numerical simulations, Börgel et al. (2018)
could show a strong coherence between the AMO climate mode and river runoff on timescales between 60 and
180 years. Accordingly, the Baltic Sea salinity and the AMO are correlated, probably due to the dominating impact
of river discharge on salinity. The river runoff leads salinity changes by about 20 years during the entire modeling
period of 850 years. Schimanke and Meier (2016) reported a similar lag of 15 years between river runoff and Baltic
Sea salinity.

According to model results, multidecadal variations in runoff (Gailiušis et al., 2011; Meier et al., 2019d) explain
about half the long-term variability of volume-averaged Baltic Sea salinity (Meier and Kauker, 2003). Radtke et
al. (2020) found that the direct dilution effect was only responsible for about one fourth of the multidecadal
variability and proposed a link between river runoff and inflow activity. Furthermore, they found that the influence
of vertical turbulent mixing is small. Salt water inflows contribute to the multidecadal salinity variability, in
particular for the bottom layer salinity. The positive trend of river runoff in the northern catchment area led to a
significant increase in the North-South salinity gradient in the Baltic Sea surface water layer (Kniebusch et al.,
2019b). Additionally, their model based study revealed a multidecadal oscillation of salinity, river runoff and
saltwater inflows of about 30 years, consistent with the long term observations.

From observations during 1982–2016, Liblik and Lips (2019) detected decreasing surface (see also Vuorinen et
al., 2015) and increasing bottom salinities, but no long term trend in the total salt budget were found (cf. Fig. 21).
Both temperature and salinity contribute to strengthening of the vertical stratification. Enhanced freshwater fluxes
combined with higher deep water salinities intensify the vertical density gradient throughout the year.



### 3.2.5.3 Stratification and overturning circulation

A direct consequence of increasing stratification is that mixing weakens between well ventilated surface waters and badly ventilated deep waters weakens, making the Baltic Sea vulnerable to deoxygenation of bottom waters (Conley et al., 2002). An increase in seasonal thermal stratification (e.g. Gröger et al., 2019) can additionally lower the vertical nutrient transport from deeper layers to the euphotic zone, thereby limiting nutrient supply and potentially affecting algal and cyanobacterial blooms, at least at the species level. The latter potential effect has not yet been thoroughly investigated. However, the hypothesis was supported by the results of Lips and Lips (2008) who found a correlation between cyanobacteria bloom intensity in the Gulf of Finland and the frequency of upwelling events along both coasts.

Since the start of regular salinity measurements at the end of the 19th century, the haline stratification has been dominated by sporadic inflows from the adjacent North Sea and variations in river discharge (Fig. 22). While no long-term trends could be demonstrated in Baltic Sea salinity during 1921-2004 (Kniebusch et al., 2019b) or in halocline depth during 1961-2007 (Väli et al., 2013), a trend towards increased horizontal salinity difference between the northern and southern Baltic Sea was found during 1921-2004 (Kniebusch et al., 2019b). Modeling studies by Kniebusch et al. (2019b) attributed this trend to increased river runoff from the northernmost catchment area. Stratification increased in most of the Baltic Sea during 1982-2016, with the seasonal thermocline strengthening by 0.33–0.39 kg m$^{-3}$ and the perennial halocline by 0.70–0.88 kg m$^{-3}$ (Liblik and Lips, 2019).

Sensitivity studies with a numerical model suggest that the basin-wide overturning circulation will decrease if the climate warms or when river runoff increases, but will tend to increase if global sea level rises (Placke et al., 2021). However, historical multidecadal variations of the overturning circulation are mainly wind-driven. Multidecadal variations in neither river runoff nor saltwater inflow had an impact, according to Placke et al. (2021).

### 3.2.5.4 Sea level

For the era of continuously operated satellite altimetry, absolute mean sea level (relative to the reference geoid) increased in the Baltic Sea. Available estimates vary depending on the exact period considered, but are broadly consistent with or slightly above the global average (3-4 mm year[-1]; Oppenheimer et al., 2019; Nerem et al., 2018). For the period 1992-2012, Stramska and Chudziak (2013) estimated an increase of 3.3 mm year[-1], and for the period 1993-2015 Madsen et al. (2019a) an increase of 4 mm year[-1] in the Baltic Sea absolute mean sea level. For the period 1886/1889-2018, the analysis of Swedish mareograph data suggest a sea level rise of about 1-2 mm year[-1] (Figs. 23 and 24). Passaro et al. (2021) showed that the increase is not uniform across the Baltic Sea but varies between about 2 mm year[-1] in the western Baltic Sea and more than 5 mm year[-1] in the Gulf of Bothnia for the period 1995-2019. The acceleration of sea level rise in the Baltic Sea was studied by Hünicke and Zorita (2016). They found that present acceleration is small and could only be detected through averaging of observations.

Sea level changes relative to the coast are more complex, since land is rising in the northern Baltic Sea, by up to about 8 mm year[-1], and sinking in the southern Baltic Sea, by about 1 mm year[-1] (Hünicke et al., 2015; Groh et al., 2017). In addition to the global mechanisms (thermal expansion due to warming and land-ice melting), sea level changes in the Baltic Sea are also affected by the changes in atmospheric circulation, water inflow from the North Sea, and changes in the freshwater budget (river runoff, precipitation and evaporation). Precipitation and river





runoff are linked to westerly winds and affect salinity and the salinity gradient across the Baltic Sea (Kniebusch
et al., 2019b) and thus the sea level height and its gradient. Stronger than normal westerly winds are associated
with increased transports across the Danish straits which leads to an increase in Baltic mean sea level. The
correlations between sea level height and westerly wind are higher in the eastern and northern parts and lower in
the southern and western parts of the Baltic Sea. Westerly winds in the region became more intense until the early
1990s, but have weakened somewhat thereafter (Feser et al., 2015). Over longer periods, no significant long-term
trend is detected (Feser et al., 2015).

The Baltic mean sea level shows a pronounced seasonal cycle with a minimum in spring and maxima in late
summer (in 1900-1930) or winter (in 1970-1998). According to Hünicke and Zorita (2008), the amplitude of the
seasonal cycle increased over the 20th century. Other authors found different periods without systematic long-term
trends (Barbosa and Donner, 2016) or even regional decreases (Männikus et al., 2020).

Baltic sea level extremes are caused by strong atmospheric cyclones, or more seldom by wind-induced
meteotsunamis (Pellikka et al., 2020) and seiches (Neumann, 1941; Wübber and Krauss, 1979). Cyclones are
associated with strong winds that cause coastal storm surges over one or two days, and if their pathway is aligned
along the west-east direction, cyclones may also increase the total volume of the Baltic Sea over one week by
pushing in water masses from the North Sea into the Baltic Sea. Coastal storm surges can then reach 20 cm above
the spatially averaged level (Weisse and Weidemann, 2017), in the eastern Gulf of Finland in Neva Bay even more.
Extreme sea levels over a predefined threshold become more frequent with rising mean sea level (Pindsoo and
Soomere, 2020). In addition, model results and analysis of observations indicate that atmospheric forcing is
responsible for the long-term increases in storm surges in some localized areas of the eastern Baltic Sea (Ribeiro
et al., 2014). The presence of sea-ice impedes the development of extreme sea levels by shielding the ocean surface
from forcing by the wind, and coastal ice protects the coast from erosion by extreme sea levels.

Storm surges caused by strong onshore winds represent a substantial hazard for the low-lying parts of the Baltic
Sea coast, in particular, the southwestern parts (Wolski et al., 2014), the Gulf of Finland (e.g. Suursaar and Sooäär,
2016), the Gulf of Riga (e.g. Männikus et al., 2019), and the Gulf of Bothnia (Averkiev and Klevanny, 2010).
Highest surges were reported for the Gulf of Finland (about 4 m in 1824) and the western Baltic Sea (more than 3
m in 1871, Wolski and Wiśniewski, 2020). For the Gulf of Riga and the western Baltic Sea values around 2 and
1-1.5 m are frequent, respectively (Wolski and Wiśniewski, 2020). Hundred-year storm surges are higher (up to
2.4 m) at the inner end of the basins, furthest away from the Baltic proper, than in center of the Baltic Sea (up to
1.2 m). No consistent long-term trend for an increase in extreme sea levels relative to the mean sea level of the
Baltic Sea has been found, in agreement with earlier assessments (BACC II Author Team, 2015). This finding is
supported by paleoclimate model studies that show no influence on extremes sea levels in the North Sea in warmer
climate periods compared to colder periods (Lang and Mikolajewicz, 2019), and by recent studies of sea level
records that suggest a pronounced decadal to multidecadal variability in storm surges relative to the mean sea level
(Marcos et al., 2015; Marcos and Woodworth, 2017; Wahl and Chambers, 2016). Although (Ribeiro et al., 2014)
argued for an increase in annual maximum sea level during 1916-2005, especially in the northern Baltic Sea, and
attributed the trends to changes in wind, these results were likely affected by the long-term internal variability.



Furthermore, extreme sea level in the Gulf of Finland, especially in Neva Bay, are very sensitive to the position of
storm tracks (Suursaar and Sooäär, 2007).
**3.2.5.5 Waves**
Instrumental wave measurements in the Baltic Sea have been made since the 1970s, first as measurement
campaigns and since the 1990s as continuous monitoring (e.g. Broman et al., 2006; Tuomi et al., 2011). As the
spatial coverage of wave measurements is still quite sparse, and there are long-term data from few locations only,
wave hindcasts have become a valuable tool for estimating the Baltic Sea wave climate (e.g. Björkqvist et al.,
2018). Lately, satellite altimeter measurements have been used to estimate the changes in the Baltic Sea wave
climate (Kudryavtseva and Soomere, 2017).

Hindcast studies (e.g. Räämet and Soomere, 2010; Tuomi et al., 2011; Björkqvist et al., 2020) are in good
agreement, and estimate the annual mean significant wave height (SWH) at 0.5-1.5 m in the open sea areas of the
Baltic Sea. Wave growth in the Baltic Sea is hampered by the shape and small size of the basins. The highest mean
values are recorded in the Baltic proper, with the longest and widest fetches. The gulfs have less severe wave
climates (Björkqvist et al., 2020). In addition, wave growth in the northern Baltic Sea in winter is limited by the
seasonal ice cover, leading to considerably lower mean and maximum values of SWH in the northernmost Gulf of
Bothnia and the easternmost Gulf of Finland.

Although the measurement and hindcast periods have so far been quite short, some studies have also analyzed
trends in Baltic Sea SWH. For example, Soomere and Räämet (2011) and Kudryavtseva and Soomere (2017)
suggest an increasing trend in SWH since the 1990s, but results are site-specific and so far rather inconclusive.

The seasonal wave climate is driven by the wind climate. The highest mean and maximum values of SWH are
reached in autumn and winter, while summer typically has the mildest wave climate. In sub-basins with long ice
season and large ice extent, such as the Bothnian Bay, the seasonal variation in the SWH is slightly different, since
waves are damped by the ice.

So far the Northern Baltic proper (NBP) holds the record measured value of Baltic SWH. In December 2004 a
SWH of 8.2 m was measured by the NBP wave buoy, with a highest individual wave of c. 14 m (Tuomi et al.,
2011; Björkqvist et al., 2018). As the spatial coverage of the wave measurements has increased and milder ice
winters have allowed late autumn and even winter measurements also in the northern parts of the Baltic Sea, 8 m
SWH has been measured also in the Bothnian Sea, in January 2019. Björkqvist et al. (2020) estimated a return
period of 104 years for this event in the present climate. Hindcast statistics have suggested that even higher
maximum values between 9.5 – 10.5 m may occur in areas and times for which wave buoy measurements are not
available (Soomere et al., 2008; Tuomi et al., 2011; Björkqvist et al., 2018).
**3.2.5.6 Sedimentation and coastal erosion**
The Glacial Isostatic Adjustment and eustatic sea level change impose a first-order control on Baltic Sea coastal
landscape change (Harff et al., 2007). In the subsiding southern Baltic Sea region, wind-driven coastal currents



and waves are the major drivers for erosion and sedimentation, especially along the sandy and clayey sections of
sandy beaches, dunes and soft moraine cliffs (Zhang et al., 2015; Harff et al., 2017).

Owing to spatial variation in aero- and hydrodynamic conditions (winds, waves and longshore currents) and
underlying geological structure (lithology, sediment composition), a diversity of morphological patterns have
developed along the Baltic Sea coast. Because of the dominant westerly winds that blow 60% of the year (Zhang
et al., 2011) and a sheltering by land in the west, wind-waves are larger in the south-eastern Baltic Sea than in the
south-western. As a result, sediment transport and dune development are more active and dynamic along the south-
eastern coast. Thus, the largest coastal dunes are found along the Polish coast, with wave length >100 m and height
> 20 m (Ludwig, 2017), while dunes along the German coast normally have wave length less than 60 m and height
below 6 m (e.g. Lampe and Lampe, 2018) . Under conditions favorable for wind-driven sand accumulation along
sandy Baltic Sea coasts, a typical cross-shore profile features one or several foredune ridges, generally with a
height of between 3 and 12 m above the mean sea level (Zhang et al., 2015; Łabuz et al., 2018). At the backshore
behind the established foredune ridges, drifting or stabilized dunes in transgressive forms, mainly parabolic or
barchanoid types, are commonly developed. The source of sediment for dune development includes fluvioglacial
sands from eroded cliffs, river-discharged sands, and older eroded dunes (Łabuz, 2015).

Because the wind-wave energy increases from west to east, so does erosion along the Baltic Sea coast. The mean
annual erosional rate of the soft moraine cliffs and sandy dunes along the north side of the southwestern Baltic Sea
coast (southern Sweden and Denmark) is 1-2 m, larger than 0.4-1 m along the south side of the southwestern Baltic
Sea coast (Germany). Erosion along the southern Baltic Sea coast increases eastward, with a mean annual rate of
0.5-1.5 m in Poland, and 0.5-4 m in Latvia, Lithuania and Russia (BACC II Author Team, 2015). Severe coastal
erosion in the Baltic Sea region is often caused by storms. The maximal storm-induced erosion increases eastward
from 2-3 m year$^{-1}$ at the southwestern Baltic Sea coast (southern Sweden, Denmark and Germany) to 3-6 m year$^{-}$
$^{1}$ along the Polish coast, and ~10 m year$^{-1}$ along the coast of Lithuania and Russia (Kaliningrad). Each storm can
erode soft Latvian cliffs 3–6 m, with a maximum of up to 20–30 m locally. Many sandy beaches along the Gulf of
Finland have recently been severely damaged by frequent storm surges, despite extensive protective measures
(BACC II Author Team, 2015).

The prevailing wind-wave pattern controls the spatial variations of not only coastline change rates but also
submarine morphologies (Deng et al., 2019). In the southwestern Baltic Sea coast where wind-wave energy is
relatively small, nearshore submarine morphology is generally featured by smooth transition from beach-
dunes/moraine cliffs to deeper water perturbed by one or two longshore bars. Morphological perturbations (e.g.
the number and amplitude of longshore bars and rip current channels) become increasingly larger toward the east
due to increased wind-wave energy. The wave incidence angle also impacts the nearshore submarine morphology,
with in general a smaller angle leading to a larger morphological heterogeneity, i.e. a larger amplitude of
perturbations. The amplitude of nearshore morphological perturbations may significantly affect coastal erosion
because rip currents act as efficient conduit for offshore sediment transport, despite that they only occur
sporadically along the Baltic Sea coast (Schönhofer and Dudkowska, 2021).



The sediments eroded from the soft moraine cliffs are composed of grain sizes from clay to pebbles. The fine-
grained sediments are mostly transported outwards to the deeper seafloor (i.e. the Baltic Sea basins), either
suspended in the water column or in a concentrated benthic fluffy layer (Emeis et al., 2002). Eroded fine-grained
sediments from the moraine cliffs have been found to contribute to a major portion (40-70%) of the Holocene
deposits in the muddy Baltic Sea basins (Porz et al., 2021). Coarser material, such as sands, stays mostly nearshore,
partly in the water, partly transported onto the beach and the dunes (Deng et al., 2014; Zhang et al., 2015).
### 3.2.5.7 Marine carbonate system and biogeochemistry
Studies summarized in BACC II showed that nitrate and phosphate concentrations in winter surface water of the
Baltic proper had increased by a factor of about three in the second half of the twentieth century, and reached a
peak between 1980 and 1990. This change was consistent with the enhanced nutrients inputs to the Baltic Sea and
caused eutrophication in the affected basin. Based on the available $CO_2$ system data, it has been estimated that the
net ecosystem production in the Baltic Sea has increased since the 1930s by a factor of about 2.5. Increase in net
ecosystem production and poor ventilation of the deep water layers due to the permanent stratification of the water
column caused significant expansion of the anoxic and hypoxic areas in the Baltic Sea. Since the 1980s, nutrients
inputs to the Baltic Sea have decreased. This led to a decrease in winter surface-water nitrate concentrations. No
decrease was, however, observed for phosphate, due to the long residence time of P in the Baltic Sea and reduced
P storage in oxygen-deficient sediments by binding to Fe-oxyhydroxides.
### 3.2.5.7.1 Oxygen and nutrients
In the Baltic Sea, hypoxia (oxygen deficiency) and even anoxia has expanded considerably since the first oxygen
measurements in 1898 (Gustafsson et al., 2012; Carstensen et al., 2014a; Fig. 25). In 2016, the maximum hypoxia
area was about 70,000 km², almost the combined area of Belgium and the Netherlands, whereas it was presumably
very small or even absent 150 years ago (Carstensen et al., 2014b; Carstensen et al., 2014a; Meier et al., 2019c;
2019d). Hypoxia was caused mainly by increasing land nutrient inputs and atmospheric deposition that led to
eutrophication of the Baltic Sea (Andersen et al., 2017; Savchuk, 2018). The impacts of other drivers like observed
warming and eustatic sea level rise were smaller, but still important (Carstensen et al., 2014a; Meier et al., 2019c).
On annual to decadal time scales, halocline variations also had considerable influence on the hypoxic area (Conley
et al., 2002; Väli et al., 2013).
Besides its detrimental effects on biota, hypoxia is responsible for the redox alterations of nitrogen and phosphorus
integral stocks reaching in the Baltic proper hundreds of thousand tonnes annually: the DIN pool is being depleted
by denitrification, while the DIP pool increases due to phosphate release in the water and sediment anoxic
environments (e.g. Savchuk, 2010; 2018 and references therein). Resulting changes of nitrate and phosphate
concentrations at the upper boundary of the halocline affect also neighboring gulfs, exporting to the Bothnian Sea
and Gulf of Finland waters with elevated phosphorus concentration (Rolff and Elfwing, 2015; Lehtoranta et al.,
2017; Savchuk, 2018).
Despite the decrease of land nutrient inputs after the 1980s, the extent of hypoxia in the Baltic Sea remains
unaltered. This is due to the long response time of the system to reductions in N and P inputs. According to recent
computations, the residence times for TN and TP in the water and sediments of the Baltic Sea combined are 9 and

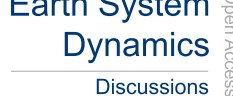
49 years, respectively (Gustafsson et al., 2017; Savchuk, 2018). Furthermore, recently observed oxygen consumption rates in the Baltic Sea are higher than earlier observed, shortening oxygen relieves from natural ventilation by oxygen-rich saltwater intrusions from the North Sea (Meier et al., 2018b). Although sediments are still the most important sinks of oxygen in the Baltic Sea, the increased rates of oxygen consumption was largely driven by water column processes with, for instance, bacterial nitrification as the most prominent. Also zooplankton and higher trophic level respiration were suggested to contribute more to oxygen consumption than 30 years ago (Meier et al., 2018b). However, the importance of the latter processes is still unknown. The present total oxygen consumption rate in the water column below 60 m depth in the Baltic proper, Gulf of Riga and Gulf of Finland is estimated to be about five times that in the period 1850-1950 (Meier et al., 2018b).

Hypoxia remains an important problem also in the Baltic Sea coastal zone (Conley et al., 2011). Coastal hypoxia most often has episodic or temporary character and is driven by the seasonal variations in organic matter supply, advective transports and water column stratification (Carstensen and Conley, 2019). The latter is mostly caused by seasonal temperature changes, but may in some areas be due to occasional inflows of saltier water and lower winds during summer, changing the vertical stratification. The coastal regions most affected by hypoxia are estuaries in the Danish straits and parts of Swedish and Finnish archipelagos located in the Baltic proper and Gulf of Bothnia (Conley et al., 2011). In contrast, hypoxia is rare along the southern and south-eastern coastline (from Poland to Estonia) due to enhanced water circulation as well as in the less productive coastal zone of the northern Baltic Sea. Despite the recently reduced nutrient inputs, bottom water oxygen concentrations have improved only in a few coastal ecosystems that have experienced the largest reductions (for instance in the Stockholm Archipelago). In most of the 33 coastal sites, evaluated by Caballero-Alfonso et al. (2015), oxygen conditions have deteriorated, especially along the Danish and Finnish coasts. This finding was explained as a coupled effect of climate changes, especially warming, which reduces oxygen solubility in water and strengthens thermal stratification as well as a delay of the system in responding to nutrient reduction.

N and P are removed from the Baltic Sea by burial in sediments, but much N is also lost by denitrification (Gustafsson et al., 2017). Coastal regions constitute an efficient nutrient filter (Almroth-Rosell et al., 2016; Asmala et al., 2017) that remove about 16% of N (by denitrification) and as much as 53% of P (by burial) delivered from land (Asmala et al., 2017). The filter effect of the coastal zone is, however, highly diverse. Denitrification rates are highest in lagoons that receive large inputs of nitrate and labile organic material, while P is most efficiently buried in archipelagos (Carstensen et al., 2020). Additionally, Hoikkala et al. (2015) argued that dissolved organic matter (DOM) plays an important role for nutrient cycling in the Baltic Sea, since more than 25% of bioavailable nutrients in riverine inputs and surface waters can be in organic form.

Furthermore, the exchange of nutrients between the coastal zone and the open sea (Eilola et al., 2012) and the role of MBIs for the phosphorus cycling (Eilola et al., 2014) were analyzed. Eilola et al. (2014) concluded that the overall impact of MBIs on the annual uplift of nutrients from below the halocline to the surface waters is small because vertical transports are comparably large also during periods without MBIs. Instead, phosphorus released from the sediments between 60 and 100 m depth in the eastern Gotland Basin contributes to the eutrophication, especially in the coastal regions of the eastern Baltic proper.



The cycling between nutrients and phytoplankton biomass was studied by Hieronymus et al. (2018) who found a
regime shift between nutrient-limited phytoplankton variations before 1950 and a less nutrient-limited regime after
1950, with a larger impact of other variations such as those in water temperature.
**3.2.5.7.2 Marine $CO_2$ system**
The marine $CO_2$ system in the Baltic Sea is greatly influenced by the production and remineralization of organic
matter, as well as inputs of organic and inorganic carbon from land (Kuliński et al., 2017). The combination of all
these factors makes Baltic Sea pH and partial pressure of $CO_2$ ($pCO_2$) highly variable in space and time (Carstensen
and Duarte, 2019). The Baltic Sea surface water is in almost permanent $pCO_2$ disequilibrium with the atmosphere
throughout the year (Schneider and Müller, 2018). In spring and summer, the surface seawater is undersaturated
with respect to atmospheric $CO_2$, as a consequence of biological production and the shallowing mixed layer depth.
Thus, seawater $pCO_2$ typically has two minima corresponding to the spring bloom and the mid-summer nitrogen
fixation period. In autumn and winter, $pCO_2$ increases due to shifting balance between autotrophy and heterotrophy
and entrainment of deeper $CO_2$-rich waters.

Remineralization of terrestrial organic matter plays an important role in shaping $pCO_2$ fields in the Baltic Sea.
Kuliński et al. (2016) found that about 20% of the dissolved organic carbon (DOC) delivered from the Vistula and
Odra rivers is bioavailable, while Gustafsson et al. (2014) even estimated that 56% of allochthonous (originating
outside the Baltic Sea) DOC is remineralized in the Baltic Sea. High inputs of terrestrial organic matter that is
subsequently partially remineralized in seawater turned the basins most affected by riverine runoff (Gulf of
Bothnia, Gulf of Finland, Gulf of Riga) to net $CO_2$ sources to the atmosphere in the period 1980-2005. This
outgassing was more than compensated by the high $CO_2$ uptake in the open Baltic proper. In 1980-2005, the whole
Baltic Sea was found to be on average a minor sink for atmospheric $CO_2$, absorbing $4.3 \pm 3.9$ g C m$^{-2}$ yr$^{-1}$, with
the rate of atmospheric $CO_2$ exchange highly sensitive to the inputs of terrestrial organic matter (Gustafsson et al.,

2014).


The high seasonal variability of $pCO_2$, enhanced by eutrophication, causes large seasonal fluctuations in surface
water pH, amounting to about 0.5 in the central Baltic Sea (Kuliński et al., 2017) and even more, often exceeding
1, in productive coastal ecosystems (Carstensen and Duarte, 2019; Stokowski et al., 2021). Furthermore, low total
alkalinity ($A_T$, measures buffer capacity), prominent in the northern basins, makes the Baltic Sea potentially
vulnerable to Ocean Acidification (OA), i.e. pH decrease caused by rising $pCO_2$ in the atmosphere and thus also
in seawater. However, Müller et al. (2016) showed that $A_T$ in the Baltic Sea has increased over time, which may
partly be due to increasing inputs from land (Duarte et al., 2013). The highest trend, 7.0 µmol kg$^{-1}$ yr$^{-1}$, found in
the Gulf of Bothnia, almost entirely mitigates the pH drop expected from rising $pCO_2$ in the atmosphere alone. In
the southern Baltic Sea, the $A_T$ increase is lower (3.4 µmol kg$^{-1}$ yr$^{-1}$) and reduces OA by about half. High seasonal
pH variability, increasing $A_T$ and variable productivity imply that OA is not measurable in the central and northern
Baltic Sea. In the Danish Straits, where no $A_T$ increase has been detected (Müller et al., 2016), a mean pH decrease
of 0.004 yr$^{-1}$ was identified in coastal waters in the period of 1972-2016 (Carstensen et al., 2018), approximately
twice the ocean trend.



Recent studies showed that the Baltic Sea $CO_2$ system functions differently from the open ocean waters. These differences include a large $CO_2$ input from remineralization of terrestrial organic matter (Gustafsson et al., 2014), a considerable contribution by organic alkalinity (Kuliński et al., 2014; Ulfsbo et al., 2015; Hammer et al., 2017), $A_T$ generation under hypoxic and anoxic conditions (Gustafsson et al., 2019; Łukawska-Matuszewska and Graca, 2018), and a borate-alkalinity anomaly (Kuliński et al., 2018), making modeling a challenge. Due to the insuffient understanding of the processes involved, state-of-the-art biogeochemical models cannot yet reproduce the positive $A_T$ trend in the Baltic Sea.

**3.2.6 Marine biosphere**

**3.2.6.1 Pelagic habitats**

**3.2.6.1.1 Microbial communities**

Microbial communities respond to increases in sea surface temperature and river runoff that enhance metabolism and augment the amount of substrate available for bacteria. By using long time-series from 1994 to 2006, increased input of riverine dissolved organic matter (DOM) in the Bothnian Bay and Bothnian Sea was shown to suppress phytoplankton biomass production and shift the carbon flow towards microbial heterotrophy (Wikner and Andersson, 2012; Paczkowska et al., 2020). (Berner et al., 2018) presented further evidence of changes in marine microbial communities.

**3.2.6.1.2 Phytoplankton and cyanobacteria**

The phytoplankton growing season has become considerably prolonged in recent decades (Kahru et al., 2016; Groetsch et al., 2016; Hjerne et al., 2019; Wasmund et al., 2019). In the Baltic proper, the duration of the growing season arbitrarily indicated by a threshold of 3 mg Chl $m^{-3}$ of a satellite-derived chlorophyll has doubled from approximately 110 days in 1998 to 220 days in 2013 (Kahru et al., 2016). In the western Baltic Sea, it now extends from February to December (Wasmund et al., 2019). Wasmund et al. (2019) analyzed data on chlorophyll a and microscopically determined biomass from 1988-2017 and found an earlier start of the growing season, which correlated with a slight increase in sunshine duration in spring, and a later end to the growing season, which correlated with warmer water in autumn. The shifts in the spring and autumn blooms led to a prolongation of the summer biomass minimum. However, time series were rather short (30 years) and trends in irradiance might be caused by internal variability (see Section 3.2.1.3). The spring phytoplankton communities have shifted from a preponderance of early-blooming diatoms to dominance by later-blooming dinoflagellates (Wasmund, 2017; Wasmund et al., 2017) and the autotrophic ciliate *Mesodinium rubrum* (Klais et al., 2011; Hällfors et al., 2013; Hjerne et al., 2019), perhaps due to reduced ice thickness and increased winter wind-speed since the 1970s (Klais et al., 2013). Wasmund (2017) suggested that the decline in the ratio between diatom and dinoflagellate biomasses between 1984 and 1991 was caused by warmer winters. Confidence in these results is, however, low.

In summer, the amount of cyanobacteria has increased and the phytoplankton biomass maximum, which in the 1980s was in spring, is now in July-August. This shift has been explained by a complex interaction between warming, eutrophication and increased top-down pressure (Suikkanen et al., 2013). There are, however, different opinions concerning the relative effects of eutrophication and climate on changes in phytoplankton biomass and community composition. In the long-term data, results vary according to area and species group (Wasmund et al.,





2011; Groetsch et al., 2016). Some studies saw evidence of eutrophication effects, modified by climate-induced
variations in temperature and salinity (Hällfors et al., 2013; Olofsson et al., 2020). Others found no explanation
for the gradual change in community composition, and concluded that the Baltic Sea phytoplankton community is
not in a steady state (Olli et al., 2011; Griffiths et al., 2020).

Cyanobacteria accumulations derived from satellite data for 1979-2018 show both short-term (two to three year)
oscillations and decadal-scale variations (Kahru and Elmgren, 2014; Kahru et al., 2018; Kahru et al., 2020).
Cyanobacteria accumulations in the Baltic proper were common in the 1970s and early 1980s, but rare during
1985–1990. They increased again starting in 1991 and, especially since 1998. In the 1980s, the annual chlorophyll
maximum in the Baltic proper was caused by the spring diatom bloom, but has in recent decades shifted to the
summer cyanobacteria bloom in July; the timing of this bloom has also advanced by about 20 days, from the end
to the beginning of July (Kahru et al., 2016).

In the Baltic proper, the hypoxia-induced decrease in N:P ratio and increase of phosphate pool left over in the
surface layer after the spring bloom led to intensification of the "vicious circle" (Vahtera et al., 2007), further
augmented by increasing water temperature, and resulted in conspicuous expansion of the surface diazotrophic
cyanobacteria accumulations, covering in the 21$^{st}$ century 150-200 thousand square kilometers (e.g. Kahru and
Elmgren, 2014; Savchuk, 2018, and references therein). Although mechanisms of the interannual oscillations of
two to three years remain unexplained (Kahru et al., 2018), there is a strong correlation of the accumulations with
hypoxia-related biogeochemical variables and water temperature at the decadal scale of five to twenty years (Kahru
et al., 2020). In the Bothnia Sea, the decreased nitrogen import and increased phosphorus import from the Baltic
proper has shifted the nutrient balance and made cyanobacteria accumulations a permanent feature (Kahru and
Elmgren, 2014; Kuosa et al., 2017) and probably also increased production and sedimentation (Ahlgren et al.,
2017; Kahru et al., 2018; Kahru et al., 2020; Kuosa et al., 2017; Lehtoranta et al., 2017; Rolff and Elfwing, 2015;
Savchuk, 2010; Vahtera et al., 2007).

Experimental evidence supports the idea that climate change can and will drive changes in the pelagic primary
production (Sommer et al., 2012), and a thorough review of benthic-pelagic coupling in the Baltic Sea
demonstrates ecosystem-wide consequences of altered pelagic primary production (Griffiths et al., 2017).
**3.2.6.1.3 Zooplankton**
Several studies have confirmed that marine copepod species have declined in abundance since the 1980s, while
euryhaline or limnetic, often small species have increased (Hänninen et al., 2015; Suikkanen et al., 2013; Kortsch
et al., 2021). The observed decline of marine taxa has been linked to the reduction in surface-water salinity since
the 1980s (Vuorinen et al., 2015), whereas the increase of brackish-water taxa has been positively influenced by
the temperature increase, directly or indirectly (Mäkinen et al., 2017). Small-scale effects on individual species
may affect reproductive success, and hence influence both populations and communities (Möller et al., 2015).



**3.2.6.2 Benthic habitats**

**3.2.6.2.1 Macroalgae and vascular plants**

Long-term changes in Baltic Sea macroalgae and charophytes have been attributed to changes in salinity, wind exposure, nutrient availability and water transparency (Gubelit, 2015; Blindow et al., 2016; Rinne and Salovius-Laurén, 2020), and biotic interactions may also play a role (Haavisto and Jormalainen, 2014; Korpinen et al., 2007). The long-term decrease of water transparency from 1936 to 2017 has been estimated to have reduced sea floor areas in the northern Baltic Sea favorable for *Fucus* spp. by 45% (Sahla et al., 2020). Overall, it is expected that climate change and its interaction with other environmental factors (e.g. eutrophication) will cause complex responses and influence carbon storage in both macroalgae and vascular plants in the Baltic Sea (Jonsson et al., 2018; Takolander et al., 2017; Röhr et al., 2016; Perry et al., 2019; Salo et al., 2020; Bobsien et al., 2021).

**3.2.6.2.2 Zoobenthos**

Soft-sediment benthic communities depend on variables that are influenced by climatic variability. On the south-western coast of Finland, amphipods have been replaced by Baltic clam *Limecola balthica* and the invasive polychaetes *Marenzelleria* spp., a change attributed to an increase in near-bottom temperature and fluctuations in salinity and oxygen (Rousi et al., 2013). Variations of zoobenthos in the Åland archipelago during 1983-2012 were associated with a salinity decline (Snickars et al., 2015), and effects related to climate change have acted as drivers for the long-term progression of zoobenthic communities (Rousi et al., 2019; Weigel et al., 2015; Ehrnsten et al., 2020).

**3.2.6.3 Non-indigenous species**

Numerous non-indigenous species have gained a stronghold in the Baltic Sea ecosystem during the past few decades, and in many cases these species have wider tolerance-ranges than the native ones, thus making them highly competitive under changing climate including warmer, and possibly less saline water, further impacted by other drivers such as eutrophication. The ecological impacts of these species may vary from filling vacant ecological niches to potentially outcompeting native species, and thus influencing the entire food web structure and functioning (Weigel et al., 2015; Griffiths et al., 2017; Ojaveer et al., 2017).

**3.2.6.4 Fish**

Sprat and herring in the Baltic Sea are influenced by multiple factors, including fisheries, predation, food availability and climatic variations. Sprat has benefited from the seawater warming (Voss et al., 2011). In 1990-2020, sprat populations were affected both by climate and top-down control, i.e. fisheries and predation by cod (Eero et al., 2016). In the 1980s, overfishing and a partly climate change-induced decline in suitable spawning habitat, 'reproductive volume', interacted to drastically reduce the cod population (Hinrichsen et al., 2011; Casini et al., 2016), with cascade effects on its main prey, sprat and herring, as well as zooplankton (Casini et al., 2008).

The various effects of temperature and salinity on sprat and cod also resulted in a spatial mismatch between these species, which contributed to an increase of sprat stocks (Reusch et al., 2018). The freshening of the Baltic Sea surface water, with the associated decline in marine copepods (Hänninen et al., 2015), contributed to a halving of weight-at-age of 3-year old herring, from 50–70 g in the late 1970s to 25–30 g in 2000s (Dippner et al., 2019).



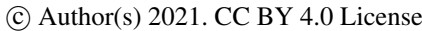


Among coastal fish, pikeperch (*Sander lucioperca*) has recently expanded its distribution northwards along the
coasts of the Bothnian Sea, apparently aided by warmer waters (Pekcan-Hekim et al., 2011). For many coastal
piscivores (perch, pike, pike-perch), as well as for cyprinids, coastal eutrophication is, however, equally or more
important than climate (Bergström et al., 2016; Snickars et al., 2015). Long-term studies illustrate that it is hard
to disentangle abiotic and biotic interactions, e.g. between fish and their food (benthos), and climate-related drivers
thus appear significant on a multidecadal time-scale across a large spatial scale (Törnroos et al., 2019).
**3.2.6.5 Marine mammals**
The breeding distributions of the ice-breeding seals in the Baltic Sea have evolved with ice coverage, with the
seals breeding where and when ice optimal for breeding occurs. Breeding ringed seals need ice throughout their
relatively long lactation period (>6 weeks), and also use ice as moulting habitat. Ringed seals prefer compact or
consolidated pack ice as it provides cavities and snowdrifts suitable for the construction of the lairs, most
importantly the breeding lair (Sundqvist et al., 2012).

Sea ice changes, along with implementation of specific management- and protection measures, have had a rapid
influence on the populations of several Baltic Sea mammal populations, in particular seals (Reusch et al., 2018).
Reusch et al. (2018) attributed these changes also to reduced exposure to harmful substances and increases in
overall fish stocks as a consequence of eutrophication (including reduced stocks of several commercial fish
species). Thus, specific climate change-related impacts on seals are hard to establish, although reconstructions of
distributional histories since the last glaciation have been attempted for some seal species (Ukkonen et al., 2014).

The availability of suitable breeding ice for ringed seals in the Bothnian Bay is decreasing (Section 3.2.4.4). The
breeding success of ringed seal was probably reduced by the exceptionally mild winter of 2007-2008 (Jüssi, 2012)
and several similar or even milder ice-winters have followed (Ilmatieteen laitos, 2020). The winters 2019–2020,
2007–2008 and 2014–2015 are the mildest in the annual ice cover statistics for the Baltic Sea (Uotila et al., 2015).
The southern breeding populations of the ringed seal in the Gulf of Finland, the Gulf of Riga and Archipelago Sea
are already facing the challenges of milder winters: the ice covered area during the breeding season has been
reduced and overlying snow for breeding lairs has been absent from the southern areas of the ringed seal breeding
range in most winters of the past decade (Ilmatieteen laitos, 2020). Thus, the only available breeding ice in the
Gulf of Finland in 2020 was found very near St. Petersburg (Halkka, 2020).

For grey seals, the lower availability of suitable breeding ice in its core distribution area has led to more breeding
on land in areas where drift ice used to be found (Jüssi et al., 2008). Grey seals are known to gather to breed in
certain sea areas regardless of the winter severity, so some land colonies may become overpopulated. As an
example, in 2016, 3,000 grey seal pups were born on three islets in the northern Gulf of Riga.

Flooding of seal haul-outs due to sea level rise will first occur in the southernmost Baltic Sea, where relative sea
level rise will be most rapid (EEA, 2019c), and haul-out sites are mainly low sand or shingle banks. In Kattegat,
relative sea level rise is estimated to be lower, and while the haul-outs in the western part are low sand and shingle
banks similar to those in the southern Baltic Sea, haul-outs in eastern Kattegat are skerries with a higher profile.



In the central and northern Baltic Sea, haul-outs are mainly skerries and here relative sea level rise is estimated to be low or even negative in the 21st century (EEA, 2019c). Yet, in recent history, winter storm surges have been observed to flood grey seal breeding colonies and push limited ice with ringed seal pups onto shore in the Gulf of Riga and Pärnu Bay. In the southern Baltic Sea and western Kattegat, increasing sea levels may turn parts of larger islands or previously inhabited islands into suitable seal haul-outs, but this is hard to project and depends, among other things, on the future management and protection of such areas.

Sea levels in the southern Baltic Sea have been rising at up to 3 mm year$^{-1}$ since the 1970s (EEA, 2019c) and the available haul-out areas have thus seen reductions already. However, during this time, the relevant harbour and grey seal populations have been recovering at high rates from past depletion (HELCOM, 2018b), with no documented or suspected effects of rising sea levels published.

The only cetacean resident in the Baltic Sea, the harbour porpoise (*Phocoena phocoena*), is a wide-spread species and seems to be rather tolerant of different temperatures as well as habitats. There are harbour porpoise populations in the waters around Greenland as well as along the coast of the Iberian Peninsula, Morocco, West Sahara and in the Black Sea. However, the Baltic proper harbour porpoise population has been shown to differ genetically (Lah et al., 2016) and morphologically (Galatius et al., 2012) from neighbouring populations, which may imply local adaptations that we are currently unaware of. Sea ice limits the range available to the Baltic proper harbour porpoise population since they need to come to the surface to breathe every 1-5 minutes. Hence, a decreasing ice cover is likely to increase the available range for the population. However, a change in the prey community resulting from climate-change related factors could potentially have serious effects on this critically endangered (Hammond et al., 2008) population of a small whale, which is dependent on constant access to prey (Wisniewska et al., 2016).

**3.2.6.6 Waterbirds**

The winter distribution of many waterbirds has extended northwards in response to the increase in temperature and the decreasing extent of sea ice cover. This can be observed as an overall increase in winter abundance of waterbirds, because part of the population of some species (mainly diving ducks) that formerly wintered further to the southwest now remain in the Baltic Sea (Pavón-Jordán et al., 2019). Many species show decreasing trends in abundance in the southern parts of their wintering ranges (typically in western and southern Europe) but increases near the northern edge of their distribution, typically the Baltic Sea region (MacLean et al., 2008; Skov et al., 2011; Aarvak et al., 2013; Lehikoinen et al., 2013; Pavón-Jordán et al., 2015; Nilsson and Haas, 2016; Marchowski et al., 2017; Fox et al., 2019). Similar shifts are seen in species that traditionally wintered in the Baltic Sea, but currently show declining wintering numbers there, as part of the population now winters in the White, Barents and Kara seas (Fox et al., 2019).

Although the community composition changes rapidly, the changes are not fast enough to track the thermal isocline shifts (Devictor et al., 2012; Gaget et al., 2021). How species respond to changes in winter temperature seems, however, to be highly species- or group-specific (Pavón-Jordán et al., 2019). Many species now winter closer to their breeding areas, shortening migration distances (Lehikoinen et al., 2006; Rainio et al., 2006; Gunnarsson et al., 2012).


Mainly owing to milder spring temperatures and related effects on vegetation and prey, many waterbirds migrate earlier in spring (Rainio et al., 2006), and hence arrive earlier in the breeding area (Vähätalo et al., 2004), and some also start breeding earlier (van der Jeugd et al., 2009). Delayed autumn migrations have also been noted, but their relation to climate change is less clear (Lehikoinen and Jaatinen, 2012).

Earlier loss of sea ice was found to improve pre-breeding body condition of female common eiders, leading to increasing fledging success in offspring (Lehikoinen et al., 2006). On the other hand, algal blooms promoted by higher seawater temperature has in some cases caused low quality in bivalve prey for common eiders, leading more birds to skip breeding (Larsson et al., 2014). Warmer seawater in winter also increases the energy expenditure of mussels, thus directly reducing their quality as prey for eiders (Waldeck and Larsson, 2013).

Most Baltic Sea waterbird species are migratory and affected by climate change also outside the Baltic Sea region, in the Arctic (breeding season) and in southern Europe and western Africa (wintering; Fox et al., 2015). This is important, given that climate warming is most pronounced in the Arctic and northern Eurasia and above average also in southern Europe and northern Africa (Allen et al., 2018).

**3.2.6.7 Marine food webs**

The entire marine food web of the Baltic Sea has been greatly impacted by climate change-related drivers that have altered the physical environment and the physiological tolerance limits of several species, by causing micro-evolution of Baltic Sea species, and by interactive effects of climate change with other environmental drivers, such as eutrophication and hypoxia/anoxia (Niiranen et al., 2013; Wikner and Andersson, 2012; Schmidt et al., 2020; Pecuchet et al., 2020).

Integrated approaches encompassing all of the ecosystem-components discussed above, are needed in order to understand and manage the linkages between large-scale and long-term changes driven by synergistic impacts of over-arching climate change-related physical and chemical drivers in combination with other factors such as eutrophication, which may complicate human adaptation to the changing marine ecosystem (Blenckner et al., 2015; Stenseth et al., 2020; Bonsdorff, 2021).

**3.3 Future climate change**

**3.3.1 Atmosphere**

**3.3.1.1 Large-scale atmospheric circulation**

In the future, the NAO is very likely to continue to exhibit large natural variations, similar to those observed in the past. In response to global warming, it is likely to become slightly more positive on average (Knudsen et al., 2011). Trends in the intensity and persistence of blocking remain uncertain (IPCC, 2014b). The AMO is expected to be very sensitive even to weak global warming, shortening the time scale of its response and weakening in amplitude (Wu et al., 2018; Wu and Liu, 2020). This will likely reduce the decadal variability of SSTs in the Northern Hemisphere. Recent studies indicate a degree of decadal predictability for blocking and the NAO influenced by the AMO (Athanasiadis et al., 2020; Wills et al., 2018; Jackson et al., 2015).





### 3.3.1.2 Air temperature

Table 7 lists the air temperature changes over the Baltic Sea catchment area and the Baltic Sea calculated from an ensemble of regional coupled atmosphere-ocean simulations (Gröger et al., 2021b). Due to the ice/snow-albedo feedback, warming is larger in winter than in summer, and the land is warming faster than the Baltic Sea (Fig. 26a). Due to its proximity to the Arctic, the Baltic Sea region including both land and sea is warming faster than the global mean figures (Section 1.5.1.1). The surface air temperature increase is expected to be largest in the northern Baltic Sea region especially in winter. These statements are true for both uncoupled atmosphere (Christensen et al., 2021) and coupled atmosphere-ocean regional climate simulations (Gröger et al., 2019; Gröger et al., 2021b).

For RCP2.6, the global annual mean surface air temperature change averaged over the simulations is 1.0°C. The corresponding global changes for RCP4.5 and RCP8.5 are 1.9 and 3.5°C, respectively. Over land in the Baltic Sea region, the warming is larger in each of the three scenarios, amounting to 1.5, 2.6 and 4.3°C, respectively (Table 7). Over the Baltic Sea, the increase is slightly smaller than over land (1.4, 2.4 and 3.9°C, respectively) but still larger than the corresponding global mean increase in surface air temperature. The latter result was expected and found in coupled atmosphere-ocean scenario simulations (Table 7), but not in all atmosphere-only runs (Christensen et al., 2021).

*Extreme Air temperatures:*

Changes in daily minimum and maximum temperatures have similar spatial patterns as the mean air temperature changes, with the expected greater warming for minimum temperature (Christensen et al., 2021). According to Christensen et al. (2021) and previous studies (BACC II Author Team, 2015), the latter result is explained by the reduced outgoing long-wave radiation under increased greenhouse gas concentrations. The long-wave radiation acts to cool the surface, especially when the ground is warmer than the air, e.g. during winter and during nights. The number of hot spells are projected to increase, in particular in the southern Baltic Sea region (Gröger et al., 2021b). In coupled atmosphere-ocean simulations, the strongest increases in the annual mean number of consecutive days of tropical nights and the annual maximum number of tropical nights (with temperature above 20°C all night) in the Baltic Sea region were projected to occur over the open sea (Gröger et al., 2021b). In contrast, projections of tropical nights with atmosphere-only models show no significant change (Gröger et al., 2021b; Meier et al., 2019a). Due to the sea ice/snow albedo feedback, the largest decline in the number of frost days was projected to occur over the northeastern Baltic Sea region, i.e. northern Scandinavia and adjacent northern Russia (Gröger et al., 2021b).

### 3.3.1.3 Solar radiation and cloudiness

There are a few studies on projected future solar radiation over Europe. Global climate models of the CMIP5 generation indicated an increase in surface solar radiation, highest over southern Europe and decreasing towards north, but still with a slight increase over the Baltic Sea (Bartók et al., 2017; Müller et al., 2019). However, some regional climate models instead showed a decrease in surface solar radiation over the Baltic Sea area, in winter by about 10% over most of the catchment (Bartók et al., 2017; Christensen et al., 2021). This change was largely attributed to increasing future cloud-cover, due to a more zonal airflow, and was accompanied by increased winter



precipitation. Thus, there are large differences in modelled surface solar radiation between global and regional
models (Bartók et al., 2017). Unknown future aerosol emissions add to the uncertainty.

Global mean energy balance components have improved with every new climate model generation. For the latest
CMIP6, models show good agreement for clear sky shortwave energy fluxes in today's climate, both between
models and compared to reference data (Wild et al., 2021). However, there are still substantial discrepancies among
the various CMIP6 models in their representation of several of the global annual mean energy balance components,
and the inter-model spread increases further on regional, seasonal and diurnal scales. Thus, future changes in solar
radiation and cloudiness remain highly uncertain, not least on the regional scale.
**3.3.1.4 Precipitation**
Precipitation in winter and spring is projected to increase over the entire Baltic Sea catchment, while summer
precipitation is projected to increase in the northern half of the basin only (Christensen et al., 2021). In the south,
summer precipitation is projected to change very little, although with a large spread between different models
including both increases and decreases. The projected increase in the north is a rather robust feature among the
regional climate models but with a large spread in the amount. Ensemble mean precipitation changes from coupled
atmosphere-ocean simulations are summarized in Table 8. For the Baltic Sea catchment area, projected annual
mean precipitation changes for the three RCP scenarios amount to 5, 9 and 15% (Table 8) and are much larger
than global averages (Section 1.5.1.2). Over the Baltic Sea, the changes are similar than over the land area (6, 8
and 16%).

Expressed by the Clausius-Clapeyron equation, warming increases the potential for extreme precipitation due to
intensification of the hydrological cycle associated with the growth of atmospheric moisture content. For Northern
Europe, regional climate models indicate an overall increase in the frequency and intensity of heavy precipitation
events in all seasons and longer wet and dry spells (Christensen and Kjellström, 2018; Rajczak and Schär, 2017;
Christensen et al., 2021, and references therein). The largest increase in the number of high precipitation days is
projected for autumn. The number of drought events per year are expected to decrease, while their length is
expected to increase (Christensen and Kjellström, 2018). Changes in more extreme events, like 10-, 20- or 50-year
events, are less certain.
**3.3.1.5 Wind**
In general, projected changes in wind speed over the Baltic Sea region are not robust among Earth System Models
(Kjellström et al., 2018; Gröger et al., 2021b). However, Ruosteenoja et al. (2019) found in CMIP5 projections a
slight but significant wind speed increase in autumn and a decrease in spring over Europe and the North Atlantic.
Furthermore, over sea areas where the ice cover is projected to diminish on average, such as the Bothnian Sea and
the eastern Gulf of Finland, the mean wind is projected to increase systematically because of a warmer sea surface
and reduced stability of the planetary boundary layer (Meier et al., 2011c; Gröger et al., 2021b; Räisänen, 2017).

Projection of the future behavior of extratropical cyclones are uncertain because changes in several drivers result
in opposite effects on cyclone activity. With global warming, the lower troposphere temperature gradient between
low and high latitudes decreases due to polar amplification. Near the tropopause and in the lower stratosphere, the





opposite is true, thus implying changes in baroclinicity (Grise and Polvani, 2014; Shaw et al., 2016; Stendel et al.,
2021). An increase in water vapour enhances diabatic heating and tends to increase the intensity of extratropical
cyclones (Willison et al., 2015; Shaw et al., 2016) and contribute to their propagation further poleward (Tamarin-
Brodsky and Kaspi, 2017; Tamarin and Kaspi, 2017). The opposite is true in parts of the North Atlantic region,
e.g. south of Greenland. For this region the North-South gradient is increasing, as the weakest warming in the
entire Northern Hemisphere is over ocean areas south of Greenland. North of this local minimum the opposite is
true. The increase in the North-South gradient over the North Atlantic may be responsible for some Earth System
Models showing an intensification of the low pressure activity and thereby high wind speed over a region from
the British Isles and through parts of north-central Europe (Leckebusch and Ulbrich, 2004; Ulbrich et al., 2008).
These projections have been confirmed by (Harvey et al., 2012). They compared the ensemble storm track response
of CMIP3 and CMIP5 model simulations and found that both projections show an increase in storm activity in the
midlatitudes, with a smaller spread in the CMIP5 simulations. In contrast to CMIP3, the CMIP5 ensemble showed
a significant decrease in cyclone track density north of 60°N. Hence, pre-CMIP3 and CMIP3 studies showed a
clear poleward shift of the North Atlantic storm track (e.g. Fischer-Bruns et al., 2005; Yin, 2005; Bengtsson et al.,
2009), whereas the CMIP5 ensemble predicts only an eastward extension of the North Atlantic storm track (Zappa
et al., 2013). The newest generation of models from CMIP6 resulted in significant reduction of biases in storm
track representation compared to CMIP3 and CMIP5, but the response to climate change is quite similar compared
to the previous assessments (Harvey et al., 2020). The eastward extension of the North Atlantic storm track seems
to be a robust result as it is found in pre-CMIP3, CMIP3 and CMIP5 simulations (Feser et al., 2015).
In summary, there is no clear consensus among climate change projections in how changes in frequency and/or
intensity of extratropical cyclones will affect the Baltic Sea region (Räisänen, 2017). However, in future climate
the frequency of severe wind gusts in summer associated with thunderstorms may increase (Rädler et al., 2019).
**3.3.1.6 Air pollution, air quality and atmospheric nutrient deposition**
The main conclusions by the BACC II Author Team (2015) concerning projections of air quality in the Baltic Sea
region still hold. The main factor determining future air quality in the region is regional emissions of air pollutants,
not changes in meteorological factors related to climate change or in intercontinental pollution transport (see e.g.
Langner et al., 2012; Hedegaard et al., 2013).
Recent post-BACC II air quality modelling studies for the Baltic Sea area are Colette et al. (2013), Varotsos et al.
(2013), Colette et al. (2015), Hendriks et al. (2016), and Watson et al. (2016). They concentrate mainly on
particulate matter (PM) and ground-level ozone ($O_3$), the pollutants most likely to be affected by changing climate
parameters. They agree with current day air quality trends in that the Baltic Sea region in general is less exposed
to air pollution than the rest of Europe.
Jacob and Winner (2009) showed that climate change is likely to increase ground-level ozone in central and
southern Europe. In a meta-analysis, Colette et al. (2015) assessed the significance and robustness of the impact
of climate change on European ground-level ozone based on 25 model projections, including some driven by SRES
(Special Report on Emission Scenarios by Nakicenovic et al., 2000) and RCP scenarios. They indicate that an
increase in ground-level ozone is not expected for the Baltic Sea region. A latitudinal gradient was found from



increase in large parts of continental Europe (+ 5 ppbv), but a small decrease over Scandinavia (up to -1 ppbv).
Studies that explicitly compared the magnitude of projected climate and anthropogenic emission changes (Langner
et al., 2012; Colette et al., 2013; Varotsos et al., 2013) all confirmed that changes in emission of ozone precursors
($NO_x$, VOCs) had the larger effect. For Northern Europe, Varotsos et al. (2013) estimated that reductions in snow
cover and solar radiation in a SRES A1B scenario lead to an ozone decrease of about 2 ppb by 2050, compared to
present conditions.
Varotsos et al. (2013) stress the importance of future biogenic isoprene emissions for ozone concentrations. In the
2050 climate, increases in ozone concentrations are associated with increased biogenic isoprene emissions due to
increased temperatures, whereas increased water vapour over the sea, as well as increased wind speeds, are
associated with decreases. Hendriks et al. (2016) emphasise that isoprene emissions may increase significantly in
coming decades if short-rotation coppice plantations are greatly expanded, to meet the increased biofuel demand
resulting from the EU decarbonisation targets. They investigate the competing effects of anticipated trends in land
use, anthropogenic emissions of ozone precursors and climate change on European ground-level ozone
concentrations and related health and environmental effects by 2050. They found that increased ozone
concentrations and associated health damage caused by a warming climate (+ 2 to 5°C across Europe in summer)
might be more than the reduction that can be achieved by cutting emissions of anthropogenic ozone precursors in
Europe.
Orru et al. (2013, 2019) and Geels et al. (2015) studied the effect of climate change on ozone-related mortality in
Europe. Orru et al. (2019) present their results on country level, including all Baltic Sea EU-countries. They
conclude that although mortality related to ground-level ozone is projected to be lower in the future (mainly due
to decrease precursor emissions), the reduction could have been larger, without climate change and an increasingly
susceptible population.
In parts of the Baltic Sea region, a considerable air pollution is due to shipping. Ship traffic in the region is
projected to increase over the coming decades, which could lead to larger emissions (i.e. $NO_x$ and PM) than today,
unless stricter air quality regulations counter this potential trend. For the Baltic Sea, a nitrogen emission control
area (NECA) will become effective in 2021. Karl et al. (2019a) designed future scenarios to study the effect of
current and planned regulations of ship emissions and the expected fuel efficiency development on air quality in
the Baltic Sea region. They showed that in a business-as-usual scenario for 2040 (SECA-0.1% and fuel efficiency
regulation effective starting in 2015), the introduction of the NECA will reduce $NO_X$ emissions from ship traffic
in the Baltic Sea by about 80% in 2040. The reduction in NOx emissions from shipping translates to a ~60%
decrease in $NO_2$ summer mean concentrations in a wide corridor around the ship routes. The coastal population of
northern Germany, Denmark and western Sweden will be exposed to less $NO_2$ in 2040 due to the introduction of
the NECA. With lower atmospheric $NO_x$ levels, less ozone will be formed, and the estimated daily maximum $O_3$
concentration over the Baltic Sea in summer 2040 will on average be 6% lower than without the NECA. Compared
to today, the introduction of the NECA will also reduce ship-related PM2.5 emissions by 72% by 2040, compared
to -48% without the NECA. Simulated nitrogen deposition on the Baltic Sea decreases 40-44% on average between
2012 and 2040. A similar study by Jonson et al. (2019) estimated that the contributions of Baltic Sea shipping to
NO$_2$ and PM2.5 concentrations, and to the deposition of nitrogen, will be reduced by 40-50 % from 2016 to 2030,
mainly as a result of NECA.
**3.3.2 Land**
**3.3.2.1 River discharge**
Climate change is likely to have a clear influence on the seasonal river flow regime, as a direct response to changes
in air temperature, precipitation and evapotranspiration (BACC II Author Team, 2015; Blöschl et al., 2017).

For areas in the northern Baltic Sea region presently characterized by spring floods due to snow melt, the floods
are likely to occur earlier in the year and their magnitude is likely to decrease owing to less snowfall, shorter snow
accumulation period, and repeated melting during winter. As a consequence, sediment transport and the risk of
inundation are likely to decrease.

In the southern part of the Baltic Sea region, increasing winter precipitation is projected to result in increased river
discharge in winter. In addition, groundwater recharge is projected to increase in areas where infiltration capacity
is not currently exceeded, resulting in higher groundwater levels. Decreasing precipitation combined with rising
temperature and evapotranspiration during summer is projected to result in drying of the root zone, increasing
demands for irrigation in the southern Baltic Sea region.

Projections with a process-oriented hydrological model suggested that, under the RCP4.5 and RCP8.5 scenarios,
the total river flow during 2069-2098 relative to 1976-2005 will increase 1-21% and 6-20%, respectively,
illustrating the large uncertainty in hydrological projections (Saraiva et al., 2019a; Meier et al., 2021b). According
to these and previous projections, the increase of river flow will mainly take place in the north, while total river
flow to the south will decrease (Stonevičius et al., 2017; Šarauskienė et al., 2017). Winter flow will increase due
to intermittent melting (Stonevičius et al., 2017). Projected discharge changes attributed to increasing air
temperature are reflected in observed trends (Section 3.2.2.1), whereas changes attributed to increasing
precipitation are necessarily not (Wilson et al., 2010).

Since the publication of BACC II (BACC II Author Team, 2015), ensemble sizes of scenario simulations with
hydrological models have increased, enabling the estimate of uncertainties in projections (e.g. Roudier et al., 2016;
Donnelly et al., 2017). Donnelly et al. (2014) focused on projecting changes in discharge to the Baltic Sea, by
using a semi-distributed conceptual hydrological model for the BSDB (Balt-HYPE), combined with a small
ensemble of climate projections under the SRES A1B and A2 scenarios. Results showed an increased overall
discharge to the Baltic Sea, with a seasonal shift towards higher winter and lower summer flows and diminished
seasonal snow-melt peaks. Efforts were made to assess the uncertainty in the model chain, and change magnitudes
were shown to be within the range of the overall uncertainty estimates, highlighting the importance of such
uncertainty assessments in effect studies to frame the quantitative model results.

Arheimer and Lindström (2015) studied future changes in annual maximum and minimum daily flows. Their
projections suggested that snow-driven spring floods in the northern–central part of Sweden may occur about one





month earlier than today and rain-driven floods in the southern part of Sweden may become more frequent. The
boundary between the two flood regimes is projected to shift northward.

Past observations (see Section 3.2.2.1) and future projections (e.g. Graham, 2004) suggest a temporal shift in the
seasonality of the river discharge, with decreasing flow in spring/summer and increasing flow in winter. Global
warming and river regulation due to hydropower production cause similar changes. However, in snow-fed rivers
globally the impact of climate change is projected to be minor compared to river regulation (Arheimer et al., 2017).
**3.3.2.2 Land nutrient inputs**
Projected changes in riverine discharge and nutrient inputs from the Baltic Sea Drainage Basin (BSDB) to Baltic
Sea coastal waters have been studied using a number of modelling frameworks in recent years. Projecting the
regional effects of future climate and environmental change on hydrology and nutrient turnover poses challenges
in terms of (i) the complex nature of the modelled system, including human influence on riverine nutrient inputs
and transport processes alike, which necessitates long projection model chains and leads to uncertainty in modelled
hydrologically driven responses, and (ii) the significance of changes in human behaviour, e.g. in terms of land
management, population, or nutrient emissions from point sources, which adds complexity to the formulation
scenarios for future change, on top of the climate change signal. Hydrological impact studies in the BSDB (and
elsewhere) therefore often explicitly use simplifying assumptions in order to reduce complexity of the modelled
system and to put focus on certain aspects of impacts of projected changes.

Hesse et al. (2015) also reported increasing discharges in a similar model study of the Vistula lagoon catchment,
using a hydrological model (SWIM), which also allows for nutrient load assessment, and climate change impact
modelling based on a climate model ensemble. On average, results showed decreasing trends for nitrogen and
phosphorus inputs, but a wide range of projections with individual ensemble members.

Hägg et al. (2014) used a split model approach to project changes in TN and TP inputs to Baltic Sea sub-basins.
Changes in discharge were estimated with a hydrological model (CSIM) combined with a climate projection
ensemble, which sampled a range of climate model and emission scenario combinations. Inputs were then
calculated with a statistical model, based on modelled discharges and population as a proxy for human nutrient
emissions, combining population change assumptions with climate projections. Results showed a general trend
towards higher nutrient inputs across the region as a result of climate change, and a significant (i.e. potentially
trend-changing) influence of human adaptation scenarios, particularly in the southern half of the BSDB.

Øygarden et al. (2014) used measurements in a number of small agricultural catchments to establish functional
relationships between precipitation, runoff, and N losses from agricultural land, and qualitatively related their
findings to projected precipitation changes across the BSDB under climate change scenarios, as well as to
mitigation measures to counter the climate-driven effects. The analyses showed a positive relationship between
runoff and N losses as well as between rainfall intensity and N losses, but stressed the wide range of feedback
loops possible between climate change effects and adaptation measures, through management or policy changes.
Such data-driven approaches avoid uncertainties related to effect-model chains at the expense of direct BSDB-
wide quantitative effect projections.



The potential effects of socio-economic adaptation under climate change conditions were investigated by Huttunen et al. (2015) in a study of Finnish catchments draining to the Baltic Sea. A national nutrient load model (VEMALA) was combined with a mini-ensemble of climate effects, and then a number of agricultural adaptation scenarios were derived, based on crop yield and policy changes, and an economic model (DREMFIA) was used to translate scenario assumptions to changes in the nutrient load model for evaluation of effects. On average, increased precipitation led to increased annual discharge and a shift from spring to winter peaks, with total nitrogen (TN) and total phosphorus (TP) inputs increasing with the discharge. Here, adaptation scenarios had less effect than climate change, with some regional variation, but significantly different load reductions were found among assessed adaptation strategies, leading to the conclusion that adaptation measures are important for overall climate change effect mitigation in the region.

The relative importance of management decisions for TN and TP load effects was studied also by Bartosova et al. (2019), using the hydrological model E-HYPE on the full BSDB. The ensemble approach combined climate and socioeconomic pathways based on IPCC fifth assessment data (Zandersen et al., 2019), where socioeconomic changes were directly translated into changes of the effect model setup. The influence of socioeconomic adaptation choices on nutrient inputs to the Baltic Sea were shown to be in the same magnitude range as climate effects, thus indicating the importance of effective mitigation strategies for the region. In order to increase this efficiency, Refsgaard et al. (2019) developed and explored the concept of spatially differentiated measures for TN load reductions in the BSDB, based on the realization that measures are not uniformly efficient over large area, and should therefore not be uniformly applied either.

### 3.3.3 Terrestrial biosphere

In the following, we focus on the European drought in 2018, to study the impact of very warm conditions on the terrestrial ecosystem, and on projections for the terrestrial ecosystems in the Arctic, because of the particularly strong climate warming in the Arctic and potentially strong feedbacks from the release of $CO_2$ and $CH_4$ in the northernmost part of the Baltic Sea region. Finally, we discuss mitigation scenarios for land use and land-cover changes associated with the Paris Agreement.

*Terrestrial ecosystems in the European drought year 2018*

The summer of 2018 saw extremely anomalous weather conditions over Europe, with high temperatures everywhere, as well as low precipitation and high incoming radiation in western, central and Northern Europe (Peters et al., 2020). These extreme weather conditions resulted in severe drought (indicated by soil moisture anomalies) in western, central and Northern Europe, including the entire Baltic Sea region. The impacts of the severe drought and heatwave in Europe in 2018 were investigated in a series of papers, ranging from individual sites to the continental scale (Peters et al., 2020).

Graf et al. (2020) studied the effects of the 2018 drought conditions on the annual energy balance at the land surface, in particular the balance between sensible and latent heat fluxes, across different terrestrial ecosystems at various sites in Europe. Graf et al. (2020) found a 9% higher incoming solar radiation compared to their reference period across the drought-affected sites. The outgoing shortwave radiation mostly followed the incoming radiation,



with an increase of 11.5%, indicating a small increase in surface albedo. The incoming longwave radiation, on the
other hand, did not change significantly, indicating that effects of higher atmospheric temperatures and reduced
cloudiness cancelled out, while outgoing longwave radiation increased by 1.3% as a result of higher land surface
temperatures. Overall, the net radiation increased by 6.3% due to the extreme drought conditions. As for the non-
radiative surface energy fluxes, the sensible heat flux showed a strong increase by 32%, while the latent heat fluxes
did not change significantly on average. Graf et al. (2020) attributed the negligible effect on latent heat fluxes to
the opposing roles of increased grass reference evapotranspiration on the one hand and soil water depletion,
stomatal closure and plant development on the other. Evapotranspiration increased where and when sufficient
water was available and later decreased only where stored soil water was depleted. As a consequence, latent heat
fluxes typically decreased at sites with a severe precipitation deficit, but often increased at sites with a comparable
surplus of grass reference evapotranspiration but only a moderate precipitation deficit. Consistent with this,
peatlands were identified as the only ecosystem with very strong increases in latent heat fluxes but insignificant
changes in sensible heat fluxes under drought conditions. Crop sites, on the other hand, showed significant
decreases in latent heat fluxes.
Lindroth et al. (2020) analysed the impact of the drought on Scandinavian forests, based on 11 forest ecosystem
sites differing in species composition, i.e. spruce, pine, mixed and deciduous. Compared to their reference year, in
2018 the forest ecosystem showed a slight decrease in evaporation at two of the sites, was nearly unchanged at
most sites and increased at two sites with pine forest. At the same time, the mean surface conductance during the
growing season was reduced 40-60% and the evaporative demand increased 15-65% due to the warm and dry
weather conditions. The annual net ecosystem productivity (NEP) decreased at most sites, but the reasons differed.
At some sites, the NEP decrease was due to an increase in ecosystem respiration (RE), while at others both RE
and the gross primary productivity (GPP) decreased, with the decrease in GPP exceeding that in RE. At six sites,
the annual NEP decreased by over 50 g C m$^{-2}$ year$^{-1}$ in 2018. Across all sites considered, NEP anomalies varied
from -389 to +74 g C m$^{-2}$ year$^{-1}$. A multi-linear regression analysis revealed that the anomalous NEP could to a
very large extent (93%) be explained by anomalous heterotrophic respiration and reduced precipitation, with most
of the variation (77%) due to the heterotrophic component.
Rinne et al. (2020) studied the effects of the drought on greenhouse gas exchange in five northern mire ecosystems
in Sweden and Finland. Due to low precipitation and high temperatures, the water table sank in most of the mires.
This led not only to a lower $CO_2$ uptake, but also to lower $CH_4$ emissions by the ecosystems. Three out of the five
mires switched from sinks to sources of $CO_2$. Estimates of the radiative forcing expected from the drought-related
changes in greenhouse gas fluxes indicated an initial cooling effect due to the reduced $CH_4$ emissions, lasting up
to several decades, followed by a warming caused by the lower $CO_2$ uptake. However, it is unknown whether these
results can be generalized to all wetlands of the Baltic Sea region.
*Terrestrial ecosystems in the Arctic region*
Climate warming has been particularly strong at high northern latitudes, and climate change projections indicate
that this trend will continue, due to the anticipated increase in anthropogenic climate forcing. This strong warming
is expected to have major consequences for terrestrial ecosystems in Arctic and sub-Arctic regions.



Zhang et al. (2013) used the Arctic version of a dynamic global vegetation model (LPJ-GUESS, Smith et al.,
2001), forced with a regionalized climate scenario (A1B anthropogenic emission scenario), to investigate land
surface feedbacks from vegetation shifts and biogeochemical cycling in terrestrial ecosystems under future climate
warming. They found marked changes in vegetation by the second half of the 21st century (2051-2080), i.e. a
poleward advance of the boundary between forests and tundra, expansion of tundra covered with tall shrubs and a
shift from deciduous trees, e.g. birch, to evergreen boreal coniferous forest. These changes in vegetation were
associated with decreases in surface albedo, particularly in winter due to the snow-masking effect, and with
increases in evapotranspiration. The reduced surface albedo would tend to enhance the projected warming (positive
feedback), while increased evapotranspiration would dampen it (negative effect). The terrestrial ecosystems
continued to act as carbon sinks during the 21st century, but at diminished rates in the second half of the century.
The initial increase in carbon sequestration, due to a longer growing season and $CO_2$ fertilisation, could be reduced
and eventually reversed by increased soil respiration and greater $CO_2$ release from increased wildfires. Peatlands
were identified as hotspots of $CH_4$ release, which would further enhance the projected warming (positive
feedback).
Using a regional Earth System Model (RCA-GUESS; Smith et al., 2011) over the Arctic region, Zhang et al.
(2014b) investigated the role that the biophysical effects of the projected future changes of the land surface play
for the terrestrial carbon sink in the Arctic region under a future climate scenario based on a high emission scenario
(RCP8.5). Two simulations were performed to determine the role of the biophysical interactions, one with and one
without the biophysical feedbacks resulting from the simulated climatic changes to the terrestrial ecosystems in
the model. In both simulations the Arctic terrestrial ecosystems continued to sequester carbon until the 2060-
2070s, after which they were projected to turn into weak sources of carbon, due to increased soil respiration and
biomass burning. The biophysical effects were found to markedly enhance the terrestrial ecosystem carbon sink,
particularly in the tundra areas. Two opposing feedback mechanisms, mediated by changes in surface albedo and
evapotranspiration, contributed to the additional carbon sequestration. The decreased surface albedo in winter and
spring notably amplified warming in spring (positive feedback), while the increased evapotranspiration led to a
marked cooling during summer (negative feedback). These feedbacks stimulated vegetation growth due to an
earlier start of the growing season, leading to changes in woody plant species and the distribution of vegetation.
In a later study, Zhang et al. (2018) found that these biophysical feedbacks play essential roles also in climate
scenario simulations with weaker anthropogenic climate forcing.
*Mitigation*
The beneficial effects of carbon sequestration by forest ecosystems on climate change may be reinforced,
counteracted or even offset by management-induced changes in surface albedo, land-surface roughness, emissions
of biogenic volatile compounds, transpiration and sensible heat flux (see above). Luyssaert et al. (2018)
investigated the trade-offs associated with using European forests to meet the climate objectives in the Paris
Agreement. A central argument of this study was that the agreement requires more than that forest management
should dampen the rise in atmospheric $CO_2$ and reduce the radiative imbalance at the top of the atmosphere. The
authors suggested two additional targets, that forest management should neither increase the near-surface
temperature nor decrease precipitation, because climate effects arising from the changes in the terrestrial biosphere
would make adaptation to climate change more demanding. Analysing different forest management portfolios in





Europe designed to maximize the carbon sink, maximize the forest albedo or reduce near-surface temperatures,
Luyssaert et al. (2018) found that only the portfolio designed to reduce near-surface temperatures accomplished
two of the objectives, i.e. to dampen the rise in atmospheric $CO_2$ and to reduce near-surface temperatures. This
portfolio featured a decrease in the area of coniferous forest in favour of a considerable increase in the area of
deciduous forest in Northern Europe, from 130,000 to 480,000 $km^2$.
**3.3.4 Cryosphere**
**3.3.4.1 Snow**
There is agreement among models that the average amount of snow accumulated in winter will decrease by over
70% in most of the Baltic Sea region. The high Scandinavian mountains, where the warming temperature will not
reach the freezing point as often as in lower-lying regions, are an exception (Christensen et al., 2021). The
reduction in snow amount is slightly larger than in maps presented by the BACC II Author Team (2015), which is
consistent with the stronger average warming projected in the RCP8.5 scenario, compared to the SRES A1B
scenario analyzed by the BACC II Author Team (2015).

For Poland, two additional downscaling experiments were made to produce reliable high-resolution climate
projections of precipitation and temperature, using the RCP4.5 and RCP8.5 scenarios (Szwed et al., 2019). The
results were used as input to a snow model (seNorge), to transform bias-adjusted daily temperature and
precipitation into daily snow conditions. The snow model projected future snow depth to decrease in autumn,
winter and spring, in both the near and far future. The maximum snow depth was projected to decrease 15-20%
by 2021-2050 and at least double that decrease by 2071–2100 (Szwed et al., 2019).
**3.3.4.2 Glaciers**
The Special Report on the Ocean and Cryosphere in a Changing Climate (IPCC, 2019a) provides the most recent
assessment of future projected glacier mass reduction under various RCPs, and treats Scandinavian glaciers
separately (Hock et al., 2019). Previous projections, summarized in the Fifth Assessment Report of the IPCC
(IPCC, 2014a; Vaughan et al., 2014), did not specifically focus on Scandinavian glaciers.

By 2100, likely (i.e. with a likelihood of 60-100%) mass losses for high-mountain glaciers are 22-44% (RCP2.6)
to 37-57% (RCP8.5) of their mass in 2015. These losses exceed global projections for glacier mass loss of 18 ±
7% for RCP2.6, and 36 ± 11% for RCP8.5 (likely ranges, IPCC, 2019a). Glaciers in Scandinavia will lose over
80% of their current mass by 2100 under RCP8.5 (medium confidence), and many are projected to disappear,
regardless of future emission scenarios (Fig. 27). Furthermore, river runoff from glaciers is projected to change
regardless of emission scenario (high confidence), and to result in increased average winter runoff (high
confidence) and in earlier spring peaks (high confidence; Hock et al., 2019).

Projections of future glacier mass loss depend crucially on climate projections providing surface air temperature
and precipitation as forcing factors in process-based glacier models. For high mountain glaciers, such as those
along the Scandinavian mountains that drain into the Baltic Sea, this is challenging, as the interplay of regional
effects such as high-mountain meteorology and elevation-dependent warming (Wang et al., 2016; Qixiang et al.,
2018) with global climate is poorly understood (Hock et al., 2019). Surface air temperatures in mountain regions



are projected to increase at an average rate of 0.3 ± 0.2°C per decade until 2050 (very high confidence), i.e. faster
than the present global average of 0.2 ± 0.1°C (Hock et al., 2019). Beyond 2050, and depending on the emission
scenario, air surface temperatures in high mountain regions are projected to either stabilize at the 2015-2050 rate
or to increase further (IPCC, 2019a).

Projected changes in surface air temperature for the period 2071–2100 (compared to 1971–2000, under various
emission scenarios) for the part of the Baltic Sea Drainage Basin that extends along the Scandinavian mountains
(SMHI, 2020; Kjellström et al., 2016) will be of great importance for the assessment of future mass loss from
glaciers draining into the Baltic Sea.
**3.3.4.3 Permafrost**
Due to recent warming more than 20% of the permafrost in the region was already lost in 1997-2018 (Figure 13;
Obu et al., 2020). As warming increases, so will loss of permafrost (Section 1.5.2). Global projections show very
limited permafrost in the region already at +2°C (Chadburn et al., 2017), but global projection (including Chadburn
et al., 2017) do not account for peatland permafrost, which can persist for centuries outside of its climate
equilibrium (Osterkamp and Romanovsky, 1999). Much of the permafrost in Baltic Sea region was very close to
its climatic boundary even before the recent acceleration of climate warming. Much of the lowland permafrost in
palsas and peat plateaus in this region is very close to the 0°C thawing point, and is likely relict permafrost,
persisting from the Little Ice Age (Sannel et al., 2016). Observations also show that lowland permafrost thaw has
been going on for decades (Åkerman and Johansson, 2008). Preliminary analyses of permafrost loss in 1997-2018
suggests that this was roughly equally divided between alpine and lowland permafrost (22 and 24%, respectively,
Figure 13), in agreement with projections of loss of all types of Baltic permafrost in the future.

Permafrost thaw by climate warming is known to affect river runoff and its loads of carbon, nutrients and
contaminants, such as mercury (Schuster et al., 2018; Vonk et al., 2015). The local effect of permafrost thaw in
alpine headwaters can be significant (Lyon et al., 2009), but in the Baltic Sea Basin alpine permafrost thaw will
likely have limited influence on the characteristics of river transport at their mouths on the Baltic Sea. This is
because the alpine permafrost in the Baltic Sea drainage basin mainly affects solid bedrock or regolith, with almost
no soil organic matter stored in permafrost (Fuchs et al., 2015). Thaw of permafrost in peatlands affects soils with
very large stocks of organic material, and has been suggested to cause large losses of peat carbon and nutrients
into aquatic ecosystems (Hugelius et al., 2020). However, these projections are highly uncertain and based on
studies of peatland thaw chronosequences in North America that may not be applicable to Fennoscandian
permafrost peatlands (though see Tang et al., 2018).

The extent of permafrost in the Baltic Sea drainage basin may decrease significantly in this century, and depending
on which warming trajectory the Earth takes, may disappear altogether in the coming century. The thaw of alpine
permafrost will have little effect on flows of water, carbon and nutrients to the Baltic Sea. Thawing peatlands may
increase the loads of carbon, nutrients and mercury to the Baltic Sea, but these projections remain highly uncertain.



**3.3.4.4 Sea ice**

Two new projections for sea ice in the Baltic Sea have been produced after BACC II (BACC II Author Team, 2015). Luomaranta et al. (2014) used simplified regression and analytical models to estimate changes in sea-ice extent (Fig. 28) and fast-ice thickness. Due to their less demanding computational approach, they could base estimates on 28 CMIP5 models. As in the Arctic Ocean (Section 1.5.2), maximum annual ice extent and thickness were both estimated to decline in the future, but some sea ice will still form every year, even by the end of the century, in agreement with earlier studies (e.g. Haapala et al., 2001; Meier, 2002; Meier et al., 2004a). Under the RCP4.5 and RCP8.5 scenarios, the modelled mean maximum ice thicknesses in Kemi were projected to be 60 cm and nearly 40 cm, respectively, in 2081-2090. However, under the RCP8.5 scenario, two models projected Kemi to be ice-free.

Höglund et al. (2017) used a more advanced approach to examine changes in sea ice conditions with a coupled ice-ocean model (Hordoir et al., 2019; Pemberton et al., 2017). They used downscaled atmospheric data from the EC-Earth and the Max Planck Institute Earth System models and simulated the response of the ice for the RCP4.5 and RCP8.5 projections. Average annual maximum ice extent at the end of the century was projected to be 90 – 100 $10^3$ km$^2$ and 30 – 40 $10^3$ km$^2$, for the medium and high emission scenarios, respectively, and ice thickness to decrease 3 – 6 cm decade$^{-1}$. Höglund et al. (2017) also projected the mobility of the ice to increase, but with little effect on future ridged ice production.

**3.3.4.5 Lake ice**

The latest model experiments demonstrate that the Baltic Sea catchment will experience a substantial reduction in lake ice cover in the future, with many lakes becoming ice covered only intermittently (Maberly et al., 2020; Sharma et al., 2019; Sharma et al., 2021; Shatwell et al., 2019). This change will commence in the south and move northwards gradually. Lithuanian and Latvian lakes will lose their ice cover after +2°C warming, and further warming will gradually move winter ice loss northwards, so that at +8°C warming, only lakes in northernmost Lapland will retain a winter ice cover (Maberly et al., 2020; Sharma et al., 2019; Sharma et al., 2021; Shatwell et al., 2019).

**3.3.5 Ocean and marine sediments**

**3.3.5.1 Water temperature**

Ocean temperatures are rising at accelerating rates (IPCC, 2019a; Section 1.5.3.2). For the end of this century, scenarios for the Baltic Sea project a sea surface temperature increase of 1.1°C (0.8-1.6°C, RCP2.6) to 3.2°C (2.5-4.1°C, RCP8.5) compared to 1976-2005 (Gröger et al., 2019; Gröger et al., 2021b), see Table 9. In brackets, the ensemble spreads indicated by the 5th and 95th percentiles are listed. These changes are slightly larger than the projected global sea surface temperature changes (Section 1.5.3.2). Other ensembles than the one by Gröger et al. (2019) give similar results that vary between 1.9°C (RCP4.5) and 2.9°C (RCP8.5) for the ensemble mean temperature increase (Meier et al., 2021b; see also Meier and Saraiva, 2020). By the end of the century, sea surface temperature changes for the RCP8.5 scenarios significantly exceed natural variability. Largest open-sea warming is found in summer in the northern Baltic Sea, due to earlier melting of the sea ice (Figs. 29 and 30). Even higher



warming of +2–6°C (the range denotes RCP2.6 and RCP8.5 scenarios) is projected for the Curonian Lagoon by
the year 2100 (Jakimavičius et al., 2018).

The main driver of interannual variations of monthly mean sea surface temperature is air temperature, through the
sensible heat fluxes (Meier et al., 2021a). The second most important drivers are cloudiness over the open sea and
latent heat and meridional and zonal wind velocities over coastal areas, the latter probably because of upwelling
(Meier et al., 2021a). In the vertical, the surface layer is warming more than the winter water, which is sandwiched
between the surface layer and the halocline. Hence, the spring and summer thermoclines are getting more intense
(Gröger et al., 2019). Water temperature trends in the deep water of those sub-basins such as Bornholm Basin and
Gotland Basin that are sporadically ventilated by salt water inflows originating from surface water are projected
to be elevated as well (Meier et al., 2021a). Projected changes of the vertical water temperature distribution are
similar than those observed since 1850 (Kniebusch et al., 2019a).

For extreme events, projections suggest, inter alia, more tropical nights over the Baltic Sea, increasing the risk of
record-breaking water temperatures (Meier et al., 2019a).
**3.3.5.2 Salinity and saltwater inflows**
Future changes in salinity will depend on changes in the wind fields over the Baltic Sea region (Lass and Matthäus,
1996), river runoff from the Baltic Sea catchment (Schinke and Matthäus, 1998) and mean sea level rise relative
to the seabed of the sills in the entrance area (Meier et al., 2017; Meier et al., 2021b). A projected increase in river
runoff will tend to decrease salinity, but sea level rise will have the opposing effect of tending to increase salinity,
because the water level above the sills at the Baltic Sea entrance would be higher, increasing the cross-sectional
area of the Danish straits. As a result, saltwater imports from Kattegat would be larger. A 0.5 m higher sea level
relative to the sill bottom at the end of the century would increase estimated Gotland Deep surface salinity by 0.7
g kg$^{-1}$ and bottom salinity by 0.9 g kg$^{-1}$ (Meier et al., 2017; Meier et al., 2021b). Due to the large uncertainty in
projected changes in wind fields over the Baltic Sea region (Section 3.3.1.5), in changes of the freshwater supply
from the catchment (section 3.3.2.1) and in global sea level rise (Section 3.3.5.4), salinity projections show a wide
spread. No robust changes were identified because the two main drivers, river runoff and sea level rise,
approximately compensate each other (Meier et al., 2021b). According to Saraiva et al. (2019b) river runoff would
increase by about 1 to 21% at the end of the century depending on the climate model under both RCP4.5 and
RCP8.5, in the ensemble mean causing a decrease in surface and bottom salinity at Gotland Deep of about 0.6-0.7
g kg$^{-1}$, with a large spread among the ensemble members. Assuming a negligible global sea level rise, the intensity
and frequency of MBIs were projected to slightly increase due to changes in the wind fields (Schimanke et al.,
2014). Hence, in ensemble studies that considered all potential drivers, no significant change in salinity were
projected as the ensemble mean (Meier et al., 2021b). In case of salinity, global climate model uncertainty was
identified to be the largest of all uncertainties (Meier et al., 2021b).
**3.3.5.3 Stratification and overturning circulation**
Model based estimations of future stratification are still rare and depend critically on how well the models project
changes in the three-dimensional distributions of temperature and salinity. A first systematic attempt using a high
resolution coupled ocean - atmosphere model and five different global climate models (Gröger et al., 2019)





explored future stratification under RCP8.5. They assumed a 10% increase in river runoff (approximately the
ensemble mean in Saraiva et al., 2019b) and an unchanged mean sea level in the North Sea at the end of the
century. The ensemble consistently indicated a basin-wide intensification of the pycnocline (by 9–35%) for nearly
the whole Baltic Sea, and a shallowing of the pycnocline depth in most regions, except the Gulf of Bothnia (Gröger
et al., 2019). The area with a pycnocline intensity > 0.05 kg m$^{-3}$m$^{-1}$ increased 23-100%. The warm season
thermocline likewise intensified in nearly the entire Baltic Sea (Gröger et al., 2019).

All ensemble members indicate a strengthening of the zonal, wind driven near-surface overturning circulation in
the southwestern Baltic Sea towards the end of the 21$^{st}$ century, whereas the zonal overturning at depth is reduced
by ~ 25% (Gröger et al., 2019). In the Baltic proper, the meridional overturning shows no clear climate change
signal. However, three out of five ensemble members indicate at least a northward expansion of the main
overturning cell. In the Bothnian Sea, all ensemble members show a significant weakening of the meridional
overturning.

As the study by Gröger et al. (2019) and previous projections (e.g. Meier et al., 2006) do not consider global sea
level rise, these scenario simulations are no longer considered plausible (Meier et al., 2021a; 2021b). Considering
all drivers of changes in salinity in the Baltic Sea (wind, river runoff, global sea level rise), neither the haline
induced stratification nor the overturning circulation is projected to change systematically among climate models
(Meier et al., 2021a). It was found that under a RCP4.5 or RCP8.5 scenario a linearly rising mean sea level by the
figures suggested by IPCC (2019b) would approximately counteract the effects of projected river runoff increases
and wind changes on salinity.
**3.3.5.4 Sea level**
Global mean and thus Baltic Sea level will continue to rise at an increasing rate. During this century, melting ice
sheets in Antarctica and Greenland are expected to contribute more to the total sea level than in the past (e.g.
Mitrovica et al., 2018). The fingerprints from melting ice sheets in Antarctica on sea level rise will be more
pronounced in the northern hemisphere and introduce large uncertainties for Baltic sea level rise. Furthermore, the
sea level in shelf seas such as the Baltic Sea will rise more strongly than one would expect from the thermostatic
expansion of the local water column only, due to spill-over effects from the open ocean (Landerer et al., 2007;
Bingham and Hughes, 2012). In addition, the long-term rate of coastal land rise is not easy to estimate accurately,
due to the limited length of Global Positioning System (GPS) measurements, and frequently revised geological
model values.

Estimates for the ensemble mean global sea level rise by 2100 ranged from 43 cm (RCP2.6) to 84 cm (RCP8.5),
with likely ranges of 29-56 cm and 61-110 cm, respectively (IPCC, 2019a), cf. Section 1.5.3.1. In particular for
RCP8.5, sea level rise projections by the fifth IPCC assessment report (IPCC, 2014a) somewhat differ from the
more recent Special Report on the Ocean and Cryosphere in a Changing Climate (IPCC, 2019a) because of the
updated contribution from Antarctica based upon new ice-sheet modeling.

The projected sea level rise relative to land in the Baltic Sea entrance area was estimated to about 80% of the
global rate (Grinsted et al., 2015; Grinsted, 2015). These results were confirmed by other studies (e.g. Kopp et al.,


2014) and summarized by Pellikka et al. (2020) who found a regional ensemble mean absolute sea level rise of
87% of the global sea level rise. Altogether, considering land uplift and eustatic sea level rise, very likely ranges
(5-95% probability) of relative sea level change under the most pessimistic IPCC emissions scenario (RCP8.5)
were projected to, e.g. 29-162 cm in Copenhagen (median 68 cm), -13 -117 cm in Stockholm (median 25 cm), and
21-151cm in St. Petersburg (median 59 cm) (Grinsted et al., 2015). For coastal sites in the northern Baltic Sea,
relative sea level changes in the Gulf of Finland in 2000–2100 were projected to be +29 cm (−22 to+92 cm), −5
cm (−66 to+65 cm) for the Bothnian Sea, and −27 cm (−72 to +28 cm) for the Bothnian Bay, where the land uplift
is larger (Johansson et al., 2014). The ranges in the latter study were estimated from the 5% and 95% cumulative
probabilities considering several published scenarios from the third and fourth IPCC assessment reports. In a recent
study based upon IPCC (2019a), Hieronymus and Kalén (2020) also estimated a sea-level fall in the northern
Baltic Sea and a 70 cm rise in the south by 2100. These upper bounds of the sea level rise projections imply a very
strong future acceleration of present rates. Current observations seems to show an acceleration, but its present
magnitude is still small (Hünicke and Zorita, 2016; see Section 3.2.5.4).

Recent efforts since the IPCC AR5 report (IPCC, 2014a) that focused on the contribution of Antarctic ice sheets
to global mean sea level rise have shown that the interaction of warming ocean water, melting the ice sheets from
below can lead to instabilities in the ice sheet dynamics. The ice sheets flowing from land into the ocean are in
contact with the ocean floor out to the grounding line. From there on outward the ocean is melting the ice from
below and the ice sheets become thinner and lighter. If the weight of the ice sheet becomes less than the weight of
the ocean water it replaces, it floats up and away. The grounding line retreats inland where the ice sheet is thicker
and the ice flow larger and reinforces the ice loss (Mercer, 1978). This and related feedback loops could lead to an
extra meter of sea level rise until the end of the century (e.g. Sweet et al., 2017). The most recent estimates based
on expert judgement (Bamber et al., 2019) for global mean sea level rise in 2100 relative to 2000, including these
potential contributions (including land water storage) are 69 cm and 111 cm for low and high sea level scenarios,
respectively. For the high sea level scenario the likely range (5 to 95%) is between 62 cm and 238 cm.

Future changes in sea level extremes in the Baltic Sea depend on future changes in mean sea level and future
developments in large-scale atmospheric conditions associated with changing wind patterns. Model projections
disagree regarding atmospheric circulation changes and therefore their relevance for extreme future sea levels
remains unclear (Räisänen, 2017). Absolute mean sea levels will continue to rise in the entire Baltic Sea, but exact
rates remain uncertain and depend on models and greenhouse gas emission scenarios (Grinsted, 2015; Hieronymus
and Kalén, 2020). Relative sea level changes will strongly vary across the Baltic Sea because of the existing spatial
gradient in glacial isostatic adjustment and the spatial inhomogeneity associated with the uncertain relative
contributions of melting from Antarctica and Greenland (e.g. Hieronymus and Kalén, 2020). For the Baltic Sea,
changing mean sea levels are expected to have larger effects on future extremes than changing atmospheric
circulation (Gräwe and Burchard, 2012). Sea ice loss in the future will further directly expose the northern Baltic
coastline to stronger storm surges.

Recent projections of extreme sea levels along Europe's coasts have considered all drivers by linear superposition,
i.e. absolute mean sea level rise and land uplift, tides (small in the Baltic Sea), storm surges and waves
(Vousdoukas et al., 2016; Vousdoukas et al., 2017). The results suggest that extreme sea levels will increase more



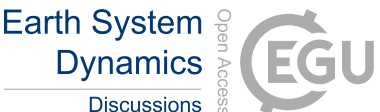

than the mean sea level, due to small changes in the large-scale atmospheric circulation, such as a northward shift
of the Northern Hemisphere storm tracks and westerlies, and increases in the NAO/AO (IPCC, 2014b). These
changes in the large-scale atmospheric circulation of the Baltic Sea region are, however, not robust among GCMs,
giving the projections of extreme sea levels by Vousdoukas et al. (2016; 2017) low confidence.
**3.3.5.5 Waves**
The few existing wave climate projections for the Baltic Sea indicate an increase in the mean wave conditions,
either in the whole area (Groll et al., 2017) or in its northern part (Bonaduce et al., 2019). This increase in the mean
conditions has been linked to two main drivers: 1) increased wind speeds and 2) reduced seasonal ice cover.

Groll et al. (2017) projected wave climate at the end of 21$^{st}$ century, based on two different scenarios. They found
a slight increase in the median wind speeds for most of the Baltic Sea area, which led to an increase of up to 15 %
in median Significant Wave Height (SWH). Using only one climate scenario, Bonaduce et al. (2019) found that
decreased wind speed in the southern Baltic Sea led to a decrease in mean SWH, whereas increased wind speeds
in the north, especially in winter, led to increased mean SWH. As neither study used multi-model ensembles of
scenario simulations (an exception for the western Baltic Sea is the work by Dreier et al. (2021), and there is high
uncertainty in the projected wind speeds and directions, which is not attributed to the decline in ice cover, the
results may not be representative. The projected changes in SWH estimates are therefore inconclusive.

Ruosteenoja et al. (2019) estimated based CMIP5 simulations that in future, mean and extreme scalar wind speeds
are not likely to significantly change in the Baltic Sea area. Hence, mean wave conditions would not change. They
also estimated that frequency of strong westerly winds will increase while strong easterly winds will become less
common. These type of changes might have more significance on the frequency of extreme SWH values and their
spatial patterns.

For extreme values, these studies give even less reliable results. The results of Groll et al. (2017), Suursaar et al.
(2016) and Bonaduce et al. (2019) all indicated large spatial variability in how the projected extremes changed. In
addition to the wind speed, extreme values are quite sensitive to wind direction, since fetch varies with direction
due to the geometry of the Baltic Sea. Mäll et al. (2020) simulated how wave conditions during three historical
Baltic Sea storms would change under climate change conditions. The results showed slight, but not significant
changes in extreme SWH values during the storms.

Future changes in seasonal sea-ice conditions in the northern Baltic Sea are more reliable, and their effect on the
wave climate easier to estimate (Rutgersson et al., 2021). Mild ice winters have already become common, and new
records of lowest annual maximum ice extent have been recorded. In the Baltic Sea, the ice season partly overlaps
with the seasons of the strongest winds, namely autumn and winter. The mean and extreme values of SWH are
therefore expected to increase in areas like the Bothnian Sea, which now typically has ice cover in winter, but will
have loose it in the future Baltic Sea climate.



### 3.3.5.6 Sedimentation and coastal erosion

As a consequence of the probably accelerating sea level rise, coastal erosion will increase regionally, to fill the increased underwater accommodation space. How much erosion will increase will depend not only on the rate of sea level rise, but also on the intensity of storms (Zhang et al., 2017).

Coastal erosion, accretion and alongshore sediment transport are primarily controlled by winds and wind-induced waves in the Baltic Sea. Projecting the future rate of coastal erosion or accretion in the Baltic Sea is highly uncertain because of a lack of consensus in the prediction of future storms. Neglecting potential change in future storms and assuming an intermediate sea level rise scenario (RCP4.5), an increment of 0.1-0.3 m year$^{-1}$ in coastline erosion has been projected for some parts of the southern Baltic Sea coast (Zhang et al., 2017; Deng et al., 2015). Due to the prevailing westerly winds, the dominant sediment transport will continue to be eastwards along most of the southern Baltic Sea coast, but with high variability along coastal sections with a small incidence angle of incoming wind-waves (Dudzińska-Nowak, 2017). It has been found that even a minor climate-change-driven rotation of the predominant wind directions over the Baltic Sea may substantially alter the structural patterns and pathways of wave-driven transport along large sections of the coastline (Viška and Soomere, 2013).

The presence of sea ice is an important factor moderating coastal erosion. Storm surges and wave run-up on the beach are much higher in ice-free periods than when there is even partial ice cover. The hydrodynamic forces are particularly effective in reshaping the shoreline when there is no ice and sediment is mobile (Ryabchuk et al., 2011). Due to global warming, both the area and duration of ice cover in the Baltic Sea will be reduced in future (Section 3.3.4.4), thus increasing coastal erosion.

Foredunes will likely continue to form on prograding coasts, but at rates influenced by the accelerating sea-level rise (Zhang et al., 2017). Foredunes may tend to become higher, but with reduced prograding rate and wavelength, if sea level rise is accelerated or storm frequency increased. If the wind-wave climate is stable, the height of coastal foredunes on a prograding coast remains stable or increases linearly with a low to intermediate rate (<1.5 mm year$^{-1}$) of sea level rise. An accelerating rate of sea level rise and/or changing storm frequency will lead to a nonlinear growth in height (following a quadratic or a higher power law; Zhang et al., 2017; Lampe and Lampe, 2018). The critical threshold that separates linear and non-linear foredune growth in response to sea level rise is likely to be reached before 2050 in the RCP8.5 scenario (Zhang et al., 2017).

Anthropogenic influence imposes further uncertainty in sediment transport and coastal erosion. Sediment transport and coastal erosion are relevant for coastal management, construction and protection strategies. In general two main types of management strategies exist for the Baltic Sea coast: 1) coastal protection by soft or hard measures; and 2) adaptation to coastal change, accepting that in some places the coast would be left in its natural state (BACC II Author Team, 2015). However, administrative efforts for coastal protection differ among Baltic Sea countries, even between neighboring states or nations. It has been found that engineering structures (e.g. piers, seawalls) may influence coastline change at a much larger spatial scale than the dimension of the structure itself.





**3.3.5.7 Marine carbonate system and biogeochemistry**
The BACC II Author Team (2015) concluded that model simulations indicated that climate change has a potential
to intensify eutrophication in the Baltic Sea. However, they also showed that the implementation of nutrient load
reductions according to the Baltic Sea Action Plan (BSAP, HELCOM, 2013a) may not only mitigate this effect
but may slightly decrease hypoxic and anoxic areas in the Baltic Sea, because the nutrient load abatement strategy
of the BSAP did not take the effect of climate change into account. In contrast, the business as usual nutrients
input scenario may increase by about 30% hypoxic area and even more than double the area affected by anoxia by
2100.


As atmospheric $CO_2$ rises, so will the concentration of $CO_2$ in the Baltic Sea surface water. This will influence the
mean future pH, while eutrophication and enhanced organic matter production/remineralization will increase the
amplitude of daily and seasonal pH fluctuations without much affecting the mean values. It was also shown that
pH in the Baltic Sea surface water will decrease, in the worst-case emission scenario (atmospheric $pCO_2$ of 850
ppm) the pH will drop by about 0.40 by 2100, while the decrease in a more optimistic emission scenario (550 ppm)
will be smaller, about 0.26.
**3.3.5.7.1 Oxygen and nutrients**
Projected warming and global mean sea level rise may worsen eutrophication and oxygen depletion in the Baltic
Sea by reducing air-sea fluxes and vertical transports of oxygen in the water column, intensifying internal nutrient
cycling, and increasing river-borne nutrient loads due to increased river runoff (Meier et al., 2011a; Meier et al.,
2012b; Meier et al., 2012c). However, the future response of deep-water oxygen conditions will depend mainly on
nutrient loads from land (Saraiva et al., 2019a, b; Meier et al., 2021b; cf. Fig. 31). In contrast to the global ocean
(see Section 1.5.4), future nutrient supplies will have a relatively larger effect on oxygen conditions and primary
production than warming. With high nutrient loads, the changing climate will have a considerable negative effect,
but if loads are kept low, climate effects can be small or negligible. Scenario simulations suggest that full
implementation of the nutrient load reductions required by BSAP will significantly improve the eutrophication
status of the Baltic Sea, irrespective of the driving global climate model (Saraiva et al., 2019b; Meier et al., 2021b)
and regional coupled climate-environmental model (Meier et al., 2018a). Despite large uncertainties of future
projections, modeling studies suggested that the future Baltic Sea ecosystem may unprecedentedly change
compared to the past 150 years (Meier et al., 2012a).

By the end of the century (2069-2098), the ensemble mean hypoxic area is projected to change only slightly under
reference (-14% for RCP4.5 and -5% for RCP8.5) and high (-2% for RCP4.5 and +5% for RCP8.5) nutrient load
scenarios, compared to 1976-2005 (Saraiva et al., 2019b). Nutrient loads in the reference scenario are the average
loads in 2010-2012. The high, or worst, scenario assumes changes caused by a 'fossil-fuelled development',
coupled to increasing river runoff (Saraiva et al., 2019a). Changes in nitrogen and phosphorus loads were estimated
from assumptions on regional population growth, changes in agricultural practices, such as land and fertilizer use,
and developments in sewage treatment (Zandersen et al., 2019; Pihlainen et al., 2020). Under the BSAP scenario,
the ensemble mean hypoxic area will be reduced by 50-60% at the end of the century, in comparison with 1976-
2005 (Saraiva et al., 2019a). The relative reductions in hypoxic area may decrease with increasing sea level (Meier
et al., 2021b).




In the same model ensemble (Saraiva et al., 2019a), BSAP implementation is projected to reduce the water column
phosphate pool in the Baltic Sea by 59% (RCP4.5) and 56% (RCP8.5) by the end of the century, and even the
reference loads would lead to a decline by 24% (RCP4.5) and 18% (RCP8.5). Also, a larger ensemble (Meier et
al., 2018a) of 8 biogeochemical models forced by outputs of 7 ESMs downscaled by 4 different RCMs projected
that the BSAP reduced phosphate concentrations in the Baltic proper, Gulf of Finland and Bothnian Sea despite
climate change, with largest reductions in surface concentrations by approximately 3 mmol m$^{-3}$ in the Gulf of
Finland. Present day nutrient loads led to small increases in surface phosphate concentration in the Baltic proper,
a small decline in the Gulf of Finland and little change in Bothnian Sea and Bothnian Bay. Little change was
predicted for DIN concentrations in the Baltic proper, whereas simulations showed an increase in the Gulf of
Finland and the Bothnian Sea, regardless whether nutrient loads were kept at present level or whether loads were
reduced.

Furthermore, future projections suggested that the sea-ice decline in the northern Baltic Sea may have considerable
consequences for the marine biogeochemistry, because of changing underwater light conditions and wave climate
(Eilola et al., 2013). Eilola et al. (2013) found that, by the end of the century, the spring bloom would start by up
to one month earlier and winds and wave-induced resuspension would increase, causing an increased transport of
nutrients from the productive coastal zone into the deeper areas.

For the Baltic proper, the internal nutrient cycling and exchanges between shallow and deeper waters were
projected to be intensified, and the internal removal of phosphorus may become weaker in future climate (Eilola
et al., 2012). These effects may counteract the efforts of planned nutrient input reductions.

Uncertainties in projections from Baltic Sea ecosystem models have recently been systematically assessed for the
first time (Meier et al., 2018a; Meier et al., 2019b; Meier and Saraiva, 2020; Meier et al., 2021b). One of the larger
sources of uncertainty is biases in global and regional climate models, in particular concerning global mean sea
level rise and regional water cycling (Meier et al., 2019b). The mechanism behind the correlation between large-
scale meteorological conditions in the different climate periods and oxygen conditions in the Baltic Sea is not well
understood and subject to ongoing research. With respect to nutrient concentrations, also uncertainties in
conditions at the North Sea boundary as well as difficulties in simulating the long-term response of the Baltic Sea
biogeochemical system to changes in nutrient inputs, play a role.

Under the BSAP scenario, mean nitrogen fixation would decrease (Meier et al., 2021b) and record-breaking
cyanobacteria blooms may no longer occur in the future, but record-breaking events may reappear at the end of
the century in a business-as-usual nutrient load scenario (Meier et al., 2019a).
**3.3.5.7.2 Marine CO$_2$ system**
The rising atmospheric pCO$_2$ due to anthropogenic emissions will increase the mean pCO$_2$ of surface seawater and
thus has the potential to lower the pH. However, the magnitude of pH changes will also depend on the development
of total alkalinity concentrations (A$_T$; Omstedt et al., 2012). Future A$_T$ changes in the Baltic Sea will be shaped by
both external inputs (riverine runoff and inflows from the North Sea) and internal generation. The latter is due to



biogeochemical processes of organic matter production and remineralization, especially under euxinic conditions.
Kuznetsov and Neumann (2013), who used $A_T$ as a tracer in a model (no internal processes included) showed that
on average $A_T$ in surface Baltic Sea waters should decrease by about 150 µmol kg$^{-1}$ by 2100, a change
corresponding to an assumed decrease in salinity. Simulations by Gustafsson et al. (2019) that include most of the
biogeochemical processes affecting $A_T$ (except S burial and Fe-oxide availability) showed that $A_T$ in the central
Baltic Sea in the "business as usual" scenario will first increase, by about 100 µmol kg$^{-1}$ by 2050, and then revert
to present levels by 2100. If BSAP is implemented, $A_T$ will decrease by about 150 µmol kg$^{-1}$ in 2100 from present
levels. Irrespective of the nutrient load scenario, pH is eventually expected to decrease in the central Baltic Sea
due to anthropogenic $CO_2$ emissions. Assuming the A1B $CO_2$ emission scenario ($pCO_2$ increase to 700 µatm by
2100), pH will drop to about 7.9 and 7.8 under "business as usual" and BSAP scenarios, respectively.
**3.3.6 Marine biosphere**
**3.3.6.1 Pelagic habitats**
**3.3.6.1.1 Microbial communities**
The effects of climate change on microbes and the functioning of the microbial loop have been studied by
experiments in which temperature, salinity, dissolved organic matter (DOM), and ocean acidification (OA) were
manipulated. In general, microbial activity and biomass increased with increasing DOM and temperature
(Ducklow et al., 2009), but effects can be mixed. For instance, an increase in DOM in the northern Gulf of Bothnia
enhanced the abundance of bacteria, whereas a temperature increase (from 12 to 15°C) decreased their abundance,
probably due to a simultaneous increase of bacterivorous flagellates (Nydahl et al., 2013).

In the southern Baltic Sea the impact of OA was limited, and the bacterial community responded primarily to
temperature and phytoplankton succession (Bergen et al., 2016). In experiments where $CO_2$ was increased and
salinity decreased (from 6 to 3), heterotrophic bacteria declined (Wulff et al., 2018). In experiments with increasing
temperature (from 16 to 18-20°C) and reduced salinity (from 6.9 to 5.9 g kg$^{-1}$), the Baltic proper microbial
community also showed mixed responses, probably due to indirect food web effects (Berner et al., 2018).
**3.3.6.1.2 Phytoplankton and cyanobacteria**
The projected increase in precipitation is expected to increase nutrient loads, especially into the northern Baltic
Sea (Huttunen et al., 2015), and together with increased internal loading of nutrients, several modelling studies
project an increased phytoplankton biomass by the end of the century (Meier et al., 2012b; Meier et al., 2012c;
Skogen et al., 2014; Ryabchenko et al., 2016).

Several mesocosm studies have investigated the effects of warming on southern Baltic Sea phytoplankton
communities. Warming accelerated the phytoplankton spring bloom and increased primary productivity (Sommer
and Lewandowska, 2011; Lewandowska et al., 2012; Paul et al., 2015). The total phytoplankton biomass still
decreased, due to negative effects of warming on nutrient flux (Lewandowska et al., 2012; Lewandowska et al.,

2014).




Ocean acidification (OA) may enhance phytoplankton productivity by increasing the $CO_2$ concentration in the water. The biomass of southern Baltic Sea autumn phytoplankton increased in mesocosms simulating OA (Sommer et al., 2015). In many experiments OA had, however, little effect on phytoplankton community composition, fatty acid composition or biovolume in spring or autumn (Paul et al., 2015; Bermúdez et al., 2016; Olofsson et al., 2019).

It has been suggested that climate change may increase the blooming of toxic species, such as the dinoflagellate *Alexandrium ostenfeldii* (Kremp et al., 2012; Kremp et al., 2016) and the cyanobacterium *Dolichospermum* sp. (Brutemark et al., 2015; Wulff et al., 2018). There are also contradictory results, indicating that OA and warming may decrease the biomass of *Nodularia* sp. and *Dolichospermum* sp. (Eichner et al., 2014; Berner et al., 2018). Several modelling studies project increases in cyanobacteria in the warmer and more stratified future Baltic Sea (Meier et al., 2011b; Andersson et al., 2015; Neumann et al., 2012; Chust et al., 2014; Hense et al., 2013), but other modelling studies project that the environmental state of the Baltic Sea will be significantly improved, and extreme cyanobacteria blooms will no longer occur if BSAP is fully implemented (Meier et al., 2018a; Meier et al., 2019a; Saraiva et al., 2019a; see Figure 32).

### 3.3.6.1.3 Zooplankton

The effects of increasing temperature and ocean acidification (OA) on zooplankton have been studied experimentally. In *Acartia* sp., a dominant copepod in the northern Baltic Sea, warming decreased egg viability, nauplii development and adult survival, and both warming and OA had negative effects on adult female size (Garzke et al., 2015; Vehmaa et al., 2016; Vehmaa et al., 2013).

In contrast, the effects of climate change on microzooplankton (MZP) seem to be mostly beneficial. Warming improved the growth rate of southern Baltic Sea MZP, which led to a reduced time-lag between phytoplankton and MZP maxima, improving the food supply to microzooplankton in warm conditions (Horn et al., 2015). (Aberle et al., 2015) showed that while protozooplankton escaped predation by slower growing copepods at low temperatures, at warmer temperatures small ciliates in particular became more strongly controlled by copepod predation.

### 3.3.6.2 Benthic habitats

### 3.3.6.2.1 Macroalgae and vascular plants

The effects of climate change on bladder wrack, *Fucus vesiculosus,* have been studied in a number of experiments. Ocean acidification (OA) appears to have a relatively small effect on macroalgae (Al-Janabi et al., 2016; Wahl et al., 2020), while temperature effects can be significant. The effects of increasing temperature are not linear, however. Growth or photosynthesis is not impaired under projected temperature increase (from 15 to 17.5°C) but at extreme temperatures (27 to 29°C), photosynthesis declines, growth ceases and necrosis starts (Graiff et al., 2015; Takolander et al., 2017). In very low salinity (2.5 g kg$^{-1}$), sexual reproduction of *F. vesiculosus* ceases (Rothäusler et al., 2018).

The direct and indirect effects of changes in temperature, salinity and pH may alter the geographic distribution of many species in the Baltic Sea. Retreat of marine species has been predicted for bladder wrack, eelgrass and blue



mussel, and up to 50 other species affiliated to these keystone species (Vuorinen et al., 2015). Species distribution
modelling has indicated that a decrease of bladder wrack will have large effects on the biodiversity and functioning
of the shallow-water communities of the northern Baltic Sea (Jonsson et al., 2018; Kotta et al., 2019). The
responses of eelgrass, *Zostera marina,* to climate change and eutrophication mitigation have recently been modeled
by Bobsien et al. (2021).

Experiments on climate change effects have been made also with other macroalgae and vascular plants. Thus, OA
increased the growth of the opportunistic green alga *Ulva intestinalis* in the Gulf of Riga (Pajusalu et al., 2013;
Pajusalu et al., 2016). Other studies showed that charophyte photosynthesis increased under high pCO2, whereas
eelgrass did not respond to the elevated $pCO_2$ alone (Pajusalu et al., 2015). Salinity decline is projected to decrease
the distribution of *Z. marina* and the red alga *Furcellaria lumbricalis*, whereas warming will probably favour
charophytes (Torn et al., 2020).
**3.3.6.2.2 Zoobenthos**
The effects of warming on invertebrates are non-linear. Respiration and growth of the isopod *Idotea balthica*
increased up to 20°C, and then decreased at 25°C (Ito et al., 2019). Many marine invertebrates, including isopods,
will also directly and indirectly suffer from decreasing salinity (Kotta et al., 2019; Rugiu et al., 2017), as well as
ocean acidification (OA). The size and time to settlement of the pelagic larvae of the Baltic clam *Limecola balthica*
(syn *Macoma balthica*) increased with OA, suggesting a developmental delay (Jansson et al., 2016), whereas OA
had no effect on the isopod *Saduria entomon* (Jakubowska et al., 2013) or larvae of the barnacle *Amphibalanus*
*improvisus* (Pansch et al., 2012).

Several modelling studies have estimated the relative effects of hydrodynamics, oxygen and food availability on
Baltic Sea zoobenthos. In previously hypoxic areas, benthic biomass was projected to increase (until 2100) by up
to 200% after re-oxygenating bottom waters, whereas in permanently oxygenated areas macrofauna may decrease
by 35% due to lowered food supply to the benthic ecosystem (Timmermann et al., 2012). It has, however, been
concluded that nutrient reductions will be a stronger driver for Baltic Sea ecosystem than climate change (Friedland
et al., 2012; Niiranen et al., 2013; Ehrnsten et al., 2019). These studies suggest that benthic-pelagic coupling will
weaken in a warmer and less eutrophic Baltic Sea, resulting in gradually decreasing benthic biomass (Ehrnsten et
al., 2020).
**3.3.6.3 Non-indigenous species**
It is often suggested that climate change will favour invasions by non-indigenous species worldwide (Jones and
Cheung, 2014). It has been shown that non-native benthic species typically occur in areas with reduced salinity,
high temperatures, high proportion of soft seabed and low wave exposure, whereas most native species show an
opposite pattern (Jänes et al., 2017). Modelled temperature and salinity scenarios suggests an increase of Ponto-
Caspian cladocerans in the pelagic community, and an increase in dreissenid bivalves, amphipods and mysids in
the benthos of coastal areas in the northern Baltic Sea by 2100 (Holopainen et al., 2016). Disentangling factors
facilitating establishment of non-native species demands long-term surveys, and data from multiple environments
in order to distinguish climate-related effects from other ecosystem-level drivers (Bailey et al., 2020). In addition,
studies on changing connectivity are needed (e.g. Jonsson et al., 2020).





**3.3.6.4 Fish**

Climate change may affect Baltic Sea fish through effects on water temperature, salinity, oxygen and pH, as well as nutrient loads, which indirectly affect food availability for fish. The responses of cod larvae to ocean acidification and warming have been studied experimentally. Some studies found no effect on hatching, survival or development rates of cod larvae (Frommel et al., 2013), while in others mortality of cod larvae doubled when exposed to high-end OA projections (RCP8.5). Several modelling studies however project low abundances of cod towards the end of the century, due to continued poor oxygen conditions (Niiranen et al., 2013; Wåhlström et al., 2020).

Climate change may also be positive for fish stocks. Warmer spring and summer temperatures have been projected to increase productivity of sprat (Voss et al., 2011; MacKenzie et al., 2012; Niiranen et al., 2013). For herring, results are more varied: both increase (Bartolino et al., 2014) and a short-term decrease (Niiranen et al., 2013) have been projected.

Multi-species modelling has also emphasized the role of climate for cod stocks. If fishing is intense but climate remains unchanged, cod declines, but not very dramatically, while if climate change proceeds as projected, cod disappeared in two models out of seven, even with the current low fishing effort (Gårdmark et al., 2013). Different scenarios yield very different outcomes, however. A medium $CO_2$ concentration scenario (RCP4.5), low nutrients and sustainable fisheries resulted in high numbers of cod and flounder, while high emissions (RCP8.5) and high nutrient loads resulted in high abundance of sprat (Bauer et al., 2018; Bauer et al., 2019). All these studies assumed a more or less pronounced decrease in salinity.

**3.3.6.5 Marine mammals**

*Ringed seal and grey seal – sea ice*

Climate change is projected to drastically reduce the extent of seasonal sea ice in the Baltic Sea (Luomaranta et al., 2014; Meier, 2006; Meier, 2015; Meier et al., 2004b). At the end of the 21st century, ice will probably in most years be confined to the Bothnian Bay, the eastern Gulf of Finland, the Archipelago sea, and the Moonsund (Sound between the Estonian mainland and the offshore western islands Saaremaa, Hiiumaa, Muhu and Vormsi) and eastern parts of the Gulf of Riga such as Pärnu Bay (Meier et al., 2004b), with corresponding changes in the breeding and moulting distribution of ringed seals. Aside from these projections, ice cover has been even more limited in all the southern areas in recent years. Extirpation of one or more of the three southern breeding ringed seal populations is possible (Sundqvist et al., 2012; Meier et al., 2004b).

The ringed seal is an obligatory ice breeder that digs lairs in the snowdrifts on offshore ice for protection of the pup (e.g. Smith and Stirling, 1975). The Baltic grey seal prefers loose floes of drift ice (Hook et al., 1972), but can also breed on land (Jüssi et al., 2008). Overall pup survival in land breeding grey seals is probably lower than for ice breeders (Jüssi et al., 2008). Absence or low quality of sea ice will adversely affect pup survival and quality in ice-breeding seals. The effects can be seen by the end of the breeding season, and beyond (Jüssi et al., 2008). Grey and ringed seals are capital breeders, i.e., their pup quality depends on effective transfer of maternal energy (fatty milk) during a short, intensive lactation period. Timing of birth for both species is strongly adapted to the availability of the optimal breeding platform, sea ice. The height of the pupping season is around February-early



March, when the extent and strength of the sea-ice is usually greatest. The immediate breeding success can be defined as survival and quality of the offspring at the end of the breeding season, but breeding conditions may have population consequences by affecting the survival and fitness of the pups throughout their lives (McNamara and Houston, 1996; Kauhala and Kurkilahti, 2020). A warming climate with higher air and water temperatures will decrease the extent of ice-cover, the ice thickness and the overlaying snow-cover as well as the stability and duration of the ice.

Loss of habitat is critical for reproductive success of the ice-associated seals, especially the ringed seal, and can eventually lead to local population decreases and changes in breeding distribution, starting in the southernmost parts of its range. The ringed seal populations breeding in the Gulf of Finland, Gulf of Riga and Archipelago Sea (SW-Finland) are already small and vulnerable to any negative changes in habitat quality.

*Harbour seal and grey seal – flooding of haul-outs*

Harbour seals and grey seals rely on undisturbed haul-out areas for key life cycle events such as breeding, moulting and resting (Allen et al., 1984; Thompson, 1989; Watts, 1996; Reder et al., 2003). In the southern Baltic Sea, relative sea levels have risen by 1 to 3 mm per year over the interval 1970-2016 (section 3.2.5.4), and increased rates of sea level rise are expected in the future (EEA, 2019c; Grinsted, 2015). A low emissions scenario for the 21st century projects an additional sea-level rise of 0.29-0.59 m, a high emissions scenario an extra 0.61-1.10 m, but substantially higher values cannot be ruled out (Grinsted, 2015; IPCC, 2019a). A high emission scenario is thus likely to flood all current seal haul-outs in the southern Baltic Sea and many important localities in Kattegat, while under a low emission scenario, most haul-outs in the southern Baltic Sea will be flooded, while others will be reduced to small fractions of their current area. In the northern and central Baltic Sea and eastern Kattegat archipelago areas, seals will have alternative islets and skerries and are not likely to be affected to the same degree as in the south and in eastern Kattegat. In parts of the Gulf of Bothnia, relative sealevels may even fall, due to post-glacial rebound (EEA, 2019c).

*Harbour porpoise*

There are no direct studies of the effects of climate change on harbour porpoises in the Baltic Sea, hence the following is based mostly on informed guesswork and on a few studies in other areas. There are a multitude of ways that changes in one parameter can affect others and we do not currently have the knowledge to predict the cumulative effects this might have on the Baltic Sea harbour porpoise population.

Harbour porpoises are present from Greenland to the African coast and the Black Sea and seem to have a rather wide thermal tolerance. Therefore, even though it is predicted that we will see a 1.2-3.2° increase in SST in the Baltic Sea (Section 3.3.5.1), it seems unlikely that this will directly affect harbour porpoise distribution, unless the Baltic Sea harbour porpoise is specifically adapted to colder temperatures. If this is the case, a northwards range shift might occur. With the expected future decrease in sea ice extent, the winter habitat available for the harbour porpoise in the northern Baltic Sea would increase.

Harbour porpoises are small cetaceans with limited capacity to store energy that mostly live in cold environments. Hence, they need to eat almost constantly (Read and Hohn, 1995; Wisniewska et al., 2016) and are therefore





expected to be tightly dependent on their prey (Sveegaard et al., 2012). Their main prey species in the Baltic proper
are cod (at least before the recent cod stock collapse), sprat, herring, gobies and sand eel (where present). Climate-
induced changes in for example SST, fronts, stratification and to some degree currents will affect the distribution,
abundance and possibly the quality of prey species, and in turn the harbour porpoise population. Their distribution
may shift as they follow their prey, and potential food shortages might lead to starvation, with possible population
effects.

It has been hypothesized that the susceptibility of marine mammals to disease may increase as temperature
increases. Higher temperatures can increase pathogen development and survival rates, facilitate transmission
among individuals and increase individual susceptibility to disease. The negative effects of disease as well as
environmental contaminants on individual fitness will obviously worsen if the animal is also under nutritional
stress.

*Seals and changes in the distribution of prey species*
Any large alteration of the ecosystem can affect the distribution of seals if there are climate-related changes in the
abundance and distribution of their main prey species, such as herring, sprat and cod, as is possible with climate
change. Such changes in top consumer distribution have been modeled in other sea areas, such as the UK
continental shelf, where the current distribution of harbour seals did not match well the projected future distribution
of their prey (Sadykova et al. 2020). There are large differences in salinity between Baltic Sea models (Saraiva et
al., 2019b), and other factors such as temperature, eutrophication, predation and competition also affect fish
distributions. Thus, future changes in abundance and distribution of seal prey species, such as herring and cod, are
hard to predict (Dippner et al., 2008; Lindegren et al., 2010; Vuorinen et al., 2015; Dippner et al., 2019).
**3.3.6.6 Waterbirds**
Climate change scenarios agree in projecting a strong temperature increase in the Arctic and sub-Arctic. This will
likely cause a northward expansion of species ranges, with colonization by new breeding and wintering species,
as well as local species declines following migration of populations to ice-free northern waters (Pavón-Jordán et
al., 2019; Fox et al., 2019).

If salinity in the Baltic Sea decreases, invertebrate species serving as prey for waterbirds (e.g. blue mussels for
common eiders) are likely to change in distribution, body size and quality as food, with consequences for the
distribution, reproduction and survival of the waterbirds that eat them (Fox et al., 2015). Predicting the
consequences of climate change for piscivorous seabirds is complex, because effects are not uniform among Baltic
Sea fish species. For example, expected increase of recruitment and abundance in an important prey species (sprat;
MacKenzie et al., 2012; Lindegren et al., 2012) as well as declining numbers of large piscivorous fish (cod) may
favour fish-eating birds, although management efforts to improve cod stocks may counteract the expected increase
in sprat and lead to population declines of their main bird predator, the common guillemot (Kadin et al., 2019).
Herring, another important prey species, is reported to be negatively affected by decreasing salinity (declining
energy content; Rajasilta et al., 2018).



A rising sea level will reduce the area of saltmarshes available for the breeding of waders and foraging by geese (Clausen et al., 2013), and other coastal habitats would likewise be affected (Clausen and Clausen, 2014). Sea level rise in combination with storms may cause loss by erosion of current coastal breeding habitats, and flood breeding sites, thus affecting the breeding success of coastal waterbirds. Climate change can also be expected to affect waterbirds in the Baltic Sea by changing the incidence of diseases and parasites (Fox et al., 2015).

**3.3.6.7 Marine food webs**

Climate change and other anthropogenic environmental drivers are expected to change entire marine food webs, from coastal to off-shore, from shallow to deep, from pelagic to benthic (sedimentary), as species-distributions are impacted, and key nodes and linkages in the food webs are altered or lost (Lindegren et al., 2010; Niiranen et al., 2013; Leidenberger et al., 2015; Griffiths et al., 2017; Kotta et al., 2019; Gårdmark and Huss, 2020). These climate-driven changes will also, when combined with societal changes, affect aquatic ecosystem services, for instance future primary production (a supportive ecosystem service) and fish catches (a provisioning ecosystem service; Hyytiäinen et al., 2021).

**4    Interactions of climate with other anthropogenic drivers**

The term "driver" in this section is defined as something affecting or being affected by another force. In this respect, climate is a force affecting other drivers, e.g. land use or shipping. On the other hand, (regional) climate may be affected by other drivers, e.g. land use or shipping. This section summarizes plausible two-way dependencies that have been described in the literature. For a deeper analysis, see Reckermann et al. (2021).

Climate change affects air and water temperature as well as precipitation, with a clear impact on land use and land cover. Growth conditions are affected by these changes, but also by political or management decisions, which may in turn be influenced by climate change (Yli-Pelkonen, 2008). Agriculture is the most important land use in the southern part of the Baltic Sea basin. Climate change strongly influences the choice of crops, as crops differ in their requirements for water availability and soil type (Fronzek and Carter, 2007; Smith et al., 2008). Still, socio-economic considerations may be even more important than climate in determining agricultural land use (Rounsevell et al., 2005; Pihlainen et al., 2020).

Land use and cover can influence the regional climate, through geophysical (albedo) and biogeochemical ($CO_2$ drawdown) effects. Bright surfaces like agricultural fields reflect more energy than dark surfaces, like forests and open waters. Thus, the type of land cover may affect regional warming, but its relative contribution is disputed (Gaillard et al., 2015; Strandberg and Kjellström, 2019). Increasing droughts with lower river flow at certain times of the year may influence water management and shipping in regulated rivers, especially in the southern catchment basins. On the other hand, extreme rain events may lead to inundations (Kundzewicz et al., 2005).

Climate change will strongly affect coastal structures through sea level rise and intensified coastal erosion. Storm surges, which run up higher as sea level rises, as well as changed currents and sediment relocations will endanger levees, groynes and other coastal structures, and have to be handled by coastal management (Le Cozannet et al., 2017; Łabuz, 2015).






We can expect a considerable increase in offshore wind energy production worldwide, in order to counteract
climate warming. Although projections of future winds are uncertain, the number of off-shore wind farms can be
expected to increase due to the politically driven shift to renewable energies, and the limited space and low
acceptance for wind mills on land. Offshore wind farms may in turn affect the regional climate by absorbing
atmospheric energy on the regional scale (Akhtar et al., 2021), but the magnitude of this effect is uncertain
(Lundquist et al., 2019).

Shipping is affected by climate change. Perils at sea for ships are all climate sensitive, ranging from storms, waves,
currents, ice conditions, visibility to sea level affecting navigational fairways. Winter navigation will be facilitated
as drastically decreasing winter sea-ice cover is projected, but search and rescue missions in winter may increase
because engine power may in the future be adapted to the lower expected ice cover. Further aspects are a potential
increase in leisure boating, a potentially temperature-dependent functioning of antifouling paints, and different
noise propagation through warmer water. The efficiency of $SO_x$ scrubbing depends on the temperature, salinity
and pH of the seawater, and eventually ends up contaminating the Baltic Sea (Turner et al., 2018). Shipping itself
affects climate through combusting fossil fuels.

Coastal processes, e.g. erosion and the translocation of sediments through erosion, currents and accretion, are
affected by climate change though sea level rise and changes in storm frequency, severity and tracks (Defeo et al.,
2009).


Climate change affects the amount of nutrients entering the sea in precipitation and by land runoff, which in turn
is affected by precipitation, air temperature and runoff pattern, e.g. Arheimer et al. (2012) and Bartosova et al.
(2019). How fertilization practices, crops grown, and land use will change in response to climate change is largely
unknown. Climate-related changes in the Baltic Sea, like warmer temperatures, changed stratification and altered
ecosystems and biogeochemical pathways may change the fate of nutrients in the sea, e.g. Kuliński et al. (2021).

There is not much evidence of a direct climate influence on the quantity and quality of submarine groundwater
discharge, but considering the driving forces (topography-driven flow, wave set-up, precipitation, sea level rise
and convection), an effect is highly plausible, but its magnitude and relevance is unknown (e.g. Taniguchi et al.,
2019).


Fisheries are strongly affected by climate change through its effect on the resources, i.e. the commercially
interesting fish populations in the Baltic Sea, mostly cod, sprat and herring (Möllmann, 2019). Climate affects
salinity and temperature in the Baltic Sea, thereby influencing the productivity of several fish species (MacKenzie
and Schiedek, 2007; Köster et al., 2016), and the resources that fisheries exploit. Growth of planktivorous species
or life stages is also affected by climatic conditions that regulate zooplankton dynamics (Casini et al., 2011; Köster
et al., 2016).

Climatic change is a plausible driver for the migration and occurrence of non-indigenous species, although there
is little direct evidence. Shipping has been identified as a major vector for the introduction of new marine species

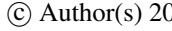



into the Baltic Sea ecosystem, through ballast water or attachment to hulls or elimination of physical barriers (e.g.
though the construction of canals between water bodies; Ojaveer et al., 2017). A northward migration of terrestrial
(Smith et al., 2008) and marine species, including fish, is documented and expected to continue (MacKenzie and
Schiedek, 2007; Holopainen et al., 2016).

Climate change affects contaminants in the Baltic Sea through an array of processes, like partitioning between
environmental phase-pairs such as air-water, air-aerosols, air-soil, air-vegetation, leading to a different distribution
between environmental compartments (Macdonald et al., 2003). Atmospheric transport and air-water exchange
can be influenced by changes in wind fields and wind speeds (Lamon et al., 2009; Kong et al., 2014). Changing
precipitation patterns influence chemical transport via atmospheric deposition (rain dissolution and scavenging of
particles, (Armitage et al., 2011) and runoff, transporting terrestrial organic carbon (Ripszam et al., 2015). As ice-
cover of lakes and the sea decreases, more organic contaminants may volatilize to the atmosphere (Macdonald et
al., 2003; Undeman et al., 2015).

Dumped military ammunition threaten the Baltic Sea in the future, as poisonous substances are expected to leak
due to advanced corrosion of hulls and containers. This process may be affected by climate, as corrosion rates
depend on temperature and oxygen, so that warming and good ventilation of dumping sites can be expected to
enhance corrosion rates (Vanninen et al., 2020). This is an urgent problem since the location of the dumped military
material is only partially known.

There is no evident direct impact of climate change on marine litter or microplastics, but there may be a connection
via increased temperature- and photolysis-dependent rates of degradation and dissolution of microplastics, and on
distribution by currents.
**5    Comparison with the North Sea region**
**5.1 The North Sea region**
Like the Baltic Sea basin, the North Sea region is both a precious natural environment and a place for settlement
and commerce for millions of people, with a rich cultural heritage. The North Sea is one of the world's richest
fishing grounds as well as one of the busiest seas with respect to shipping and infrastructure for oil and gas
extraction, and of enormous economic value. In recent years the area has also become a major site for wind energy,
with many large offshore wind farms.

As climate change is expected to have profound effects on North Sea ecosystems and economic development, an
independent, voluntary, international team of scientists from across the region compiled the North Sea Region
Climate Change Assessment (NOSCCA; Quante and Colijn, 2016). The NOSCCA approach is similar to BACC
in format and intention. The assessment provides a comprehensive overview of all aspects of a changing climate,
discussing a wide range of topics including past, current and future climate change, and climate-related changes
in marine, terrestrial and freshwater ecosystems. It also explores the impact of climate change on some socio-
economic sectors, such as fisheries, agriculture, coastal zone management, coastal protection, urban climate,
recreation/tourism, offshore activities/energy, and air pollution.






The North Sea is a semi-enclosed marginal sea of the North Atlantic Ocean, situated on the north-west European
shelf. It opens widely into the Atlantic Ocean at its northern boundary, with a smaller connection to the Atlantic
Ocean via the Dover Strait and English Channel in the south-west. To the east it connects to the Baltic Sea. The
Kattegat, a transition zone between the North and Baltic seas, is located between the Skagerrak and the Danish
straits. Comprehensive reviews of North Sea physical oceanography are provided by Otto et al. (1990), Rodhe
(1998) and Sündermann and Pohlmann (2011). Physical-chemical-biological interaction processes within the
North Sea are reviewed by Rodhe et al. (2006) and Emeis et al. (2015), and a description of the North Sea marine
ecosystem was compiled by McGlade (2002).

Among the most striking differences between the North and Baltic seas is the wide, direct opening of the North
Sea to the North Eastern Atlantic, allowing free exchange of matter, heat and momentum between the two seas.
As a result the North Sea water has a much higher salinity than the Baltic Sea. The North Sea dynamics are greatly
influenced by tides, while Baltic Sea tides are much weaker than in the North Sea, where tidal amplitudes vary
spatially from a few decimeters to several meters. In addition to the wind-driven circulation, which dominates the
mean cyclonic current system, North Sea tidal currents show non-vanishing residual currents (due to nonlinear
processes), which cannot be neglected. Tidal currents cause strong mixing. Low pressure systems often travel from
the Atlantic with minimum blockage and cause strong storm surges, which are the greatest potential natural hazards
for coastal communities in the North Sea region.

Only selected examples from NOSCCA will be presented here. In general, the North Sea region already
experiences a changing climate and projections indicate that further, partly accelerating, changes are to be expected
(warming of air and water, changing precipitation intensities and patterns, sea level rise, seawater acidification).
Changes in ecosystems (marine, coastal, terrestrial) are observed, and are projected to strengthen, with degree
depending on scenario. Observational as well as modelling studies have revealed a large natural variability in the
North Sea region (from annual to multidecadal time scales), making it difficult to identify regional climate change
signals and impacts for some parameters. Projecting regional climate change and impacts for the North Sea region
is currently limited by the small number of regional coupled model runs available and the lack of consistent
downscaling approaches, both for marine and terrestrial impacts. The wide spread in results from multi-model
ensembles indicates the present uncertainty in the amplitude and spatial pattern of the projected changes in sea
level, temperature, salinity and primary production. For moderate climate change, anthropogenic drivers such as
changes in land use, agricultural practice, river flow management or pollutant emissions often seem more
important for impacts on ecosystems than climate change.
**5.2 A few selected and highly aggregated results from NOSCCA**
*Atmosphere:* Observations reveal that the near-surface atmospheric temperature has increased everywhere in the
North Sea region, especially in spring and in the north. The rise was faster over land than over the sea. Linear
trends in the annual mean land temperature are about +0.39°C per decade for the period 1980–2010. Generally,
more warm extremes and fewer cold ones were observed. A north-eastward shift in storm tracks was observed, in
agreement with projections from climate models forced by increased greenhouse gas concentrations. Overall,
precipitation has increased in the northern North Sea region and decreased in the south, summers have become


warmer and drier and winters have become wetter. Heavy precipitation events have become more extreme. A
marked further mean warming of 1.7–3.2°C is projected for the end of the 21st century (2071–2100, with respect
to 1971–2000) for different scenarios (RCP4.5 and RCP8.5, respectively), with stronger warming in winter than
in summer and particularly strong warming over southern Norway.

*North Sea*: There is strong evidence of surface warming in the North Sea, especially since the 1980s (Fig. 33).
Warming is greatest in the south-east, exceeding 1°C since the end of the 19th century. Absolute mean sea level
in the North Sea rose by about 1.6 mm/year over the past 100–120 years, in agreement with the global rise. The
North Sea is a sink for atmospheric carbon dioxide ($CO_2$); this uptake declined over the last decade, due to lower
pH and warmer water. Models consistently project the surface water to warm further by the end of the century (by
about 1–3°C; A1B scenario). Exact numbers are not given due to differences in spatial averaging and reference
periods from published studies. Coherent findings from published climate change studies include an overall rise
in sea level, an increase in ocean acidification and a decrease in primary production. Uncertainties are large for
projected changes in extreme sea level and waves as well as for decreases in net primary production, which range
from 1 to 36 %.

*Rivers:* To date, no significant trends in response to climate change are apparent for most individual rivers
discharging into the North Sea. Nevertheless, climate models project increased socio-economically important risks
for the region, due to more intense hydrological extremes in the North Sea region, such as flooding along rivers,
droughts and water scarcity. The exposure and vulnerability of cities in the North Sea region to changes in extreme
hydrometeorological and hydrological conditions are expected to increase, due to greater urban land use and rising
urban populations.

*Ecosystems:* Long-term knowledge from exploitation of the North Sea indicates that climate affects marine biota
in complex ways. Climate change influences the distribution of all taxa, but other factors (fishing, biological
interactions) are also important. The distribution and abundance of many species have changed. Warm-water
species have become more common and species richness has increased. Among coastal ecosystems, estuaries and
most mainland marshes will survive sea-level rise, while back-barrier salt marshes with lower suspended sediment
concentrations and tidal ranges are probably more vulnerable. Plant and animal communities can suffer habitat
loss in dunes and salt marshes through high wave energy, and are affected by changes in temperature and
precipitation and by atmospheric deposition of nitrogen. Lakes in the North Sea region have experienced a range
of physical, chemical and biological changes due to climatic drivers over past decades. Lake temperatures have
increased, ice-cover duration has decreased. For terrestrial ecosystems there is strong empirical evidence of
changes in phenology in many plant and animal taxa and northward range expansions of mobile heat-loving
animals. Climate change projections and effect studies suggest a northward shift of vegetation zones, with
terrestrial net primary production likely to increase in the North Sea region, due to warmer conditions and longer
growing seasons.

*Socio-economic effects:* The assessments of climate change effects on the different socio-economic sectors in the
North Sea region find that adaptation measures are essential for all of them, e.g. for coastal protection and in
agriculture. For North Sea fisheries, the rapid temperature rise is already being felt in terms of shifts in species





distribution and variability in stock recruitment. In agriculture an increased risk of summer drought and associated
effects will be a challenge, particularly in the South. In general, extreme weather events are likely to more often
severely disrupt crop production. Offshore and onshore activities in the North Sea energy sector (dominated by
oil, gas and wind) are highly vulnerable to extreme weather events, in terms of extreme wave heights, storms and
storm surges. All coastal countries around the North Sea with areas vulnerable to flooding by storm surges are
preparing for the challenges expected due to climate change, but coastal protection strategies differ widely from
country to country. Due to uncertainty concerning the extent and timing of climate-driven impacts, current coastal
zone adaptation plans focus on no-regret measures.
**5.3 Some differences in climate change effects between the North and Baltic Seas**
Many of the climate change signals in the Baltic and North Seas show similar behaviours and trends. But there are
also some notable differences between the two regions, which are listed below. They are based on findings reported
in the appropriate chapters of the recent assessments BACC II (BACC II Author Team, 2015) and NOSCCA
(Quante and Colijn, 2016).
• In recent decades, the surface air temperature in the Baltic Sea and North Sea regions rose in a similar
way, on the order of 1 °C in the past century. Projections of the surface air temperature as obtained by
Euro-CORDEX downscaling for a moderate scenario (RCP4.5) indicate a stronger winter and spring
warming (> 1 K) at the end of the century (2071-2100) relative to present day (1971-2000) for most parts
of the Baltic Sea region than for the western part of the North Sea region. In the summer and autumn
months the projected warming is at the same level.
• The North Sea is vigorously ventilated by the Atlantic (overturning time ~1 to 4 years). Therefore, climate
change signals from the Atlantic are rapidly transferred to the North Sea, while climate change in the
North Sea can be expected to be damped by the large thermal inertia of the Atlantic Ocean. By contrast,
the Baltic Sea is more prone to changes in mean meteorological conditions as its connection to the World
Ocean is very narrow.
• Projected changes in mean precipitation show a distinctive difference between the two sea regions for the
summer (JJA) and autumn months (SON). In the Baltic Sea region the mean precipitation for a RCP4.5
scenario is projected to increase for most land areas (5 to 25%), whereas no noticeable change (5 to -5%)
is projected along the western and southern shores of the North Sea region.
• SST is currently rising and projected to rise further for both sea areas, but the spatial pattern of the SST
increase is different. In the southern North Sea SST rises more than in the northern North Sea, while SST
warming trends are higher in the north-eastern part of the Baltic Sea (Bothnian Sea and Gulf of Finland)
than in the southern part. These spatial differences are explained by water depth (North Sea) and the ice-
albedo feedback (Baltic Sea). In addition, the northern North Sea is affected by Atlantic water inflow at
the western side of the Norwegian trench.
• The coastal regions of the North Sea experience increases in both mean sea level (MSL, as measured by
satellites) and relative mean sea level (RMSL, as measured by tide gauges). Trends in RMSL vary
significantly across the North Sea region due to the influence of vertical land movement (uplift in northern
Scotland, Norway and Denmark, and subsidence elsewhere). But the trend of RMSL is still positive
everywhere in the North Sea coasts. In contrast, sea levels relative to land along the northern Baltic Sea
coast are sinking because land levels continue rising, due to post-glacial rebound since the last ice age.


The northern Baltic Sea will experience considerable land rise also in future. As a result, the sea level will probably continue to decrease relative to land in this region. As positive trends in RMSL are more relevant for coastal protection, all countries around the North Sea with coastal areas vulnerable to flooding due to storm surges face similar challenges, while in the Baltic Sea region coastal protection is of greater concern for the countries in the south.

- The frequency of sea ice occurrence in the North Sea has decreased since about 1961, with a similar development in the western Baltic Sea. In contrast, ice still forms in the northern Baltic Sea, where it will remain a prominent feature for many years, covering about 50 to 200 x $10^3$ km$^2$, with high interannual variability, even though a linear trend of 2% decrease per decade is reported.

## 6 Knowledge gaps and research needs

Knowledge gaps and research needs have been intensively discussed within the grand challenge working groups of Baltic Earth and are summarized by the BEARs (Lehmann et al., 2021; Kuliński et al., 2021; Rutgersson et al., 2021; Weisse et al., 2021; Christensen et al., 2021; Gröger et al., 2021a; Meier et al., 2021a; Viitasalo, 2021; Reckermann et al., 2021).

In summary, we conclude that the processes that control the variability of salinity in the Baltic Sea and its entire water and energy cycles are still not fully understood (Lehmann et al., 2021). The time-dependence of the haline stratification and its links to climate change are in special need of further study. Salinity dynamics is important for its dominant role in stratification, concerning both mixing conditions and ecosystem composition and functioning. The environmental and biological factors favoring certain biogeochemical pathways through complex interactions, the pools of dissolved organic matter, and sediment biogeochemical processes are poorly understood (Kuliński et al., 2021). Although initial studies on the coastal filter capacity have been made, coastal zone models for the entire Baltic Sea and an overall estimate of bioavailable nutrients and carbon loads from land to the open sea do not exist (Kuliński et al., 2021). Considering the large internal variability, investigations of changes in extremes are limited because high-resolution observational time series are too short and model ensembles too small (Rutgersson et al., 2021). Global mean sea level rise, land uplift and wind field changes control sea level of the Baltic. However, the future evolution of these drivers, which are needed for projections, is rather uncertain (Weisse et al., 2021). Furthermore, databases for coastline changes and erosion and basin-scale models of coastal change under sea-level rise do not exist (Weisse et al., 2021).

Fully coupled regional ESMs for the Baltic Sea including the various compartments of the Earth system, atmosphere, land, ocean, sea ice, waves, terrestrial and marine ecosystems are under development but are not yet available for dynamical downscaling (Gröger et al., 2021a). The numerical estimation of water and energy cycles suffers from both model deficiencies and natural variability. For climate projections, large ensembles of regional atmosphere models are available but only one ensemble with 22 members utilized a coupled atmosphere-ice-ocean model (Christensen et al., 2021). In the past, the ocean ensembles have had too few members to address well the uncertainty related to the large multidecadal variability in the ocean (Meier et al., 2021a). Furthermore, the global sea level rise needs to be considered when making salinity projections (Meier et al., 2021b). The large uncertainty in future projections of salinity fundamentally affects the projections of the marine ecosystem (Viitasalo, 2021).





The response of food web interactions to climate change are largely unknown. The uncertainties of scenario
simulations with coupled physical-biogeochemical ocean models were discussed by Meier et al. (2018a; 2019b;
2021b). They found that in addition to natural variability the largest uncertainties are caused by (i) poorly known
current bioavailable nutrient loads from land and atmosphere and uncertain assumptions about future loads, (ii)
uncertainties of models including global sea level rise, and (iii) poorly known long-term future greenhouse gas
emissions.

Finally, the regional Earth system is driven by multiple drivers, of which climate change is just one. Multi-driver
studies are just beginning to be made and only a few have yet been published (Reckermann et al., 2021).

In the following, we list a few selected knowledge gaps related to the variables addressed by this study.
**6.1 Large-scale atmospheric circulation**
The interactions between atmospheric modes of variability of importance for the Baltic Sea region are still not
well known. For instance, while climate models are able to simulate the main features of the NAO, the frequency
of blocking over the Euro-Atlantic sector is still underestimated (IPCC, 2014b). Since observational records are
relatively short, our understanding of the AMO and its possible changes depends largely on models, and these
cannot be reliably evaluated for time scales longer than the AMO period (Knight, 2009). However, while possible
changes in these climate phenomena do contribute to the uncertainty in near-term climate projections, they are not
the main driver of the projected warming over Europe by the end of the century (Cattiaux et al., 2013; IPCC,
2014b).
**6.2 Air temperature**
Temperature and its extremes are to a large extent determined by the large-scale circulation patterns. There is
limited knowledge primarily concerning changes in large-scale atmospheric circulation patterns in a changing
climate, as mirrored by climate model discrepancies. Nevertheless, the heat cycle of the Baltic Sea region is
probably better understood than the water cycle.
**6.3 Solar radiation and cloudiness**
Multidecadal variations in surface solar radiation (SSR) are generally not well captured by current climate model
simulations (Allen et al., 2013; Storelvmo et al., 2018). The extent to which the observed variations in SSR are
caused by natural variation in cloudiness induced by atmospheric dynamic variability (Stanhill et al., 2014; Parding
et al., 2014), or by anthropogenic aerosol emissions (Wild, 2012; Ruckstuhl et al., 2008; Philipona et al., 2009;
Storelvmo et al., 2018), or perhaps additional causes, is not understood. Future cloudiness trends in global and
regional models differ in their sign (Bartók et al., 2017).
**6.4 Precipitation**
Even if climate scenarios are becoming more frequent and there is now a growing ensemble of relatively high-
resolution regional climate scenarios for Europe, they still represent only a subset of the global climate model
projections assessed by the IPCC. This means that the uncertainties of future climate change in the Baltic Sea
region are not fully captured at the horizontal resolution needed for detailed studies of climate change effects in





the region (Christensen et al., 2021). Very high-resolution so called "convective-permitting" climate models operating at grid spacing of 1-3 km are lacking for the Baltic Sea region. In other regions, such models have better agreed with observations of precipitation extremes and sometimes also given a larger climate change signal than the more traditional "high-resolution" models operating at c. 10 km grid spacing (Christensen et al., 2021). Land use change and cover, including changes in forests, can induce both local and downwind precipitation change (Meier et al., 2021c), and need to be included in projections.

### 6.5 Wind

Historical wind measurements suffer from inhomogeneity and records too short for detecting changes, considering the large internal variability in the Baltic Sea region. Projected changes are not robust among the few available downscaled ESMs.

### 6.6 Air pollution

The spatially and time resolved air quality status of a region is often assessed by means of model systems, typically with emission, meteorological and chemistry transport submodels. These model systems, used for the calculation of atmospheric concentrations and deposition of pollutants, need further developments and validation. Uncertainties are often connected to the emission segment of the modelling chain. Improvements of the implemented time profiles for the different emission sectors are especially necessary (Matthias et al., 2018). For projections of air quality with climate change models, more work is needed to establish a set of emission scenarios for air pollutants consistent with regional socio-economic pathways, like those developed by Zandersen et al. (2019). The shipping sector is currently a considerable source of air pollution in the Baltic Sea region. More research and development is needed on new fuel types and emission factors for air pollutants, relevant for politically and technologically driven abatement measures. To better address exposure and health impacts of shipping emissions more studies are required like those of Ramacher et al. (2019) and Barregard et al. (2019), especially at the harbour and city scale. Better knowledge and reduced uncertainties will improve quantification of air pollution as part of the environmental imprint of shipping in the Baltic Sea region, as developed by Moldanová et al. (2021).

### 6.7 River discharge

Precipitation from regional atmosphere models is biased and the bias correction methods applied for hydrological modeling affect the sensitivity of hydrological models to climate change (Donnelly et al., 2014). Natural variability and model uncertainties may explain the large spread in current river discharge projections (Roudier et al., 2016; Donnelly et al., 2017). The values of the parameters of a hydrological model are normally found through calibration against historical data and are always associated with uncertainty. This uncertainty will translate into uncertainty in the projected changes.

### 6.8 Nutrient inputs from land

The time scales for exchange of the nutrient pools in soils are not well known (McCrackin et al., 2018). Long-term observations do not exist. Future projections of river discharge and nutrient inputs in the Baltic Sea drainage basin agree on key aspects (e.g. increased annual discharge), but also highlight the uncertainty of the projections. To improve assessments, studies should be designed to allow explicit semi-quantitative comparisons of the effects of

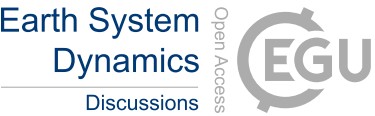

the incorporated change factors, e.g. climate, land management, policy. In the case of nitrogen inputs, the effect
of changes in anthropogenic atmospheric deposition should also be included in future projections.

**6.9 Terrestrial biosphere**

Terrestrial ecosystems in the Baltic Sea region are governed by human activities, both changes in climate due to
anthropogenic climate forcing and anthropogenic changes in land use and land cover. In return, terrestrial
ecosystems affect climate by altering the composition and the energy and water cycles of the atmosphere.
Biophysical interactions between the land surface and the atmosphere have been incorporated into regional ESMs,
in order to assess the impacts of changes in land use and land cover on regional climate and terrestrial ecosystems.
Still, biogeochemical processes related to the carbon cycle are lacking, as are explicit forest management actions
(Lindeskog et al., 2021), while explicit descriptions of some disturbances (e.g. wildfires, major storms, insect
attacks) are under development in ESMs. Only when all these interactions are incorporated can the effects of
national or international (e.g. in the European Union) climate policies on regional climate and terrestrial
ecosystems be fully assessed for compliance with the goals of the Paris Agreement.

**6.10 Snow**

A general decrease in snow-cover duration in the Baltic Sea region is well documented, especially for the southern
part. Changes in snow depth due to climate warming are much more unclear. Some evidence of increasing snow
depth in recent decades have been reported from the northern part of the study region and from mountainous areas.
Changes in sea-effect snowfall events during present climate are unknown.

**6.11 Glaciers**

It is presently not known how glacier-fed lakes react to competing environmental drivers, such as the general
Arctic warming, and the simultaneous warming-triggered lake cooling caused by increased inflow of cold glacier
meltwater, potentially carrying high sediment, nutrient, and organic matter loads. Understanding changing lake
thermal regimes and vertical mixing dynamics as well as timing and duration of seasonal ice cover is important
because ecological, biological, chemical processes, including carbon-cycling, will be affected (Lundin et al., 2015;
Smol et al., 2005; Jansen et al., 2019). Since Scandinavian glaciers are predicted to decline 80% in volume by
2100 under RCP8.5, Scandinavian glacier-fed lakes could be used as natural observatories, where changes in
processes, timescales, and effects in response to competing drivers can be studied before they occur at other glacial
lake sites, where glaciers melt more slowly (Kirchner et al., 2021).

**6.12 Permafrost**

Thawing permafrost peatlands may potentially release large amounts of organic matter, nutrients and greenhouse
gases to aquatic systems locally, but the timing and magnitude of such releases remain highly uncertain.

**6.13 Sea ice**

While the extent of the sea ice cover is well observed, observations of ice thickness are scarce. Ice thickness is
regularly monitored only at a few coastal sites with fast ice. Long records of the various ice classes, such as ridged
ice, do not exist. Sea ice models do not represent sea ice classes correctly. Since the last assessment by the BACC
II Author Team (2015) only two new scenario simulation studies on sea ice were published.



### 6.14 Lake ice

Research is required to better understand the reasons for regional and temporal differences in the patterns of change in lake ice phenology and its relationship to large-scale climatic forcing. There is a need to better understand how loss of lake ice cover modifies gas exchange between lake and atmosphere, mixing of the water column, biogeochemical cycling, and ecosystem structure and function. The socioeconomic and cultural importance of winter ice also deserves further research.

### 6.15 Water temperature

The causes of the pronounced natural variability of Baltic Sea temperature and its connection to large-scale patterns of climate variability is not well known. The occurrence of marine heatwaves is projected to increase. However, only a few studies of their impacts on the marine ecosystem exists. Furthermore, sea surface temperature trends also depend on coastal upwelling, which affects large areas of the Baltic Sea surface (Lehmann et al., 2012). Projected changes in upwelling are, however, very uncertain (Meier et al., 2021a).

### 6.16 Salinity and saltwater inflows

Salinity change depends on wind, river discharge, net precipitation on the sea and global sea level rise. Due to considerable uncertainty in all drivers and the different signs in the response of salinity to these drivers, the relative uncertainty in salinity projections is large, and larger ensembles of scenario simulations are needed (Meier et al., 2021b). This knowledge gap is compounded by the uncertainty of whether saltwater inflows from the North Sea will change. As salinity is a very important variable for the circulation in the Baltic Sea and for the marine ecosystem, projections for the Baltic Sea are a priority.

### 6.17 Stratification and overturning circulation

Stratification depends on mixing as well as on gradients in water temperature and salinity, making changes in stratification uncertain. Mixing processes such as thermal and haline convection, entrainment, double diffusive convection or boundary mixing are not fully understood. Initial results on the sensitivity of the vertical overturning circulation rely on model studies only. Hence, more measurements on the fine-structure of horizontal and vertical turbulence are needed.

### 6.18 Sea level

The regional variability of processes which drive sea-level changes, along with their uncertainties and relative importance over different timescales, display long-term developments that still require an explanation and are a challenge to planning by coastal communities (Hamlington et al., 2020). For instance, the annual cycle in Baltic Sea mean sea level (winter maxima minus spring minima) shows a basin-wide widening in the period 1800-2000 (Hünicke and Zorita, 2008). The precise mechanisms responsible for this effect are not yet completely understood, although it seems strongly controlled by atmospheric forcing (Barbosa and Donner, 2016). Furthermore, at the longer time-scales relevant for anthropogenic climate change, Baltic Sea and North Atlantic sea levels are strongly affected by the very uncertain future melting of the Antarctic ice sheet. Current estimates are mostly based on heuristic expert knowledge, as models are still under development. This is probably the largest knowledge gap affecting projections of future Baltic sea-level rise (Bamber et al., 2019). Finally, long-term relative sea level





trends are strongly affected by the vertical land movement due to glacial isostatic adjustment. This can be as large as, or even larger, than global sea-level rise. Currently, it is estimated from relatively short GPS measurements and from geo-elastic models. Both are uncertain, as point GPS measurements are strongly affected by other geological and anthropogenic effects on vertical land velocities and results from model geo-elastic models are often revised. In addition, the glacial isostatic adjustment may affect the flow intensity of river runoff into the northern Baltic Sea (coastal regions rising relative to inland regions), the effects of which, e.g. on salinity and water levels, have not been explored.

### 6.19 Waves

The lack of long-term instrumental wave measurements and gaps in the data due to the ice season complicate the analysis of extreme values. Although wave hindcasts provide a good alternative, the accuracy naturally does not match that of measured data. Furthermore, Björkqvist et al. (2020) showed that the calculation of return periods of extreme events may depend on the sampling frequency. Adding sampling variability typical for in-situ measurements to simulated hindcast data, will result in consistently shorter estimates of return periods for high significant wave heights than using the original hindcast data.

### 6.20 Sedimentation and coastal erosion

We lack a comprehensive understanding of alongshore sediment transport and its associated spatial and temporal variability along the Baltic Sea coast. In general, an eastward transport dominates along most of the southern Baltic Sea coast due to the prevailing westerly winds. However, the intensity of secondary transport induced by easterly and northerly winds is much less understood. Its combination with storm surges will expose sand dunes and cliffs to the greatest erosional impact, further complicating understanding (Musielak et al., 2017). Due to the orientation of the coastline, transport along some parts of the Baltic Sea coastline is very sensitive to the angle of incidence of the waves. For example, the incidence angle of westerly wind-waves at the western part of the Wolin Island in Poland (Dudzińska-Nowak, 2017) and the coast of Lithuania and Latvia (Soomere et al., 2017) is very small and even a slight change in the wind direction (e.g. by 10 degrees) could lead to a reversal of the direction of alongshore transport. Coastline changes at these sections vary greatly, and will hence be extremely sensitive to future changes in wind wave climate (Viška and Soomere, 2013). Another knowledge gap in understanding coastal erosion in response to future climate change concerns the impact of water levels and the submergence of the beach. Water level plays a key role in dune toe erosion and also limits aeolian sand transport on the beach. The relationship between the intensity of the forcing (wave energy, run-up) and the morphological response (erosion at the beach and dunes) during storms is not straightforward (Dudzińska-Nowak, 2017; Zhang et al., 2017). At some sites (e.g. Miedzyzdroje), dune erosion is well correlated with maximum storm surge level and storm frequency, but at others (e.g. Swinoujscie), the beach morphology is more important in determining the effect of erosion than the storm surge level.

### 6.21 Oxygen and nutrients

There are significant knowledge gaps related to the identification and quantification of oxygen sinks and sources in the Baltic Sea. In particular, more understanding is required on the dynamics of seawater inflows from the North Sea, the role of mixing processes in the ventilation of the deep water, rates of oxygen consumption in water column and sediments and how they depend on climate change. Knowledge gaps also exist concerning the transport and



transformations of DOM (including terrestrial DOM) and better quantification of the processes occurring in the
microbial loop is needed to understand the nutrient (but also C and O) dynamics in the Baltic Sea.

The direct effects of climate change are likely to be detectable first in the coastal zone, e.g. indicated by increasing
seasonal hypoxia due to warming. However, long-term records from the coastal zone are rare. More important
could be the intensification of the proposed hypoxia-related "vicious circle" in the Baltic proper, due to the
warming of both surface and deep-water layers in the Baltic proper (Savchuk, 2018; Meier et al., 2018b). The
consequent expansion of cyanobacteria blooms and increased nitrogen fixation in the Baltic proper and
neighboring basins could further counteract nitrogen load reductions and maintain hypoxia, with all its detrimental
effects. However, there are still no biogeochemical and ecosystem models capable of producing reliable long-term
scenario simulations of these processes, with sufficient confidence and precision (see Meier et al., 2018a; Meier
et al., 2019b).

### 6.22 Marine $CO_2$ system


Due to the high spatial and temporal variability of air-sea carbon fluxes, it is not known whether the Baltic Sea as
a whole is a net sink or a net source of $CO_2$. The source of the alkalinity increase observed in the Baltic Sea is still
unclear. Plausible hypotheses indicate increased weathering in the catchment and processes related to anoxic
remineralization of organic matter. There is high uncertainty in quantifying sediment/water fluxes of C, N and P,
which are important bottlenecks for understanding the dynamics of the marine $CO_2$ system and the C, N, P and $O_2$
cycling generally, especially in the deep water layers. The lack of system understanding is particularly evident in
the Gulf of Bothnia. Fransner et al. (2018) suggested that non-Redfieldian stoichiometry in phytoplankton
production could explain $pCO_2$ fields in these basins, but confirmation by observations is still lacking.

### 6.23 Marine biosphere


### 6.23.1 Lower trophic levels


The summer cyanobacteria bloom in the Baltic proper, and increasingly in recent years in the Bothnian Sea, is
considered one of the main problems of Baltic Sea eutrophication, and the nitrogen fixation it carries out is an
important process in Baltic ecosystem models (Munkes et al., 2021). It has long been considered limited by the
availability of phosphorus (Larsson et al., 1985; Granéli et al., 1990). It is therefore remarkable that it is not
possible to predict inter-annual variations in cyanobacteria blooms observed by satellites from water chemistry
(Kahru et al., 2020).

There are significant knowledge gaps related to the quantification of nitrogen fixation and the fate of the fixed
nitrogen in the Baltic Sea pelagic zone. Direct nitrogen fixation measurements were until recently dogged by
method problems, and even if these are now hopefully largely resolved (Klawonn et al., 2015), the enormous
patchiness of cyanobacteria blooms remains a huge problem. The alternative approach of directly measuring the
increase in total combined nitrogen during the bloom (Larsson et al., 2001) requires very high precision, also
suffers from patchiness problems (Rolff et al., 2007), and has not been much used. Finally, the amount of nitrogen
fixed can be estimated by modelling, based on uptake of $CO_2$ or phosphorus and assuming a Redfield N:P or C:N
ratio. This theoretically highly attractive approach (Eggert and Schneider, 2015) is hampered by the possibility of





non-Redfieldian ratios, and has made some biologists skeptical by predicting high nitrogen fixation in spring, when there are not sufficient known nitrogen-fixing autotrophs in the water to carry out this nitrogen fixation.

While total nitrogen in the water column clearly increases during the summer cyanobacterial bloom, just a couple of months later this increase seems largely to have disappeared, even though sediment traps find little evidence that it has settled out of the upper mixed layer. Are sediment trap measurements gross underestimates, or are there unidentified sites of denitrification or other overlooked nitrogen sinks in the water column? Nitrogen fixation is a central process in Baltic ecosystem models, and better observationally based estimates of processes in the nitrogen cycle are required for assessing their credibility (Munkes et al., 2021).

### 6.23.2 Marine mammals

Seal and porpoise foraging distribution and the relation of seals to haul-out sites is not well known. The requirement of sea ice for successful breeding of ringed seals has not been sufficiently assessed. Land-breeding of grey seals is not monitored regularly in most Baltic countries. The effects of interspecific competition on distributions are not known. Range contraction can be conceptualized as three stages (Bates et al., 2014): performance decline, population decrease and local extinction, all of which should be studied. For example, studies on performance decline, such as physiological conditions that reduce reproductive potential (Helle, 1980; Jüssi et al., 2008; Kauhala et al., 2017; Kauhala et al., 2019) are important (Bates et al., 2014). Breeding success of Baltic Sea ringed seals in normal winters is poorly known, as the lairs in pack ice snowdrifts are rarely found. Likewise, observations of the effects of poor ice-conditions on breeding success of ringed seals in mild winters are very limited, but the lack of protection against harsh weather and predators is assumed to be highly negative.

### 6.23.3 Waterbirds

The complex interaction between many primary parameters affected by climate change makes it hard to identify which environmental changes are actually causing changes in waterbird populations. It is currently not known in detail how shifts in distribution and timing of migration match the availability and quality of food, and thus the importance of potential temporal mismatches between food availability and requirements is unknown. In addition, changes in waterbird distribution are likely to alter inter- and intraspecific competition. Resolving these issues requires investigation of effects at other levels of the food web (e.g. loss of bivalves from areas of reduced salinity, species and size-class composition of fish communities) and their consequences for waterbirds. So far, knowledge of climate change effects on waterbirds in the Baltic Sea are mostly restricted to ducks (including diving and dabbling ducks), with much less known for other quantitatively important components of the waterbird community, i.e. divers, grebes, waders, gulls and auks. Interactions between fish and piscivorous waterbirds in particular need more attention. Responses to climate change are likely to vary between waterbird species and groups. There is still little information on which species are most affected (negatively or positively) by changes in climatic conditions and the uncertainty is therefore large on how species in future waterbird assemblages will interact and the consequences for the functioning of the Baltic Sea. To gain a better understanding on how single species (or groups with similar ecology, often closely-related) will respond to climate change is critical for projecting effects of climate change on waterbirds around the Baltic Sea.





**6.23.4 Marine food webs**

Some changes observed in marine food webs have been partly by attributed to warming, brightening and sea ice decline on long time-scales. Other drivers, such as eutrophication or fisheries, may however predominate and many records are too short to allow attributing the observed changes to climate change. Although effects of warming, ocean acidification and dissolved organic matter on some ecosystem functions have been identified in mesocosm experiments, changing food web interactions are still impossible to project. It is, however, important to include the marine biosphere in management-strategies for tackling the complex interactive aspects of climate change-related effects on the marine ecosystem and human adaptations to them (Andersson et al., 2015; Stenseth et al., 2020).

**7 Key messages**

The following lists selected key messages from this assessment that either confirm the conclusions of previous assessments or are novel (marked with NEW). The estimated level of confidence based upon agreement and evidence (see Section 2.3) of each key message refers to whether a systematic change in the considered variable was detected and attributed to climate change. Climate change is here defined as the change in climate due to human impact only (BACC II Author Team, 2015; see Section 2.1). Key messages referring to observed or simulated changes in the Baltic Sea region caused by other drivers than climate change (e.g. afforestation, eutrophication, fisheries, etc.) are not classified by a confidence level. A summary of all key messages related to climate change is presented in Table 10 and Figure 34.

**7.1 Past climate changes**

- Large-scale circulation: The AMO has undergone frequency changes, but its influence on climate variability in the Baltic Sea region remained similar, independent of the dominant frequency [NEW].
- Air temperature: During the Holocene, The Baltic Sea region experienced periods as warm as the 20th century, such as the mid-Holocene Optimum and the Medieval Warm Period. The implied rate of change was, however, much slower than the present. The past warming signal was regionally markedly heterogeneous, mostly along a west-east gradient [NEW].
- Oxygen: The previous warm periods were accompanied by oxygen deficiency in the deeper waters of the Baltic Sea, which cannot be attributed to eutrophication and was likely a result of climate forcing [NEW].

**7.2 Present climate changes**

- Large-scale atmospheric circulation: Systematic changes in large-scale atmospheric circulation related to climate change could not be detected [low confidence]. The AMO is an important driver of climate variability in the Baltic Sea region, affecting *inter alia* the correlation of regional climate variables with the NAO.
- Air temperature: Linear trends of the annual mean temperature anomalies during 1876−2018 were +0.10 °C decade$^{-1}$ north of 60°N and +0.09 °C decade$^{-1}$ south of 60°N in the Baltic Sea region [high confidence]. This is larger than the global mean temperature trend and slightly larger than estimated in the earlier BACC reports [NEW]. The warm spell duration index has increased during 1950-2018 [medium confidence]. Statistically significant decreases in winter cold spell duration index across the period 1979−



2013 have been widespread in Norway and Sweden, but less prevalent in eastern Finland, while changes
in summer cold spell have been small in general [medium confidence].
• Solar radiation and cloudiness: Various satellite data products suggest a small but robust decline in
cloudiness over the Baltic Sea region since the 1980s [low confidence, NEW]. However, whether this
signal is an indicator of a changing climate or due to internal variability is unknown.
• Precipitation: Since 1950, annual mean precipitation has generally increased in the northern part of the
Baltic Sea region. There is some evidence of a long-term trend [low confidence]. However, long-term
records suffer from inhomogeneity due to the increasing number of rain gauges. Frequency and intensity
of heavy precipitation events have increased [medium confidence]. Drought frequency has increased
across southern Europe and most of central Europe since 1950, but decreased in many parts of Northern
Europe [low confidence].
• Wind: Owing to the large internal variability, it is unclear whether there is an overall trend in mean wind
speed. There has been an increase in the number of deep cyclones over central and Northern Europe since
the late 1950s, but no evidence for a long-term trend [low confidence].
• Air pollution: The influence of climate change on air pollution is small and undetectable, given the
dominance of other human activities [low confidence]. Land-based emissions are declining due to
emission control measures, but some emissions from the shipping sector may be increasing.
• River discharge: For the period 1900-2008, no trend in total river discharge was found, but there was a
pronounced 30-year variability. Data for some rivers in the northern Baltic Sea catchment indicate a long-
term trend, but the confidence in these reconstructions is low. Since the 1970s, the total river winter
discharge is increasing, perhaps due to warming or river regulations [low confidence]. Due to earlier
snow-melt, driven by temperature increases in the region and a decreasing frequency of arctic air mass
advection, high flow events in the Baltic Sea region shifted from late March to February,. In Sweden,
trends in the magnitude of high flow events over the past 100 years are not statistically significant [low
confidence, NEW].
• Riverine nutrient loads: The effect of changing climate on riverine nutrient loads is small and not
detectable [low confidence].
• Terrestrial biosphere: Combining all vegetation types in the entire Baltic Sea region, satellite observations
suggest an advancement of the growing season by 0.30 day/year over the period 2000-2016. The most
important driver of the advancement of the growing season is spring mean temperature, with an
advancement rate of 2.47 day/°C of spring warming [medium confidence, NEW]. Observations and
model results suggest cooling trends in daily minimum and warming trends in daily maximum
temperatures in response to deforestation, and the opposite tendencies for afforestation. [NEW]
• Snow: The decrease in snow cover has accelerated in recent decades, except in the mountain areas and
the north-eastern part of the Baltic Sea region [high confidence, NEW]. On average, the number of days
with snow cover has declined by 3–5 days per decade, [high confidence]. Mean and maximum snow
depth has also decreased, most clearly in the southern and central part of the region [high confidence].
Whether sea-effect snowfall events have changed is unknown [low confidence].
• Glaciers: Inventories of all Scandinavian glaciers, available only since 2006, show that they have lost 20
Gt of ice (~8% of their total mass) during 2006-2015. Atmospheric warming is very likely the primary
driver of glacier mass loss [high confidence]. [NEW]



- Permafrost: Recent warming has caused losses of over 20% of the original 6200 km$^2$ of permafrost in the Baltic Sea catchment area during 1997-2018 [medium confidence, NEW].

- Sea ice: Long-term decreases in sea ice in the Baltic Sea have exceeded the large natural climatological variability and can only be attributed to global climate change [high confidence]. In addition, unprecedented mild ice seasons have occurred in the last ten years, and 100-year trends in sea ice cover showed an accelerated decline in 1921-2020 compared to 1910-2011 [high confidence, NEW].

- Lake ice: Warming in the Baltic Sea catchment during recent decades has resulted in earlier ice break-up, later freeze-up, and hence shorter ice cover duration on the lakes in the region. [high confidence, NEW]

- Water temperature: Monitoring data, satellite data and model-based historical reconstructions indicate an increase in annual mean sea surface temperature averaged over the Baltic Sea of 0.4–0.6 °C decade$^{-1}$ or ~1 – 2 °C since the 1980s [high confidence]. During 1856–2005, reconstructed SSTs increased by 0.03 and 0.06 °C decade$^{-1}$ in northeastern and southwestern areas, respectively. Hence, recent warming trends have accelerated tenfold [NEW]. Long-term measurements at Tvärminne, on the north coast of the Gulf of Finland, indicate that marine heat waves have increased since 1926 [low confidence].

- Salinity and saltwater inflows: The record of major Baltic inflows (MBIs) has been revised and the earlier reported decreasing trend is now seen as artifactual. On centennial time-scales, there are no statistically significant trends in salinity averaged over the Baltic Sea (1920-2008) or in MBIs (1887-2017), but pronounced multidecadal variability, with a period of about 30 years. Model results suggest that a decade of decreasing salinity, like the 1983-1992 stagnation, happens about once a century due to natural variability. Due to increased river runoff in the northern catchment, the North-South gradient in sea surface salinity likely increased in 1900–2008 [low confidence, NEW].

- Stratification and overturning circulation: No long-term trend in stratification was detected, but during 1982-2016 stratification increased in most of the Baltic Sea, with the seasonal thermocline and the perennial halocline strengthening by 0.33–0.39 and 0.70–0.88 kg m$^{-3}$, respectively [low confidence, NEW].

- Sea level: Since 1886 the mean sea level in the Baltic Sea relative to the geoid has increased by about 1-2 mm per year, similar to the global mean rate [high confidence]. However, in the northern Baltic Sea rapid land uplift causes a relative sea level decrease [high confidence]. Although an acceleration of the mean sea level rise at individual stations could not yet be detected, the all-station-average-record showed an almost statistically significant acceleration [low confidence, NEW]. Basin-wide, no statistically significant, long-term changes in extreme sea levels relative to the mean sea level of the Baltic Sea could be documented [low confidence, NEW].

- Waves: Wave hindcasts and observations are too short for studies of climate-relevant trends [low confidence].

- Sedimentation and coastal erosion: Dominance of mobile sediments makes the southern and eastern coasts more vulnerable to wind-wave induced transport than other Baltic Sea coasts [high confidence]. Prevailing westerly winds lead to mainly west-east sediment transport and an alternation of glacial till cliffs (sources), sandy beaches and spits (sinks). No statistically significant, long-term changes were found [low confidence, NEW].



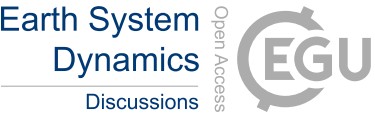

- Oxygen and nutrients: Reconstructions of oxygen conditions in the Baltic Sea for the period 1898-2012 suggest a tenfold increase of the hypoxic area, with current values of up to 70,000 km$^2$. This increase was attributed mainly to increased nutrient loads, with a minor contribution from climate warming [low confidence, NEW]. Furthermore, recently estimated oxygen consumption rates in the Baltic Sea are higher than observed before, reducing the duration of improved oxygen conditions after natural ventilation events by oxygen-enriched saltwater inflows.

- Marine $CO_2$ system – air-sea exchange: In the period 1980-2005, sub-basins affected by high riverine runoff and related high loads of terrestrial organic matter (e.g. Gulf of Bothnia) were found to be on average a source of $CO_2$ to the atmosphere. This outgassing was more than compensated by the high $CO_2$ uptake by the open waters of the Baltic proper [medium confidence, NEW].

- Marine CO2 system – alkalinity: During 1900-2015, a long-term trend in alkalinity was observed, with largest increases in the Gulf of Bothnia, where it almost entirely cancelled the pH decrease expected from rising atmospheric $pCO_2$. The smaller alkalinity increase in the southern Baltic Sea compensated ocean acidification by about 50%. Due to the high seasonal variability in pH, large interannual variability in productivity and the identified alkalinity trend, no acidification was measurable in the central and northern Baltic Sea [medium confidence, NEW].

- Microbial communities: Long-term time series from 1994 to 2006 show that increased riverine dissolved organic matter suppresses phytoplankton biomass production and shifts the carbon flow towards heterotrophic microbes [low confidence, NEW].

- Phytoplankton and cyanobacteria: The growing season for phytoplankton and cyanobacteria has lengthened significantly in the past few decades [medium confidence] and the ratio between diatom and dinoflagellate biomasses declined during the past century, probably due to warmer winters [low confidence, NEW]. The annual chlorophyll maximum, in the 1980s associated with the spring diatom bloom, has shifted to coincide with the summer cyanobacteria bloom [low confidence, NEW]. Although inter-annually oscillating, surface cyanobacteria accumulations became a recurrent summer feature of the southern Bothnian Sea in the 2010s [medium confidence, NEW].

- Macroalgae: Long-term changes in Baltic Sea macroalgae and charophytes have been attributed to changes in salinity, wind exposure, nutrient availability and water transparency as well as biotic interactions [low confidence, NEW]. However, the role of climate change is unclear.

- Zoobenthos: Increasing near-bottom temperature may partially explain the spreading of non-indigenous species, such as polychaetes of the genus *Marenzelleria*. The effects on zoobenthos are primarily synergistic, through e.g. eutrophication and hypoxia [low confidence, NEW].

- Fish: Changes in temperature, salinity and species interactions can affect the stocks of cod, sprat and herring. However, the dominant driver is the fishery. For coastal fish, the distribution of pikeperch expanded northwards along the coasts of the Bothnian Sea, apparently due to the warming waters. For many coastal fish species eutrophication is, however, equally or more important than climate change [low confidence, NEW].

- Marine mammals: Populations of ice-breeding seals, especially southern populations of the ringed seal, have likely suffered from the sea ice decline [medium confidence]. However, this is based on occasional ringed seal moult counts that indicating no population growth, while monitoring data on reproductive success are missing.





- Waterbirds: Many waterbird species have shifted their wintering range northwards [high confidence]. They now migrate earlier in spring [medium confidence]. Effects of warming sea temperature are inconsistent, because both positive and negative effects on foraging conditions and food quality have been found [low confidence]. Most migrating Baltic Sea waterbirds are also affected by climate change outside the Baltic Sea [medium confidence].

- Marine food webs: Significant alterations in food web structure and functioning such as the shift from early diatom to later dinoflagellate dominated blooms have been observed. However, the causes of these changes are unknown [low confidence].

**7.3 Future climate changes**

- Large-scale circulation: Projections suggest a more zonal flow over Northern Europe and a northward shift in the mean summer position of the westerlies at the end of the century [low confidence].

- Air temperature: Coupled atmosphere-ocean regional climate models project an increase in annual mean air temperature by between 1.5 and 4.3°C over the Baltic Sea catchment area at the end of the century. The range indicates ensemble mean values for RCP2.6 and RCP8.5 scenarios. On average, air over surrounding land will warm about 0.1 to 0.4°C more than the air over the Baltic Sea [high confidence, NEW]. A bias-adjusted median estimate of increase in warm spell duration index in Scandinavia for the period 2071-2100, compared to 1981-2010, was about 15 days under RCP8.5, with an uncertainty range of about 5-20 days [medium confidence]. The cold spell duration index in Northern Europe is projected to decrease in the future, with a likely range of from -5 to -8 days per year by 2071-2100, compared to 1971-2000 [medium confidence].

- Solar radiation and cloudiness: Projections for solar radiation and cloudiness differ systematically in sign between global and regional climate models, indicating high uncertainty [low confidence, NEW].

- Precipitation: Annual mean precipitation is projected to increase over the entire Baltic Sea catchment at the end of the century [medium confidence]. The signal is robust for winter among the various regional climate models but is highly uncertain for summer in the south. The intensity and frequency of heavy rainfall events are projected to increase. These increases are even larger for convection-resolving models [high confidence, NEW]. Projections show that the number of dry days in the southern and central parts of the Baltic Sea basin increases mainly in summer [low confidence].

- Wind: Changes in wind over the Baltic Sea region are highly uncertain [low confidence]. Over sea areas where the average ice cover is projected to diminish, such as the Bothnian Sea and the eastern Gulf of Finland, the mean wind is projected to increase because of a warmer sea surface and reduced stability of the planetary boundary layer [low confidence].

- Air pollution: The impact of climate change on air quality and atmospheric deposition is smaller than the assumed impact of future changes in emissions [low confidence].

- River discharge: River runoff is projected to increase 2–22% in RCP4.5 and 7–22% in RCP8.5. River discharge is projected to increase to the northern and decrease to the southern sub-basins [low confidence]. High flows are projected to decrease in spring and increase in autumn and winter due to earlier snow melt and more winter rain. Over much of continental Europe, an increase in intensity of high flow events is projected with increasing temperature [low confidence].




- Land nutrient inputs: The impact of climate change on land nutrient inputs is smaller than the impact of changes in land management, populations and nutrient point-source releases. In any given river, larger runoff would lead to larger nutrient inputs [low confidence].

- Terrestrial growing season: Projections suggested that decreasing surface albedo in the Arctic region in winter and spring will notably amplify the future warming in spring (positive feedback), while the increased evapotranspiration will led to a marked cooling during summer (negative feedback). These feedbacks will stimulate vegetation growth, due to an earlier start of the growing season, leading to compositional changes in woody plants and the distribution of vegetation. Arctic terrestrial ecosystems could continue to sequester carbon until the 2060-2070s, after which the terrestrial ecosystems are projected to turn into weak sources of carbon due to increased soil respiration and biomass burning [low confidence, NEW].

- Terrestrial carbon sequestration: Mitigation scenarios that decrease the fraction of coniferous forest in favour of deciduous forest, and increase the area of deciduous forest in Northern Europe from 130,000 to 480,000 km$^2$, were projected to reduce near-surface temperatures and give maximum carbon sequestration. [NEW]

- Snow: Projections under RCP8.5 suggest a reduction of the average snow amount by more 70% for most areas, with the exception of the high Scandinavian mountains, where the warming temperature does not reach the freezing point as often as in lower-lying regions [high confidence]. Sea-effect snowfall events in future climate have not been investigated yet.

- Glaciers: Scandinavian glaciers will lose more than 80% of their current mass by 2100 under RCP8.5, and many are projected to disappear, regardless of future emission scenarios [high confidence, NEW]. Furthermore, river runoff from glaciers is also projected to change regardless of the emission scenario, and to result in increased average winter runoff and in earlier spring peaks [high confidence, NEW].

- Permafrost: In the future climate, the on-going loss of permafrost in the Baltic Sea catchment will very likely accelerate [high confidence].

- Sea ice: Regional climate projections consistently project shrinking and thinning of Baltic Sea ice cover [high confidence], but still estimate that some ice will be formed even in mildest future winters. However, those estimates are based on a limited number of ensembles and may not represent future climate variability correctly.

- Lake ice: The observed trends of earlier ice break-up, later freeze-up, and shorter ice cover duration on lakes in the region are projected to continue with future warming, and lakes with intermittent winter ice will consequently become increasingly abundant [high confidence, NEW].

- Water temperature: Coupled atmosphere-ocean regional climate models project an increase in annual mean SST of between 1.2 and 3.2℃, averaged for the Baltic Sea the end of the century. The range indicates ensemble mean values for RCP2.6 and RCP8.5 scenarios. Warming will be largest in summer in the northern Baltic Sea [high confidence]. Under both RCP4.5 and RCP8.5, record-breaking summer mean SSTs were projected to increase at the end of the century [medium confidence, NEW]. However, due to the pronounced internal variability there might be decades in the near future without record-breaking events.

- Salinity and saltwater inflows: An increase in river runoff or westerly winds will tend to decrease salinity, but a global sea level rise will tend to increase it, because an enlarged cross-sectional area of the Danish



Straits will increase the saltwater imports from the Kattegat. Due to the large uncertainty in projected
river runoff, wind and global sea level rise, salinity projections show a wide spread, from increasing to
decreasing salinities, and no robust changes were identified [low confidence, NEW].
• Stratification and overturning circulation: Considering all potential drivers of changes in salinity in the
Baltic Sea (wind, river runoff, net precipitation, global sea level rise), neither the haline-induced
stratification nor the overturning circulation is projected to change [low confidence, NEW]. Projections
consistently show that the seasonal thermocline during summer will intensify across nearly the whole
Baltic Sea [high confidence, NEW].
• Sea level: Future absolute sea level in the Baltic Sea will continue to rise with the global mean sea level
[high confidence]. Its regional manifestation is, however, modulated by the future melting of Antarctica,
which affects the Baltic Sea more strongly than the melting of Greenland [low confidence]. Using current
estimates, the regional mean sea level is projected to rise by about 87% of the global mean sea level. Land
uplift is roughly known but difficult to estimate accurately in practice, as many regional geological factors
blur the signature of the glacial isostatic adjustment. Trends in sea level extremes will be determined by
the changing mean sea level and possible future changes in storminess. The uncertainty in the latter driver
is very large [low confidence].
• Waves: The projected decrease in seasonal sea ice cover will have considerable effects on the wave
climate in the northernmost Baltic Sea [high confidence, NEW]. Otherwise, there are no conclusive
results on possible changes in the wave climate and wave extremes, because of the uncertainty about
changes in wind fields [low confidence].
• Sedimentation and coastal erosion: Changes in sea level, wind, waves and sea ice all affect sediment
transport and coastal erosion. Hence, available projections are highly uncertain [low confidence, NEW].
• Oxygen and nutrients: The future response of deep water oxygen conditions will mainly depend on future
nutrient inputs from land [medium confidence]. However, coastal hypoxia might increase due to warming
of the water in shallow areas [medium confidence]. Implementation of the BSAP will lead to declining
phosphorus concentrations [medium confidence, NEW].
• Marine $CO_2$ system: Due to anthropogenic emissions, atmospheric $pCO_2$ will rise, and consequently also
the mean $pCO_2$ of Baltic surface seawater, which has the potential to lower pH [high confidence].
However, the magnitude of the pH change also depends on alkalinity trends, which are highly uncertain
[low confidence]. Hence, projections for the Baltic Sea are different from the global ocean.
• Microbial communities: The impact of climate change on microbes and the functioning of the microbial
loop have been studied experimentally. In the northern Gulf of Bothnia, adding DOM increased the
abundance of bacteria, whereas a temperature increase (from 12 to 15°C) reduced their abundance [low
confidence, NEW].
• Phytoplankton and cyanobacteria: The effect of climate change on phytoplankton and cyanobacteria
blooms is larger under high nutrient concentrations, but nutrient loads are the dominant driver. If the
BSAP is fully implemented, the projected environmental status of the Baltic Sea will be significantly
improved, and extreme cyanobacteria blooms will be rare or absent [low confidence, NEW].
• Zooplankton: Experimental studies suggested improved conditions for microzooplankton due to warming
but negative effects on some larger zooplankton species [low confidence, NEW].



- Macroalgae and vascular plants: The direct and indirect effects of changes in temperature, salinity and pH are likely to change the geographic distribution of Baltic Sea macrophytes. However, neither experimental studies nor past observed changes provide conclusive projections for the effects of climate change [low confidence, NEW].

- Zoobenthos: In a warmer and less eutrophic Baltic Sea, benthic-pelagic coupling will be weaker, resulting in decreasing benthic biomass [low confidence, NEW].

- Non-indigenous species: Climate change may favour invasions of non-indigenous species. However, it is impossible to project which species may enter the Baltic Sea in future [low confidence].

- Fish: Projected changes in temperature and salinity will affect the stocks of cod, sprat and herring. However, nutrient loads and especially fishing mortality are also important drivers. Although multi-driver modeling studies have been performed, the impact of climate change is uncertain [low confidence, NEW].

- Marine mammals: Mild winters are known to negatively affect Baltic ringed seals (*Phoca hispida botnica*) because without their sea ice lair, the pups are more vulnerable to weather and predators, and it has been projected that the growth rates of ringed seal populations will decline in the next 90 years. Also for grey seals (*Halichoerus grypus*), it has been suggested that reduced ice cover in combination with (partly climate-driven) changes in the food web, may affect their body condition and birth rate [low confidence].

- Waterbirds: The northward distributional shifts of waterbirds are expected to continue [medium confidence]. Effects on waterbird food will be manifold, but consequences are difficult to predict [low confidence]. The rising sea level and erosion are expected to reduce the availability of breeding habitats [low confidence].

- Marine food webs: Significant alterations in food web structure and functioning can be expected, since species distributions and abundances are expected to change with warming sea water. The consequences are difficult to project, as research into the long-term dynamics of food webs is still scarce [low confidence].

## 8   Concluding remarks

We found that

1. The overall conclusions of the BACC I and BACC II assessments remain valid.

2. However, new coupled models (atmosphere-ice-ocean, atmosphere-land), larger ensembles of scenario simulations (CORDEX), new mesocosm experiments (warming, ocean acidification, and dissolved organic matter), extended monitoring (glaciers, satellite data) and homogenized records of observations (MBIs) have led to new insights into past and future climate variability.

3. Improved paleoclimate simulations of the Holocene, new dendroclimatological reconstructions of the past 1000 years and new climate regionalizations have added regional details (east-west gradients over the Baltic Sea region) and improved our understanding of internal variability (sea level extremes, stagnation periods) and the remote impact of low-frequency North Atlantic variability on the Baltic Sea region (AMO, Baltic Sea salinity). New sediment cores suggest that hypoxia during the Medieval Climate Anomaly was caused by climate variability, rather than by human influence, as claimed earlier.



4. Natural variability of many variables of the Earth system is larger than previously realized, requiring larger model ensembles for convincing future projections. Although the first, relatively large ensemble of scenario simulations utilizing a regional coupled atmosphere-ice-ocean model has become available, uncertainty estimates are still incomplete.

5. New regional ESMs including additional components of the Earth system are under development. However, the simulated water cycle is still biased.

6. The first complex multiple-driver study with focus on present and future climates addressing for instance eutrophication of the Baltic Sea, fisheries and climate change has become available and an overall assessment of the various drivers in the Baltic Sea region is part of the BEARs. However, further research on the interplay between drivers is needed.

7. More research on changing extremes was performed, acknowledging that the impact of changing extremes may be more important than that of changing means. However, most observational records are either too short or too heterogeneous for statistical studies of extremes.

8. The climate change signal is still confined to increases in observed air and water temperatures, to decreases in sea and lake ice, snow cover, permafrost and glacier mass, to the rise in mean sea level, and to variables directly related to temperature and the cryosphere, such as ringed seal habitats. Compared to the previous BACC report, changes in air temperature, sea ice, snow cover and sea level were shown to have accelerated.

9. Intensive research on the land-sea interface focussing on the coastal filter has been performed and nutrient retention in the coastal zone was estimated for the first time. Uncertainty concerning the bioavailability of nutrient loads was identified as one of the foremost challenges for marine biogeochemistry. However, a model for the entire Baltic Sea coastal zone is still missing and the effect of climate change on the coastal filter capacity is still unknown.

10. In contradiction to earlier results, observed MBIs have no declining trend. Due to the uncertainties in projections of the regional wind, regional precipitation and evaporation, river discharge and global mean sea level rise, projections of salinity in the Baltic Sea are uncertain and it remains unknown whether the Baltic Sea will become less or more salty. As salinity is a crucial variable for the marine ecosystem and for Baltic circulation, projections for the Baltic Sea as a whole are regarded as uncertain.

11. The Baltic Sea may become more acidic in the future, but the decrease in pH may partly be compensated by an alkalinity increase, as in the past. Hence, past changes in Baltic carbonate chemistry were different from the global ocean acidification, and pH changes may differ also in future.

12. Large marine food web changes were observed, which could partly be attributed to warming, brightening and sea ice decline. However other factors also play important roles, and many records are too short for attribution studies.

**Author contributions**

| Chapter | Title | Authors |
| --- | --- | --- |
| 1 | **Introduction** | |
| 1.1 | Overview | H.E.M. Meier |
| 1.2 | The BACC process | M. Reckermann, H.E.M. Meier |





| 1.3 | Summary of BACC I and BACC II key messages | H.E.M. Meier |
|---|---|---|
| 1.4 | Baltic Sea Region characteristics | K. Myrberg |
| 1.5 | Global climate change | M. Gröger |
| 2 | **Methods** | |
| 2.1 | Assessment of literature | H.E.M. Meier |
| 2.2 | Climate model data | H.E.M. Meier |
| 2.3 | Uncertainty estimates | H.E.M. Meier |
| 3 | **Current state of knowledge** | |
| 3.1 | **Past climate change** | E. Zorita |
| 3.2 | **Present climate change** | |
| 3.2.1 | Present climate change - Atmosphere | |
| 3.2.1.1 | Large-scale circulation | M. Stendel, C. Frauen, F. Börgel, H.E.M. Meier, M. Kniebusch |
| 3.2.1.2 | Air temperature | A. Rutgersson |
| 3.2.1.3 | Solar radiation | T. Carlund, A. Rutgersson |
| 3.2.1.4 | Precipitation | J. Käyhkö, E. Kjellström |
| 3.2.1.5 | Wind | M. Stendel |
| 3.2.1.6 | Air pollution, air quality and atmospheric nutrient deposition | M. Quante |
| 3.2.2 | Present climate change - Land | |
| 3.2.2.1 | River discharge | J. Käyhkö, G. Lindström |
| 3.2.2.2 | Land nutrient inputs | O.P. Savchuk, B. Müller-Karulis |
| 3.2.3 | Terrestrial biosphere | W. May, P.A. Miller |
| 3.2.4 | Present climate change - Cryosphere | |
| 3.2.4.1 | Snow | J. Jaagus |
| 3.2.4.2 | Glaciers | N. Kirchner |
| 3.2.4.3 | Permafrost | G. Hugelius |
| 3.2.4.4 | Sea ice | J.J. Haapala |
| 3.2.4.5 | Lake ice | J. Käyhkö |
| 3.2.5 | Present climate change – Ocean and marine sediments | |
| 3.2.5.1 | Water temperature | C. Dieterich, H.E.M. Meier |
| 3.2.5.2 | Salinity and saltwater inflows | V. Mohrholz, H.E.M. Meier, K. Myrberg, A. Lehmann |





| 3.2.5.3 | Stratification and overturning circulation | M. Gröger, H.E.M. Meier, K. Myrberg |
|---|---|---|
| 3.2.5.4 | Sea level | B. Hünicke, E. Zorita, C. Dieterich, R. Weisse |
| 3.2.5.5 | Waves | L. Tuomi |
| 3.2.5.6 | Sedimentation and coastal erosion | W. Zhang |
| 3.2.5.7 | Marine carbonate and biogeochemistry | K. Kulinski, J. Carstensen, B. Müller-Karulis, O. Savchuk |
| 3.2.6 | Marine biosphere | M. Viitasalo, E. Bonsdorff, R. Elmgren, A. Galatius, M. Ahola, V. Dierschke, I. Carlen, M. Frederiksen, E. Gaget, A. Halkka, M. Jüssi, D. Pavon-Jordan |
| 3.3 | **Future climate change** | All |
| 3.3.1 | Future climate change - Atmosphere | |
| 3.3.1.1 | Large-scale circulation | A. Rutgersson, F. Börgel, C. Frauen |
| 3.3.1.2 | Air temperature | A. Rutgersson, E. Kjellström, O.B. Christensen |
| 3.3.1.3 | Solar radiation | A. Rutgersson, T. Carlund |
| 3.3.1.4 | Precipitation | E. Kjellström, O.B. Christensen |
| 3.3.1.5 | Wind | M. Stendel, H.E.M. Meier |
| 3.3.1.6 | Air pollution, air quality and atmospheric nutrient deposition | M. Quante |
| 3.3.2 | Future climate change - Land | |
| 3.3.2.1 | River discharge | J Käyhkö |
| 3.3.2.2 | Riverine nutrient loads | R. Capell, A. Bartosova |
| 3.3.3 | Terrestrial biosphere | W. May, P.A. Miller |
| 3.3.4 | Future climate change - Cryosphere | |
| 3.3.4.1 | Snow | O.B. Christensen |
| 3.3.4.2 | Glaciers | N. Kirchner |
| 3.3.4.3 | Permafrost | G. Hugelius |
| 3.3.4.4 | Sea ice | J.J. Haapala |
| 3.3.4.5 | Lake ice | J. Käyhkö |
| 3.3.5 | Future climate change – Ocean and marine sediments | |
| 3.3.5.1 | Water temperature | C. Dieterich, H.E.M. Meier, M. Gröger |
| 3.3.5.2 | Salinity and saltwater inflows | H.E.M. Meier |
| 3.3.5.3 | Stratification and overturning circulation | M. Gröger, C. Dieterich, H.E.M. Meier |





| 3.3.5.4 | Sea level | B. Hünicke, E. Zorita, C. Dieterich, R. Weisse |
|---|---|---|
| 3.3.5.5 | Waves | L. Tuomi |
| 3.3.5.6 | Sedimentation and coastal erosion | W. Zhang |
| 3.3.5.7 | Marine carbonate and biogeochemistry | K. Kulinski, J. Carstensen, B. Müller-Karulis, O. Savchuk |
| 3.3.6 | Marine biosphere | M. Viitasalo, E. Bonsdorff, R. Elmgren, A. Galatius, M. Ahola, V. Dierschke, I. Carlen, M. Frederiksen, E. Gaget, A. Halkka, M. Jüssi, D. Pavon-Jordan |
| 4 | **Interactions with other drivers** | M. Reckermann |
| 5 | **Comparison with the North Sea region** | M. Quante |
| 6 | **Knowledge gaps** | H.E.M. Meier and All |
| 7 | **Key messages** | H.E.M. Meier and All |
| 8 | **Concluding remarks** | H.E.M. Meier and All |
| Figures and Tables | Analysis of observed time series | M. Kniebusch |
| Figures and Tables | Analysis of scenario simulations | M. Kniebusch, H.E.M. Meier, C. Dieterich, M. Gröger, O.P. Savchuk, G. Lindström, N. Kirchner, E. Zorita, G. Hugelius |

**Acknowledgements**
The research presented in this study is part of the Baltic Earth Assessment Reports project of the Baltic
Earth program (Earth System Science for the Baltic Sea region, see http://www.baltic.earth). Glacier mass
loss data were obtained from the SITES Data Portal (https://data.fieldsites.se/portal/, see World Glacier
Monitoring Service (WGMS, 2020) and Swedish Infrastructure for Ecosystem Science (SITES, 2021a, b, c).
We thank Berit Recklebe for technical support and preparation of the reference list.



**Figures**

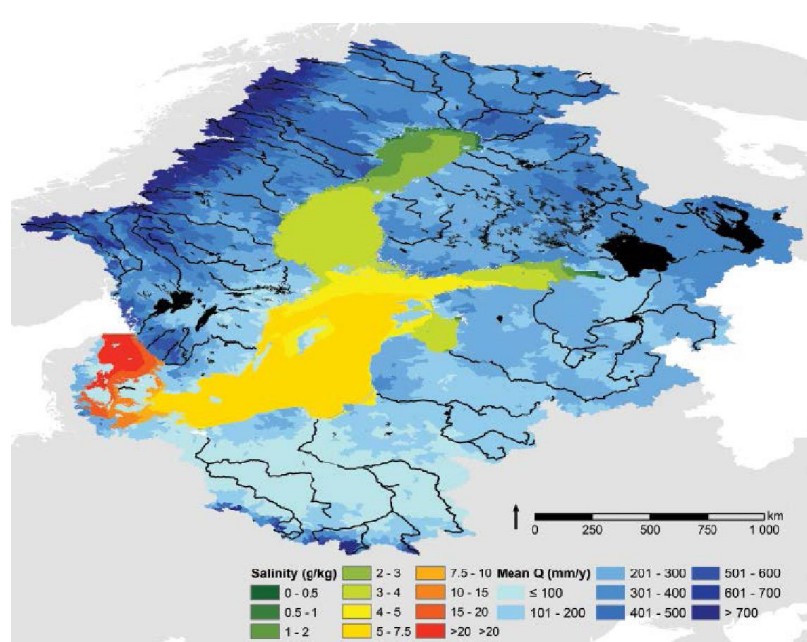


**Figure 1:** The Baltic Sea and its catchment area, showing climatological mean salinity (in g kg$^{-1}$) and river runoff
(in mm year$^{-1}$). (Source: Meier et al., 2014)



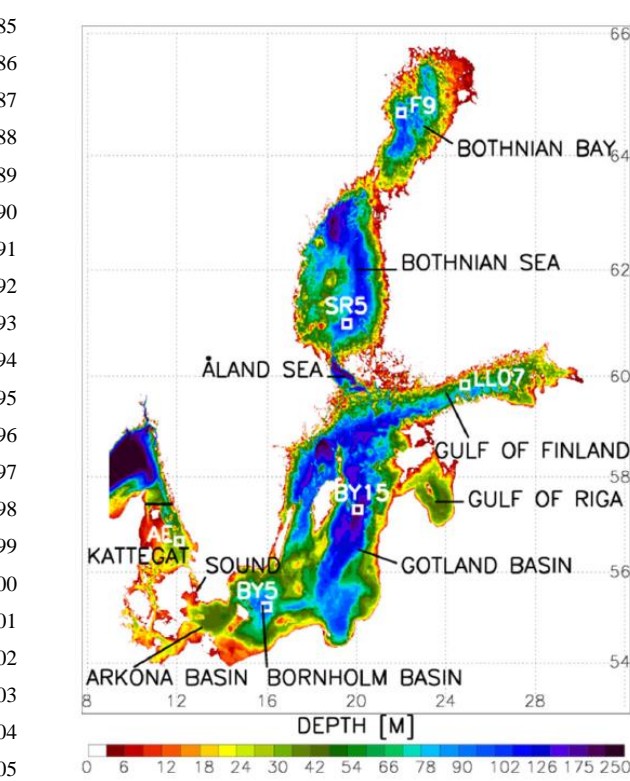

**Figure 2:** Bottom topography of the Baltic Sea and locations of the monitoring stations Arkona Deep (BY2), Bornholm Deep (BY5), Gdansk Deep (BMPL1), Gotland Deep (BY15), Northern Deep (OMTF 0286), Landsort Deep (BY15), and Åland Sea (F64). The Baltic proper comprises the Arkona Basin, Bornholm Basin and Gotland Basin.




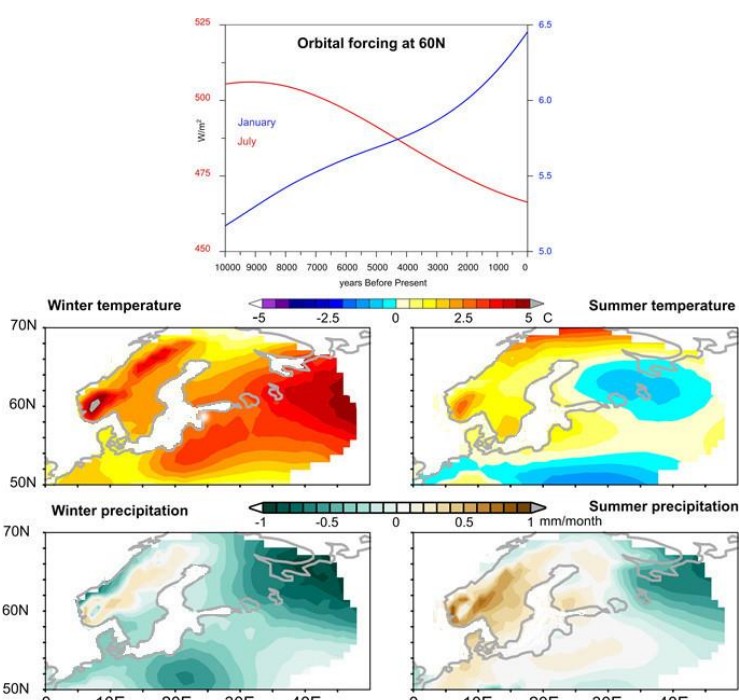


**Figure 3**: Orbital forcing (irradiance) at 60ºN in January and July (derived from Laskar et al., 2004) and the
anomalies of reconstructed seasonal temperature and precipitation compared to preindustrial climate (Mauri et al.,
2015) in the Baltic Sea region at the Mid-Holocene Optimum (6000 before present).



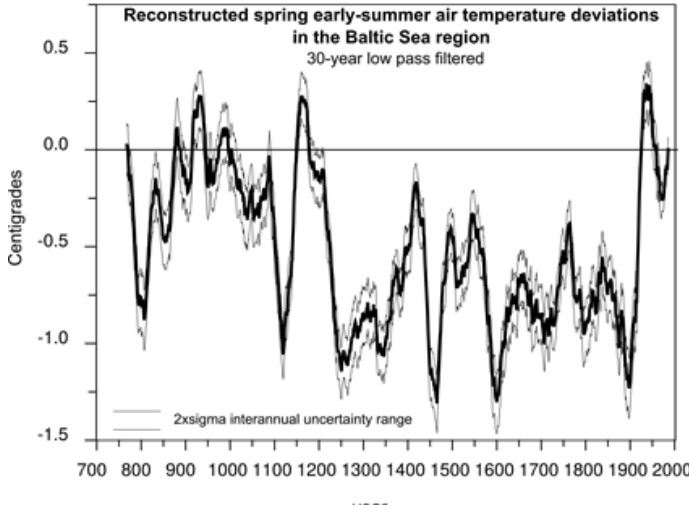


**Figure 4:** Reconstructed spring-early-summer air temperature in the Baltic Sea region (land areas in the box 0-
40ºE x 55-70ºN, deviations from the 20th century mean) derived from Luterbacher et al. (2016).The record is
smoothed by a 30-year low-pass filter. The approximate uncertainty range has been estimated here from the data
provided by the original publication at interannual and grid-cell scale.


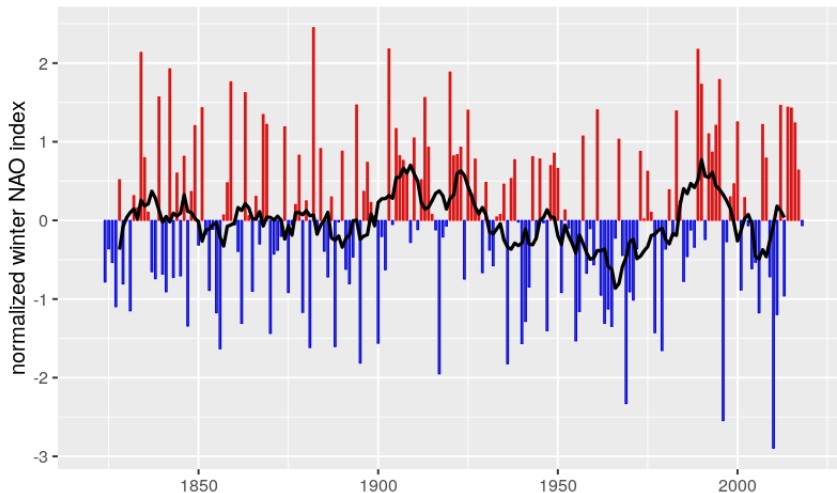



**Figure 5:** Normalized winter (December through March; DJFM) mean NAO index during 1821/22-2018/19. Red:
positive, blue: negative, black: 10-year running mean. Normalization: (data-mean (data)/standard deviation (data).
(Data source: https://crudata.uea.ac.uk/cru/data/nao/nao.dat, compiled by Madline Kniebusch, Leibniz Institute for
Baltic Sea Research Warnemünde)





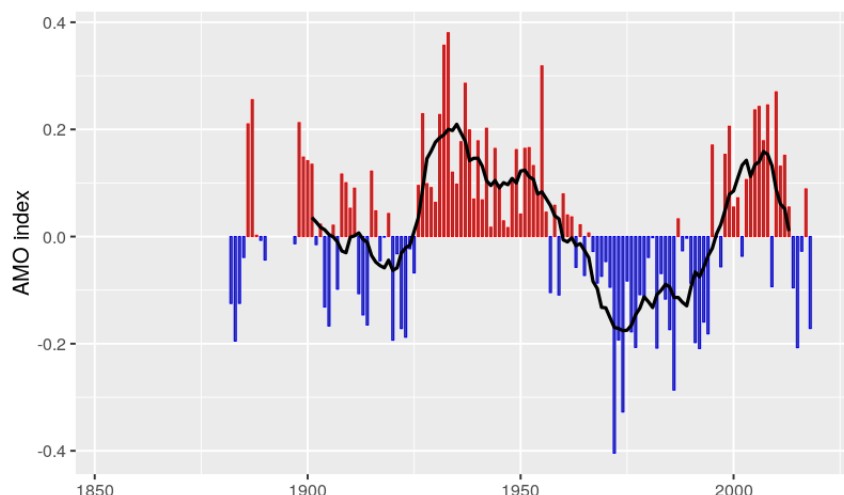


**Figure 6:** Normalized annual mean AMO index during 1882-2018. Red: positive, blue: negative, black: 10-year
running mean. Normalization: (data-mean (data)/standard deviation (data). (Data source:
https://climexp.knmi.nl/data/iamo_hadsst_ts.dat, compiled by Madline Kniebusch, Leibniz Institute for Baltic Sea
Research Warnemünde)



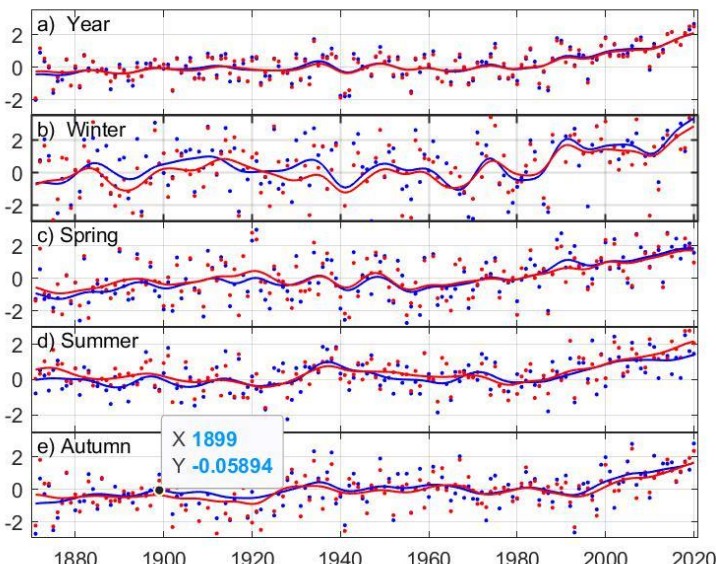


**Figure 7:** Annual and seasonal mean near-surface air temperature anomalies for the Baltic Sea basin for
1871−2020, taken from the CRUTEM4v dataset (Jones et al., 2012), compiled by Anna Rutgersson, Uppsala
University. Blue, red: Baltic Sea basin region north and south, respectively, of 60°N. Dots: individual years.
Smoothed curves: variability on timescales longer than 10 years.

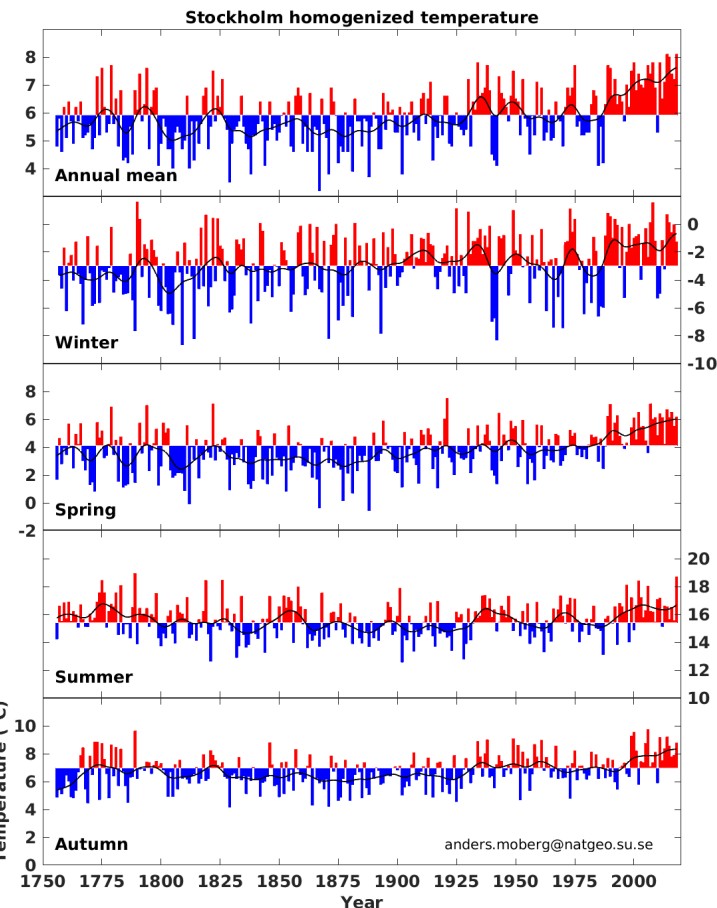


**Figure 8:** Homogenized annual and seasonal mean temperature in Stockholm during 1756-2018 measured at Bolin

Centre, Stockholm. Each colored bar show the annual mean temperature, in red or blue, depending on whether the

temperature is above or below the average during the reference period 1961-1990. The black curve represents

smoothed 10-year mean temperatures. (Source: https://bolin.su.se/data/stockholm-historical-temps-monthly)





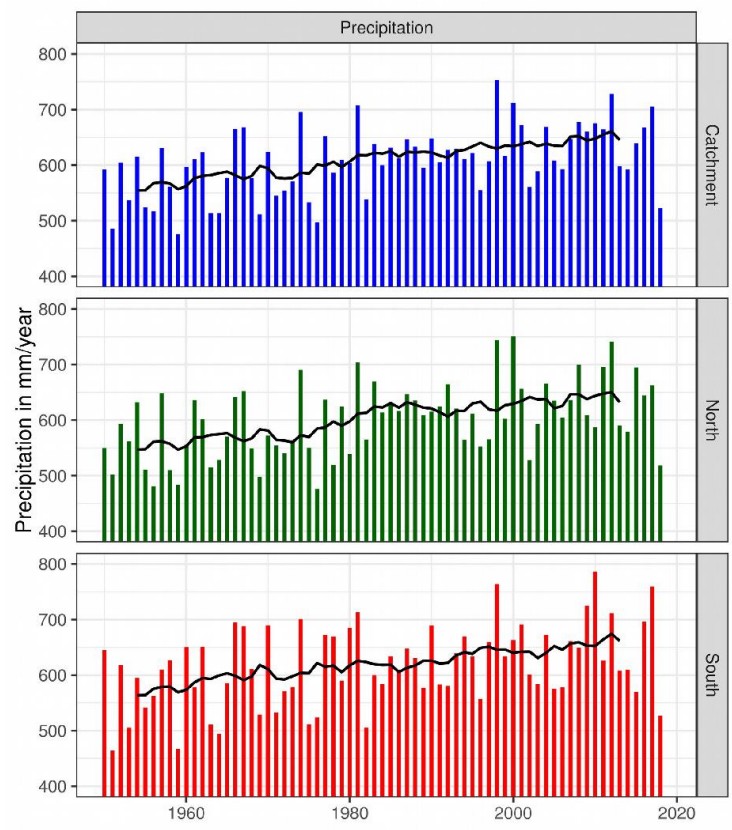

4149

**Figure 9:** Mean annual precipitation over land in mm year[-1] in the Baltic Sea catchment area during 1950-2018. Blue: whole catchment area, green: North of 59°N, Red: South of 59°N. Bars: annual sum, black: 10-year running mean. (Data source: http://surfobs.climate.copernicus.eu/dataaccess/access_eobs.php#datafiles, compiled by Madline Kniebusch, Leibniz Institute for Baltic Sea Research Warnemünde). Trends: 1.44 mm year[-1] (total), 1.51 mm year[-1] (North), 1.37 mm year[-1] (South), significant on 99% using the phase-scrambling method (Kniebusch et al., 2019b).

4156



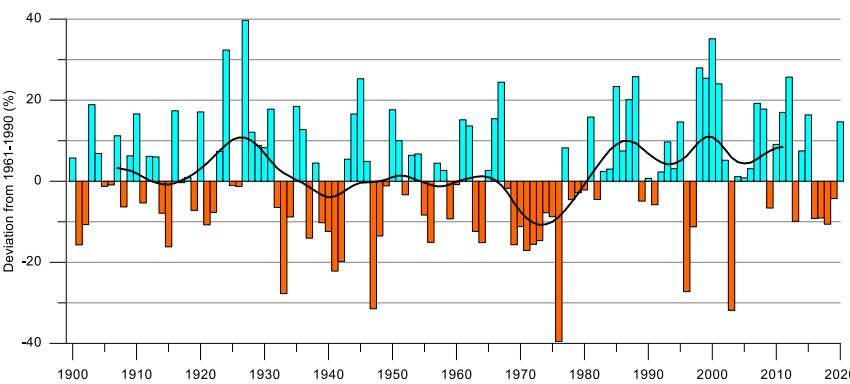

**Figure 10:** Area weighted river runoff anomalies relative to 1960-1990 (in %) from Sweden to the Baltic Sea. The black solid curve denotes Gaussian filtered data with a standard deviation of three years. (Source: Göran Lindström, Swedish Meteorological and Hydrological Institute).



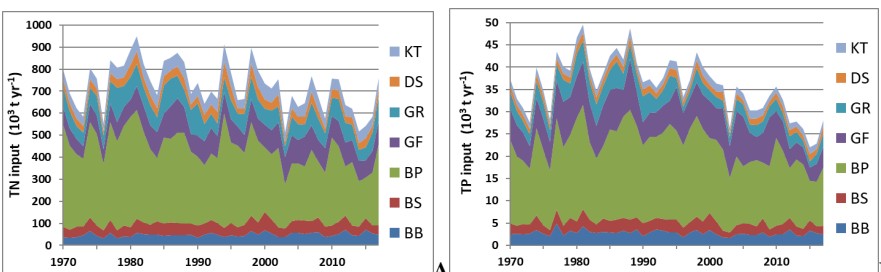

**Figure 11:** Long-term dynamics (1970–2017) of annual nitrogen (A) and phosphorus (B) land inputs to the major Baltic Sea basins: BB - Bothnian Bay; BS – Bothnian Sea; BP - Baltic proper; GF - Gulf of Finland; GR - Gulf of Riga; DS – Danish straits; KT – Kattegat. Time (years) is on the horizontal axis. (Source: O.P. Savchuk, Stockholm University)





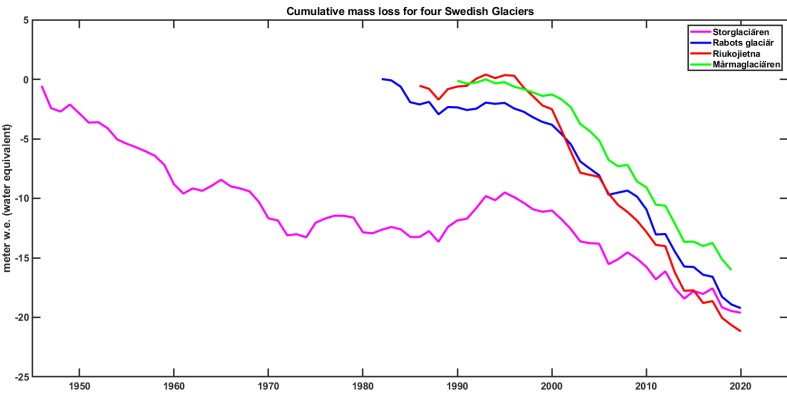

4168

**Figure 12:** Cumulative mass loss for four Swedish glaciers: Storglaciären (since 1946), Rabots glaciär (since 1982,
no data for 2004 and 2007 and hence interpolated), Riukojietna (since 1986, data for 2004 interpolated), och
Mårmaglaciär (since 1990, no data for 2020). Data are accessible from the SITES Data Portal,
https://data.fieldsites.se/portal/ (Swedish Infrastructure for Ecosystem Science, 2021a, c, b). (Source: Nina
Kirchner, Stockholm University)

4174


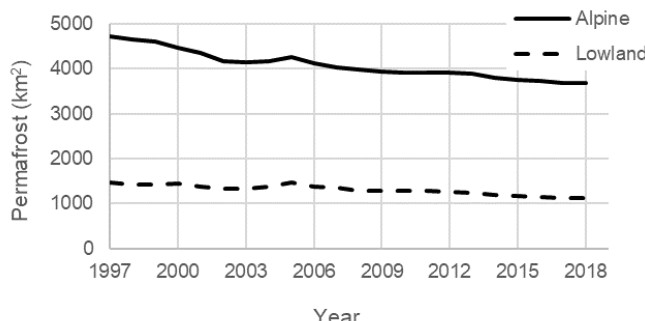

**Figure 13:** Modeled permafrost extent of alpine (> 700 m a.s.l.) and lowland permafrost for the years 1997-2018 in the Baltic Sea drainage basin. Permafrost data from (Obu et al., 2020), extent of catchment from (Hannerz and Destouni, 2006) and elevation data from USGS Global Multi-resolution Terrain Elevation Data 2010 (GMTED2010). Analyses performed at 1 km resolution in an equal area projection. (Source: Gustav Hugelius, Stockholm University)





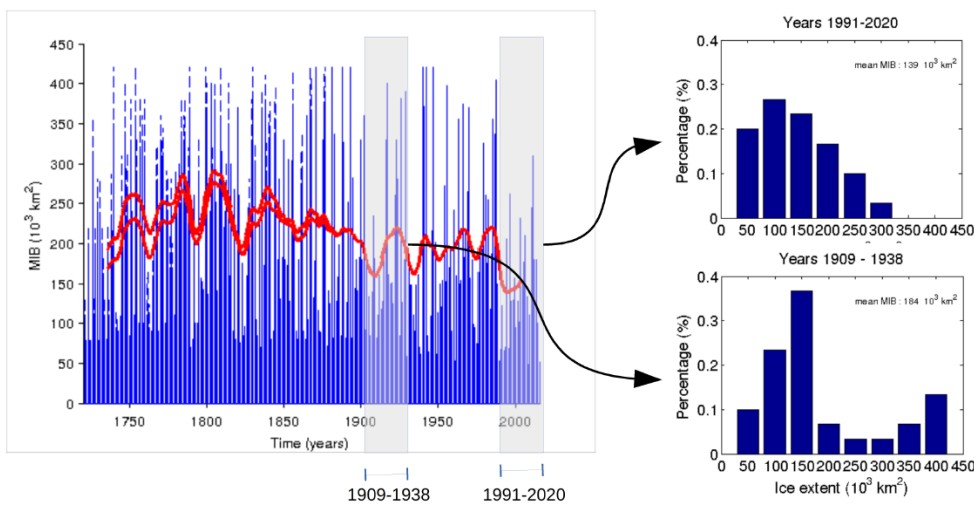

1909-1938    1991-2020

**Figure 14:** Left: Annual maximum sea ice extent of the Baltic Sea (MIB) in km$^2$ during 1720-2020. Blue bars: annual, red: 15-year running mean. Right: 30-year distribution functions of MIB during 1909-1938 and 1991-2020. (Data sources: https://www.eea.europa.eu/data-and-maps/daviz/maximum-extent-of-ice-cover-3#tab-chart_1, website Finnish Meteorological Institute: https://en.ilmatieteenlaitos.fi/ice-season-in-the-baltic-sea). (Source: Jari Haapala, Finnish Meteorological Institute)





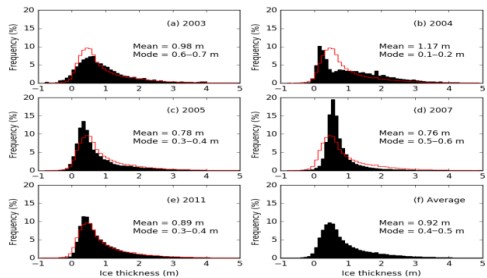

**Figure 15:** An average sea ice thickness distribution in the Bay of Bothnia. Statistics is based on helicopter electromagnetic measurements conducted in winters 2003, 2004, 2005, 2007 and 2011 (Source: Ronkainen et al., 2018).



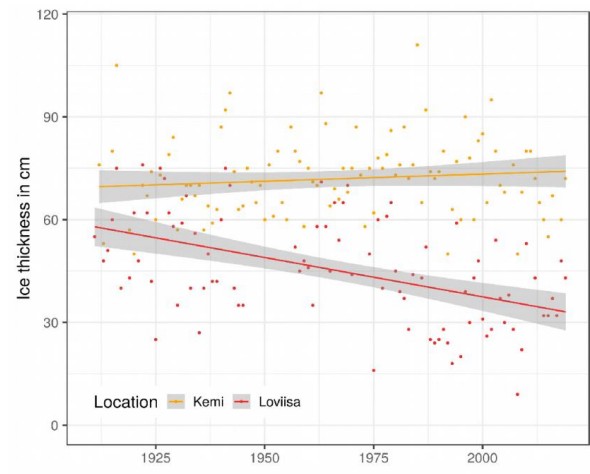


**Figure 16:** Level ice thickness at Kemi, Finland and Loviisa, Finland during 1912-2019. Points: annual mean

values, lines: linear trend with 95% confidence intervals (Data source: Jari Haapala, Finnish Meteorological

Institute).







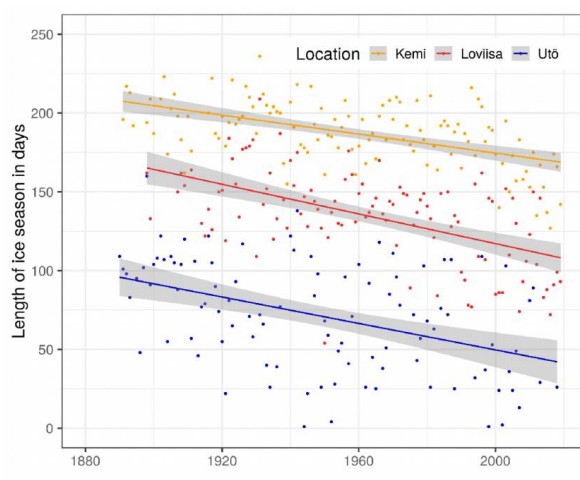


**Figure 17:** Length of the ice season in days at Kemi, Loviisa and Utö (Finland) during 1890-2019. Points: annual
mean, lines: linear trend with 95% confidence intervals (Data source: Jari Haapala, Finnish Meteorological
Institute).



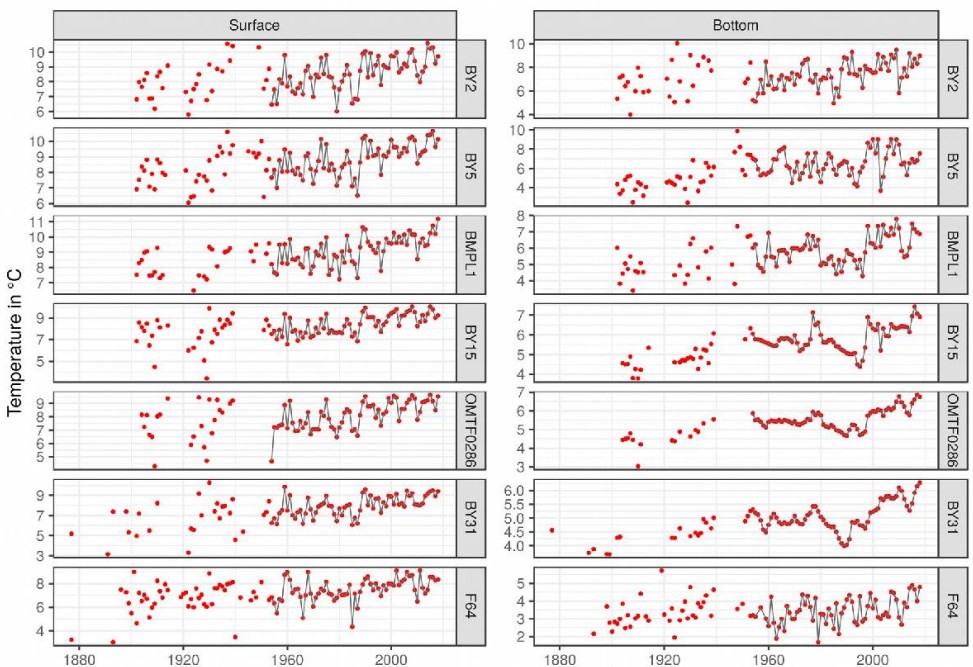


**Figure 18:** Annual mean values of de-seasonalized daily sea surface (left) and bottom (right) temperature (red points) at seven monitoring stations during 1877-2018. For the location of the stations see Figure 2. The grey lines show the period when every station has data for every year (1954-2018). For Figures 18, 19, 21 and 22, ICES data (https://ocean.ices.dk/HydChem/) for temperature and salinity (bottle data, i.e. from specific depths) were used. Post processing of the data was done following Radtke et al. (2020) in order to overcome possible seasonal biases due to missing values in the observations. Therefore, gaps were statistically filled using a GAMM model (general additive mixed models) taking the seasonality into account. (Source: Madline Kniebusch, Leibniz Institute for Baltic Sea Research Warnemünde)




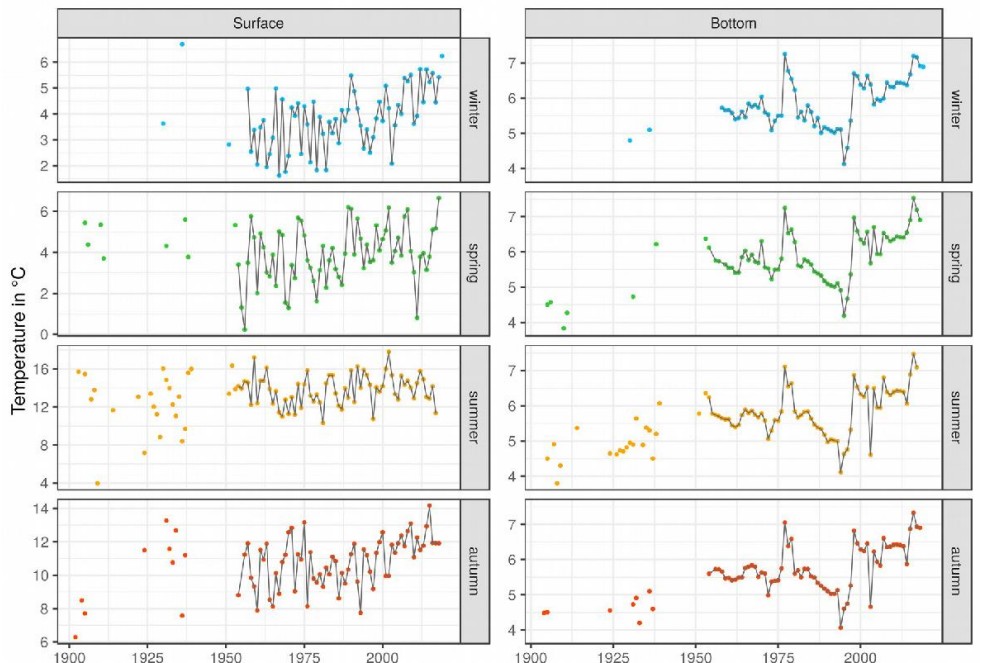


**Figure 19:** Seasonal mean sea surface and bottom temperature values during 1877-2018 at Gotland Deep (BY15).
Blue: winter, green: spring, yellow: summer and orange: autumn. The grey lines show the period when every
station has data for every year (1954-2018). (Source: Madline Kniebusch, Leibniz Institute for Baltic Sea Research
Warnemünde)



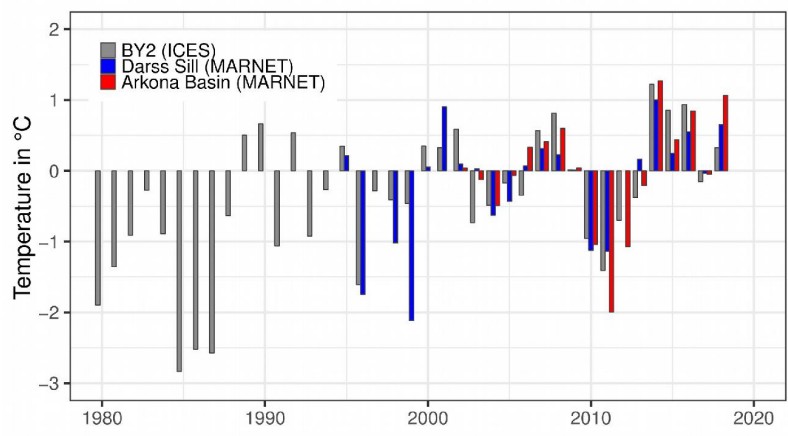


**Figure 20:** Annual sea surface temperature anomalies to the reference period 2002-2018 of de-seasonalized
measurements at BY2 and the MARNET stations Darss Sill and Arkona Basin during 1980-2018. (Source:
Madline Kniebusch, Leibniz Institute for Baltic Sea Research Warnemünde)









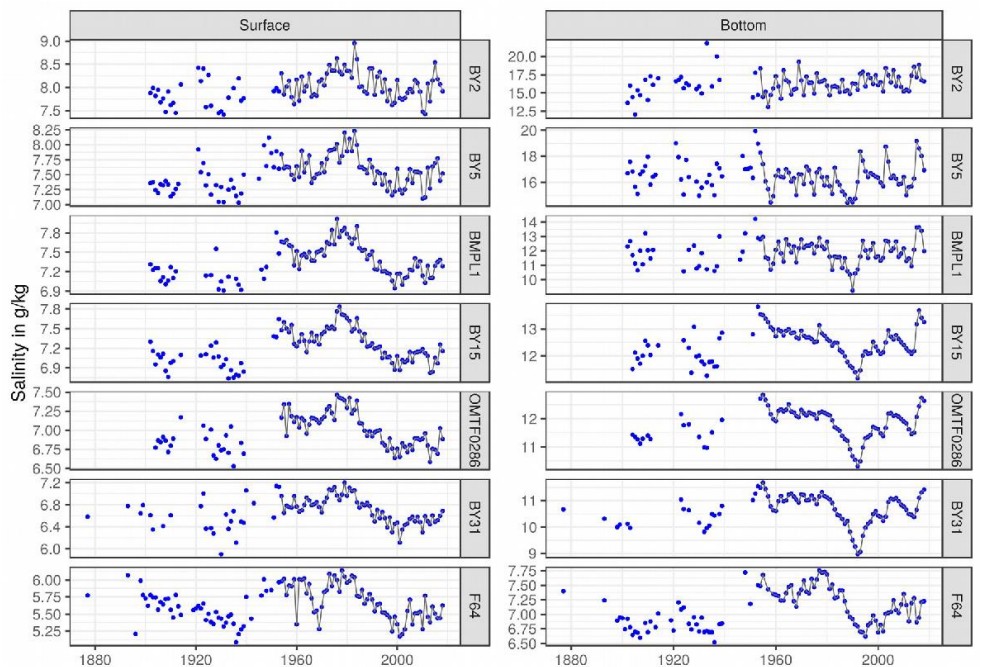


**Figure 21:** Annual mean values of de-seasonalized daily sea surface (left) and bottom (right) salinity (blue points) at seven important stations during 1877-2018. The grey lines show the period when every station has data for every year (1954-2018). (Source: Madline Kniebusch, Leibniz Institute for Baltic Sea Research Warnemünde)







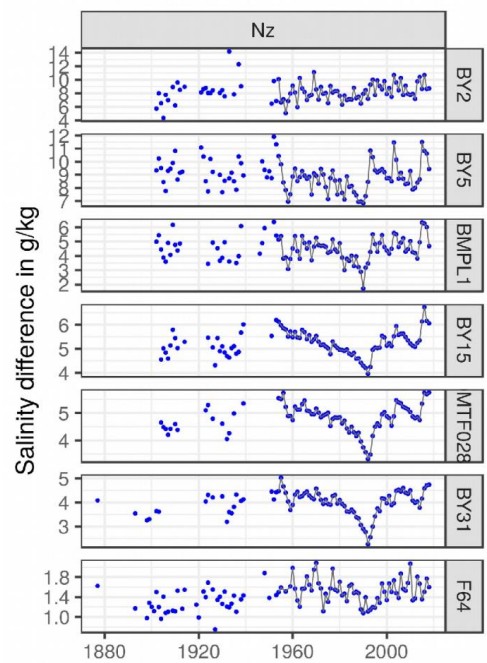


**Figure 22:** Difference between bottom and surface salinity as a measure for the vertical stratification (blue points) during 1877-2018. Only time steps when both values were available are considered. The grey lines show the period when every station has data for every year (1954-2018). (Source: Madline Kniebusch, Leibniz Institute for Baltic Sea Research Warnemünde)










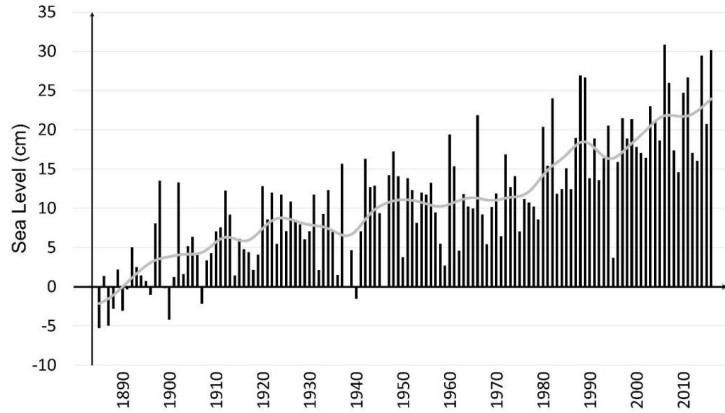


**Figure 23:** Annual mean sea level changes in centimeters for 14 Swedish mareographs since 1886. The data are
corrected for land uplift. The grey line shows a smoothed curve. (Source: Swedish Meteorological and
Hydrological Institute)



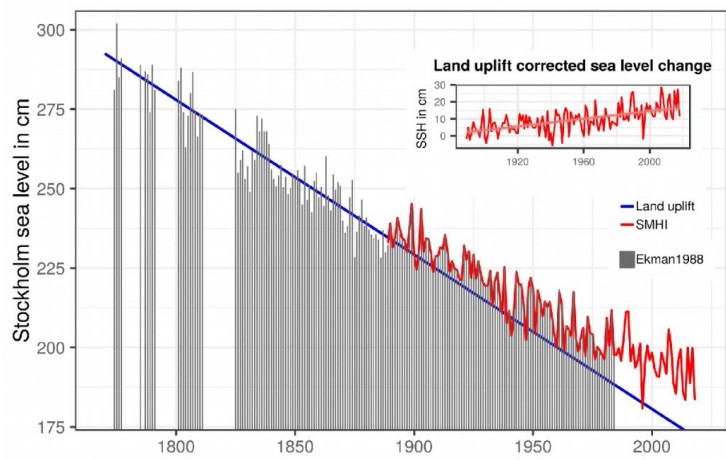


**Figure 24:** Annual mean sea level in Stockholm during 1774-2018. Grey bars: historic time series from Ekman
(1988), red: SMHI data (RH2000, 1889-2018), blue: trend computed for 1774-1884 (estimated land uplift: 4.9 mm
year[-1]) and extrapolated until 2018. The SMHI data has been bias corrected (mean difference during overlapping
time period) to make both time series comparable. Sea level rise of SMHI data corrected by estimated land uplift
amounts 1.13 mm year[-1] during 1889-2018. (Source: Madline Kniebusch, Leibniz Institute for Baltic Sea Research
Warnemünde)





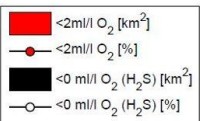

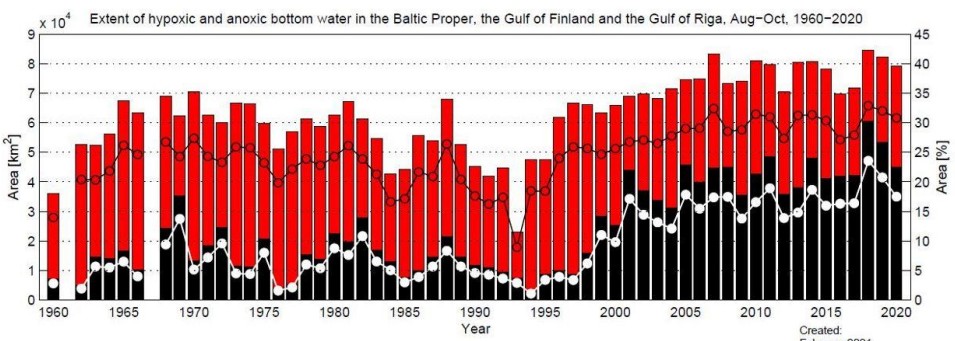


**Figure 25:** Extent of hypoxic (< 2 mL $O_2$ $L^{-1}$) and anoxic (< 0 mL $O_2$ $L^{-1}$) bottom water (in $km^2$) in the Baltic proper, Gulf of Finland and Gulf of Riga during cruise in August-October 1960-2020. (Source: Swedish Meteorological and Hydrological Institute)



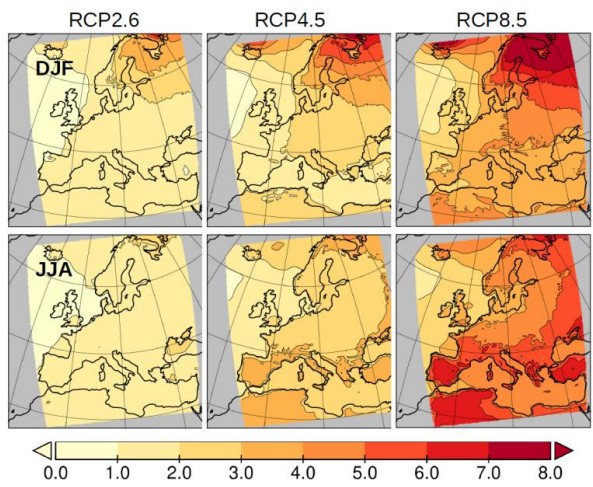


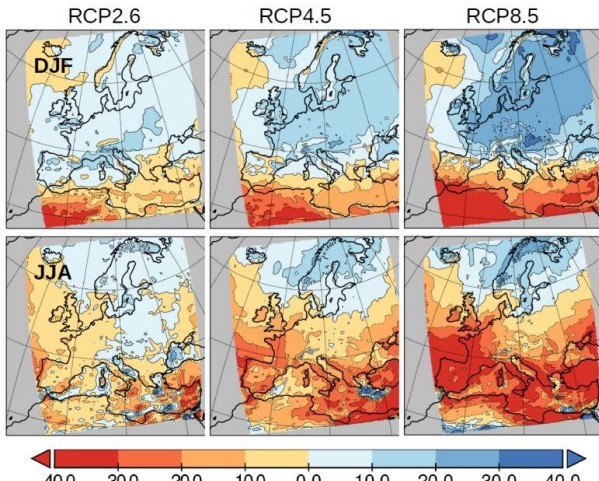


**Figure 26:** (a) Ensemble mean 2 m air temperature change (ºC) between 2070-2099 and 1970-1999 for winter
(December through January, upper panels) and summer (June through August, lower panels) under RCP2.6,
RCP4.5 and RCP8.5. (b) as (a) but for precipitation change (mm day$^{-1}$). Eight different Earth System Models are
used. (Data source: Gröger et al., 2021b)



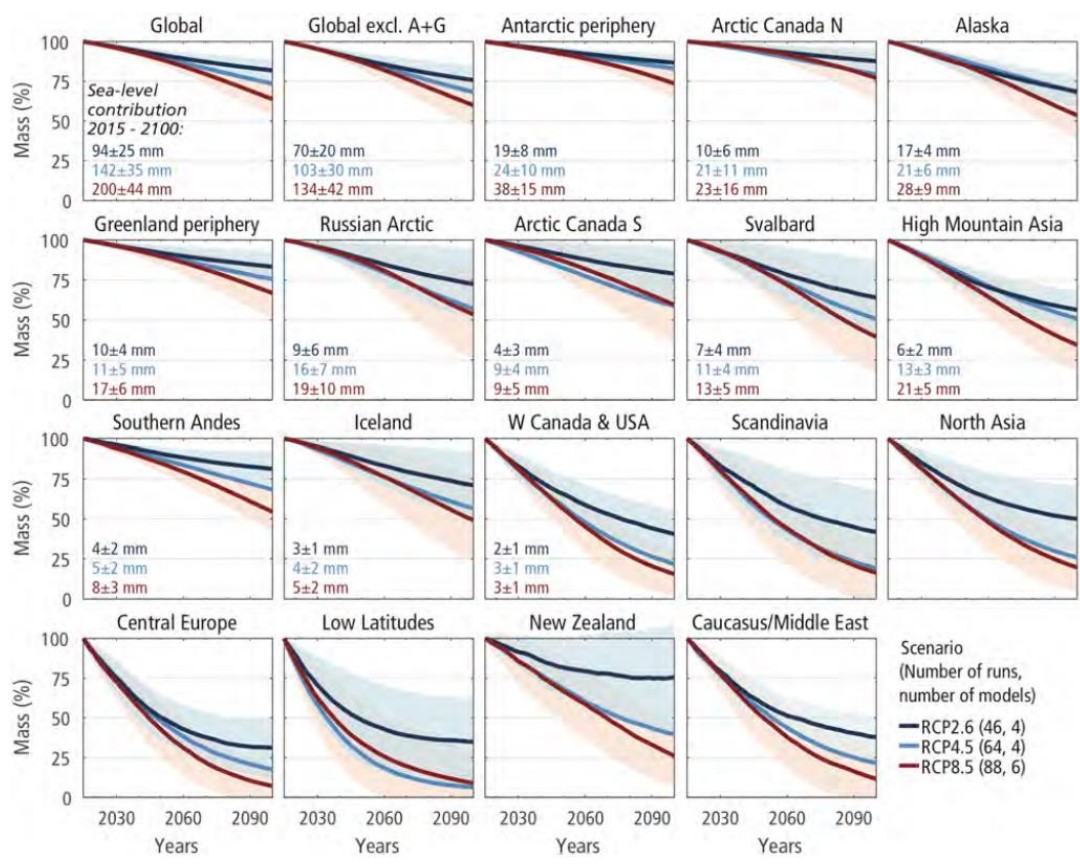

**Figure 27:** Mean projected glacier mass evolution between 2015 and 2100 relative to each region's glacier mass

in 2015 (in %) and ± 1 standard deviation under RCP2.6, RCP4.5 and RCP8.5. (Source: Hock et al., 2019)

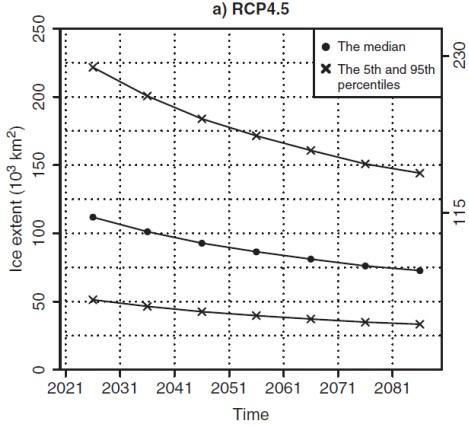

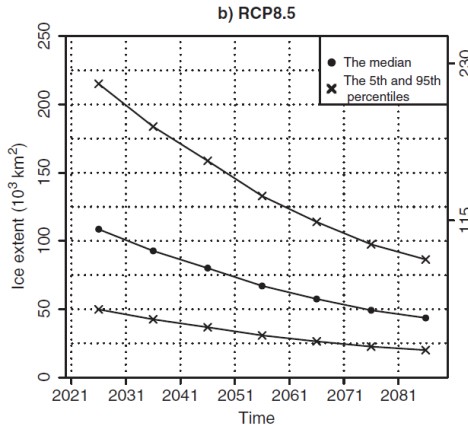


**Figure 28:** Median, 5th and 95th percentiles of the annual maximum ice extent of the Baltic Sea estimated from 28 CMIP5 models. The vertical axis shows upper class limits for mild and average ice winters. (a) RCP4.5, (b) RCP8.5. (Source: Luomaranta et al., 2014)




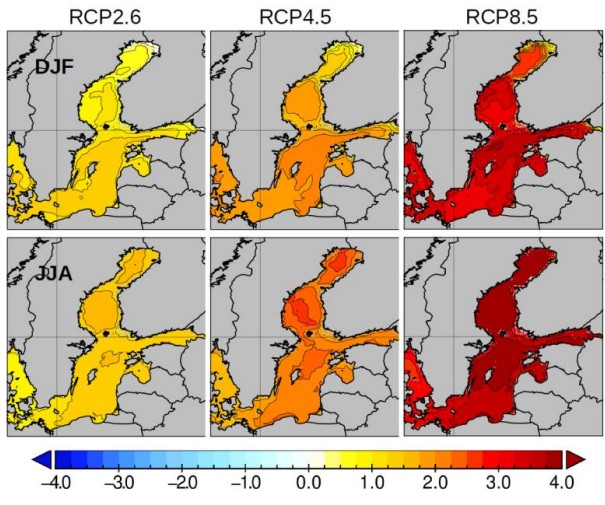



**Figure 29:** (a) Ensemble mean sea surface temperature change (°C) between 2070-2099 and 1970-1999 for winter
(December through January, upper panels) and summer (June through August, lower panels) under RCP2.6,
RCP4.5 and RCP8.5. (b) as (a) but for the standard deviation of the change, i.e. the ensemble spread (°C). Eight
different Earth System Models are used. (Source: Gröger et al., 2019)



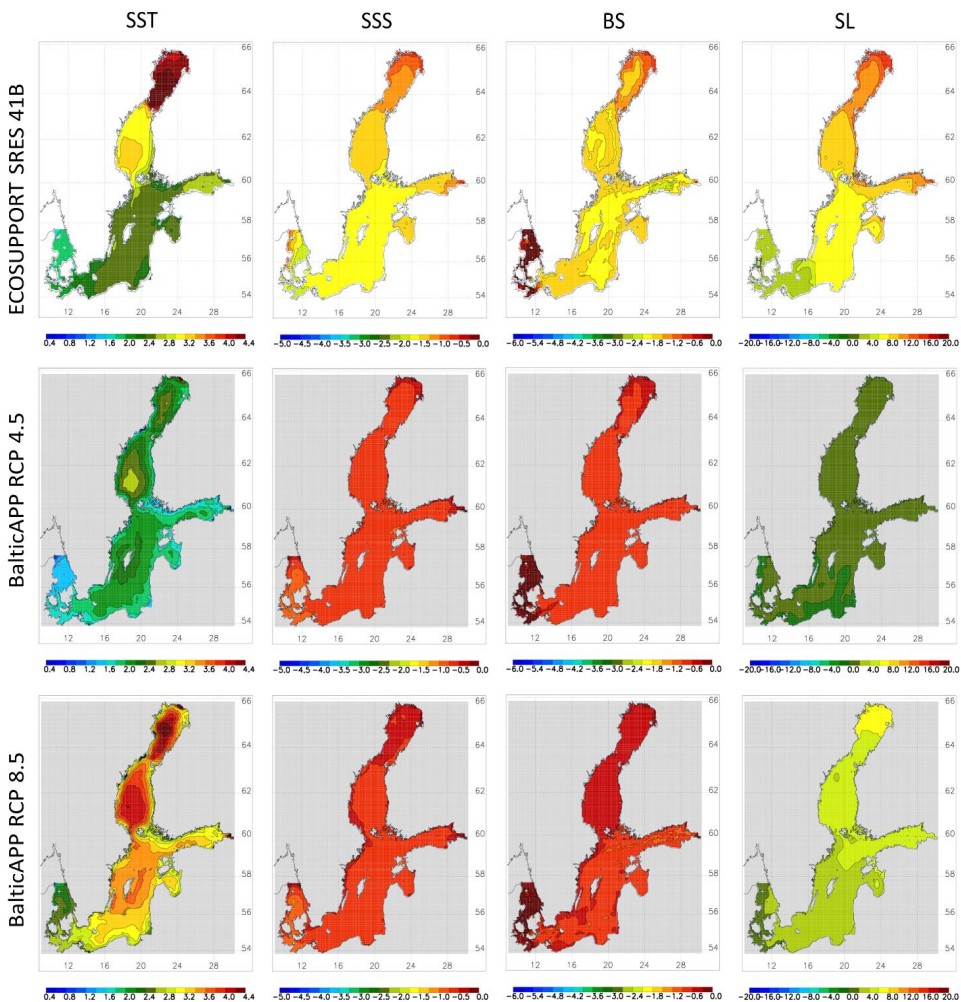


**Figure 30:** From left to right changes of summer (June – August) mean sea surface temperature (SST; °C), annual
mean sea surface salinity (SSS; g kg⁻¹), annual mean bottom salinity (BS; g kg⁻¹), and winter (December –
February) mean sea level (SL; cm) between 1978-2007 and 2069-2098 are shown. From top to bottom results of
the ensembles by Meier et al. (2011a) under the A1B/A2 greenhouse gas emission scenario (white background),
and by Saraiva et al. (2019b), RCP 4.5 (grey background) and RCP 8.5 (grey background) are depicted.


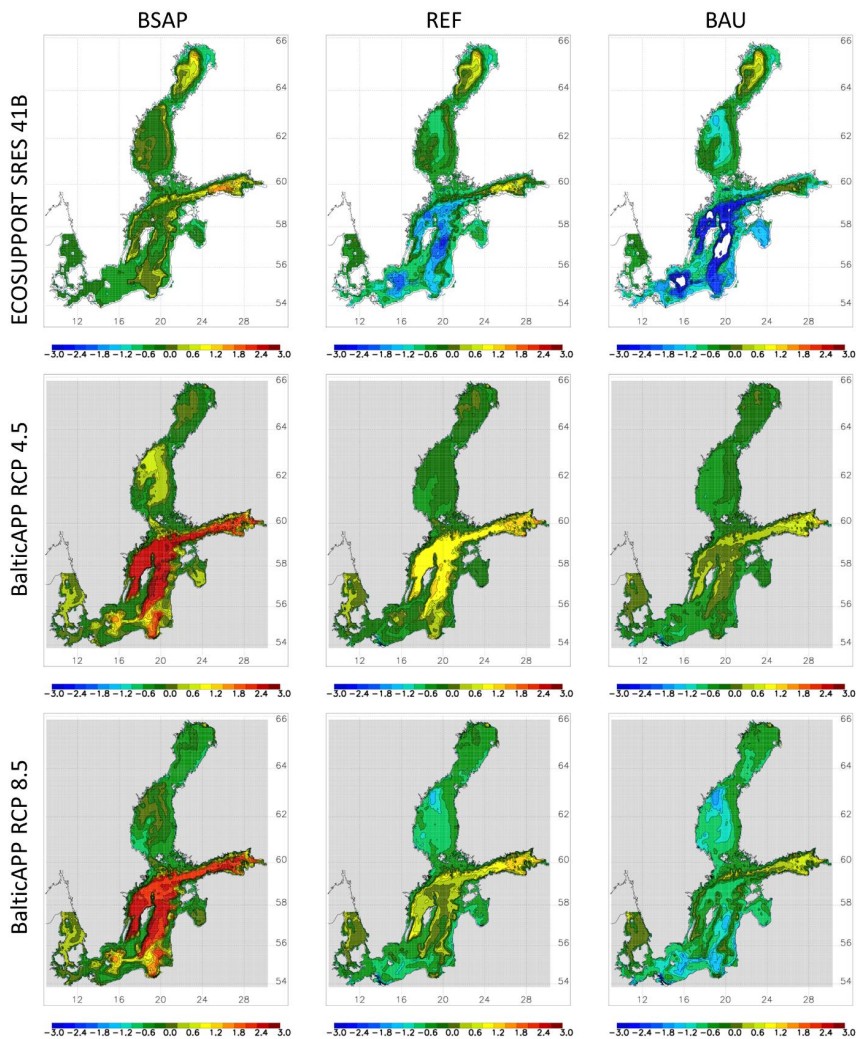

**Figure 31:** Ensemble mean summer (June – August) bottom dissolved oxygen concentration changes (mL L$^{-1}$)
between 1978-2007 and 2069-2098. From left to right results of the nutrient load scenarios Baltic Sea Action Plan
(BSAP), Reference (REF) and Business-As-Usual (BAU) are shown. From top to bottom results of the ensembles
by Meier et al. (2011a) under the A1B/A2 greenhouse gas emission scenario (white background), and by Saraiva
et al. (2019b), RCP 4.5 (grey background) and RCP 8.5 (grey background) are depicted.

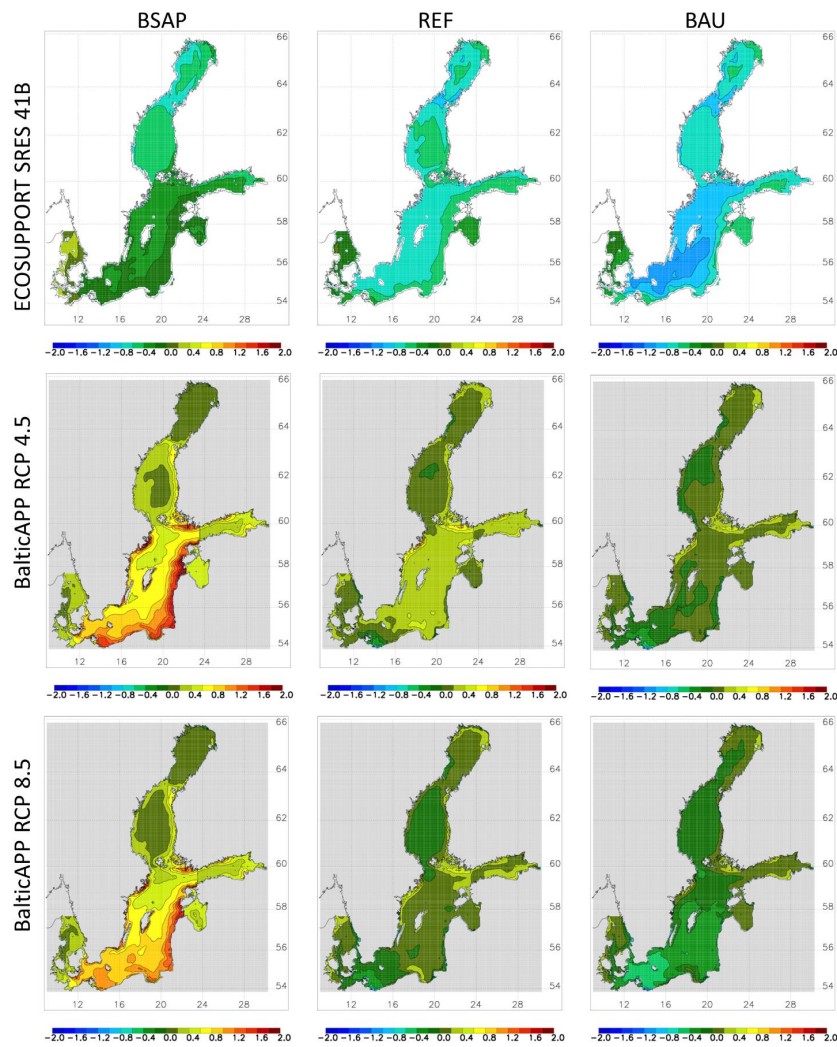

**Figure 32:** As Figure 30 but for annual mean Secchi depth changes (m). Secchi depth changes indicate changes in water transparency caused by phytoplankton and detritus concentration changes.





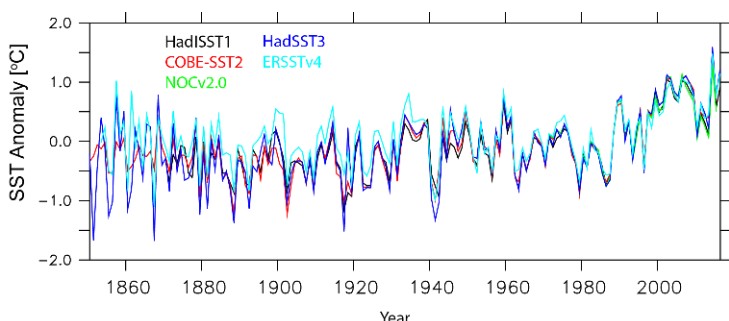

4294

**Figure 33:** Sea surface temperature anomaly in the Greater North Sea region from 1870 to 2016 (relative to the mean 1971 to 2000), according to different data sets. (Huthnance et al., 2016), updated by Elizabeth Kent, Southampton)

4298





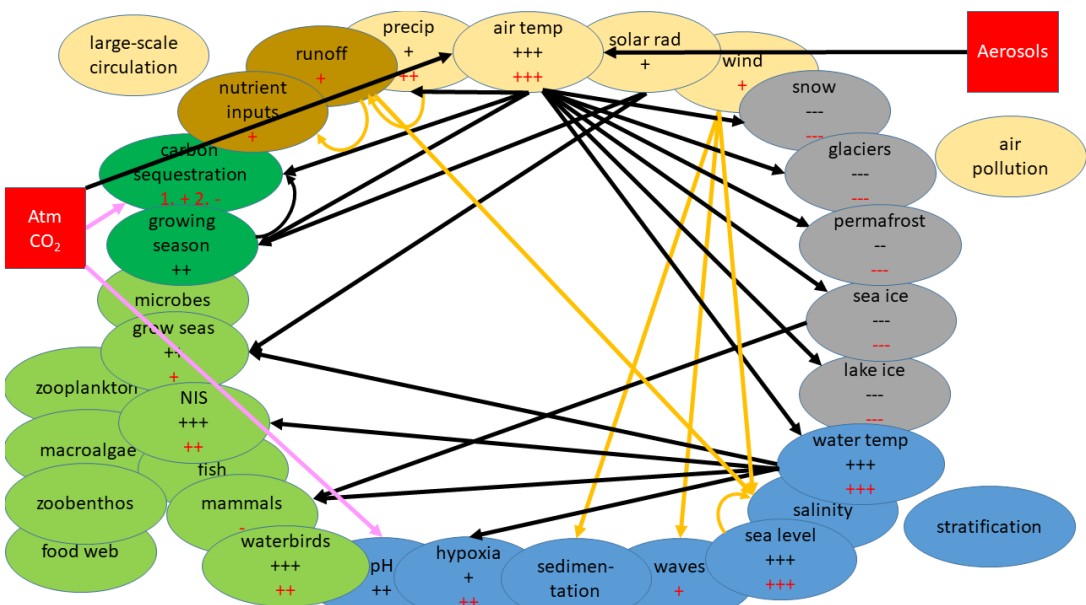

**Figure 34:** Synthesis of the knowledge on present and future climate changes. Shown are the anthropogenic climate changes in 33 Earth system variables (bubbles) of the atmosphere (yellow), land surface (brown), terrestrial biosphere (dark green), cryosphere (grey), ocean and sediment (blue), and marine biosphere (light green). The sign of a change (plus/minus) is shown together with the level of confidence denoted by the number of signs, i.e. one to three signs correspond to low, medium and high confidence levels, as the result of the literature assessment reflecting consensus and evidence following the IPCC definitions (Section 2.3). Sign colours indicate the direction of past (black) and future (red) changes following Table 10. Uncertain changes (+/-) are not displayed. Investigated external anthropogenic drivers of the Earth system are shown as red squares, i.e. greenhouse gases, in particular CO$_2$, and aerosol emissions. Climate change attribution relationships with sufficiently high confidence are shown by arrows (black: heat cycle, orange: water cycle, pink: carbon cycle). Projections of carbon sequestration of Arctic terrestrial ecosystems for the 21$^{st}$ century showed first increased uptake and later a carbon source (Section 3.3.3), denoted by 1. + 2. -.






**Tables**

**Table 1:** Variables of this assessment and further reference (1: Lehmann et al., 2021; 2: Kuliński et al., 2021; 3:
Rutgersson et al., 2021; 4: Weisse et al., 2021; 5: Reckermann et al., 2021; 6: Gröger et al., 2021a; 7: Christensen
et al., 2021; 8: Meier et al., 2021a; 9: Viitasalo, 2021)

| Number | Variable | Past and present climates | | Future climate | |
|---|---|---|---|---|---|
| Atmosphere | | | | | |
| 1 | Large-scale circulation | 3.2.1.1 | 3 | 3.3.1.1 | 3, 7 |
| 2 | Air temperature | 3.1.2, 3.1.3, 3.1.4 | | 3.3.1.2 | 7 |
| | Warm spell | 3.2.1.2 | 3 | | 3 |
| | Cold spell | | 3 | | 3 |
| 3 | Solar radiation and cloudiness | 3.2.1.3 | | 3.3.1.3 | 7 |
| 4 | Precipitation | 3.1.2, 3.1.3, 3.1.4 | | 3.3.1.4 | 7 |
| | Heavy precipitation | 3.2.1.4 | 3 | | 3 |
| | Drought | | 3 | | 3 |
| 5 | Wind | 3.2.1.5 | | 3.3.1.5 | 7 |
| | Storm | | 3 | | 3 |
| 6 | Air pollution, air quality and atmospheric deposition | 3.2.1.6 | | 3.3.1.6 | |
| Land | | | | | |
| 7 | River discharge | 3.2.2.1 | | 3.3.2.1 | 8 |
| | High flow | | 3 | | 3 |
| 8 | Land nutrient inputs | 3.2.2.2 | | 3.3.2.2 | 8 |
| Terrestrial biosphere | | | | | |
| 9 | Land cover (forest, crops, grassland, peatland, mires) | 3.2.3 | 6 | 3.3.3 | |
| 10 | Carbon sequestration | | | 3.3.3 | |
| Cryosphere | | | | | |
| 11 | Snow | 3.2.4.1 | | 3.3.4.1 | 7 |
| | Sea-effect snowfall | | 3 | | 3 |
| 12 | Glaciers | 3.2.4.2 | | 3.3.4.2 | |





| 13 | Permafrost | 3.2.4.3 | | 3.3.4.3 | |
|----|------------|---------|---|---------|---|
| 14 | Sea ice | 3.2.4.4 | | 3.3.4.4 | 8 |
| | Extreme mild winter | | 3 | | 3 |
| | Severe winter | | 3 | | 3 |
| | Ice ridging | | 3 | | 3 |
| 15 | Lake ice | 3.2.4.5 | | 3.3.4.5 | |
| | Ocean and marine sediments | | | | |
| 16 | Water temperature | 3.2.5.1 | | 3.3.5.1 | 8 |
| | Marine heat wave | | 3 | | 3 |
| 17 | Salinity and saltwater inflows | 3.2.5.2 | 1 | 3.3.5.2 | 8 |
| 18 | Stratification and overturning circulation | 3.2.5.3 | 1 | 3.3.5.3 | 8 |
| 19 | Sea level | 3.2.5.4 | 4 | 3.3.5.4 | 8 |
| | Sea level extreme | | 3 | | 3 |
| 20 | Waves | 3.2.5.5 | 4 | 3.3.5.5 | |
| | Extreme waves | | 3 | | 3 |
| 21 | Sedimentation and coastal erosion | 3.2.5.6 | 4 | 3.3.5.6 | |
| 22 | Oxygen and nutrients | 3.1.4 3.2.5.7.1 | 2 | 3.3.5.7.1 | 8 |
| 23 | Marine $CO_2$ system | 3.2.5.7.2 | 2 | 3.3.5.7.2 | |
| | Marine biosphere | | | | |
| 24 | Pelagic habitats: Microbial communities | 3.2.6.1.1 | 2, 9 | 3.3.6.1.1 | 9 |
| 25 | Pelagic habitats: Phytoplankton and cyanobacteria | 3.2.6.1.2 | 2, 3, 9 | 3.3.6.1.2 | 3, 9 |
| 26 | Pelagic habitats: Zooplankton | 3.2.6.1.3 | 9 | 3.3.6.1.3 | 9 |
| 27 | Benthic habitats: Macroalgae and vascular plants | 3.2.6.2.1 | 9 | 3.3.6.2.1 | 9 |
| 28 | Benthic habitats: Zoobenthos | 3.2.6.2.2 | 9 | 3.3.6.2.2 | 9 |
| 29 | Non-indigenous species | 3.2.6.3 | 9 | 3.3.6.3 | 9 |
| 30 | Fish | 3.2.6.4 | 9 | 3.3.6.4 | 9 |





| 31 | Marine mammals | 3.2.6.5 | 9 | 3.3.6.5 | 9 |
| 32 | Waterbirds | 3.2.6.6 | 9 | 3.3.6.6 | 9 |
| 33 | Marine food web | 3.2.6.7 | 9 | 3.3.6.7 | 9 |






**Table 2:** Comparison of the Baltic Sea with other intra-continental seas and large lakes.

| Basin | Area | Mean depth | Mean salinity | Fresh water budget | Ice cover on average | Location Centre |
|---|---|---|---|---|---|---|
| Unit | $10^3$ km$^2$ | m | g kg$^{-1}$ | | | Lat Long |


| Basin | Area | Mean depth | Mean salinity | Fresh water budget | Ice cover on average | Location Centre |
|---|---|---|---|---|---|---|
| Baltic Sea | 393 | 54 | 7½ | + | Half | 60°N 20°E |
| Black Sea | 436 | 1197 | 20 | + | Northeast | 43°N 35°E |
| Gulf of Ob | 41 | 12 | 5 | + | All | 73°N 74°E |
| Chesapeake Bay | 12 | 6 | 15 | + | Shores | 38°N 76°W |
| Hudson Bay | 1232 | 128 | 30 | + | All | 58°N 85°W |
| Red Sea | 438 | 491 | 40 | – | None | 22°N 38°E |
| Persian Gulf | 239 | 25 | 40 | – | None | 27°N 52°E |
| Caspian Sea | 374 | 211 | 12 | 0 | North | 43°N 50°E |
| Lake Superior | 82 | 149 | < 0.1 | 0 | All | 48°N 88°W |




**Table 3:** The main characteristics of the physical features of the European Seas (Leppäranta and Myrberg, 2009;
Sündermann and Pohlmann, 2011; www.ospar.org; British Oceanographic Data Centre). Greater North Sea –
being the neighbouring sea area to the Baltic Sea - is shown as a sub-region of the NE-Atlantic, but other European
sub-regions are not listed (from Myrberg et al., 2019).

| Basin | Area $10^3$ km² | Mean depth m | Mean salinity g kg⁻¹ | Fresh water budget | Ice cover on average | Tides | Water residence time (years) |
|---|---|---|---|---|---|---|---|
| | | | | | | | |
| Baltic Sea | 393 | 54 | 7.4 | Pos. | 37 % [1] | Weak | 40 |
| Black Sea | 422 | 1 200 | 18 | Pos. | Northeast only | Weak | 3 000 |
| Greater North Sea | 750 | 80 | 34–35 | Pos. | No | Strong | Not applicable |
| Mediterranean Sea | 2 970 | 1 500 | 38 | Neg. | No | Weak/ Moderate | 80-100 |
| NE Atlantic shelf | 13 500 [2] | 1 500 | 34–35 | Not applicable | No | Strong | Not applicable |
| 1) Mean maximal ice cover between 2000-2017, see Fig. 3.2.4.4.1 | | | | | | | |
| 2) defined as the OSPAR convention area, incl. the Greater North Sea | | | | | | | |






**Table 4:** Linear surface air temperature trends (K decade$^{-1}$) for the period 1876−2020 over the northern (>60°N) and southern (<60°N) Baltic Sea basin (1878−2020 is selected for comparison with (Rutgersson et al., 2014), with an equally long time period). Bold: significance at $p < 0.05$. Data from the updated CRUTEM4v dataset (Jones et al., 2012).

|  | Annual | Winter | Spring | Summer | Autumn |
|---|---|---|---|---|---|
| North | **0.10** | **0.11** | **0.14** | **0.08** | **0.08** |
| South | **0.10** | **0.13** | **0.10** | **0.09** | **0.09** |



**Table 5:** Average (2013-2017) riverine and coastal nutrient inputs ($10^3$ t N (P) yr$^{-1}$) to the major basins of the
Baltic Sea (Source: Oleg P. Savchuk, Stockholm University). For abbreviated basin names see Figure 11.

|  | BB | BS | BP | GF | GR | DS | KT | Entire BS |
|---|---|---|---|---|---|---|---|---|
| TN river | 48 | 40 | 251 | 85 | 73 | 32 | 46 | 575 |
| TN coast | 3.6 | 4.3 | 6.5 | 9.3 | 0.5 | 2.8 | 2.0 | 29 |
| TP river | 2.4 | 1.7 | 11.8 | 3.4 | 2.0 | 1.1 | 1.3 | 24.0 |
| TP coast | 0.1 | 0.2 | 0.5 | 0.4 | 0.1 | 0.2 | 0.1 | 1.6 |






**Table 6**: Mass balances for the Swedish glaciers Storglaciären, Rabots glaciär, Mårmaglaciär, and Riuokjietna.
General references are given as footnotes in connection with balance years and long-term monitoring intervals,
respectively. Selected specific references are given as footnotes in connection with the glacier names, and include
also neighboring glaciers Kårsa glacier and Kebnepakteglaciär. (Source: Nina Kirchner, Stockholm University)

| | Recent mass balance years. Gains and losses in mm w.e. (millimeter water equivalent). Note that the unit mm w.e. is interchangeable with the unit kg m$^{-2}$ | | | Long-term mass balance, losses per year in mm w.e. | |
|---|---|---|---|---|---|
| | 2015/ 2016[3] | 2016/ 2017[3] | 2017/ 2018[3] | 1980-2010[4] | 1985-2015[5] |
| Storglaciären[6,7,8] | -240 | +470 | -1600 | -113 | -153 |
| Rabots glaciär[9,10] | -650 | -170 | -1680 | -394 | -465 |
| Mårmarglaciär | -370 | +260 | | -430 | -460 |
| Riuokjietna | -1060 | +150 | | -592 | -592 |


---

[3] World Glacier Monitoring Service, 2020

[4] Blunden and Arendt, 2015

[5] Hartfield et al., 2018

[6] Mercer, 2016

[7] Holmlund and Holmlund, 2019

[8] Kirchner et al., 2019

[9] Brugger and Pankratz, 2015

[10] Williams et al., 2016





**Table 7**: Air temperature ($T_{2m}$) changes (°C) between 1976 - 2005 and 2069 - 2098 averaged over each season and
annual mean over the Baltic Sea catchment area and over the Baltic Sea calculated from nine regionalized ESMs
(Data source: Gröger et al., 2021b, compiled by Christian Dieterich, Swedish Meteorological and Hydrological
Institute). In addition to the ensemble mean change, the 5th and 95th percentiles indicating the ensemble spread are
listed (in brackets).

|  | Annual | Winter | Spring | Summer | Autumn |
|---|---|---|---|---|---|
| Total land |  |  |  |  |  |
| RCP2.6 | 1.5 (1.2, 2.0) | 2.1 (1.5, 3.3) | 1.5 (1.2, 2.0) | 1.3 (0.8, 2.1) | 1.3 (0.9, 1.8) |
| RCP4.5 | 2.6 (1.6, 3.2) | 3.2 (2.1, 4.2) | 2.4 (1.5, 3.3) | 2.1 (1.3, 3.1) | 2.3 (1.4, 2.8) |
| RCP8.5 | 4.3 (3.5, 5.2) | 5.0 (3.4, 6.3) | 3.8 (3.1, 4.5) | 3.7 (2.5, 5.0) | 3.8 (2.6, 4.8) |
| Land north of 60ºN |  |  |  |  |  |
| RCP2.6 | 1.7 (1.4, 2.4) | 2.5 (1.9, 3.1) | 1.7 (1.2, 2.3) | 1.4 (0.8, 2.3) | 1.5 (1.1, 2.1) |
| RCP4.5 | 2.9 (2.0, 3.7) | 4.0 (2.9, 5.0) | 2.8 (1.8, 3.8) | 2.3 (1.3, 3.4) | 2.5 (1.7, 3.2) |
| RCP8.5 | 4.9 (3.9, 5.9) | 6.0 (4.2, 7.5) | 4.2 (3.5, 5.1) | 3.9 (2.8, 5.1) | 4.2 (2.9, 5.3) |
| Land south of 60ºN |  |  |  |  |  |
| RCP2.6 | 1.4 (1.0, 1.8) | 1.7 (1.1, 3.4) | 1.3 (0.9, 1.7) | 1.3 (0.9, 1.9) | 1.2 (0.7, 1.6) |
| RCP4.5 | 2.2 (1.3, 2.8) | 2.6 (1.5, 4.0) | 2.2 (1.3, 2.9) | 2.0 (1.1, 3.0) | 2.1 (1.2, 2.7) |
| RCP8.5 | 3.9 (3.2, 4.7) | 4.2 (2.9, 5.7) | 3.4 (2.9, 4.0) | 3.5 (2.2, 4.9) | 3.5 (2.3, 4.5) |
| Baltic Sea |  |  |  |  |  |
| RCP2.6 | 1.4 (1.2, 1.9) | 1.9 (1.3, 2.8) | 1.5 (1.1, 1.9) | 1.2 (0.6, 1.8) | 1.2 (0.9, 1.7) |
| RCP4.5 | 2.4 (1.4, 2.9) | 2.9 (1.8, 3.7) | 2.5 (1.5, 3.1) | 2.0 (1.2, 2.7) | 2.1 (1.2, 2.7) |
| RCP8.5 | 3.9 (3.1, 4.8) | 4.6 (3.2, 5.8) | 3.9 (3.0, 4.9) | 3.5 (2.4, 4.6) | 3.6 (2.6, 4.6) |




**Table 8**: Relative precipitation changes (%) between 1976 - 2005 and 2069 - 2098 averaged over each season and annual mean over the Baltic Sea catchment area and over the Baltic Sea calculated from nine regionalized ESMs (Data source: Gröger et al., 2021b, compiled by Christian Dieterich, Swedish Meteorological and Hydrological Institute). In addition to the ensemble mean change, the 5[th] and 95[th] percentiles indicating the ensemble spread are listed (in brackets).

| | Annual | Winter | Spring | Summer | Autumn |
|---|---|---|---|---|---|
| Total land | | | | | |
| RCP2.6 | 5 (2, 14) | 7 (1, 22) | 8 (2, 12) | 3 (-2, 13) | 4 (-4, 12) |
| RCP4.5 | 9 (6, 14) | 12 (4, 24) | 13 (8, 17) | 4 (1, 11) | 6 (-5, 12) |
| RCP8.5 | 15 (11, 22) | 22 (11, 38) | 20 (7, 26) | 5 (-4, 15) | 13 (-1, 18) |
| Land north of 60ºN | | | | | |
| RCP2.6 | 6 (2, 15) | 7 (2, 23) | 8 (0, 13) | 5 (1, 17) | 5 (-5, 14) |
| RCP4.5 | 11 (7, 18) | 13 (6, 27) | 15 (2, 21) | 9 (4, 14) | 8 (-3, 17) |
| RCP8.5 | 19 (12, 30) | 22 (12, 41) | 24 (7, 35) | 13 (-1, 30) | 17 (1, 26) |
| Land south of 60ºN | | | | | |
| RCP2.6 | 5 (0, 13) | 7 (-1, 22) | 7 (3, 13) | 2 (-5, 10) | 3 (-7, 11) |
| RCP4.5 | 7 (4, 11) | 12 (1, 22) | 12 (6, 20) | 1 (-5, 11) | 4 (-8, 11) |
| RCP8.5 | 12 (8, 18) | 21 (9, 35) | 18 (7, 26) | -1 (-14, 9) | 9 (-3, 17) |
| Baltic Sea | | | | | |
| RCP2.6 | 6 (0, 15) | 5 (-3, 15) | 4 (-1, 8) | 8 (0, 22) | 5 (-3, 13) |
| RCP4.5 | 8 (3, 13) | 9 (-4, 20) | 11 (1, 17) | 6 (-1, 16) | 6 (-3, 15) |
| RCP8.5 | 16 (8, 23) | 18 (3, 31) | 19 (-3, 32) | 10 (-9, 22) | 15 (4, 26) |





**Table 9**: Sea surface temperature (SST) changes (°C) between 1976 - 2005 and 2069 - 2098 averaged over each season and annual mean over the Baltic Sea calculated from nine regionalized ESMs (Data source: Gröger et al., 2021b, compiled by Christian Dieterich, Swedish Meteorological and Hydrological Institute). In addition to the ensemble mean change, the 5[th] and 95[th] percentiles indicating the ensemble spread are listed (in brackets).

| Baltic Sea | Annual | Winter | Spring | Summer | Autumn |
|---|---|---|---|---|---|
| RCP2.6 | 1.1 (0.8, 1.6) | 1.0 (0.9, 1.4) | 1.1 (0.9, 1.6) | 1.2 (0.6, 1.7) | 0.9 (0.7, 1.6) |
| RCP4.5 | 1.8 (1.1, 2.5) | 1.7 (1.0, 2.3) | 1.9 (1.2, 2.6) | 2.0 (1.2, 2.6) | 1.8 (1.1, 2.4) |
| RCP8.5 | 3.2 (2.5, 4.1) | 3.0 (2.3, 3.8) | 3.2 (2.5, 3.9) | 3.4 (2.4, 4.5) | 3.1 (2.4, 4.1) |





**Table 10:** Summary of key messages about the impact of global warming on selected variables. The sign of a change (plus/minus) is listed together with the level of confidence denoted by the number of signs, i.e. one to three signs correspond to low, medium and high confidence levels. +/- means no detected or projected change due to climate change. Key messages of this assessment that are new compared to the previous assessment by BACC II Author Team (2015) are marked and a brief explanation is provided in the neighboring column. (NA = North Atlantic)

| Number | Variable | Present climates | | Future climate | |
|---|---|---|---|---|---|
| Atmosphere | | | | | |
| 1 | Large-scale circulation | +/- | Remote influence of the multi-decadal variability in the NA on the Baltic Sea | +/- | Impact of warming Arctic with declining sea ice might be relevant |
| 2 | Air temperature<br>Warm spell<br>Cold spell | +++<br>++<br>-- | Accelerated warming | +++<br>++<br>-- | Greater confidence due to increased ensemble size, coupled atmosphere-ocean models |
| 3 | Solar radiation | + | Comparison between various satellite products | +/- | GCM and RCM systematically differ |
| 4 | Precipitation<br>Heavy precipitation<br>Drought north (south) of 59ºN | +<br>++<br>-(+) | | ++<br>+++<br>-(+) | convection-resolving models became available |
| 5 | Wind<br>Number of deep cyclones | +/-<br>+ | | +<br>+/- | Small systematic increase in winter in the northern Baltic where the sea ice will melt |
| 6 | Air pollution, air quality and atmospheric deposition | +/- | | +/- | |
| Land | | | | | |
| 7 | River discharge<br>High flow[11] in the north (south) | +/-<br>+/- (+/-) | Dataset of observed time series for the past century from Sweden merged with high-resolution dynamic model | +<br>- (+) | Changing seasonality (decrease of river discharge in spring, increase in winter) |

---

[11] Based upon annual maximum river discharges of daily data for Sweden with 10- and 100-year repeat periods (Roudier et al., 2016) and for Finland with 100-year repeat period (Veijalainen et al., 2010)



| | | | | | |
|---|---|---|---|---|---|
| | | | projections of the upcoming century | | may affect the occurrence of floods[12,13] |
| 8 | Land nutrient inputs | +/- | | + | |
| Terrestrial biosphere | | | | | |
| 9 | Growing season in the Baltic Sea region | ++ | Study based on satellite data available | +/- | No new study |
| 10 | Carbon sequestration in northern terrestrial ecosystems | +/- | | +, later - | First increasing sinks. Weak sources of carbon after 2060-2070s due to increased soil respiration and biomass burning |
| Cryosphere | | | | | |
| 11 | Snow<br>Sea-effect snowfall | ---<br>+/- | | ---<br>+/- | |
| 12 | Ice mass of glaciers | --- | Since 2006 inventories of all Scandinavian glaciers have become available | --- | High-resolution projections of Scandinavian glaciers available |
| 13 | Permafrost | -- | High-resolution modeling | --- | |
| 14 | Sea ice cover<br>Extreme mild winter<br>Extreme severe winter<br>Ice ridging | ---<br>+++<br>---<br>+/- | | ---<br>+++<br>---<br>- | |
| 15 | Lake ice | --- | Systematic assessment available | --- | Projections for global lake ice available |
| Ocean and marine sediments | | | | | |
| 16 | Water temperature<br>Marine heat wave | +++<br>+ | Accelerated warming | +++<br>+++ | Increasing number of record-breaking summer mean SST events and number of heat waves |
| 17 | Salinity and saltwater inflows | +/- | Homogenous data of saltwater inflows, north-south salinity | +/- | Uncertainty sources of salinity due to wind, river discharge and global sea |

---

[12] Roudier et al., 2016

[13] Veijalainen et al., 2010





| | | | | | |
|---|---|---|---|---|---|
| | | | gradient has increased | | level rise changes were assessed |
| 18 | Stratification and overturning circulation | +/- | Systematic study of monitoring data since the 1980s | +/- | Intensified seasonal thermoclines during summer but no change of the halocline and overturning circulation |
| 19 | Absolute sea level | +++ | | +++ | |
| | Storm surge relative to the mean sea level | +/- | Paleoclimate study on sea level extremes did not show systematic changes in changing climate, dissensus in the literature[14,15] | +/- | Dissensus in the literature[16,17] |
| 20 | Waves | +/- | | + | Small increase in winter in the northern Baltic Sea |
| | Extreme waves | +/- | | + | |
| 21 | Sedimentation and coastal erosion | +/- | | +/- | First modeling studies available |
| 22 | Hypoxic area | + | Warming contributed to the historical spread of hypoxia in the deep water and in the coastal zone, sediment cores suggest that changing climate caused hypoxia during the Medieval Climate Anomaly instead of agriculture | ++ | Oxygen decline in the coastal zone due to warming |
| 23 | $CO_2$ uptake | ++ | New observations and modeling, positive alkalinity trends identified | +/- | |
| | pH southern (northern) Baltic Sea | --(+/-) | | +/- | |

---

[14] Ribeiro et al., 2014

[15] Marcos and Woodworth, 2017

[16] Vousdoukas et al., 2016

[17] Vousdoukas et al., 2017





| Marine biosphere | | | | | |
|---|---|---|---|---|---|
| 24 | Microbial communities | + | In the northern Baltic Sea increased riverine dissolved organic matter suppressed phytoplankton biomass production and shifts the carbon flow towards microbial heterotrophy | +/- | Increase of dissolved organic matter and temperature will enhance and decrease the abundance of bacteria, respectively |
| 25 | growing season of phytoplankton (cyanobacteria) | ++ | | +/- | Warming causes prolonged and intensified cyanobacteria blooms but the nutrient control is dominating |
| | cyanobacteria biomass | +/- | | + | |
| | ratio between diatom and dinoflagellate biomasses since 1901 | - | new indicator for the environmental status developed | +/- | |
| 26 | Zooplankton | +/- | | +/- | Increasing microzooplankton biomass |
| 27 | Macroalgae and vascular plants | +/- | Systematic studies on benthic ecosystems | +/- | |
| 28 | Zoobenthos | +/- | Systematic studies on benthic ecosystems, spreading of non-indigenous such as polychaete *Marenzelleria* spp. | +/- | Weaker benthic-pelagic coupling and decreasing benthic biomass in a warmer and less eutrophic Baltic |
| 29 | Non-indigenous species | + | | ++ | |
| 30 | Fish | +/- | Food web modeling including fisheries | +/- | Multi-driver (climate change, eutrophication, |





| | | | | | fisheries) food web projections were performed |
|---|---|---|---|---|---|
| 31 | Populations of marine mammals | +/- | | - | |
| 32 | Waterbird migration | +++ | Northward shift of the wintering range of waterbirds | ++ | Controlled by food availability |
| 33 | Marine food web | +/- | | +/- | |

4363



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
