# Peer review of "Climate Change in the Baltic Sea Region: A Summary 1"

_Earth System Dynamics, 2021_

## Author Comment (AC1)

Earth Syst. Dynam. Discuss., referee comment RC1
https://doi.org/10.5194/esd-2021-67-RC1, 2021 ©
Author(s) 2021. This work is distributed under the
Creative Commons Attribution 4.0 License.

[Figure]

**Comment on esd-2021-67**

Jouni Räisänen (Referee)
* * *
Referee comment on "Climate Change in the Baltic Sea Region: A Summary" by H. E. Markus Meier et al., Earth Syst. Dynam. Discuss., https://doi.org/10.5194/esd-2021-67-RC1, 2021
* * *
**Answers to reviewer no. 1 (Dr. Jouni Räisänen) in red**

**1. General comments**

This review summarizes our current knowledge on climate change in the Baltic Sea region, both for the pre-instrumental past, the instrumental era, and the rest of the 21st century. It also covers other environmentally and societally important topics such as biological changes in the Baltic Sea and coastal erosion, considering other anthropogenic drivers such as nutrient input in addition to the effects of climate change. It consolidates the work done by several tens of scientists in the Baltic Earth Assessment Reports (BEAR) project, building on nine more specialized BEAR review articles as well as a large volume of other recent literature.

This review is a highly valuable body of work. It will most likely become the new default source of information on Climate change in the Baltic Sea area, similarly to the first (2008) and second (2015) Baltic Sea basin climate change assessments that were published in book format. Furthermore, the review was a pleasure to read because the structure is well organized, and most parts of the text are very well written.

Thank you very much for the thorough review and excellent comments. We will follow your suggestions and will revise the manuscript accordingly. We are impressed by your work.

Due to my own background in meteorology and atmospheric climate change, I am in a better position to evaluate the substance on these parts of the text than, for example, marine biochemistry. Where I am not an expert I have largely focussed on the presentation, commenting on text where I suspect that my difficulties to understand are affected by unclear writing.

- Even for the atmospheric part, I have few general comments on the substance, except for two that may be somewhat difficult to address in practice:

- With respect to the IPCC assessment cycle, this review comes out at a slightly unfortunate time. In a few months from now, the natural science part of the IPCC 6th assessment report will be fully available, and this may make the readers of this review feel that some of the results are already outdated. Fortunately, based on the IPCC Summary for Policymakers published in August 2021, the main conclusions on future climate change will not change. However, I will point in my detailed comments a couple of new results and other features in the new IPCC assessment that should be incorporated in this review.

We agree.

The analysis of projected future climate changes builds heavily on regional climate model (RCM) simulations in the EURO-CORDEX project. This is a pragmatic choice since global climate model simulations are too coarse to give geographically detailed information (e.g., differences in warming between land and the Baltic Sea) on climate change in the area. Nevertheless, a larger-scale (whole Baltic Sea drainage basin) comparison between the EURO-CORDEX simulations and the wider set of CMIP5 (or CMIP6) simulations would have been of interest, to check how well the uncertainty implied by the variation between the CMIP5 models is captured in EURO-CORDEX, as well as for any systematic differences between the global and regional models.

Thanks for this very good suggestion. Given that we are synthesizing only already existing literature, such a large-scale comparison is, however, out of the scope of the paper. We have explained the problem (in Ch 2.2), added a few references to previous studies also including the EURO-CORDEX ensemble and highlighted the need for further studies along these lines also in the "Knowledge gap" section.

Furthermore, we will add to L582

The choice of working with regional climate model projections, downscaling a limited subset of all available CMIP5 GCMs, implies that the resulting ensemble may not represent all available data properly. Previous studies of parts of the 72-member EURO-CORDEX RCP8.5 ensemble (a sparsely filled GCM-RCM matrix with in total 11 RCMs downscaling 12 GCM projections) assessed here, and presented in more detail by Christensen et al. (2021), illustrated this hypothesis.

By investigating 18 of these RCM simulations (8 RCMs downscaling 9 GCMs), Kjellström et al. (2018) found that the 9-member GCM ensemble showed lower temperature response for northern and eastern Europe compared to the entire CMIP5 ensemble. In addition, it was found that the RCMs can – to some degree – alter the results of the driving GCMs (as also discussed by Sørland et al., 2018). In a more recent study, Coppola et al. (2021) investigated a 55-member ensemble with the same 11 RCMs downscaling the same 12 GCMs as assessed by Christensen et al. (2021). They compared the 55-member ensemble to the driving 12 GCMs and concluded that the RCMs modify the results. In their analysis, Coppola et al. (2021) also considered a set of 12 CMIP6 GCMs finding that these show a

stronger warming signal than the 12 CMIP5 GCMs. This was related to the higher equilibrium climate sensitivity in several global models of the new generation.

The sentence starting on l3456 will be expanded and split into

For climate model projections, even if large ensembles of high-resolution regional atmosphere models are becoming increasingly available, the coverage of the underlying global climate model ensembles is still small implying that detailed conclusions on uncertainty and/or robustness in details of future climate change and its impacts cannot easily be drawn. In addition, only one ensemble with 22 members utilized a coupled atmosphere-ice-ocean model (Christensen et al., 2021).

References:

Sørland, S., Lüthi, D., Schär, C. and Kjellström, E., 2018. Bias patterns and climate change signals in GCM-RCM model chains. Environ. Res. Lett., 13, 074017, DOI: 10.1088/1748-9326/aacc77.

Coppola, E., Nogherotto, R., Ciarlo, J.M., Giorgi, F., Somot, S., Nabat, P., Corre, L., Christensen, O.B., Boberg, F., van Meijgaard, E., Aalbers, E., Lenderink, G., Schwingshackl, C., Sandstad, M., Sillmann, J., Bülow, K., Teichmann, C., Iles, C., Kadygrov, N., Vautard, R., Levavasseur, G., Sørland, S.L., Demory, M.-E., Kjellström, E. and Nikulin, G., 2021. Assessment of the European climate projections as simulated by the large EURO-CORDEX regional climate model ensemble. J. Geophys. Res.: Atmospheres, 126, e2019JD032356, DOI: 10.1029/2019JD032356

Considering the reader-friendliness of this extensive review, I have two suggestions:

- If technically possible, a table of contents in the beginning of the article would make the orientation easier. We will add a table of contents.
- A list of acronyms might also help the reader. Most (although not all) acronyms are defined appropriately in the text but, as the review is long, it would be useful to have them all in the same place. We will add a list of acronyms.

More detailed comments on the substance and presentation follow below. After them, minor technical comments (language, typos etc.) are presented. Naturally, the division between the two categories is not always clear-cut.

**2. Comments on substance and presentation**

- L227: occurred more than two thousand years ago OR started more than two thousand years ago? started more than two thousand years ago
-

L256: a forced component related to, inter alia, volcanic eruptions ( ) and anthropogenic
aerosols? See, for example: Watanabe, M. and Tatebe, H. 2019: Reconciling roles of
sulphate aerosol forcing and internal variability in Atlantic
multidecadal climate changes. Climate Dynamics, 53, 4651–4665
https://doi.org/10.1007/s00382-019-04811-3 We will rephrase the sentence to be
more precise and include the suggested citation:

The AMO consists of an unforced component which is the results of atmosphere-ocean
interactions (e.g. Wills et al., 2018) and a forced component. It has been shown that
external forcing such as solar activity, ozone, and volcanic and anthropogenic aerosols
are also important drivers altering variance and phase of the AMV (Mann et al., 2021;
Mann 257 et al., 2020, Watanabe and Tatebe, 2019).

- L266-267: observed shift in the storm tracks - to which direction? New text:
- Increasing winter temperatures in the Baltic Sea have also been linked to an observed
  shift northward in the storm tracks (BACC II Author Team, 2015).
- L279-280: … may influence the atmospheric circulation in a way that leads to
additional precipitation over the Baltic Sea region - in its positive or negative phase? We
will add that CMIP6 models suggested increased precipitation during the negative phase
of the AMO: However, a recent model study suggested that variations in the AMO may
influence atmospheric circulation that leads to additional precipitation during positive
AMO phases over the Baltic Sea region (Börgel et al., 2018). However, it should be noted
that the ensemble mean response of the CMIP6 control runs shows an increase in
precipitation during negative AMO phases (Börgel et al. 2021, under review).

L362-364: IPCC AR6 summary of policymakers was published in August 2021. It might
be a good idea to read it and check whether anything in Section 1.5 needs to be
updated based on it, in addition to the couple of examples mentioned below. We will add
- updates from IPCC AR6 to Section 1.5.
- L369: 1981-2005 is a somewhat unusual choice for a baseline (e.g., 1981-2010 and
  1986-2005 are more common). Does it have a specific meaning here? Correct. It was
a mistake and it should be 1986-2005 for Euro-CORDEX data
L385-391: It might be good to note that the RCP scenarios have been replaced by SSP
(Shared Socioeconomic Pathways) scenarios in IPCC AR6. The lowest of these, SSP1-
1.9, which was designed to limit the global warming to 1.5°C above the preindustrial,
has lower radiative forcing than RCP2.6. For the other SSP scenarios, the effective
radiative forcing tends to be slightly higher than for the nominally corresponding RCP
scenarios (e.g., SSP5-8.5 is slightly higher than RCP8.5). Thus, the range covered by
the SSP scenarios is even wider than that for the RCPs. See the draft of IPCC AR6
Technical Summary, p. 21-23. We will add a paragraph about the new SSPs.
- L405: by a factor of 2.2 to 2.4, as a multi-model mean value? Yes, we will add this
- information
L413: Although the difference in heat capacity plays some role in the land-sea warming
contrast during the transient phase of warming, it is not its main reason. As first shown
by Joshi et al. (2008), the overall land-sea contrast is to a large extent caused by the
dryness of land surfaces, which makes it impossible for evaporation to increase as much
in a warmer climate as it does over the oceans. The mechanistic pathway involves
atmospheric dynamics and is probably too complicated to be described here. Reference:
Joshi, M.M., Gregory, J.M., Webb, M.J. et al. Mechanisms for the land/sea warming
contrast exhibited by simulations of climate change. Climated Dynamics 30,
455–465 (2008). https://doi.org/10.1007/s00382-007-0306-1 We will add this
explanation and refer to the mentioned article by Joshi et al. (2008).
- L420: Please reiterate that the normalization is by the global mean (and not local)
  warming. We will add "Note that this normalization is by the global mean warming as
- already mentioned above"

L422: I would suggest omitting "Under RCP8.5", although it is present in the IPCC sentence. The basic features of precipitation change are the same for all the RCP scenarios, although the signal-to-noise ratio increases with stronger forcing. We will skip "Under RCP8.5"

L439-440: The IPCC SROCC "High mountain" region is very inhomogeneous, covering mountain regions all over the world. Giving an average snow depth decrease for such a heterogeneous area does not seem meaningful. OK, but this is a number directly of SROCC.

L447-461: For sea level change, it would be useful to also give longer-term projections, since many people do not realize that the 21st century sea level rise is just the beginning. Citing IPCC AR6 Summary for Policymakers, p. 28: "In the longer term, sea level is committed to rise for centuries to millennia due to continuing deep ocean warming and ice sheet melt, and will remain elevated for thousands of years (high confidence). Over the next 2000 years, global mean sea level will rise by about 2 to 3 m if warming is limited to 1.5°C, 2 to 6 m if limited to 2°C and 19 to 22 m with 5°C of warming, and it will continue to rise over subsequent millennia (low confidence)". Good suggestion. We will add this information.

L478: global net primary productivity? Yes. Global net primary productivity. "global" wil be added

L495-L506: A key result in IPCC AR6 that should be cited here is the narrowed uncertainty range of ECS, allowed by improved scientific understanding and accumulation of new data. Largely based on the review by Sherwood et al. (2020), the IPCC now gives a likely range of 2.5-4 degrees for the ECS. Reference: Sherwood, S. C., Webb, M. J., Annan, J. D., Armour, K. C., Forster, P. M., Hargreaves, J. C., et al. (2020). An assessment of Earth's climate sensitivity using multiple lines of evidence. Reviews of Geophysics, 58, e2019RG000678. https://doi.org/10.1029/2019RG000678
We will add this important information.

L510-513: Divide into two sentences: … were selected (Table 1). Scientific peerreviewed publications and reports of scientific institutes since 2013 on past, present and future climate changes in these variables were assessed ...OK.

L576: What does "regionalizations" of Global Climate Model (GCM) or ESM simulations mean? Does it refer to dynamically downscaled GCM / ESM simulations or just regional features of the GCM / ESM simulations themselves? It refers to dynamical downscaling. We will explain it better.

L588: internally consistent results OK.

L618: drivers of climate and environmental variability? OK.

L668-670: Here the reader gets the impression that, in addition to the global / Northern Hemisphere mean changes, the changes in the Baltic Sea region can also be explained by volcanic and solar forcing, without the need to additionally consider longterm internal variability. Is this what is meant (and if yes, is this well established)? Good comment. Internal variability is important and we will rephrase the sentences.

L718-720: Is the unit correct? Figures 6 and 9 in Mauri et al. (2014) suggest one order of magnitude larger precipitation anomalies, as does the text (although not the figures) in Mauri et al. (2015). The scales in Mauri et al. (2014) cannot be correct. For instance, Fig 9 displaying the reconstructed winter precipitation anomalies for the Midholocene would indicate that in Ireland, Spain, and Syria (about -50 mm/month) precipitation would have been dramatically reduced. Current winter precipitation in those regions is about 80-100 mm/month.

L774-776: Suggested rewording: "The temperatures in the Medieval Warm Period and the Contemporary Warm Period are similar within their respective uncertainties"
OK.

L833: considering the last few years of the NAO time series in Fig. 5: from the mid-1990s to the early 2010s, there was ... OK.

L836-837: Perhaps this article should also be cited: Scaife, A.A., Smith, D. A signal-
tonoise paradox in climate science. npj Clim Atmos Sci 1, 28 (2018).
https://doi.org/10.1038/s41612-018-0038-4. The article suggests that the
atmospheric circulation in climate models might not be sensitive enough to (e.g.)
changes in sea surface conditions.

Thank you. We will add the reference.

-
-
-
-
- L868-869: Mention the period: 1979-2018. OK.

  L878-879: influence of the AMO on the warming of Baltic Sea SSTs during 1980-2008?
- Thank you. Much better sentence.
- L905: annual number of days defined to belong in warm spells? OK.

  L922-923: The wording is unclear. High values in dimming or global radiation? A
- minimum in dimming or global radiation? OK. We will rephrase the sentence.
- L928: Rather: The CERA20C reanalysis. Reanalyses are not pure model simulations.
  OK.
- L983: with the 95th percentile of wet-day precipitation amounts ranging … (or how
  were typical amounts defined)? OK.

  L984-985: Simpler language: An index for the annual maximum five-day precipitation
- OK.
- L991-994: Another concern about wind trends in reanalyses, especially over land: are
  the effects of land use change on surface roughness appropriately included? Ok. We
  will add a sentence.

- L1005-1006: 7-11% per decade (if real) would amount to a rather large (22-34%)
  change in 31 years. I don't think this is a weak trend in an absolute sense, regardless
- of the signal-to-noise ratio. We agree and will revise the text "Common to all
  reanalysis datasets is a weak upward trend in the number of moderately deep and
  shallow cyclones, but a decrease in the number of deep cyclones, in particular for the
- period 1989-2010."

  L1026: more attention than what? We will replace the text "The importance of the
- stratospheric polar vortex for storm track changes has recently also received attention
  (Zappa and Shepherd, 2017)."

  L1027-1028: Should be: a poleward and downstream displacement of precipitation
  relative to the cyclone centre. However, this was a highly idealized simulation with a
  globally uniform increase in SST, and its relevance for the current review article seems
  therefore low. We agree and we will delete the sentence.

  L1104: population exposed to a large number of days with high ozone concentrations
- (or something similar)? OK.

  L1167-1168: Suggested reordering of words: However, the regional impacts of
- precipitation change on both the observed and projected changes in stream flow are
  still unclear. We will rephrase the paragraph:
  The observed temperature increases have affected stream flow in the northern Baltic
  Sea region for 1920-2002 in a manner corresponding well to the projected
  consequences of a continued rise in global temperature in term of increasing winter
  time discharges (Hisdal et al., 2010). However, the regional impacts of precipitation
- change on both the observed and projected changes in stream flow are still unclear as
- the combined effects of changes in precipitation and temperature are still not well
  known (Stahl et al. 2010).

- L1262: Does "emissions of black carbon" refer to the effect of forest fires? Perhaps this
- should be clarified. Yes, we will write "*emissions of black carbon from forest fires*".
  L1285-1286: This sensitivity is surprisingly weak. Given that in the Baltic Sea area
  temperature increases by about 6°C in 30 days in spring, a uniform warming of 1°C
- should lead to a 5-day advance in the start of the growing season, assuming that this
- is primarily determined by temperature. One possible explanation is that the spring
- temperatures in Jin et al. refer to the 3-month period preceding the start of the
  growing season. The sensitivity (as estimated from interannual variability) would likely
- be stronger if temperatures just before the start of the growing season were used as

-
-

the predictor. We are not sure if the analogy to the mean seasonal cycle is valid here. It is correct that Jin et al. define "*spring*" as the 3-month period before the start of the growing season. That might well affect the strength of the sensitivity. Adding some information about the definition of the seasons as used in the paper might, thus, be helpful. In line 1286 we will add "*In the study, spring was defined as the three months preceding the start of the growing season.*" Also, we will revise the statement in lines 1288 and 1289 into "*change the sensitivity to climate conditions in the previous summer and winter seasons*" for clarification.

L1287: 0.18 days for each 1 cm decrease in precipitation in the 3-month period preceding the start of the growing season? Yes, we will write "*...cm$^{-1}$ decrease in precipitation*"

L1287-1292: Do Jin et al. present a plausible mechanism for the apparent effect of summer and winter temperatures on the onset of the following growing season? If not, I would be sceptical, and would rather refrain from discussing these weaker winter and summer effects in this review, particularly as the statistical analysis was based on only 17 years of data. Jin et al. speculated on the mechanisms of the effects of winter and summer warming leading to a delay of the following growing season based on the scientific literature. They also acknowledged that the mechanisms are unclear. The authors didn't discuss limitations due to the limited data record. Given the projected future warming in all seasons, we would prefer to keep the statements and add a note of caution regarding the possible mechanisms in line 1292 "*As mechanisms for the delayed start of the growing season in Fennoscandia in relation to a warming in the preceding summer and winter seasons the authors suggested the effects on plant dormancy. A winter warming could, for instance, prolong the chilling accumulation required to break winter dormancy of trees. Later summer temperatures, on the other hand, could affect bud dormancy initiation, while reduced soil moisture associated with higher summer and autumn temperatures could delay the leafing and flowering of plants. However, details of the involved mechanisms are yet unknown.*"

L1352-1356: Long sentence, divide to two: ... southwest Europe. However, their analyses also ... OK.

L1425-1426: It should be mentioned that the study of Irannezhad et al. was based on a temperature-index snowpack model using daily temperature and precipitation as input, rather than directly on snow observations. OK. This information will be added.

L1449-1453: Long sentence. Divide to two: ... reference glaciers. This means that ... OK. We will divide the sentence.

L1460-1465: I feel that the summer 2016/2017 case is discussed in unnecessary detail. It would be enough to just write: "For example, slightly positive mass balances were observed for ... in 2016/2017 as a result of a cold summer (references)". OK.

L1480: (permafrost p > 0.8). This definition does not seem essential for the text. OK. The definition will be deleted.

L1502: since the late 19th century, or even earlier for MIB? You are right. We will add this information.

L1545-1546: The sentence implicitly suggests that there is a category between "severe" and "average". Is there, or should "severe" be "extremely severe"? We will modify "no severe and no extremely severe ice winters"

L1550: towards low values? "Towards zero" is problematic since no nearly-zero MIB has been observed this far. OK.

-
-

L1555: Only one severe or only one extremely severe? Which winter? It is 2011. According to a 3-level scale of the Finnish Meteorological Institute it was a severe ice winter.

L1607: The total volume-averaged warming? No, at monitoring stations. We will rephrase.

L1638: changes in Baltic Sea marine heatwaves? Correct. We will add "Baltic Sea marine"

L1740: A maximum rate of 10 mm/year was mentioned on L661. It would be great to reach internally consistent numbers. We agree and corrected the figures.

L1748-1749: from the 1960s to the early 1990s? Correct. We added "from the 1960s to the early 1990s"

L1761: 20 cm sounds like a very modest number. We will be more precise and rephrased the paragraph.

L1772: about 4 m in St. Petersburg? This number is not representative for the Gulf of Finland as a whole. Yes, we will add St. Petersburg.

L1938: phosphorus released during the MBIs (or more generally)? No, phosphorus released from the sediments under anoxic conditions in general. We will rephrase the sentence.

L2108: Do "changes" refer to increases in seal population, or a more complicated mix of increases and decreases? For clarification the first paragraphs will be rewritten:

During the 20th century the marine mammals of the Baltic Sea experienced large declines in abundance because of hunting of seals, bycatch of porpoises and exposure to harmful substances causing reduced fertility

The breeding distributions of the ice-breeding seals in the Baltic Sea have evolved with ice coverage, with the seals breeding where and when ice optimal for breeding occurs. Breeding ringed seals need ice throughout their relatively long lactation period (>6 weeks), and also use ice as moulting habitat. Ringed seals prefer compact or consolidated pack ice as it provides cavities and snowdrifts suitable for the construction of the lairs, most  importantly the breeding lair (Sundqvist et al., 2012).

Implementation of specific management- and protection measures, have had a profound positive influence on the populations of several Baltic Sea mammal populations, in particular seals (Reusch et al., 2018). Reusch et al. (2018) attributed these changes also to reduced exposure to harmful substances and increases in overall fish stocks as a consequence of eutrophication (including reduced stocks of several commercial fish species).

Specific climate change-related impacts on seals are hard to establish, although reconstructions of distributional histories since the last glaciation have been attempted for some seal species (Ukkonen et al., 2014). Along with the warming winters the availability of suitable breeding ice for ringed seals in the Bothnian Bay is decreasing (Section 3.2.4.4).

L2202-2209: The essence of future climate change is global greenhouse gas induced warming. Compared with this, changes in large-scale atmospheric circulation will likely be a minor issue for most purposes. Therefore, although it is logical to retain the same order of subsections in 3.3 as in 3.2 (thus starting with the atmospheric circulation), please make it clear that changes in atmospheric circulation are not the primary cause of projected future warming. We will add the requested clarification at the beginning of

the section 3.3.1.1: "Continued greenhouse-gas induced warming is the key driver for future climate change and changes in atmospheric circulation are relatively less important (IPCC, 2021). However, as the regional climate in the Baltic Sea region is strongly governed by the large-scale circulation of the atmosphere it is important to also consider changes in it when assessing future regional climate (e.g. Kjellström et al., 2018)."

L2220: Please mention the periods between which the climate changes are calculated. We will add the periods.

L2229-2230: Is there any information on the magnitude of this difference? We will calculate numbers from the scenario simulation data and put these into the revised manuscript because they do not exist from the literature.

L2231-2233: This explanation oversimplifies the dynamics of diurnal temperature range (DTR) changes, which originate from a multitude of factors (e.g., Lindvall, J. & Svensson, G, 2015: The diurnal temperature range in the CMIP5 models. Clim Dyn 44, 405–421). In addition to the processes discussed in the mentioned paper, it should be noted that, in the middle of the winter when there is little solar radiation, the genuine diurnal temperature range in northern Europe is very small. However, differences between the daily maximum and minimum temperatures can still be substantial due to synoptic-scale weather variability. Factors that reduce the temperature variability on synoptic time scales (e.g., reduced temperature gradients between the Atlantic Ocean and Eurasia) therefore also likely contribute to the apparent decrease in DTR. We will replace

"Changes in daily minimum and maximum temperatures have similar spatial patterns as the mean air temperature changes, with the expected greater warming for minimum temperature (Christensen et al., 2021). According to Christensen et al. (2021) and previous studies (BACC II Author Team, 2015), the latter result is explained by the reduced outgoing long-wave radiation under increased greenhouse gas concentrations. The long-wave radiation acts to cool the surface, especially when the ground is warmer than the air, e.g. during winter and during nights."

with

"Changes in daily minimum and maximum temperatures have similar spatial patterns as the mean air temperature changes, although with greater warming for minimum temperature (Christensen et al., 2021). Such a decrease of the diurnal temperature range can have a number of explanations. Lindvall and Svensson (2015), based on an ensemble of CMIP5 GCMs suggested: increasing downwelling longwave radiation due to larger greenhouse gas concentrations, increased cloudiness at high latitudes, changes in the hydrological cycle and in changes in shortwave incoming radiation. In addition to these, we note that the difference in diurnal temperature range in Northern Europe is small in winter implying that differences between daily maximum and minimum temperatures also substantially depend on synoptic-scale variability and air mass origin. As a consequence, reduced temperature variability on synoptic time scales, resulting from reduced temperature gradients between the Atlantic Ocean and Eurasia, may be a reason."

Lindvall J, Svensson G: The diurnal temperature range in the CMIP5 models. Clim Dyn 44, 405–421 (2015). https://doi.org/10.1007/s00382-014-2144-2

L2237-2238: What is the origin of this difference? Is there less warming over the Baltic Sea in the uncoupled simulations? Or is there a cold bias in the baseline SSTs, which precludes the uncoupled models to reach warm enough Baltic Sea temperatures to exceed the tropical night threshold even after a greenhouse-gas-induced warming?

-
-

Investigations are currently under way but not yet finished nor is this published. Therefore we would like to avoid any speculative statements as explanation at this time.

L2239-2241: In addition to the magnitude of the warming that is affected by the ice/snow albedo feedback, the baseline climate may also play a role. Further southwest, where the winters are milder, there are less frost days to start with, and therefore less room for a further decrease in the future. We agree and we will modify the text.

L2242-2250: This article should be cited: Boé, J., Somot, S., Corre, L. et al. Large discrepancies in summer climate change over Europe as projected by global and regional climate models: causes and consequences. Clim Dyn 54, 2981–3002 (2020). https://doi.org/10.1007/s00382-020-05153-1. A quotation from the abstract: "The RCMs generally simulate a much smaller increase in shortwave radiation at surface, which directly impacts surface temperature. In addition to differences in cloud cover changes, the absence of time-varying anthropogenic aerosols in most regional simulations plays a major role in the differences of solar radiation changes". In other words: The RCP scenarios include a decrease in aerosol emissions, which enhances the warming and the increase in solar radiation in the global climate models. However, this effect is absent from many of the RCMs, in which the aerosols stay constant with time. We will add the reference and some text.

L2253: Good agreement with each other or with observations? Both. We will rephrase the sentence.

L2272: Does this also apply to dry spells in the northern parts of the area? We will replace (starting at l2269)

"Expressed by the Clausius-Clapeyron equation, warming increases the potential for extreme precipitation due to intensification of the hydrological cycle associated with the growth of atmospheric moisture content. For Northern Europe, regional climate models indicate an overall increase in the frequency and intensity of heavy precipitation events in all seasons and longer wet and dry spells (Christensen and Kjellström, 2018; Rajczak and Schär, 2017; Christensen et al., 2021, and references therein). The largest increase in the number of high precipitation days is projected for autumn. The number of drought events per year are expected to decrease, while their length is expected to increase (Christensen and Kjellström, 2018). Changes in more extreme events, like 10-, 20- or 50-year events, are less certain."
with
"Expressed by the Clausius-Clapeyron equation, warming increases the potential for extreme precipitation due to intensification of the hydrological cycle associated with the growth of atmospheric moisture content. For Northern Europe, regional climate models indicate an overall increase in the frequency and intensity of heavy precipitation events in all seasons (Christensen and Kjellström, 2018; Rajczak and Schär, 2017; Christensen et al., 2021, and references therein). The largest increase in the number of high precipitation days is projected for autumn. Changes in more extreme events, like 10-, 20- or 50-year events, are less certain. Changes in dry spells is another feature of an intensified hydrological cycle. Coppola et al. (2021), finds increasing number of consecutive dry days without precipitation for countries south of the Baltic Sea while no changes were reported in the north."

L2288: implying competing lower and upper tropospheric effects on changes in baroclinicity? We agree and we will change the sentence accordingly.

-
-

L2292: North-South temperature gradient Correct. We will add "temperature".

L2294-2306: Please clarify whether this text refers to the storm track activity in winter, or also in the other seasons? We will add "in winter".

L2306: What about CMIP6 (cf. Fig. 3 in Harvey et al., 2020)? We will add "The response of CMIP6 models is similar to CMIP5 models, but it is considerably larger, probably due to the larger climate sensitivity in the CMIP6 models (Harvey et al., 2020)."

L2342-2345: Perhaps it would be good to notify that this is an exception to the general rule (emission changes dominate over the effects of climate change) articulated on L2313-2314. We will add in Line 2345: "This is an example for domination of the climate change effect over a pollutant emission reduction effect, an exception to the statement made at the beginning of this section."

L2401-2403: Suggested rewording for clarity: the simulated changes were not larger in magnitude than their uncertainty We will modify the sentence.

L2426: "also" and "similar" are out of context since Hesse et al. (2015) is the first study mentioned. We agree and deleted "also" and "similar".

L2568-2569: By which time? Under which emission scenario(s)? See answer to comment in L2568 below.

L2568-2573: Overall, the relative decrease in snow amount is projected to be smaller in the colder (northern and eastern) than in the milder (southern and western) parts of the Baltic Sea region. See, for example: Räisänen, J.: Snow conditions in northern Europe: the dynamics of interannual variability versus projected long-term change, The Cryosphere, 15, 1677–1696, https://doi.org/10.5194/tc-15-1677-2021, 2021. We will replace

"There is agreement among models that the average amount of snow accumulated in winter will decrease by over 70% in most of the Baltic Sea region. The high Scandinavian mountains, where the warming temperature will not reach the freezing point as often as in lower-lying regions, are an exception (Christensen et al., 2021). The reduction in snow amount is slightly larger than in maps presented by the BACC II Author Team (2015), which is consistent with the stronger average warming projected in the RCP8.5 scenario, compared to the SRES A1B scenario analyzed by the BACC II Author Team (2015)."

with

"Generally, the relative decrease in snow amount is projected to be smaller in the colder northern and eastern parts of the Baltic Sea region than in the milder southern and western parts (e.g. Räisänen, 2021). There is agreement among the EURO-CORDEX RCMs that the average amount of snow accumulated in winter will decrease by around 50% for land areas north of 60N for the RCP8.5 scenario by 2071-2100 relative to 1981-2010 (Christensen et al., 2021). South of this, the corresponding decrease is almost 80%. The reduction in snow amount is slightly larger than in maps presented by the BACC II Author Team (2015), which is consistent with the stronger average warming and, in the northern part of the area, smaller precipitation increases, projected in the RCP8.5 scenario, compared to the SRES A1B scenario analyzed by the BACC II Author Team (2015)."

L2580: at least double: Is this true even under the RCP4.5 scenario? The change by 2071-2100 must be strongly scenario dependent. We will replace

"The maximum snow depth was projected to decrease 15-20% by 2021-2050 and at least double that decrease by 2071–2100 (Szwed et al., 2019)."

with

"The maximum snow depth was projected to decrease 15-25% by 2021-2050 in both scenarios. By 2071-2100, decreases under RCP4.5 and RCP8.5 were estimated to be 18-34% and 44-60% respectively (Szwed et al., 2019)."

L2587-2588: If the first sentence refers to high-mountain areas in general, and not Scandinavia, it seems irrelevant for this review. We agree but the sentence provides context. Hence, we would like to keep the sentence as an introduction/overview.

L2602-2603: This formulation seems inaccurate. Citing Hock et al. (2019): "Beyond mid-21st century, atmospheric warming in mountains will be stronger under a high greenhouse gas emission scenario (RCP8.5) and will stabilise at mid-21st levels under a low greenhouse gas emission scenario (RCP2.6)." Thus, depending on the scenario, the range of temperature projections varies from a stabilization to mid-century temperatures to accelerated warming. We will reformulate the sentence. "Beyond 2050, air surface temperatures in high mountain regions are projected to increase under a high emission scenario (RCP8.5), and to stabilize at 2015-2050 levels under a low emission scenario (RCP2.6) (IPCC, 2019a)."

L2737-2738: It would also be good to mention that ice melt in Greenland has a relatively modest effect on sea level in the Baltic Sea. We will add this information.

L2747: are somewhat lower? Correct. We will add lower.

L2754-2757: Are these numbers also for the change from 2000 to 2100? Yes. We will better explain. "For the period 2090-2099 relative to 1980-1999 and based on the SRES A1B scenario, the projected absolute sea level rise in the Baltic Sea was estimated to be about 80% of the global increase (Grinsted et al., 2015). These results were confirmed by other studies for other scenarios and slightly different reference periods (e.g. Kopp et al. 2014; Grinsted, 2015), and summarized by Pellikka et al. (2020) who, for the period 2000-2100, documented an ensemble mean absolute sea level rise in the Baltic Sea of about 87% of the global mean sea level rise."

L2762-2763: Is this estimate also based on the RCP8.5 scenario? Yes. We will better explain.

L2873-2874: A confusing sentence. Why does the fact that the BSAP did not take climate change into account imply that the hypoxic and anoxic area may decrease? For clarification we will rephrase the paragraph.

L2882-2883: Are these numbers also based on BACC II? Yes, we will add the reference BACC II Author Team (2015)

L2885: Should the increased precipitation be also mentioned (cf. increased river runoff on L2887)? Yes, we will add increased precipitation.

L2934: biases and different future changes? Correct. "Larger sources of uncertainty are global and regional climate model uncertainties,…"

L2950-2957: Based on L1972-1973, total alkalinity has increased this far. What explains the contrast between the past and projected future changes? We will clarify the apparent contradiction.

According to observations, alkalinity has increased in the Baltic Sea (L1972-1973). We do not know, however, if this increase will continue in the future. All available projections for future changes in alkalinity (mentioned in L2950-2957) are highly uncertain, as they address usually only few out of multiple factors shaping the alkalinity pool in the Baltic, namely: changes in salinity, river runoff, weathering in the catchment, organic matter production (eutrophication) and remineralization (especially

at the low redox conditions), processes in sediments (including pyrite and vivianite formation).

L3047-3049: What was assumed about climate change and anthropogenic nutrient input in this projection? The cited studies assumed various differing scenarios. Hence, a comparison is impossible.

L3163: between Baltic Sea climate projections, rather than Baltic Sea models? Correct. We will change to projections.

L3193-3196: Can anything be said about the likely direction (positive or negative) of these changes, or are they too case-specific or uncertain for any generalization? We will add to the text (including one new reference):

Certain marine species, e.g., cod and bladderwrack, may decline in both distribution and abundance (Gårdmark et al. 2013; Takolander et al. 2017), whereas others, e.g. sprat and certain mainly costal freshwater -fish, may increase (MacKenzie et al. 2012; Bergström et al. 2016). An increase in cyanobacteria blooms has also been projected, especially for the Central Baltic Sea (Meier et al. 2011b; Funkey et al. 2014), while increased flow of DOC may reduce both primary and secondary production in the northernmost low-saline areas with pronounced brown-water runoff (Wikner & Andersson 2012; Figueroa et al. 2021). The responses also depend on human intervention, i.e. the success of nutrient reduction schemes, and are most probably non-linear (Hyytiäinen et al. 2019; Ehrnsten et al. 2020). However, it can be summarised that – if only climate change is accounted for – most studies tend to project a decline in the overall state of the ecosystem, and a long-term decline in the provision of associated ecosystem services to humans is likely if the climate change is not significantly mitigated.

New reference: Figueroa, D., E. Capo, M. V. Lindh, O. F. Rowe, J. Paczkowska, J. Pinhassi and A. Andersson. 2021. Terrestrial dissolved organic matter inflow drives temporal dynamics of the bacterial community of a subarctic estuary (northern Baltic Sea). Environmental Microbiology 23: 4200-4213.

L3198: Based on this definition, even something that is not affecting anything else could count as a driver. Perhaps something like this: … is defined as something affecting something else, although a driver itself may be affected by other drivers. Thank you for this comment. We will re-phrase the definition.

L3212: reflect more solar radiation / solar radiation energy We will change the sentence accordingly.

L3227: atmospheric kinetic energy Kinetic will be added.

L3235-3236: Problem in sentence structure (The efficiency … eventually ends up) We will use two sentences.

L3408-3410: Please clarify whether these are annual or seasonal mean (JJA and SON) values. We will add "seasonal mean".

L3431: Which period does this magnitude of trend represent? We will add "for the 21st century"

L3483: The strong dependence of even local temperature changes on the evolution of greenhouse gas emissions and the feedbacks that determine the global climate sensitivity should also be mentioned. Another "black swan" uncertainty, of unknown

-
-

importance, is the extent to which larger-than-expected decreases in the AMOC could potentially counteract the effects of global warming in Northern Europe. As shown by

Fig. 12.9 in IPCC (2014b), there in fact was one model with a cooling of Northern Europe in the CMIP5 ensemble. The recent suggestion that the AMOC may be more sensitive to anthropogenic climate change than current climate models indicate (Boers, N., 2021: Observation-based early-warning signals for a collapse of the Atlantic Meridional Overturning Circulation. Nature Climare Change, 11, 680–688, https://doi.org/10.1038/s41558-021-01097-4) may also be relevant in this context.

- We will add this discussion and references to the knowledge gaps.
  L3492-3493: Also refer to Boé et al. (2020) (cf. comment 65), about the lack of
- timevarying aerosol forcing in RCMs. OK.
  L3551-3552: Also mention that these increases are not projected to continue, according to climate model projections. Whether or not there is a discrepancy between the observed and projected trends is not known. We will add this information to the discussion of knowledge gaps.
- L3764: Please specify "long-term". Does this refer to a positive trend starting before 1950? Long-term refers to the period 1950-2018. Information will be added.
- L3776-3777: A positive or negative long-term trend? A positive trend for the period
- 1921-2004 (Kniebusch et al., 2019, Geophys. Res. Lett.).
  L3780: I could not find this result (shift from later March to February) earlier in the manuscript. Furthermore, I was surprised to learn that the highest snow-melt floods take place so early in spring. Is this representative for the Baltic Sea region as a whole? We revised the sentence "occurred about a month earlier"
- L3934: By which time by more than 70%? We will add "between 1981-2010 and 2071-
- 2100"
  L3958: Why will increasing westerly winds decrease salinity? If this is just because westerly winds are typically associated with more precipitation and thus runoff, "or westerly winds" is redundant. Increasing westerly wind will block the freshwater outflow. Consequently, the saltwater inflow is reduced due to mass conservation (see Meier and Kauker, 2003). We will add this explanation to section 3.3.5.2
- Figure 3: Is the unit of precipitation change correct (cf. comment 21)? Yes, see our
- answer above.
- Caption of Figure 7: Please mention the baseline period (1961-1990?) New text: Figure 7: Annual and seasonal mean near-surface air temperature anomalies for the
- Baltic Sea basin for 1871−2020, taken from the CRUTEM4v dataset (Jones et al., 2012), compiled by Anna Rutgersson, Uppsala University. Baseline period is 1961-
- 1990. Blue, red: Baltic Sea basin region north and south, respectively, of 60°N. Dots: individual years. Smoothed curves: variability on timescales longer than 10 years.
  L4144-4145: "measured at Bolin Centre" appears suspicious. The Bolin Centre did not exist in the 18th century and is distributed among several locations. We will delete the phrase.
  Caption of Figure 14: Please also explain why there are two red curves and the dashed bars in the early parts of the time series. We will add "The dashed bars represent the error range of the early observations. The error range of the 30-year moving average is indicated by two red curves, converging into one when high quality data became available."
  L4190-4191: Suggestion for the beginning of the caption: Sea ice thickness distribution in the Bay of Bothnia in five winters ((a)-(e)), and its five-winter average (f), also shown as a red line in (a)-(e). Thank you. We rephrased "Sea ice thickness distribution in the Bothnian Bay estimated from helicopter electromagnetic measurements during February-March in five winters ((a)-(e)) and its five-winter average (f), also shown as a red line in (a)-(e)."
- L4195: In which month, or is this the winter maximum? The values are too large to be annual means as the caption suggests. February to March. See our answer above.
- L4274-4275: Should be: "Eight different dynamically downscaled Earth System Model simulations are used" OK.
- Figure 34: NIS = Non-indigenous species? Is the abbreviation defined somewhere? We
- will explain the abbreviation in the figure caption. "The abbreviation NIS stands for non-indigenous species."
- Some of the numeric values (e.g., the area of the Black Sea) differ between Tables 2 and 3. Please ensure that the numbers are consistent. We will correct the tables. The number for the Black Sea are: area 436 km2, mean depth 1197 m.

L4341, L4347 and 4353: Does "nine regionalized ESMs" mean "nine dynamically downscaled ESM simulations"? Yes, we will replace the word "regionalization".

**3. Technical comments** Thank you. We will correct the text accordingly.

- L98: Reckermann et al. (2021)
- L191: Add the proper reference to BACC II.
- L198: cannot yet be described?
- L237: regime varies / regimes vary
- L385: 8.3 or 8.5?
- L404: Please give a reference entry to IPCC
- AR4. L441: reduction varies / reductions vary
- L448: 2014a or 2014b?
- L479: warming and *changes* in stratification, light etc.
- L506: Delete "that"
- L518: and even in more general terms?

- L595: These results are reproduced in Figure 27.
- L600: was previously neglected
- L629-630: the response of climate?
- L661: continues
- L736: the fifth IPCC report. AR6 is already partly published.
- L737: CMIP5?
- L788: tend to be smaller than those reconstructed
- L795: and influence / which influences
- L841: the impact … is still under debate
- L866: weak effect
- L888: 1876-2020 or 1878-2020 (cf. caption of Table 4)?
- L894: particularly / in particular
- L898: has been observed
- L1052: except for
- L1216-1217: ... during 2000-2010 ... input to the catchment were exported ...
- L1385: $2 \times 10^5$, not $2 \times 105$
- L1459-1460: Regional and local deviations ... are, however, expected
- L1460-1461: were the result of a cold summer
- L1492: Wording: models indicate? Projections typically refer to the future.
- L1509: latest six? (2015, 2016, 2017, 2018, 2019, 2020)
- L1525: measurements in plural
- L1536: built structures
- L1575: Delete "be"
- L1664: "process" in singular.
- L1705-1706: "weakens" is repeated twice.
- L1891-1892: Please define the acronyms DIN (dissolved inorganic nitrogen?) and DIP (dissolved inorganic phosphorus?)
- L2039: Bothnian Sea
- L2179-2180: ... algal blooms ... have in some cases caused
- L2234: number ... is projected to increase
- L2266: similar to those over the land area
- L2274: number ... is expected
- L2283: (Räisänen, 2017) is missing from the list of references.
- L2285: Projections ... are uncertain?
- L2295: higher wind speeds?
- L2350: decreased precursor emissions
- L2587: 66-100% (cf. IPCC definitions)?
- L2612: "global projections" in plural
- L2679: latent heat flux?
- L2685: similar to those observed
- L2706-2707: no significant changes ... were projected / no significant change ... was projected
- L2764: seem to show
- L2768: Delete "the interaction off"?
- L2812: based on CMIP5 simulations
- L2830: have lost
- L2875: may increase the hypoxic area by about 30%
- L2947: of total
- L3058: scenarios suggest
- L3240: through sea level rise
- L3264: through the construction
- L3462: responses ... are / response ... is
- L3580: causes … are not well known
- L3581: studies ... exist
- L3759: summer cold spells
- L3878: that indicate?

- L3924: will lead
- L3946: Limited number of ensembles or limited number of ensemble members?
- L3952: in the end of the century?
- L4170: "och" should be "and"
- Figure 14: The percentages in the two right-hand panels should be multiplied by 100.
  - Thank you. We will revise the figure.
- L4251: in $10^4$ km$^2$?
- L4257: December through February
- L4258: The unit of precipitation change must be per cent, not mm/day.
- L4267: The right-hand-side vertical axis?
- L4273: December through February

L4327-4328: 1876-2020 or 1878-2020?

Resource [2020] www.kudi.org

---

## Author Comment (AC2)

Earth Syst. Dynam. Discuss., referee comment RC2
https://doi.org/10.5194/esd-2021-67-RC2, 2021

[Figure]

**Comment on esd-2021-67**

Donald Boesch (Referee)
* * *
Referee comment on "Climate Change in the Baltic Sea Region: A Summary" by H. E. Markus Meier et al., Earth Syst. Dynam. Discuss., https://doi.org/10.5194/esd-2021-67-RC2, 2021
* * *
**Answers to reviewer no. 2 (Dr. Donald Boesch) in red**

This paper is a *tour de force* compendium on the latest scientific results relevant to understanding recent and future climate change on the Baltic Sea Region.  The authors and the Baltic science community are to be commended in taking this on in a way that builds on and updates the two previous BACC assessments.  It is particularly effective that the assessment is linked with the efforts of HELCOM.  It sets a high standard for climate change assessments for regional seas in other parts of the world.

Thank you very much for the thorough review and excellent comments. We will follow your suggestions and will revise the manuscript accordingly. We are impressed by your review work.

This summary paper brings together and depends on the results of nine specialty papers or BEARs, which I have not reviewed.  Nor have I been charged with reviewing the consistency of the summary with the BEARs, but trust the authors to ensure that consistency.  The construct wherein each environmental variable is treated under present climate change, future climate change, and knowledge gaps and research needs results in some redundancy.  This could be alleviated somewhat by reference to the corresponding previous section without repeating narrative and references.  The section on concluding remarks does help bring these all together.

Within Baltic Earth, we performed prior to submission an internal review with two reviewers that were not involved in the BEARs to guarantee consistency.

The list of knowledge gaps and research needs is rather daunting, and rather depressing as it seems that virtually everything uncertain and unknown, and equally so.  Clearly, this is not the case. Concluding that section with a brief consideration of the knowledge gaps and research needs that are most critical to determining the future Baltic and are most potentially resolvable with concerted research would be help.

We will add a concluding paragraph.

The words uncertain, uncertainty, and uncertainties are used some 104 times in the paper, and often incautiously. Frequently, there are better terms to describe the nature of these so-called uncertainties. They may be a result of inadequate knowledge rather that inherent uncertainty or they might actually be deep uncertainties. In particular, when future changes depend on steps society might take to limit greenhouse gas emissions, these seem not so much as uncertain but yet to be determined. Some fine-tuning of the uncertainty language would help.

We will fine-tune the uncertainty language.

While, the paper indicated it will follow the terminology used by IPCC concerning degree of confidence in statements, as it does in section on key messages, it is not as very careful when it comes to the use of the term "likely" some 64 times and doesn't seem to differentiate among as-likely-as-not, likely, very likely or virtually certain as per the use of likelihood terms in IPCC parlance. Similarly, the term unknown is used quite a bit, without differentiating among completely unknown, largely unknown, not fully known, or incompletely understood. These might be more accurate descriptors in places.

We will check and possibly be more specific in the revised version. However, for many variables the information about confidence levels does not exist at the regional scale because large ensembles do not exist. We will explain this fact in the introduction and modify the definition of our terminology.

While this paper was developed prior to the release of the IPCC Sixth Assessment in August 2021 it is appearing after this release. Virtually all the literature cited used earlier GCM results, although some results based on CMIP6 models are discussed. It is impossible and unreasonable that this paper attempt to incorporate or compare Sixth Assessment models and conclusions in any great detail, it would be useful if the authors wrote brief comments about the extent to which conclusions might be affected by the new IPCC assessment, perhaps after section 1.5.5. My sense is that they wouldn't dramatically affect the conclusions. Recent literature (e.g. Hausfather and Peters, 2020, as cited in this paper) makes the point that the RCP8.5 pathway and the associated 4°C warming during this century is highly unlikely to occur and, in fact, the IPCC AR6 essentially admits this. Something between RCP7.0 and RCP4.5 is probably the maximum warming without substantial mitigation. Perhaps this point can be made more strongly in this paper. In fact, it would be informative to mention in key places where mitigation measures would affect key climate drivers, if and as society significantly reduces emissions and the use of fossil fuels (e.g. this would affect N deposition, shipping, plastics, etc.).

Following the comments of both reviewers, differences between CMIP5 and CMIP6 and how these differences may affect our results will be discussed.

**Specific comments:**

What are the current Baltic Earth Grand Challenges, a listing or, at least, a reference (line 128).

We will add the Grand Challenges and a reference.

The regional weather regimes vary (not varies) (line 237)

Will be corrected.

Freshwater (not fresh water) as an adjective (lines 289, 310, 311 1137, 4319, 4325)
Fresh water (not freshwater) as a noun (line 1257)

Thank you. Will be corrected.

Farther (not further) north, as this relates to distance (line 424)

Will be corrected.

analyzed by IPCC (2014b; 2019b) are assessed. (lines 576-577).

Will be corrected.

Were compared by Christensen et al. (2021) (lines 587-588)

Will be corrected.

Regions farther (not further) north as this refers to distance (line 706)

Will be corrected.

Because IPCC AR6 is now out, reference to the last IPCC report is confusing.  Should be specific as to what assessment/report/CMIP this refers (lines 736)

Will be specified.

Weak (not weal) effect (line 866)

Will be corrected.

Is this global radiation or radiation at the three sites? (918-919)

It is global radiation as stated in the text.

Not significantly different from what?  Does mean there was no trend from 2000 to 2014 or 2000 was not different from 2014? (line 1096)

A trend in the $O_3$ mean concentration in Northern Europe from 2000 to 2014 could not be identified, given the internal variability. We will rephrase the sentence.

Eutrophication of what, terrestrial ecosystems or surface water? (line 1111)

We will rephrase the sentence "for eutrophication of open water bodies in all European countries".

$O_3$ (not O3). (line 1125)

Will be corrected.

Kniebusch et al. (2019b) also identified. (line 1155)

Will be corrected.

It is not clear how temperature increases affected stream flow if precipitation increases are unclear, by increased evapotraspiration? (line 1165)

Very good point. We will rephrase the paragraph:

The observed temperature increases have affected stream flow in the northern Baltic Sea region for 1920-2002 in a manner corresponding well to the projected consequences of a continued rise in global temperature in term of increasing winter time discharges (Hisdal et al., 2010). However, the regional impacts of precipitation change on both the observed and projected changes in stream flow are still unclear as the combined effects of changes in precipitation and temperature are still not well known (Stahl et al. 2010).

Should this be 0.18 day $(°C)^{-1}$ as on the previous line? (line 1287) . .

No, this is correct as it refers to the change in precipitation ("drying"). We will clarify and write "...$cm^{-1}$ decrease in precipitation"
. Drainage Basin (as defined by Vogt et al, 2007; etc.) (line 1437) as

Will be corrected.

reported by Hock et al. (2019) is . . . (line 1439)

Will be corrected.

Farther (not further) south as this refers to distance (line 1476).

Will be corrected.

Is 139 $10^3$ km$^2$ correct? (line 1553)

Yes, but we will change to 139,000 km$^2$.

Comma after Lehmann et al. (2017). (line 1669)

Will be corrected.

Farthest (not furthest) as this refers to distance. (line 1775)

Will be corrected.

Meaning of "shortening oxygen" is unclear. (line 1902)

Will be rephrased.

It doesn't seem to follow how the less productive coastal zone of the northern Baltic Sea explains why hypoxia is rare along the southern and south-eastern coastline.  (lines 1917-1919)

Will be rephrased. "counteracting vertical oxygen supply and natural ventilation by oxygen-rich saltwater intrusions from the North Sea"

Berner et al. (2018) presented further . . . (line 1995)

Will be corrected.

This is a long and complicated sentence, recommending breaking it into two (lines 2195-2199)

Integrated approaches encompassing all of the ecosystem-components discussed above are needed in order to understand and manage the linkages among large-scale and long-term climate effects. These are driven by synergistic interactions of climate change-related physical and chemical drivers with other factors, such as eutrophication or large-scale fisheries, which complicate human adaptation to the changing marine ecosystem (Niiranen et al., 2013; Blenckner et al., 2015; Hyytiäinen et al., 2019; Stenseth et al., 2020; Bonsdorff, 2021).

What about weaking of the polar vortex causing greater meandering of the jet stream? (line 2203)

This is really interesting, but we think that this topic is not within the scope of this paper. To our understanding there is no scientific consensus. No change of the text.

The changes are MORE similar that over the land area. (line 2266)

Will be corrected. It should be "Over the Baltic Sea, the changes are similar to those over the land area"

Projection . . . IS uncertain because .  . . (line 2285)

Will be corrected.

Farther (not further) poleward as this refers to distance (line 2290)

Will be corrected.

The future scenarios for shipping do not include a future where the use of hydrocarbon fuels or at least emissions of $CO_2$ are greatly restricted to meet GHG reduction requirements.  Could the authors speculate what this might mean? (lines 2353)

This is a valid point. The current available studies for shipping in the Baltic Sea do not include the strong fossil fuel emission reductions IMO is postulating (i.e. 50% less greenhouse gas emissiones by 2050). The IMO secretary-general states in the foreword of the Fourth IMO Greenhouse Gas Study: "The Study demonstrates that whilst further improvement of the carbon intensity of shipping can be achieved, it will be difficult to achieve IMO's 2050 GHG reduction ambition only through energy-saving technologies and speed reduction of ships."

New synthetic fuels are needed or alternative propulsion. We are just running our CTM for scenarios addressing those requirements. Results will be available in spring next year, earliest.

In a synthesis paper for Shipping in the Baltic Sea, we will show that under current legislation and quite strong energy efficiency assumptions (stronger than the EEDI from IMO), CO2 emissions in the BS will drop down to only about 78% in 2040 compared to the value in 2014. Other measures are needed, while the use of LNG as fuel is a good solution for reducing air pollutants like $NO_X$, $SO_2$ and PM, $CO_2$ emissions remain considerable. Methane slip during transport and operation can even compensate for the reduced $CO_2$ emissions. This paper will not be published in time for this summary.

We will close the paragraph with a more general remark (line 2369):

"The pollutant concentrations reported in this section may drop to yet not known lower values if the shipping sector is (partly) successful in meeting the IMO target in greenhouse gas emission reduction of 50% by 2050. This reduction is only possible if low-carbon alternative fuels will be introduced, employing a high energy efficiency as it is already considered in scenarios used in Karl et al. (2019a) will not be sufficient. The new fuels will also lead to altered emissions of pollutants."

In this section, is the use of "likely" versus "very likely" consistent with IPCC? (line 2372)

We will change "may have an influence" because it is difficult to state this cause-and-effect relationship for all systems of the hydrosphere.

This paragraph refers to both mitigation measures and adaptation measures.  This is confusing in light of the way those terms are used in climate change assessments. (lines 2439-2244)

We agree and we will use "anthropogenic measures".

Here again adaptation and mitigation are both used in a somewhat confusing way (lines 2450-2464)

We will rephrase the paragraphs.

This sentence is unclear. (line 2556-2557)

We suggest the following alternative formulation: "*The authors argued that a more comprehensive assessment of forest management as a strategy to achieve the goals of the Paris Agreement should go beyond the reduction of atmospheric $CO_2$ and, thus, the reduction of the radiative imbalance at the top of the atmosphere. They suggested…*"

To which models are you referring?  Do you mean under both RCP2.6 and RCP8.5 emissions pathways?  (line 2568-2569)

We will rephrase the first paragraph to clarify the changes under the various RCPs.

Shouldn't this be median -25 cm? (line 2756)

Yes, will be corrected.

Is this also assuming a RCP8.5 pathway?  (lines 2762-2763)

Yes, we will clarify.

There is a need for a reference for this paragraph, I assume it is BACC II Author Team (2015).  These conclusions about decreased pH are contradicted somewhat by the previous discussion in section 3.2.5.7.2. (lines 2878-2883)

We will add the BACC II Author Team 2015 reference and clarify the apparent contradiction.

Aberle et al. (2015) showed . . . (lines 3009-3010)

Will be corrected.

The concept of retreat of marine species may not be clear for readers unfamiliar with the Baltic Sea, perhaps this can be more accurately stated as reduced penetration of marine species into the Baltic Sea. (line 3024)

We will add new text in line 3024

… many species in the Baltic Sea. Because of the projected decline in salinity, a reduced penetration of marine species, such as bladderwrack, eelgrass and blue mussel, into the Baltic Sea has been predicted (Vuorinen et al., 2015). A large number of other species is affiliated with such keystone species, and species distribution modelling has indicated that, e.g., a decrease of bladderwrack will have large effects…

Here again, it seems to be assumed that shipping will continue to depend on the use of fossil fuels. (lines 3236-3237)

New text: The efficiency of SOx scrubbing depends on the temperature, salinity and pH of the seawater, and eventually ends up contaminating the Baltic Sea (Turner et al., 2018). Shipping itself affects climate through combusting fossil fuels, although the emissions can be expected to be reduced with the expected increase of renewable energy within the European Union (EC, 2021)

EC, 2021. Directive of the European parliament and of the council amending Directive (EU) 2018/2001 of the European Parliament and of the Council, Regulation (EU) 2018/1999 of the European Parliament and of the Council and Directive 98/70/EC of the European Parliament.

Isn't it more accurate to state that how these practices will change in response to climate change is yet to be determined? (line 3245-3246)

We will rephrase this sentence to emphasize the research needs in this respect.

Perhaps state that there is YET little direct evidence THAT THIS IS OCCURRING? (lines 3261-3262)

Thank you, as for the previous comment, we will rephrase the sentence.

It is mentioned earlier that warmer temperatures should allow the establishment of more nonindigenous species.  This bears repeating here. (line 3263).

We will add the information.

Won't microplastics also be greatly affected by societal decisions about the use of plastics, in part influenced by efforts to decarbonize? (line 3283-3285)

Thank you, that is a good point; changes in the use of plastics and regulations in response to the problem can be expected but the effects are uncertain. We will add a short sentence in this respect.

Should the authors continue to use Celsius rather than Kelvin here? (line 3399)

Yes, we decided to use Celsius in the entire manuscript will change here.

. . . strongly affected by whether warming is allowed to proceed to the point of destabilizing Antarctic ice sheets. (line 3605)

We will rephrase the sentence.

What does it mean to have low confidence in a statement that changes could not be detected? (line 3750)

The confidence level refers to our knowledge about the change in large-scale circulation and not to the specific statement. We will explain the confidence levels better.

If this trend is almost statistically significant, why isn't it medium confidence, just less that a 95% threshold for high confidence? (line 3829)

We agree and change to medium confidence.

Why is there only low confidence in the statement that larger runoff would lead to larger nutrient inputs? (lines 3920-3921)

We agree and change to medium confidence.

---

## Author Response (AR1)

**List of changes and comments**

1) In the revised version, we followed the advice of both reviewers and all comments have been addressed. For details, see the published response to reviewer 1 and 2.
2) In addition, some typos and unclear sentences were corrected.
3) All changes are marked in the version with tracked changes.
4) High-resolution versions of the figures will be provided separately.
5) Figure 2 will be updated and missing monitoring stations will be added.